# Discovering Interpretable Algorithms by Decompiling Transformers to RASP

Xinting Huang [* 1]   Aleksandra Bakalova [* 1]   Satwik Bhattamishra [2]   William Merrill [3]   Michael Hahn [1]

## Abstract

Recent work has shown that the computations of Transformers can be simulated in the RASP family of programming languages. These findings have enabled improved understanding of the expressive capacity and generalization abilities of Transformers. In particular, Transformers have been suggested to length-generalize exactly on problems that have simple RASP programs. However, it remains open whether trained models actually implement simple interpretable programs. In this paper, we present a general method to extract such programs from trained Transformers. The idea is to faithfully re-parameterize a Transformer as a RASP program and then apply causal interventions to discover a small sufficient sub-program. In experiments on small Transformers trained on algorithmic and formal language tasks, we show that our method often recovers simple and interpretable RASP programs from length-generalizing transformers. Our results provide the most direct evidence so far that Transformers internally implement simple RASP programs.[1]

## 1. Introduction

Understanding the computations in trained Transformers is a central goal of mechanistic interpretability. Over the past few years, a complementary theoretical line has sharpened this question by giving symbolic characterizations of Transformer computation. In particular, the RASP language (Weiss et al., 2021) and its dialects (e.g. Strobl et al., 2025; Yang & Chiang, 2024; Zhou et al., 2024) simulate broad classes of Transformer architectures, and have been used to explain when and why models generalize on specific tasks. A recurring thesis in this literature is that *simple programs imply robust generalization*: if a task admits a short RASP program, then a Transformer trained on shorter inputs will generalize to longer inputs (RASP length generalization conjecture, Zhou et al., 2024; Huang et al., 2025). Yet, this story has a missing piece: Theoretical results show that Transformers *can* realize RASP-like computations, but they do not show that trained models *do* so in a human-interpretable way. Interpretability methods such as circuit discovery (e.g. Conmy et al., 2023) identify sparse computation graphs supporting particular behaviors, but the resulting objects are *circuits* rather than *programs*: they are usually tied to specific input lengths and input templates, and do not directly yield an algorithm uniformly across input lengths. Lindner et al. (2023) map RASP programs *to* Transformers, but the inverse problem—recovering a compact symbolic program *from* a trained Transformer—has remained open. This gap is especially salient for the length-generalization conjecture: if length-generalizing models succeed because they internally implement short RASP programs, we would like to *extract* those programs and inspect them.

In this work we introduce a general decompilation pipeline for recovering interpretable RASP-like programs from trained Transformers. Our starting point is a faithful reparameterization: we define a target dialect, **Decompiled RASP (D-RASP)**, whose primitives mirror Transformer computation while exposing intermediate variables in interpretable spaces such as token and position bases. Under a linearization assumption for layer normalization, we show that a GPT-2 style Transformer can be translated into a D-RASP program that exactly preserves its input–output behavior. Faithful translation alone does not yield interpretability: the naive program is exponentially large in depth, reflecting the many paths through the residual stream. Our key methodological contribution is thus a second step discovering a *minimal sufficient sub-program*. Using causal interventions, we prune program components while preserving behavioral match to the original model, and replace learned components with simple primitives when possible. The result is often a short, readable program that implements the same algorithmic behavior as the trained Transformer.

We validate this *reparameterize-and-simplify* approach on

*Equal contribution [1]Saarland Informatics Campus, Saarland University [2]University of Oxford [3]Allen Institute for AI. Correspondence to: Xinting Huang <xhuang@lst.uni-saarland.de>, Aleksandra Bakalova <abakalov@lst.uni-saarland.de>.

*Proceedings of the 43$^{rd}$ International Conference on Machine Learning*, Seoul, South Korea. PMLR 306, 2026. Copyright 2026 by the author(s).

[1]Code link: https://github.com/lacoco-lab/decompiling_transformers

Blog post: https://lacoco-lab.github.io/home/decompiling_transformers

small GPT-2 style models trained on algorithmic and formal-language benchmarks. On models that length-generalize, decompilation often recovers compact programs aligning with known hypotheses from theory and interpretability—for example, histogram-based majority computation (cf Weiss et al., 2021), induction-head-based copying (Olsson et al., 2022; Zhou et al., 2024), and bracket counting in bounded-depth Dyck languages (Yao et al., 2021; Wen et al., 2023). In contrast, on non-length-generalizing models, decompilation does not succeed, suggesting that these models rely on more entangled, non-program-like mechanisms.

Our results provide direct evidence that, at least in the controlled setting of small models trained on formal problems, length-generalizing Transformers can internally implement simple RASP-like algorithms—and that these algorithms can be extracted automatically as symbolic programs.

**The paper is organized as follows:**  Section 2 introduces D-RASP, our target RASP dialect. Section 3 describes the program extraction method, which includes a faithful translation (Step 1) as well as pruning and simplification (Step 2). Section 4 reports the experimental results and extracted programs, Section 5 shows how our approach fits into the existing literature, Section 6 discusses the implications of our research, and Section 7 concludes the paper.

## 2. Decompiled RASP (D-RASP)

Here, we define our target dialect of RASP, **Decompiled RASP** (D-RASP). It is centered around selectors, aggregation, and element-wise operations as the original RASP (Weiss et al., 2021), but with adapted definitions. Let $\Sigma$ be the (finite) alphabet, $\Sigma = \{\sigma_1, \ldots, \sigma_{|\Sigma|}\}$, and let $T \in \mathbb{N}$ be the maximum context size of the model. We also use $N$ to denote the length of a given input ($1 \leq N \leq T$)

D-RASP has two types of variables: The first type is **activation variables**, $v \in \mathbb{R}^{d \times N}$ where $d = d(v)$ is specific to this variable and specified in the program. For a variable $v$, we write $d(v)$ for its dimensionality, so that $v \in \mathbb{R}^{d(v) \times N}$. In our decompiled programs, typical values for $d(v)$ may be $|\Sigma|$ (for encoding token information) or $T$ (for encoding position information). We write $v(i)$ for $v_{.,i} \in \mathbb{R}^{d(v)}$. The second type is **selector variables**, $\alpha \in \mathbb{R}^{N \times N}$; these determine the aggregation of information across positions.

Each program has access to two initial variables holding one-hot vectors for positions and tokens (Figure 2):

$$\text{pos}(i)_j = \delta_{j,i} = \mathbb{I}[j = i] \qquad \text{pos} \in \mathbb{R}^{T \times N}$$
$$\text{token}(i)_j = \delta_{\sigma_j, x_i} = \mathbb{I}[\sigma_j = x_i] \quad \text{token} \in \mathbb{R}^{|\Sigma| \times N}$$

where, for an input string $x \in \Sigma^N$, $x_i \in \Sigma$ indicates the token at position $i$. Other variables are created by operations.

**Selectors and Aggregation** Self-attention operations are captured using `select` and `aggregate` operators. Given activation variables $v_k, v_q$, we produce a **selector** $\alpha \in \mathbb{R}^{N \times N}$

$$\alpha = \texttt{select}\Big(\texttt{k=}v_k, \texttt{q=}v_q, \texttt{op=}\boldsymbol{A}\Big) \quad (\boldsymbol{A} \in \mathbb{R}^{d(v_q) \times d(v_k)})$$
$$\alpha = \texttt{select}\Big(\texttt{k=}v_k, \texttt{op=}\boldsymbol{b}\Big) \qquad (\boldsymbol{b} \in \mathbb{R}^{1 \times d(v_k)}),$$

evaluating to the tensor ($1 \leq s \leq i \leq N$):

$$\alpha(i, s) = v_q(i)^\top \boldsymbol{A}\, v_k(s) \quad \text{or}$$

$$\alpha(i, s) = \boldsymbol{b}\, v_k(s)$$

The $\boldsymbol{A}, \boldsymbol{b}$ tensors are part of the program; they might be one of a set of hard-coded constructs from a library of primitives (e.g., the identity matrix), or something arbitrary. Selectors can only be used as input to an **aggregation** operation:

$$v = \texttt{aggregate(s=}\alpha_1 + \cdots + \alpha_p, \texttt{v=}w)$$

where $\alpha_i$ are selector variables, $w$ an activation variable; $v$ an activation variable; with output defined as

$$v(i) = \sum_{j \leq i} a_{i,j} w(j) \qquad \text{where}$$

$$a_{i,s} = \frac{\exp(\alpha(i, s)_1 + \cdots + \alpha(i, s)_p)}{\sum_{s' \leq i} \exp(\alpha(i, s')_1 + \cdots + \alpha(i, s')_p)}$$

Next, **elementwise operations** capture MLPs. They take a set of variables and produce a a new variable:

$$v = \texttt{element\_wise\_op}(v_1, \ldots, v_s, \texttt{func=}f) \quad (1)$$

where $f : \mathbb{R}^{d(v_1) \times \cdots \times d(v_s)} \to \mathbb{R}^{d(v)}$ is an arbitrary function provided with the program; $v$ and the inputs $v_j$ are activation variables. It evaluates to the tensor $v \in \mathbb{R}^{d(v) \times N}$:

$$v(i) = f(v_1(i), \ldots, v_s(i)) \quad (2)$$

At the end, we compute **next-token logits**, $p = \texttt{project(inp=}v, \texttt{op=}\boldsymbol{A})$ defined as

$$p(j) = \boldsymbol{A} \cdot v(j) \quad (3)$$

where $\boldsymbol{A} \in \mathbb{R}^{|\Sigma| \times d(v)}$, $d(p) = |\Sigma|$; or $p = \texttt{project(op=}\boldsymbol{b})$ defined as $p = \boldsymbol{b}$ ($\boldsymbol{b} \in \mathbb{R}^{|\Sigma|}$). We obtain the output as

$$\texttt{prediction}(j) := \Phi(p_1(j) + \cdots + p_s(j)) \quad (4)$$

where $\Phi \in \{\texttt{softmax}, \texttt{sigmoid}\}$ depending on the task; each $d(p_i) = |\Sigma|$ and $d(output) = |\Sigma|$.

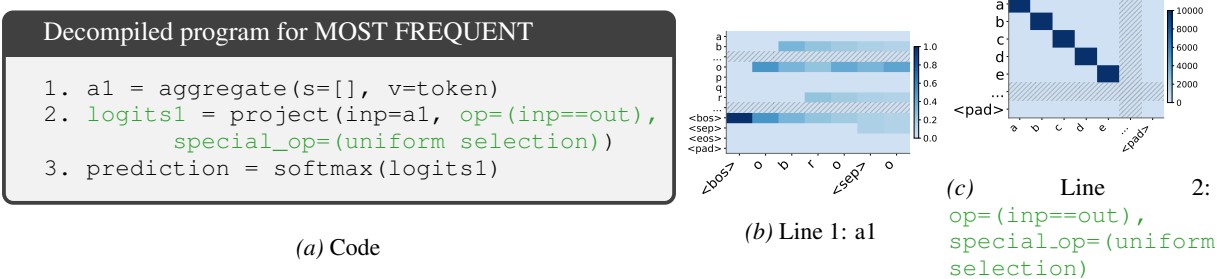

**(a) Code**

```
1. a1 = aggregate(s=[], v=token)
2. logits1 = project(inp=a1, op=(inp==out),
        special_op=(uniform selection))
3. prediction = softmax(logits1)
```

Decompiled program for MOST FREQUENT

**(b) Line 1: a1**

**(c) Line 2:** `op=(inp==out), special_op=(uniform selection)`

*Figure 1.* Program (a) for finding the most frequent character in a string, extracted from a real transformer. An example input is `BOS o b r o SEP o`; the model is trained to predict the last token "o". Line 1 computes, at each position, the relative frequencies of symbols at preceding positions ($a1 \in \mathbb{R}^{|\Sigma| \times N}$, in (b). x-axis = input string, y-axis = variable dims., by aggregating over $token \in \mathbb{R}^{|\Sigma| \times N}$ (Figure 2b). In Line 1, s=[] indicates that *no* selector is used, i.e., the weights $a_{i,j}$ in aggregation are constant. Line 2 projects a1 to output logits via the matrix in (c) (rows = input dims., columns = output dims.). For ordinary tokens (covered by op=), each token receives an output logit proportional to its frequency, corresponding to a temperature-scaled identity matrix. Logits for special tokens are excluded (`special_op=(uniform selection)`), precluding outputting BOS/SEP on length-1 strings (App. Figure 43b has logits1). Overall, the program assigns the highest output logit to the most frequent non-BOS/SEP token. The program is extracted from a 1-layer 4-head transformer (App. J.6); an equivalent program results from a 4-layer 4-head transformer (App. J.7).

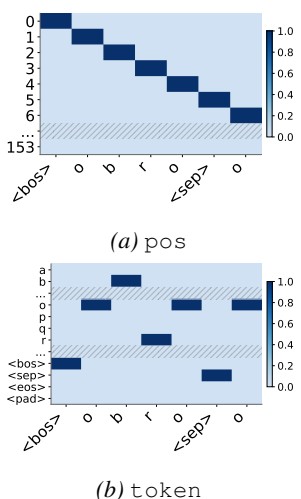

**(a) pos**

**(b) token**

*Figure 2.* Initial variables on the example input used in Figure 1. x-axis = input string $\in \Sigma^N$; y-axis = activation dimensions

**Primitives** D-RASP programs can use arbitrary tensors $A, b$ for selectors and next-token projections, and arbitrary maps $f$ for elementwise operations. We provide a library of tensors $A, b$ for selector and next-token logits, and elementwise operations $f$. We provide the full library in Appendices G and H. Examples for $A$ in select include the identity matrix ("k == q") and an off-by-one shifted identity matrix ("k == q-1"). We find it useful to define separate primitives for normal tokens and special tokens (BOS, EOS, SEP, PAD), provided using separate op= (for normal tokens) and special_op= (for special tokens) arguments. We only do this as syntactic sugar when using primitives; for a general non-primitive matrix $A$, we just write op=$A$. Examples for $f$ in element_wise_op include a "hard-max" operation ($f_{harden}(x) := e_{\arg\max_j x}$, i.e., creating a one-hot vector at the maximal component of the vector $x$). In decompilation, where possible, we will attempt to use these primitives, as detailed below.

**Comparison to other RASP dialects** D-RASP differs from other RASP dialects (e.g. Weiss et al., 2021; Yang & Chiang, 2024; Strobl et al., 2025) in using softmax for ag-

gregation, in keeping with Transformers. This is useful for enabling direct and faithful translation from Transformers; it enables both non-uniform attention (unlike C-RASP, Yang & Chiang (2024)) and attention over an unbounded number of position (unlike B-RASP, Yang et al. (2024)). The most relevant comparison for us here is to C-RASP, which has been linked to length generalizability of transformers (Yang & Chiang, 2024; Yang et al., 2025; Huang et al., 2025; Jobanputra et al., 2025; Chen et al., 2025; Jiang et al., 2025). Superficially, there is a major difference: C-RASP programs only produce counts and boolean outputs, whereas D-RASP produces general real-valued activations. However, if we slightly alter the semantics of D-RASP so that the outcome of aggregate or element_wise_op is rounded to some $p$-bit precision (which in practice happens on real-world hardware), and also enforce a rationality constraint on the parameters, then we identify a fragment equivalent to C-RASP (Theorem 2.1, proof in App. C). Merrill & Sabharwal (2023) show that log-precision transformers can be expressed in the logic FO(M). As C-RASP defines a strict subset of FO(M) (Huang et al., 2025), this fragment also defines a strict subset for FO(M).

**Theorem 2.1** (Correspondence of D-RASP and C-RASP). *Let $\mathcal{D}$ be the class of D-RASP programs not using pos where all tensor entries $A_{ij}, b_i$ (for the parameters of select, project) are in $\{-\infty, \} \cup \{\log q : q \in \mathbb{Q}_+\}$. Then the programs in $\mathcal{D}$, using the rounded semantics, define exactly the same functions as C-RASP.*

## 3. Decompilation Method

Our pipeline applies to GPT-2 style models with absolute positional encodings. Although modern LLMs typically use different positional encoding strategies, we focus on this

architecture because the theory of length generalization is especially well developed for it (Zhou et al., 2024; Huang et al., 2025; Izzo et al., 2025); we further discuss this in FAQ 1.

Here, we describe our decompilation method. Our method consists of two steps: reparamaterization and simplification. The first part translates the transformer into a D-RASP program. The second one simplifies the program while empirically aiming to preserve faithfulness.

### 3.1. Step 1: From Transformers to Programs

We view a transformer as a length-preserving function that maps a sequence $x \in \Sigma^*$ with $|x| = N$ ($N \leq T$) to a sequence of next-token logits in $\in \mathbb{R}^{|\Sigma| \times N}$. Our reparameterization applies to transformers satisfying:

**Assumption 3.1** (Linear Layer Norm Assumption). A transformer $T$ satisfies the *Linear Layer Norm Assumption* (LLNA) iff each layer norm operation can be replaced with a linear operation for some constant parameter $\gamma'$ with negligible change to the input-output behavior:

$$\frac{x - \overline{x}}{\sqrt{\sigma^2(x) + \epsilon}} \gamma + \beta \quad \Rightarrow \quad (x - \overline{x}) \gamma' + \beta \qquad (5)$$

While we generally allow this assumption to hold approximately, for Theorem 3.2 we assume exact equality between the left- and right-hand sides of (5).

The motivation for this assumption comes from Baroni et al. (2025), who empirically show that trained language models can be fine-tuned to satisfy LLNA with little performance drop. We also find empirically that LLNA tends to hold for length-generalizable models (Tables 1, 2), which are precisely the class of models for which prior work suggests that short RASP programs should exist (Zhou et al., 2024; Huang et al., 2025). We refer the reader to FAQ 2 for further discussion of LLNA.

We then show:

**Theorem 3.2.** *Consider a GPT-2-style transformer satisfying LLNA with exact equality between left- and right-hand sides of (5). There is a D-RASP program defining the same input-output map; it can be explicitly obtained from the transformer's parameters. Indeed, the residual stream at each layer can be recovered linearly from a set of variables in the D-RASP program.*

We provide the full formal proof in Appendix B.3. Here, we give an intuitive discussion of the construction. Each path in the transformer leading up to a given layer is represented by a variable. Here, we provide an example. The variables pos, token simulate the input layer, which can be recovered as $\boldsymbol{E}\text{token} + \boldsymbol{P}\text{pos}$, where $\boldsymbol{E} \in \mathbb{R}^{d \times |\Sigma|}$ is the token embedding matrix, and $\boldsymbol{P} \in \mathbb{R}^{d \times T}$ is the position

embedding matrix. We simulate the first layer as follows. For each attention head $h = 1, \ldots, H$, we define four selectors, describing the four possible key-query interactions between the two variables from the input layer:

$$\alpha_{1,h} = \texttt{select}(k = \text{pos}, q = \text{token}, op = \boldsymbol{E}^T \boldsymbol{Q}_{1,h}^T \boldsymbol{K}_{1,h} \boldsymbol{P})$$
$$\alpha_{1,h}^{'} = \texttt{select}(k = \text{token}, q = \text{token}, op = \boldsymbol{E}^T \boldsymbol{Q}_{1,h}^T \boldsymbol{K}_{1,h} \boldsymbol{E})$$
$$\alpha_{1,h}^{''} = \texttt{select}(k = \text{pos}, q = \text{pos}, op = \boldsymbol{P}^T \boldsymbol{Q}_{1,h}^T \boldsymbol{K}_{1,h} \boldsymbol{P})$$
$$\alpha_{1,h}^{'''} = \texttt{select}(k = \text{token}, q = \text{pos}, op = \boldsymbol{P}^T \boldsymbol{Q}_{1,h}^T \boldsymbol{K}_{1,h} \boldsymbol{E})$$

where $\boldsymbol{K}_{l,h}, \boldsymbol{Q}_{l,h} \in \mathbb{R}^{d_h \times d}$ project the residual stream onto the key or query activation of the $h$-th head in the $l$-th layer.

We further define two output variables, reflecting aggregation of token or position counts:

$$v_{\langle pos, \langle 1,h \rangle \rangle} = \texttt{aggregate}(\alpha_{1,h} + \alpha_{1,h}^{'} + \alpha_{1,h}^{''} + \alpha_{1,h}^{'''}, \text{pos})$$
$$v_{\langle tok, \langle 1,h \rangle \rangle} = \texttt{aggregate}(\alpha_{1,h} + \alpha_{1,h}^{'} + \alpha_{1,h}^{''} + \alpha_{1,h}^{'''}, \text{token})$$

By construction, the output of the original attention head can be recovered as[2]

$$\boldsymbol{V}_{1,h} \boldsymbol{P} v_{\langle pos, \langle 1,h \rangle \rangle} + \boldsymbol{V}_{1,h} \boldsymbol{E} v_{\langle tok, \langle 1,h \rangle \rangle} \qquad (6)$$

The pre-MLP residual stream in layer 1 is recovered as the sum of these terms for all attention heads, plus $\boldsymbol{E}\text{token} + \boldsymbol{P}\text{pos}$. The $2H$ variables defined in layer 1, plus token, pos jointly serve as input to an element_wise_op representing the MLP, mapping to $\mathbb{R}^d$. We thus have represented the residual stream in layer 1 in terms of $2 + 2H + 1$ variables. This construction can be repeated recursively layer by layer. Now, by induction, any layer can be described as a linear combination of variables $v_1, \ldots, v_k$, with some coefficient matrices $\boldsymbol{C}_1, \ldots, \boldsymbol{C}_k, (\boldsymbol{C}_i \in \mathbb{R}^{d \times d(v_i)})$ obtained from the transformer parameters: $\sum_{i=1}^{k} \boldsymbol{C}_i v_i$. For each of these variables in the top layer, we add a project statement, with $A$ defined in terms of the original model's unembedding matrix $\boldsymbol{U} \in \mathbb{R}^{|\Sigma| \times d}$: $p_i = \texttt{project}(v_i, \text{op=}\boldsymbol{U}\boldsymbol{C}_i)$ for $i = 1, \ldots, k$. By construction, the original next-token logits equal $\sum_{i=1}^{k} p_i \in \mathbb{R}^{|\Sigma| \times N}$. We show a full example (one-layer transformer) in Appendix Figure 9. The discussion above disregards layer norm. Under LLNA, the weights $\gamma'$ and $\beta$ can be absorbed into the op= tensors.

Notably, the size of the program guaranteed by Theorem 3.2 is exponential in the original transformer's size (FAQ 11), necessitating a simplification step to make it tractable, which we turn to next.

### 3.2. Step 2: Discovering Minimal Sufficient Programs

In Step 2, we aim to simplify the program by (i) pruning components that are causally irrelevant, (ii) simplifying the

---

[2]For simplicity of notation, we subsume the $\boldsymbol{V}$ and $\boldsymbol{O}$ matrices of GPT-2 into a single $\boldsymbol{V} \in \mathbb{R}^{d \times d}$ matrix (App. B.1).

matrices and vectors that appear in the program by referring, where possible, to primitives from the library. We measure faithfulness by quantifying the *match accuracy*, i.e., the fraction of input instances where the program gives the same response as the original model. Formally, the match accuracy is the proportion of inputs where the pruned model (with MLP replaced) makes the same prediction (next token with highest probability) as the original model in *all* token positions on which the model receives training signal. Throughout, we deem decompilation to be *causally faithful* when the match accuracy is $\geq 90\%$.

**Step 2.1: Causal Pruning & Linearizing Layer Norm** For each `aggregate` operator, we try to replace selector inputs with 0 or with a key-only selector if such replacements have negligible causal effect. For each `element_wise_op` operator, we try to remove $v_s$ inputs; each removed input is replaced with a constant absorbed into the transformation $f$. Meanwhile, we also try to remove inputs from the final softmax, replacing them with a constant absorbed into the $bias$. Variables that do not enter into downstream components are removed. We show an example in Appendix, Figure 9.

Removing parts of the program that are not causally relevant to the final output can be viewed as removing edges in the model's computation graph. This corresponds to a problem addressed by causal ablation methods, which are standard in interpretability research, particularly in circuit discovery (Conmy et al., 2023; Li & Janson, 2024; Bhaskar et al., 2024). We base our implementation on Li & Janson (2024), pruning edges using trainable gates with an objective that combines the KL divergence between the pruned and original models with a sparsity penalty.

However, the exponential size of the program guaranteed by Theorem 3.2 makes causal pruning nontrivial in our setting. To avoid unfolding the full program and to make pruning scalable, we instead unfold and prune the program in a multistage procedure. Full details are provided in Appendix F.

Meanwhile, we also train $\gamma'$ (5) for each layer norm to minimize the KL divergence. We then replace all layer norms in a model with their linearized form:

$$\text{LayerNorm}(x) \approx \boldsymbol{L}x + \beta, \qquad \boldsymbol{L} = \left(\boldsymbol{I} - \frac{\boldsymbol{1}\boldsymbol{1}^T}{d(x)}\right)\gamma'$$

where $\boldsymbol{1} \in \mathbb{R}^{d(x)}$ is a vector with all 1s. $\boldsymbol{L}$ parameters are absorbed into the attention and MLP parameters (App. B.2).

**Step 2.2: Explaining Per-Position Transformations** We try to explain `element_wise_op` by replacing them with known primitives. Before that, we try to simplify per-position transformations $v = \texttt{element\_wise\_op}(v_1, \ldots, v_s, \texttt{func}=f)$ as a sum of single-input operations $w_j = \texttt{element\_wise\_op}(v_j, \texttt{func}=f_j)$ $(j = 1, \ldots, s)$,

where $f_1, \ldots, f_s$ are re-fitted as MLPs to recover the original downstream model outputs as $v(i) \approx \sum_j w_j(i)$. The motivation for this is that single-input operations are likely to be, in general, easier to interpret. Any downstream appearance of $v$ is replaced by $w_1, \ldots, w_s$; these can then be individually pruned using causal pruning. In practice, we integrate this process with the causal pruning stage.

We then try to replace `element_wise_op` operations with primitives in our library shown in Appendix G. Given $v = \texttt{element\_wise\_op}(x, \texttt{func}=f)$, we replace it with new $w = \texttt{element\_wise\_op}(x, \texttt{func}=f_{primitive})$. $w$ has dimension determined by the selected $f_{primitive}$. $w$ stays interpretable since $f_{primitive}$ and input $x$ are interpretable. For example, if it is $f_{no-op}(x) = x$ and $x$ is `token`, then the output dimension is $|\Sigma|$ instead of $d$. We fit a matrix $\boldsymbol{C} \in \mathbb{R}^{d \times d(w)}$ to minimize $\|v - \boldsymbol{C}w\|_F^2$, where $\boldsymbol{C}$ has a closed-form solution. We then replace all downstream occurrences of $v$ by $w$, and absorb $\boldsymbol{C}$ into the matrices associated with those downstream occurrences. For example, when $v$ appears in $\texttt{project}(v, \boldsymbol{A})$, we replace it by $\texttt{project}(w, \boldsymbol{A}\boldsymbol{C})$; similar change applies when $v$ appears in `select` or `element_wise_op`. In sum, whereas the nonlinearity originally applies to activations $\in \mathbb{R}^d$, it now maps between interpretable variables.

If no primitive achieves high enough match accuracy, we interpret the `element_wise_op` by inspecting its inputs and outputs as follows. On the input side, we store the interpretable activation variables that are fed into the `element_wise_op`. On the output side, if the operation feeds into unembedding, then we can linearly transform it to have output dimension $|\Sigma|$ and see which output tokens are promoted given different activation variable as input; if it feeds as $k$ (similarly $q$) into `select`, we check what activation variables of $q$ (similarly $k$) are associated with different inputs; in other words, we inspect pairs of activation variables that result in high attention logit. This technique can be viewed as a principled adaptation of LogitLens (nostalgebraist, 2020), but unlike that technique also ensures that downstream effects through paths other than the residual stream are correctly accounted for (App. G.5).

**Step 2.3: Explaining tensors in `select` and `project`** While the $\boldsymbol{A}$ and $\boldsymbol{b}$ tensors in the `select` and `project` operators, and the bias in the final softmax yielded by Theorem 3.2 tend to be interpretable (Figs. 6, 5, 7) due to their interpretable dimensions (FAQ 7), we aim to automatically explain them further to reduce the burden of human interpretation.

We replace these tensors with a matrix from the library, whenever this is causally faithful. When this is not possible, we optimize the matrices to be sparse and have integer values. See Appendix H for the library of primitives and details of the optimization process.

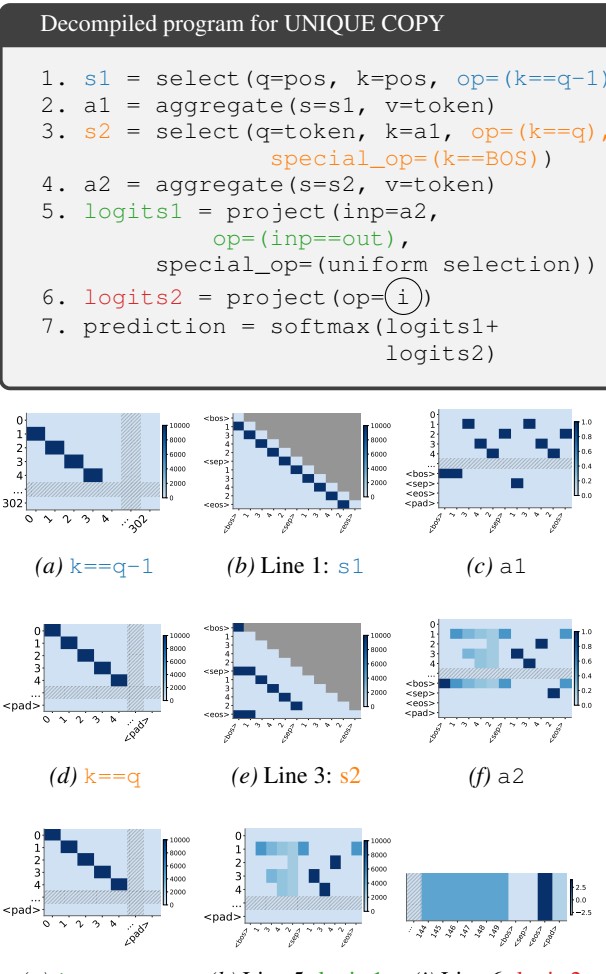

*Figure 3.* An extracted program for copying a string without repetitions: The input | BOS | 1 | 3 | 4 | 2 | SEP | is completed with the string | 1 | 3 | 4 | 2 | EOS |. The primitive "k == q-1" (line 1) selects the position immediately preceding the query position; it is given by the matrix in (a) (rows = query dimensions; columns = key dimensions) and yields the selector in (b) (x-axis = key positions; y-axis = query positions). The variable a1 (c) (x-axis = positions in the input; y-axis = dimensions of the variable) stores the previous token at each position, as a one-hot vector. The variables token and a1 jointly encode the bigrams appearing in the string, which is sufficient for copying a string without repetitions. The variable a2 (f) then stores the token to be output next. It is obtained by finding the position $j$ where a1(j) equals token(i), and retrieving token(j) into a2(i). Here, the primitive "k==q" selects a matching token; it is given by the matrix in (d). In these cases, there is special behavior on the special tokens, ensuring that the transformer produces the first token in the given sequence after ⟨sep⟩. a2 is directly forwarded into logits1 (h) by a scaled identity matrix (g). The bias in line 6 (i) favors EOS in the absence of other outputs, ensuring EOS is output at the end. The program recapitulates the "induction head" motif; importantly, our decompilation pipeline provides it fully automatically.

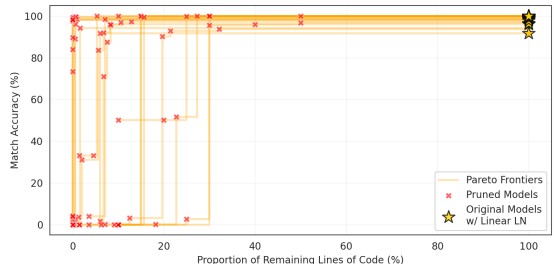

*Figure 4.* Causal Pruning (Step 2.1): Pareto frontiers over lines of code vs. match accuracy for models satisfying LLNA.

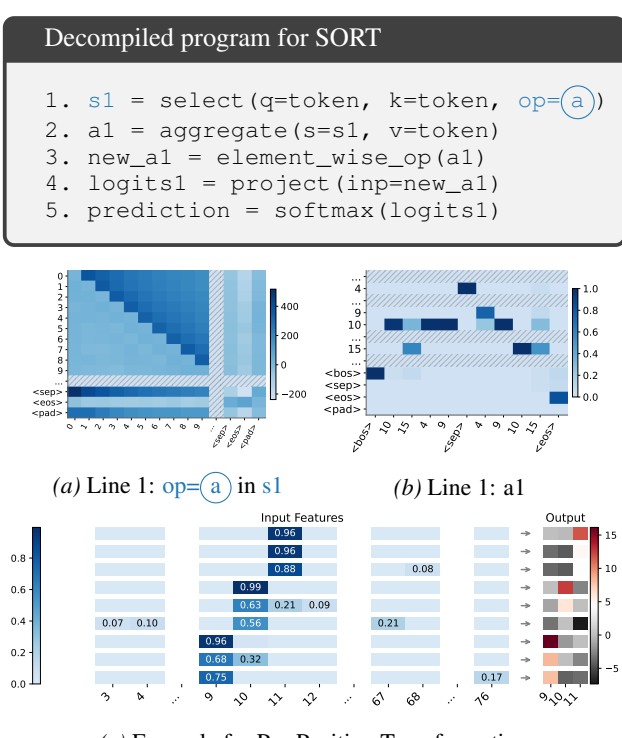

*Figure 5.* An extracted program for sorting a list of integers: The input | BOS | 10 | 15 | 4 | 9 | SEP | is completed with the string | 4 | 9 | 10 | 15 | EOS |. Line 1 uses a select operation given by the matrix in (a). Entries are largest when $k$ is slightly larger than $q$, and then fall off. As a result, a1 collects tokens bigger than the current one, and among those favoring the smallest one (b). SEP always collects the smallest number to start counting. Line 3 performs a nonlinear operation $f : \mathbb{R}^{|\Sigma|} \to \mathbb{R}^d$, not replaced by a primitive. We inspect behavior on twelve sample inputs a1(i) ∈ $\mathbb{R}^{|\Sigma|}$, each denoted by a row in (c, left), together with the associated output vector new_a1(i) ∈ $\mathbb{R}^{|\Sigma|}$ (c, right). The operation "hardens" the input, selecting the dimension for which entries are largest in the input (e.g., 9, 10, 11 in the examples), effectively making the vector one-hot. When there is no number bigger than the current one, the attention is diffuse (as shown in the matrix in (a), entries are very similar when k is ≤ q), making the input a1 also diffuse. The same operation in Line 3 then maps this type of inputs to EOS (Fig. 51 shows that EOS is promoted when none of the non-BOS entry contains decisively high value). App. J.8 has more heatmaps.

# 4. Results

We train small GPT2-like models on a suite of algorithmic problems based on Huang et al. (2025); Zhou et al. (2024), and on a battery of finite-state languages from Bhattamishra et al. (2020) (list in Appendix D). We view the algorithmic problems as sequence prediction, predicting a single token or a sequence ending in EOS. For the formal languages, we instead have the model at each step output the set of allowed next tokens (Bhattamishra et al., 2020; Sarrof et al., 2024; Huang et al., 2025), which we implement by feeding the logits into sigmoid to obtain a validity label $\in [0, 1]$ for each symbol. As formal languages have no inherent probability distribution over strings, this is a simpler strategy than language modeling.

Following the setup of Huang et al. (2025), we train models at input lengths $\leq 50$ and evaluate length generalization at input lengths in $[51, 150]$, because we are interested in how length-generalization relates to interpretability of models. Random offsets ensure that all position embeddings are trained. See details in Appendix D.2. The decompilation pipeline (Step 2 in Sec. 3) is run on inputs of length $\leq 150$, the match accuracy is measured at length $\leq 150$ on random i.i.d. samples. For each task, we train models with different hyperparameters (e.g., number of layers and heads $\in \{1, 2, 4\}$, model dimensions $\in \{16, 64, 256\}$, etc.). We keep at least one representative model for each task, and for some algorithmic tasks also have additional models with different architecture and different length-generalization performance. For each task, we keep the model with best length generalization performance across all models trained with different hyperparameters. For each model, we trace out a Pareto frontier (match accuracy vs. program length) by varying the sparsity coefficients in Step 2.1. We declare decompilation to be successful if some programs have match accuracy $\geq 0.9$, and select the shortest one.

**When does Decompilation succeed?** We expect decompilation to be successful only for models that actually internally implement short D-RASP programs (FAQ 6). Prior work (Zhou et al., 2024; Huang et al., 2025) conjectured a connection between the existence of (short) RASP programs and length-generalization; accordingly, we expect only length-generalizable models to be decompilable.

Eight **algorithmic tasks** showed length generalization at least at some hyperparameters, three did not. Results (App. Table 1) show that the length-generalizing models can be decompiled, and models that do not length-generalize could not be decompiled. A similar trend is also observed for **formal languages** (App. Table 2). 14 languages showed length generalization; decompilation succeeded on 9 of them. 3 languages did not show length generalization, and decompilation failed. Thus, on the formal languages, decompilation

succeeds in the majority of length-generalizing models.

Moreover, we find that, when decompilation succeeds, we can usually vastly cut down the size of programs compared to the initial D-RASP reparameterization (Figure 4). For instance, for finding the most frequent character in a string, we trained 1-layer 4-head and 4-layer 4-head models, with D-RASP translations of 56 and 16,201,616 lines, respectively. Causal pruning results in equivalent 3-line programs in both cases. In contrast, for non-length-generalizing models, even if keeping layer norm on, pruning usually has only very limited success (App. Figure 12). Thus, interestingly, even though all models are transformers of similar sizes, generalizable models tend to be more interpretable.

The choice of architecture (Figure 10), random seed (Table 3), and training checkpoint (Figure 11) all affect a model's length-generalization performance, and thus its decompilability. While models trained with different random seeds (Appendix E) and pruned with different sparsity penalties (Appendix E) can generally yield different programs, the extracted programs are often similar in practice. We discuss the decompilability of other trained models in Appendix E.

## 4.1. Decompilation recovers interpretable algorithms

We show all decompiled programs in App. J–K. We show a program extracted for **finding the most frequent** character in a string in Figure 1: a select+aggregate operation aggregates a histogram of tokens, which directly determines the output logits (Figure 1(c)), so that the most frequent symbol has the highest output logit.

We show a program extracted from a transformer trained to **copy a string in which each token is unique** (Unique Copy) in Figure 3. In this case, all operations can be expressed in terms of predefined primitives. The first select+aggregate annotates each position with the directly preceding symbol; the second one then corresponds to an "induction head" (Olsson et al., 2022): it retrieves whichever prior symbol was preceded by the current symbol. This indeed is the RASP mechanism hypothesized to perform copying without repetition in Zhou et al. (2024); Huang et al. (2025). For a transformer trained to copy *backwards*, we find a corresponding program implementing an "anti-induction" head operation (App. J.12).

For transformers trained to **copy strings in which each bigram is unique**, we find (i) a program that encodes trigram statistics into a variable of dimension $|\Sigma|^3$ via a per-position operation, and (ii) a different program that only uses selection/aggregation operations (App. J.9, J.10).

We show an extracted program for **sorting a list of integers** in Figure 5. Here, operations are not expressed in terms of primitives, but the output of decompilation is nonethe-

less interpretable: there is a `select` operation favoring the smallest keys larger than the query, and a per-position operation making the aggregated histogram vector one-hot.

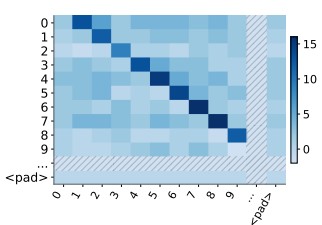

In a **counting** task (Zhou et al., 2024), the model completes e.g. ⟨bos⟩ 12 16 ⟨sep⟩ with 12 13 14 15 16⟨eos⟩. Incrementing counts while counting is done by a projection matrix acting directly on `token`:  $p_1 =$ `project(token)`, `op=` Figure 6) ); it maps "0" to "1", "1" to "2", etc. Simultaneously, `aggregate` operations control behavior at the beginning and end of counting (App. J.4).

*Figure 6.* See text. Rows denote input dimensions; Columns denote output dimensions.

We next discuss formal languages. For a few **strictly local languages**, where each symbol determines the next possible symbols, no `aggregate` is needed. For instance, for $a^+b^+c^+d^+e^+$, the program obtains the output logits as `project(token)` where the `op=` matrix (Figure 7) maps each symbol to the possible next symbols: "a" and "b" can follow "a"; "b" and "c" can follow "b"; "e" and EOS can follow "e".

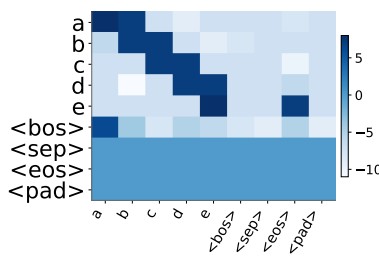

*Figure 7.* See text. Rows denote input dimensions; Columns denote output dimensions

The benchmark includes **bounded-depth Dyck languages** at various depths. The models for D2, D4, D12 were decompiled; the model for D3 failed in the layer norm linearization. The algorithms are similar, but the shortest program (Figure 8) is found for D4, i.e., the language of well-nested strings over one pair of parentheses (opening "a" and closing "b") with depth at most four (equivalently, the formal language $(a(a(a(ab)^*b)^*b)^*b)^*)$: uniform aggregation creates the histogram of BOS, opening, and closing brackets; a per-position activation then performs a threshold calculation: e.g., EOS is allowed only when a and b are balanced. At two other depths (D2, D12), the aggegation intriguingly assigns different `select` weights to "a" and "b"; this nonuniformity cascades into somewhat more complicated variants of the same algorithm (App. K.1,K.2).

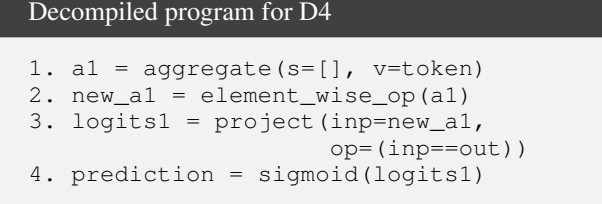

Decompiled program for D4

```
1. a1 = aggregate(s=[], v=token)
2. new_a1 = element_wise_op(a1)
3. logits1 = project(inp=new_a1,
                     op=(inp==out))
4. prediction = sigmoid(logits1)
```

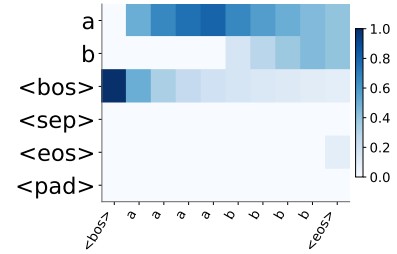

*(a)* Line 1: `a1`

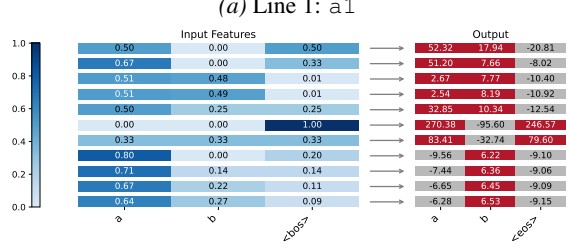

*(b)* Line 2: `element_wise_op`

*Figure 8.* Decompiled program for depth-4 Dyck language, on sample input | BOS | a | a | a | a | b | b | b | b | EOS |. The program at each step outputs the set of allowed next characters. `a1` contains the relative frequencies of a, b, BOS. Line 2 applies an element-wise operation, whose output is directly forwarded to next-token predictions in Line 3 via the identity matrix (`project(...,op=(inp==out))`). We show the input-output behavior in line 2 on 11 sample inputs `a1` $\in \mathbb{R}^{|\Sigma|}$ in (b). "a" is allowed (i.e., receives positive logit) only when #a-#b $< 4 \cdot$ #BOS; "b" is allowed when #a > #b, and "EOS" is allowed when the string is exactly balanced (#a = #b). More in App. K.3.

## 5. Related Work

**Symbolic Representations of Transformers** Translations from transformers to logics from the theoretical literature on transformers, such as logic with majority quantifiers (Merrill & Sabharwal, 2023), C-RASP (Yang et al., 2025), and first-order logic (Li & Cotterell, 2025), usually create one formula for each dimension of the residual stream, which means that the size of the program will grow with the hidden dimension, and the meaning of the individual formulas is not directly interpretable, as the dimensions of the residual stream are not in general individually interpretable. A key advantage of our D-RASP translation over these prior theoretical translations is that the size of the resulting program is independent of the transformer's hidden dimension, with embedding, query, key, value, unembedding transformations all absorbed into matrices; in many cases, these

have interpretable dimensions such as $|\Sigma|$. This is a powerful property especially when models are overparameterized for optimization reasons.

Tracr (Lindner et al., 2023) *compiles* RASP programs to transformers. Multiple works aim to recover RASP programs from Tracr-generated models (Baker, 2023; Thurnherr & Riesen, 2024; Langosco et al., 2024). Our work differs by decompiling models *trained* with standard methods, rather than models specifically constructed to mirror RASP algorithms. Friedman et al. (2023); Lai-Dang et al. (2025); Zhang et al. (2026) train transformer variants amenable to symbolic interpretation, e.g. with discrete representations and attention weights. Our work instead applies to normal transformers trained with standard methods.

**Comparison to Circuit Discovery**    The most prominent approach to understanding the internal computations of Transformers is **circuit discovery** (e.g. Elhage et al., 2021; Wang et al., 2023; Conmy et al., 2023). This approach views the transformer's computation as a computation graph and finds a subgraph faithfully representing the computation of the full transformer. The major difference to our work is that we discover *programs* rather than *computation graphs*. With this, we address a major challenge to circuit discovery, namely providing abstractions of model behavior that are valid across a broad range of input strings. The reason is that computation graphs discovered by circuit discovery are tightly linked to specific input templates; generalizing beyond individual templates is nontrivial. Programs over sequences are more natural representations of algorithms across input lengths, whereas circuits are generally specific to a given input length. Decompilation addresses this challenge, at least in the setting of small transformers trained on algorithmic problems: programs apply to the full input space. An example is the unique copy task (Figure 3), where standard circuit discovery would lead to a circuit where the final prediction is connected to all positions with redundant connections, as the bigram relevant to the next prediction might be located anywhere in the input. In contrast, programs abstract these connections into a single aggregation operation. This difference is even more pronounced in the context of length generalization: a circuit is tied to a specific input length, whereas our method allows us to find length-generalizing RASP programs.

## 6. Discussion

Our results support and strengthen the RASP length generalization conjecture (Zhou et al., 2024; Huang et al., 2025): Prior work suggests that the presence of length generalization tracks (C-)RASP expressiveness (Huang et al., 2025). Our work strengthens this link, confirming that, when transformers show length generalization on synthetic tasks, they in fact often internally implement simple RASP algorithms.

Concerning the applicability of our method, we emphasize that we do not claim that it can be successfully applied to any trained model. Some models may internally implement heuristic, non-length-generalizable solutions, for which no short D-RASP program exists and thus none can be extracted. We argue that one should not expect non-length-generalizing models to be decompilable into RASP in the first place, since prior work suggests that such models are unlikely to implement interpretable and concise RASP algorithms (Zhou et al., 2024; Huang et al., 2025). We view the inability to decompile such models not as a failure of the method, but as an interesting scientific insight about transformers.

It is, however, possible that the method may sometimes fail even when a model implements a short D-RASP algorithm, for example if LLNA does not hold or if the MLPs are not splittable. Empirically, this appears to be uncommon, as LLNA tends to hold for length-generalizing models. Developing more general solutions to these limitations would be a promising direction for future work.

An intriguing question is whether our method can also be used for interpreting LLMs as a complement to circuit discovery. It appears unlikely that language models as a whole would be expressible as simple RASP programs due to the extremely rich variation of their training data. However, they might develop interpretable sub-programs for individual algorithmic tasks. The presence of induction heads in real-world language models (e.g. Olsson et al., 2022; Wang et al., 2023; Yin & Steinhardt, 2025), effectively a subprogram isomorphic to the program in Figure 3, suggests this might be the case. Testing this is a nontrivial and interesting question for future work, for which the present work provides foundations.

## 7. Conclusion

We introduce a method for extracting simple programs from transformers trained on individual synthetic tasks. Our results constitute the most direct evidence so far that length-generalizing transformers trained on algorithmic and formal language problems often implement interpretable RASP programs. It is an interesting question if this also applies to language models, which are effectively trained to perform a large variety of tasks simultaneously. It is plausible that language models might implement modular sub-programs for different subtasks; testing this is an interesting problem for future research.

## Impact Statement

This paper presents work whose goal is to advance the field of Machine Learning. There are many potential societal consequences of our work, none of which we feel must be specifically highlighted here.

## Acknowledgments

This research is funded by the Deutsche Forschungsgemeinschaft (DFG, German Research Foundation) – Project-ID 232722074 – SFB 1102 "Information Density and Linguistic Encoding"; Project-ID 471607914 – GRK 2853/1 "Neuroexplicit Models of Language, Vision, and Action". MH thanks the participants of the Dagstuhl Seminar "Theory of Neural Language Models" for useful discussion.

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

# A. FAQ

1. *Why use APE, rather than positional encodings more popular in modern LLMs?*

   We use APE mainly because theoretical understanding of length generalization in terms of RASP is best-developed for APE (Zhou et al., 2024; Huang et al., 2025; Izzo et al., 2025). We also note that a lot of interpretability research has taken GPT-2 as a reference (e.g. Wang et al., 2023; Hanna et al., 2023; Baroni et al., 2025), and that APE subsumes NoPE; in fact our programs in App. J, K often don't include `pos`.

2. *What is the need for having LLNA? Isn't it limiting the applicability of the method?*

   We believe LLNA is reasonable for the following reasons:

   (a) When LLNA holds, the decompilation is faithful to the original model. We preserve the input-output behavior of the underlying network, making sure that if a program exists, it is *causally faithful* to the *original* model, even though its layer norm (LN) is linearized.

   (b) LLNA tends to hold for length-generalizable models, which are exactly the class of models for which the RASP programs are suggested to exist. Thus, while theoretically LLNA could lead to inability of extracting a program, this would empirically concern cases where a short D-RASP program is anyways not implemented by a model. Therefore, we believe that our method is *not limited by* this assumption.

3. *Why isn't the method applied to LLMs?*

   The goal of this paper is to answer a foundational question about transformers, i.e., whether they learn interpretable RASP-like algorithms. While this has been suggested by theoretical studies of transformer's learning abilities (Zhou et al., 2024; Huang et al., 2025), our work provides the most direct empirical test of this idea so far. Our work thus targets transformers specifically trained on individual algorithmic problems. Language models in contrast are trained on highly varied data, and effectively are exposed to many different kinds of algorithmic tasks. It appears unlikely that language models as a whole would be expressible as simple RASP programs due to the extremely rich variation of their training data (e.g., grammar, world knowledge). It is an interesting question if language models also develop interpretable sub-programs for individual algorithmic tasks. The presence of induction heads in real-world language models (e.g. Olsson et al., 2022; Wang et al., 2023; Yin & Steinhardt, 2025), effectively a subprogram isomorphic to the program in Figure 3, suggests this might be case. Testing this is a nontrivial and interesting question for future work, for which the present work provides foundations.

4. *Why can't one just use existing circuit discovery methods?*

   While circuit discovery and D-RASP decompilation are methodologically closely connected, their outputs differ in a few meaningful and important ways. A D-RASP program by design assigns an interpretable operation to each of the attention heads and MLPs that stay part of the program, while circuit discovery methods only show which pathways are important in a computation graph and do not address a question of interpreting the operations. Moreover, operations in a D-RASP program act across all tokens in the input sequence, treating tokens and positions as variables, while circuit discovery methods are highly template-dependent and aligned to specific positions in the input (e.g. Haklay et al., 2025). Importantly, existing circuit discovery methods define circuits usually on component level, which is not aligned with variables in D-RASP. A vertex in component-level circuits can be an aggregation of many D-RASP variables, making it harder to perform interpretation.

5. *Why not build on existing translations from transformers to logic?*

   As explained in the Discussion section, existing translations typically create one formula for each dimension in the residual stream, which in general are not individually interpretable. While it is conceivable that pruning might cut this down, our strategy circumvents such a blow-up. For instance, it directly results in isomorphic programs for the "Most Frequent" task (Figure 1, 256 dimensions in residual stream of original transformer) and the binary variant "Majority" (App. J.1, 16 dimensions in residual stream of original transformer).

6. *Can decompilation be applied to any model? When is decompilation expected to work?*

   We emphasize that *we do not claim that the method can be successfully applied to any trained model.* Some models might be internally implementing a heuristic-based non-length-generalizable solution, and a short D-RASP program might not exist for those models, and thus cannot be extracted. We argue that one *should not* hope to decompile non-generalizing models into RASP in the first place, since these would not be expected to implement interpretable and

concise RASP algorithms. We view the inability to decompile non-generalizing models not as a failure of a method, but as an interesting *scientific insight* about transformers.

Decompilation requires LLNA to hold. In practice, it also requires the model to allow pruning of a large fraction of the program lines (i.e., to have a sparse sufficient sub-program), as small programs ease interpretation. Our findings suggest that these properties are often satisfied in length-generalizing models, but not in models that fail to length-generalize.

7. *Is there a guarantee that all variables will end up with interpretable dimensionalities such as $|\Sigma|$?*

   Theoretically, there can be cases where the decompilation will result in variables of dimension equal to that of the original model, e.g., when the output of an MLP feeds into attention and then again into an MLP. We do not observe such cases in our experiments, where decompiled variables all have interpretable dimensions such as $|\Sigma|, T$.

8. *Isn't the decompilation pipeline complicated? Are all steps of the decompilation pipeline needed?*

   The pipeline has three steps: (i) causal pruning and linearizing layer norm, (ii) explaining per-position transformations, (iii) explaining tensors in `select` and `project`. The first steps is needed in order to obtain problems of manageable size. The second and third steps are helpful in automating part of the interpretation. Whenever no primitive is found, interpretation via manual inspection is still feasible.

9. *How are the hyperparameters justified?*

   In most cases, we observed in preliminary experiments that the choice of hyperparameters does not significantly affect the result. This is supported by the fact that we use the same set of hyperparameters for all models. For sparsity coefficient, we always sweep over various values to trace out a Pareto frontier (Figure 4). An important hyperparameter is the accuracy threshold, which defines whether the program is decompiled or not, which we chose based on common sense.

10. *Why do we need a new RASP dialect? Why not decompile into an existing dialect?*

    The most important feature of D-RASP is that it uses softmax (unlike RASP, B-RASP, C-RASP, S-RASP), which makes it suited to faithfully representing transformer computations.

11. *Isn't the exponential size of the D-RASP programs from Theorem 3.2 a big problem?*

    The D-RASP program has exponential number of lines only after the *reparametrization* step. This is not a problem for interpreting final programs as we perform *simplification*, which significantly reduces the number of lines (Figure. 4) This is also not a big problem for pruning, because we do not unroll all lines of D-RASP to perform pruning. rather, pruning is done in a efficient manner, where we use scalable gradient-based approach, and in initial stages prune over groups of lines instead of single line. This is outlined in the main paper, and described in detail in App. F.

12. *How does D-RASP compare to Transformer Programs (Friedman et al., 2023; Lai-Dang et al., 2025) and the Discrete Transformer (Zhang et al., 2026)?*

    On a technical level, there are some similarities; in particular, our definition of separate variables for each path through the residual stream is similar to the disentangled residual stream where the output of the attention heads is concatenated rather than added to the residual stream. That said, there are key differences: first, our decompilation procedure is designed to apply to normal transformers trained using standard methods; second, causal pruning enables small programs despite a theoretically exponential number of paths through the residual stream.

13. *Why don't we train and decompile the model on the same length? Would that make more models decompilable?*

    We also considered decompiling models to programs that match the original model only on training lengths. Indeed for some non-length-generalizing models, this enables decompilation (though the program might not match the model's behavior beyond training lengths). But there are some other models that are still not decompilable, meaning that their inner computational mechanism may be complex and uninterpretable even if we only consider model behavior on a limited length range on which the model does perform well. An important consideration is that programs obtained by decompiling on only the training length so would not help us on understanding why generalization fails, as they do not match model behavior on longer length. See details in Appendix E.

# B. D-RASP Construction

## B.1. Notation for Transformers

Here, we introduce our notation for GPT-2-style transformers, closely following Huang et al. (2025). By LLNA, we assume layer norm has been linearized and absorbed into the parameters (Section B.2). For simplicity of notation, we omit biases in attention, and merge value and output matrices $V$, $O$ into a single $V$ matrix per head. Our implementation, however, takes all such details into account faithfully.

**Computation of Activations and Outputs**  If $L$ is the number of layers, then we write the output of layer $l = 1, \ldots, L$ at position $i = 1, \ldots, N(T)$ as $\boldsymbol{y}_i^{(l)} \in \mathbb{R}^d$. We set

$$\boldsymbol{y}_i^{(0)} = \boldsymbol{E}_{x_i} + \boldsymbol{P}_i \quad i = 1, \ldots, |x| \tag{7}$$

where $x_i \in \Sigma$ is the input symbol at position $i$, $\boldsymbol{E}$ is the word embedding matrix, and $\boldsymbol{P}$ is the position embedding matrix. Attention logits, at query position $i$ and key position $j$ are computed as

$$a_{i,j}^{(l,h)} = (\boldsymbol{y}_i^{(l-1)})^T \boldsymbol{Q}_{l,h}^T \boldsymbol{K}_{l,h} \boldsymbol{y}_j^{(l-1)} \ \text{ for } 1 \le j \le i \le |x|; \ l = 1, \ldots, L; \ h = 1, \ldots, H \tag{8}$$

We assume standard softmax attention:

$$\boldsymbol{Y}_i^{(l)} := \boldsymbol{y}_i^{(l-1)} + \sum_{h=1}^{H} \frac{\sum_{j=1}^{i} \exp\left(a_{i,j}^{(l,h)}\right) \boldsymbol{V}_{l,h} \boldsymbol{y}_j^{(l-1)}}{\sum_{j=1}^{i} \exp\left(a_{i,j}^{(l,h)}\right)} \tag{9}$$

After each attention block, the activations are passed through a one-layer MLP:

$$\boldsymbol{y}_i^{(l)} := \boldsymbol{Y}_i^{(l)} + \psi_l(\boldsymbol{Y}_i^{(l)}) \tag{10}$$

where $\psi_l$ is an MLP.

A transformer maps strings $x \in \Sigma^*$ ($|x| \le T$) to vectors of next-token prediction logits, $\in \mathbb{R}^{|\Sigma| \times |x|}$, obtained as $\boldsymbol{U}\boldsymbol{y}_i^{(L)}$ ($i = 1, \ldots, |x|$) for the unembedding matrix $\boldsymbol{U} \in \mathbb{R}^{|\Sigma| \times d}$.

## B.2. Reparameterizing Layer Norm when LLNA is Satisfied

When LLNA is satisfied (for some constant parameter $\gamma'$):

$$\frac{x - \overline{x}}{\sqrt{\sigma^2(x) + \epsilon}} \gamma + \beta \quad \Rightarrow \quad (x - \overline{x}) \gamma' + \beta$$

we can replace all layer norm operations in a model with their linearized form:

$$\text{LayerNorm}(x) \approx \boldsymbol{L}x + \beta, \qquad \boldsymbol{L} = \left(\boldsymbol{I} - \frac{\boldsymbol{1}\boldsymbol{1}^T}{d(x)}\right)\gamma'$$

where $\boldsymbol{1}$ is a vector of all ones. $\boldsymbol{L}$ parameters can be absorbed into the attention parameters:

$$\boldsymbol{Q}_{l,h} \leftarrow \boldsymbol{Q}_{l,h}\boldsymbol{L}_{l,h}; \qquad \boldsymbol{K}_{l,h} \leftarrow \boldsymbol{K}_{l,h}\boldsymbol{L}_{l,h}; \qquad \boldsymbol{V}_{l,h} \leftarrow \boldsymbol{V}_{l,h}\boldsymbol{L}_{l,h}$$

and similarly for the MLP.

## B.3. Detailed Definition of D-RASP Translation

Here, we prove Theorem 3.2, using the notation from Appendix B.1.

Let $L$ be the number of layers of the transformer. Let $H$ be the number of heads.

By a "path" $p$, we refer to a sequence over $\{1, \ldots, L\} \times \{1, \cdots H\}$ that is strictly decreasing in the first component. We set $layer(p) := \max\left(\{0\} \cup \{i : (i, h) \in p\}\right)$; that is, the maximum layer involved in the path.

We use $\langle \dots \rangle$ to indicate ordered tuples.

We will accumulate a set of variables defined in each layer. Some variables will be in token space, others will be in position space, and (when necessary) others will be left in the model's original activation space.

**Definition B.1.** We define a set $\mathcal{V}$ indexing the variables of the D-RASP program as follows. Let $start = \{tok, pos, mlp_1, \dots, mlp_L\}$. Then $\mathcal{V}$ is the smallest set such that:

1. For each $x \in start$, $\langle x, \langle \rangle \rangle \in \mathcal{V}$.

2. For any $l \in [L], h \in [H]$, whenever $layer(\langle x, p \rangle) < l$ and $\langle x, p \rangle \in \mathcal{V}$, then $\langle x, \langle (l, h) \rangle \oplus p \rangle \in \mathcal{V}$.

    where

3. $layer(\langle tok, p \rangle) := layer(p)$

4. $layer(\langle pos, p \rangle) := layer(p)$

5. $layer(\langle mlp_k, p \rangle) := \max(k, layer(p))$

In particular, each $v \in \mathcal{V}$ is a tuple consisting of $x \in start$ and a path $p$.

For each $\langle i, p \rangle \in \mathcal{V}$, our D-RASP program will have a variable $v_{\langle i, p \rangle}$. We will instantiate the variables so that, for each $p$,

$$d(v_{\langle tok, p \rangle}) = |\Sigma|$$
$$d(v_{\langle pos, p \rangle}) = N$$
$$d(v_{\langle mlp_k, p \rangle}) = d$$

Using the variables $v_{\langle i, p \rangle}$, we define the following program: First, $v_{\langle tok, \langle \rangle \rangle} = \texttt{token(en)}$ and $v_{\langle pos, \langle \rangle \rangle} = \texttt{pos}$ are already given in the program. We need to define the other variables. We first define as auxiliary quantities:

$$\mathcal{T}(token) = \boldsymbol{E} \in \mathbb{R}^{d \times |\Sigma|}$$
$$\mathcal{T}(position) = \boldsymbol{P}[:, :N] \in \mathbb{R}^{d \times N}$$
$$\mathcal{T}(mlp_l) = \mathrm{I}_{d \times d} \in \mathbb{R}^{d \times d}$$

**Defining Selector Matrices**   Selectors are defined as follows. For layer $l$, head $h$ and each $r, s \in \mathcal{V}, layer(r), layer(s) < l$, we add the line

---

Line defining a selector

$\alpha_{l,h,r,s} = \texttt{select(k=}v_r\texttt{, q=}v_s\texttt{, op=}\boldsymbol{A}_{l,h,r,s}\texttt{)}$

---

Intuitively, we have a different term in the selection for each interaction of existing variables. Here, for the tuple $\langle i_k, p_k \rangle, \langle i_q, p_q \rangle \in \mathcal{V}$ we define the selector matrices:

$$\boldsymbol{A}_{l,h,\langle i_q,p_q \rangle,\langle i_k,p_k \rangle} := \mathcal{T}(i_q)^T \cdot \left[ \prod_{\langle l',h' \rangle \in p_q} \boldsymbol{V}_{l',h'} \right]^T \cdot \boldsymbol{Q}_{l,h}^T \cdot \boldsymbol{K}_{l,h} \cdot \left[ \prod_{\langle l',h' \rangle \in p_k} \boldsymbol{V}_{l',h'} \right] \cdot \mathcal{T}(i_k) \tag{11}$$

where the product $\prod_{\cdot \in p}$ is computed in decreasing order along the entries of the path. For each $i \in start, layer(p) < l$, $h$ we add the line:

---

Line performing aggregation

$v_{\langle i, \langle \langle l,h \rangle \rangle \oplus p \rangle} = \texttt{aggregate(}\sum_{r,s} \alpha_{l,h,r,s}\texttt{, } v_{\langle i,p \rangle}\texttt{)}$

---

where the sum runs over all $r, s$ for which we above defined $\alpha_{l,h,r,s}$.

**Running Example: One-Layer One-Head Model**   For a one-layer one-head model, we obtain the following set of variables indexed by tuples $\langle x, p \rangle \in \mathcal{V}$:

| | |
|---|---|
| $v_{\langle pos, \langle\rangle\rangle}$ | Associated layer: 0 |
| $v_{\langle tok, \langle\rangle\rangle}$ | Associated layer: 0 |
| $v_{\langle pos, \langle\langle 1,1\rangle\rangle\rangle}$ | Associated layer: 1 |
| $v_{\langle tok, \langle\langle 1,1\rangle\rangle\rangle}$ | Associated layer: 1 |
| $v_{\langle mlp_1, \langle\rangle\rangle}$ | Associated layer: 1 |

We first obtain the following selectors and aggregations – recall $\texttt{pos} = v_{\langle pos, \langle\rangle\rangle}$ and $\texttt{token} = v_{\langle tok, \langle\rangle\rangle}$:

---
**Line of generated program**

$1.\alpha_{1,1,v_{\langle tok,\langle\rangle\rangle},v_{\langle tok,\langle\rangle\rangle}} = \texttt{select}(\texttt{q}=v_{\langle tok,\langle\rangle\rangle}, \texttt{k}=v_{\langle tok,\langle\rangle\rangle}, \texttt{op}=\boldsymbol{A}_{1,1,\langle tok,\langle\rangle\rangle,\langle tok,\langle\rangle\rangle})$
$2.\alpha_{1,1,v_{\langle tok,\langle\rangle\rangle},v_{\langle pos,\langle\rangle\rangle}} = \texttt{select}(\texttt{q}=v_{\langle tok,\langle\rangle\rangle}, \texttt{k}=v_{\langle pos,\langle\rangle\rangle}, \texttt{op}=\boldsymbol{A}_{1,1,\langle tok,\langle\rangle\rangle,\langle pos,\langle\rangle\rangle})$
$3.\alpha_{1,1,v_{\langle pos,\langle\rangle\rangle},v_{\langle tok,\langle\rangle\rangle}} = \texttt{select}(\texttt{q}=v_{\langle pos,\langle\rangle\rangle}, \texttt{k}=v_{\langle tok,\langle\rangle\rangle}, \texttt{op}=\boldsymbol{A}_{1,1,\langle pos,\langle\rangle\rangle,\langle tok,\langle\rangle\rangle})$
$4.\alpha_{1,1,v_{\langle pos,\langle\rangle\rangle},v_{\langle pos,\langle\rangle\rangle}} = \texttt{select}(\texttt{q}=v_{\langle pos,\langle\rangle\rangle}, \texttt{k}=v_{\langle pos,\langle\rangle\rangle}, \texttt{op}=\boldsymbol{A}_{1,1,\langle pos,\langle\rangle\rangle,\langle pos,\langle\rangle\rangle})$
$5.v_{\langle tok,\langle\langle 1,1\rangle\rangle\rangle} = \texttt{aggregate}(\texttt{s}=\alpha_{1,1,v_{\langle tok,\langle\rangle\rangle},v_{\langle tok,\langle\rangle\rangle}} + \alpha_{1,1,v_{\langle tok,\langle\rangle\rangle},v_{\langle pos,\langle\rangle\rangle}} + \alpha_{1,1,v_{\langle pos,\langle\rangle\rangle},v_{\langle tok,\langle\rangle\rangle}} + \alpha_{1,1,v_{\langle pos,\langle\rangle\rangle},v_{\langle pos,\langle\rangle\rangle}}, \texttt{v}=v_{\langle tok,\langle\rangle\rangle})$
$6.v_{\langle pos,\langle\langle 1,1\rangle\rangle\rangle} = \texttt{aggregate}(\texttt{s}=\alpha_{1,1,v_{\langle tok,\langle\rangle\rangle},v_{\langle tok,\langle\rangle\rangle}} + \alpha_{1,1,v_{\langle tok,\langle\rangle\rangle},v_{\langle pos,\langle\rangle\rangle}} + \alpha_{1,1,v_{\langle pos,\langle\rangle\rangle},v_{\langle tok,\langle\rangle\rangle}} + \alpha_{1,1,v_{\langle pos,\langle\rangle\rangle},v_{\langle pos,\langle\rangle\rangle}}, \texttt{v}=v_{\langle pos,\langle\rangle\rangle})$

---

With simplified variable naming, as used in our decompiled programs, we get the following equivalent program (suppressing the `op=` arguments):

---
**Line of generated program**

```
1. s1 = select(q=token, k=token)
2. s2 = select(q=token, k=pos)
3. s3 = select(q=pos, k=token)
4. s4 = select(q=pos, k=pos)
5. a1 = aggregate(s=s1+s2+s3+s4, v=token)
6. a2 = aggregate(s=s1+s2+s3+s4, v=pos)
```

---

**Defining elementwise operations**   We set

---
**Line performing per-position operation**

$$v_{\langle \text{mlp}_l, \langle\rangle\rangle} = \texttt{element\_wise\_op}(v_s(i) : s \in \mathcal{V} - \{v_{mlp,l}\}, layer(s) \leq l, \texttt{func}=f)$$

---

where $f$ outputs the output vector of the original MLP $f_{MLP,l}$ of the $l$-th layer:

$$f\left(v_{(s,p)}(i) : (s,p) \in \mathcal{V}, s \neq mlp_l, layer((s,p)) \leq l\right) := f_{MLP,l}\left(\sum_{(s,p)\in\mathcal{V}, s\neq mlp_l, layer((s,p))\leq l} s\right) \tag{12}$$

**Running Example: One-Layer One-Head Model**   In our running example, we have the line:

---
**Line of generated program**

$7. v_{\langle mlp_1,\langle\rangle\rangle} = \texttt{element\_wise\_op}(v_{\langle pos,\langle\rangle\rangle}, v_{\langle tok,\langle\rangle\rangle}, v_{\langle pos,\langle\langle 1,1\rangle\rangle\rangle}, v_{\langle tok,\langle\langle 1,1\rangle\rangle\rangle}, \texttt{func}=f_{MLP,1})$

---

or in the simplified notation (we generally suppress the $f$ argument):

> **Line of generated program**
>
> ```
> 7. m = element_wise_op(token+pos+a1+a2)
> ```

**Treating Unembedding Matrices**    We define the unembedding projections as

$$\boldsymbol{R}_{\langle i,p \rangle} = \boldsymbol{U} \left[ \prod_{\langle l',h' \rangle \in p} V_{h',l'} \right] \mathcal{T}(i) \tag{13}$$

for each $\langle i, p \rangle$ with $layer(p) \leq L$, and set

> **Line performing per-position operation**
>
> $p_{\langle i,p \rangle} = \text{project}(v_{\langle i,p \rangle})$    for each $layer(\langle i,p \rangle) \leq L$, op=$\boldsymbol{R}_{\langle i,p \rangle}$)

The output is obtained as the softmax of all $p_{\langle i,p \rangle}$ variables, without bias.

**Running Example: One-Layer One-Head Model**    In our running example:

> **Lines of generated program (simplified naming)**
>
> 8. $p_{\langle tok,\langle \rangle \rangle}$ = project$(v_{\langle tok,\langle \rangle \rangle},$ op=$\boldsymbol{U}\mathcal{T}(token))$
> 9. $p_{\langle pos,\langle \rangle \rangle}$ = project$(v_{\langle pos,\langle \rangle \rangle},$ op=$\boldsymbol{U}\mathcal{T}(pos))$
> 10. $p_{\langle tok,\langle\langle 1,1 \rangle\rangle \rangle}$ = project$(v_{\langle tok,\langle\langle 1,1 \rangle\rangle \rangle},$ op=$\boldsymbol{U}\mathcal{T}(token))$
> 11. $p_{\langle pos,\langle\langle 1,1 \rangle\rangle \rangle}$ = project$(v_{\langle pos,\langle\langle 1,1 \rangle\rangle \rangle},$ op=$\boldsymbol{U}\mathcal{T}(pos))$
> 12. $p_{\langle mlp_1,\langle \rangle \rangle}$ = project$(v_{\langle mlp_1,\langle \rangle \rangle},$ op=$\boldsymbol{U}\mathcal{T}(mlp_1))$
> 13. prediction = softmax$(p_{\langle tok,\langle \rangle \rangle} + p_{\langle pos,\langle \rangle \rangle} + p_{\langle tok,\langle\langle 1,1 \rangle\rangle \rangle} + p_{\langle pos,\langle\langle 1,1 \rangle\rangle \rangle} + p_{\langle mlp_1,\langle \rangle \rangle})$

or, with simplified variable naming (suppressing the op= arguments):

> **Lines of generated program (simplified naming)**
>
> ```
> 8.  logits1 = project(token)
> 9.  logits2 = project(pos)
> 10. logits3 = project(a1)
> 11. logits4 = project(a2)
> 12. logits5 = project(m)
> 13. prediction = softmax(logits1+logits2+logits3+logits4+logits5)
> ```

### B.4. Proof of Theorem 3.2

**Theorem B.2** (Restated from Theorem 3.2)**.** *Consider a GPT-2-style transformer strictly satisfying LLNA (i.e. ). There is a D-RASP program defining the same input-output map; it can be explicitly obtained from the transformer's parameters. Indeed, the residual stream at each layer can be recovered linearly from a set of variables in the D-RASP program.*

*Proof.* We show that the D-RASP program described in Appendix B.3 satisfies the claims made in the theorem. The key idea is that, in each layer, we can linearly recover the residual stream from appropriate variables in the program. The key claim is the following linear decomposition of the residual stream into variables:

$$\boldsymbol{y}_i^{(l)} = \sum_{(x,p) \in \mathcal{V}:layer(\langle x,p \rangle) \leq l} \left( \prod_{\langle l',h' \rangle \in p} \boldsymbol{V}_{l',h'} \right) \cdot \mathcal{T}(x) \cdot v_{\langle x,p \rangle}(i) \tag{14}$$

where the product is performed in decreasing order according to the order of layers in the elements of the path $p$. Equation 14 shows that one can recover the original transformer's computations linearly. Once this is established, we can recover the original transformer's output by taking $l = L$ and applying the unembedding matrix $U$ on both sides, and applying the definition of $\boldsymbol{R}_{(i,p)}$ (13):

$$
\begin{aligned}
\boldsymbol{U}\boldsymbol{y}_L &= \sum_{layer(\langle i,p\rangle)\leq L} \boldsymbol{U}\left(\prod_{\langle l',h'\rangle\in p} V_{l',h'}\right)\mathcal{T}(i)v_{\langle i,p\rangle} \\
&= \sum_{layer(\langle i,p\rangle)\leq L} \boldsymbol{R}_{(i,p)}v_{\langle i,p\rangle} \\
&= \sum_{layer(\langle i,p\rangle)\leq L)} \texttt{project}(v_{\langle i,p\rangle}, \texttt{op=}(\boldsymbol{R}_{\langle i,p\rangle}))
\end{aligned}
$$

It remains to prove Equation 14. The proof proceeds by induction over the layers. We first consider the input:

$$
\boldsymbol{y}_i^{(0)} = \boldsymbol{E}_{x_i} + \boldsymbol{P}_i = \mathcal{T}(token)\cdot\texttt{token} + \mathcal{T}(position)\cdot\texttt{pos} \tag{15}
$$

where we use that the only variables at layer $\leq 0$ are $\texttt{token}$ and $\texttt{pos}$; $\texttt{pos} = v_{\langle pos,\langle\rangle\rangle}$; $\texttt{token} = v_{\langle tok,\langle\rangle\rangle}$.

We next perform the inductive step, assuming the claim has been shown for all layers $< l$. We first consider the attention logits of each head $h$ in layer $l$ (Equation 8):

$$
a_{i,j}^{(l,h)} = (\boldsymbol{y}_i^{(l-1)})^T\boldsymbol{Q}_{l,h}^T\boldsymbol{K}_{l,h}\boldsymbol{y}_j^{(l-1)} \quad \text{for } 1 \leq j \leq i \leq N \tag{16}
$$

We now rewrite this as follows, using the inductive hypothesis:

$$
\begin{aligned}
a_{i,j}^{(l,h)} &= (\boldsymbol{y}_i^{(l-1)})^T\boldsymbol{Q}_{l,h}^T\boldsymbol{K}_{l,h}\boldsymbol{y}_j^{(l-1)} \\
&= \left(\sum_{(x,p)\in\mathcal{V}:layer(\langle x,p\rangle)\leq l-1}\left(\prod_{\langle l',h'\rangle\in p} \boldsymbol{V}_{l',h'}\right)\cdot\mathcal{T}(x)\cdot v_{\langle x,p\rangle}(i)\right)^T \boldsymbol{Q}_{l,h}^T\boldsymbol{K}_{l,h}\left(\sum_{(y,q)\in\mathcal{V}:layer(\langle y,q\rangle)\leq l-1}\left(\prod_{\langle l',h'\rangle\in p} \boldsymbol{V}_{l',h'}\right)\cdot\mathcal{T}(y)\cdot v_{\langle y,q\rangle}(j)\right) \\
&= \left(\sum_{(x,p)\in\mathcal{V}:layer(\langle x,p\rangle)\leq l-1} v_{\langle x,p\rangle}(i)^T\mathcal{T}(x)^T\left(\prod_{\langle l',h'\rangle\in p} \boldsymbol{V}_{l',h'}\right)\right)^T \boldsymbol{Q}_{l,h}^T\boldsymbol{K}_{l,h}\left(\sum_{(y,q)\in\mathcal{V}:layer(\langle y,q\rangle)\leq l-1}\left(\prod_{\langle l',h'\rangle\in q} \boldsymbol{V}_{l',h'}\right)\cdot\mathcal{T}(y)\cdot v_{\langle y,q\rangle}(j)\right) \\
&= \sum_{(x,p)\in\mathcal{V}:layer(\langle x,p\rangle)\leq l-1 \ (y,q)\in\mathcal{V}:layer(\langle y,q\rangle)\leq l-1} v_{\langle x,p\rangle}(i)^T\mathcal{T}(x)^T\left(\prod_{\langle l',h'\rangle\in p} \boldsymbol{V}_{l',h'}\right)^T \boldsymbol{Q}_{l,h}^T\boldsymbol{K}_{l,h}\left(\prod_{\langle l',h'\rangle\in q} \boldsymbol{V}_{l',h'}\right)\cdot\mathcal{T}(y)\cdot v_{\langle y,q\rangle}(j) \\
&= \sum_{(x,p)\in\mathcal{V}:layer(\langle x,p\rangle)\leq l-1 \ (y,q)\in\mathcal{V}:layer(\langle y,q\rangle)\leq l-1} v_{\langle x,p\rangle}(i)^T\boldsymbol{A}_{l,h,(x,p),(y,q)}^T v_{\langle y,q\rangle}(j)
\end{aligned}
$$

The last term indeed is the sum of all selectors created for head $(l,h)$ under "Line defining a selector" in Appendix B.3 (compare (11)). Thus, the sum of the selectors associated to a head $(l,h)$ exactly recovers the head's attention logit.

As a consequence, for each $\langle x,p\rangle$ with $\langle x,p\rangle < l$, we recover the output of the head as a linear combination of the variables

associated to a path ending in $(l, h)$, i.e., variables of the form $v_{\langle x, \langle\langle l,h \rangle\rangle \oplus p\rangle}$:

$$\sum_{x,p} \left( \prod_{\langle l',h' \rangle \in \langle\langle l,h \rangle \oplus p\rangle} \boldsymbol{V}_{l',h'} \right) \mathcal{T}(x) v_{\langle x, \langle\langle l,h \rangle\rangle \oplus p\rangle}(i)$$

$$= \boldsymbol{V}_{l,h} \cdot \sum_{x,p} \left( \prod_{\langle l',h' \rangle \in p} \boldsymbol{V}_{l',h'} \right) \mathcal{T}(x) v_{\langle x, \langle\langle l,h \rangle\rangle \oplus p\rangle}(i)$$

$$= \boldsymbol{V}_{l,h} \cdot \sum_{x,p} \left( \prod_{\langle l',h' \rangle \in p} \boldsymbol{V}_{l',h'} \right) \mathcal{T}(x) \cdot \texttt{aggregate} \left( \sum_{r,s} \alpha_{l,h,r,s} v_{\langle x,p \rangle} \right)(i)$$

$$= \boldsymbol{V}_{l,h} \cdot \sum_{x,p} \left( \prod_{\langle l',h' \rangle \in p} \boldsymbol{V}_{l',h'} \right) \mathcal{T}(x) \sum_{j=1}^{i} \mathrm{softmax} \left( \sum_{r,s} \alpha_{l,h,r,s} \right)_{i,j} v_{\langle x,p \rangle}(j)$$

$$= \boldsymbol{V}_{l,h} \cdot \sum_{j=1}^{i} \mathrm{softmax} \left( \sum_{r,s} \alpha_{l,h,r,s} \right)_{i,j} \sum_{x,p} \left( \prod_{\langle l',h' \rangle \in p} \boldsymbol{V}_{l',h'} \right) \mathcal{T}(x) v_{\langle x,p \rangle}(j)$$

$$= \sum_{j=1}^{i} \mathrm{softmax} \left( \sum_{r,s} \alpha_{l,h,r,s} \right)_{i,j} \boldsymbol{V}_{l,h} \cdot \boldsymbol{y}_j^{(l-1)} \qquad \text{(i.e., the output of the head)}$$

where $\mathrm{softmax}(\sum_{r,s} \alpha_{l,h,r,s})_{i,j}$ denotes the attention weight with query position $i$ and key position $j$. We thus recover

$$\boldsymbol{Y}_i^{(l)} = \sum_{(x,p): x \neq mlp_l, layer((x,p)) \leq l} \left( \prod_{\langle l',h' \rangle \in p} \boldsymbol{V}_{l',h'} \right) \cdot \mathcal{T}(x) \cdot v_{\langle x,p \rangle}(i) \qquad (17)$$

Taken together with the definition of $v_{\langle mlp_l, \langle\rangle \rangle}$, (14) follows for layer $l$. This concludes the proof. $\qquad \square$

### B.5. Example of D-RASP Translation

In Figure 9, we show a full program provided by Theorem 3.2 on a one-layer one-head transformer. To give an idea of the extent of pruning, we use strike-through show which parts of the program remain (plain) or not (strike-through) in the program shown in Figure 1.

Original:

```
1. s1 = select(q=token, k=token)
2. s2 = select(q=token, k=pos)
3. s3 = select(q=pos, k=token)
4. s4 = select(q=pos, k=pos)
5. a1 = aggregate(s=s1+s2+s3+s4, v=token)
6. a2 = aggregate(s=s1+s2+s3+s4, v=pos)
7. m = elementwise_op(token+pos+a1+a2)
8. logits1 = project(token)
9. logits2 = project(pos)
10. logits3 = project(a1)
11. logits4 = project(a2)
12. logits5 = project(m)
13. prediction = softmax(logits1+logits2+logits3+logits4+logits5)
```

After pruning:

```
1. s1 = select(q=token, k=token)
2. s2 = select(q=token, k=pos)
3. s3 = select(q=pos, k=token)
4. s4 = select(q=pos, k=pos)
5. a1 = aggregate(s=s1+s2+s3+s4, v=token)
6. a2 = aggregate(s=s1+s2+s3+s4, v=pos)
7. m = elementwise_op(token+pos+a1+a2)
8. logits1 = project(token)
9. logits2 = project(pos)
10. logits3 = project(a1)
11. logits4 = project(a2)
12. logits5 = project(m)
13. prediction = softmax(logits1+logits2+logits3+logits4+logits5)
```

*Figure 9.* See text.

## B.6. Size of D-RASP Translation

The size of the D-RASP translation given by Theorem 3.2 is given by:

```
def determine_total_line(num_layer, num_head, split_mlps):
    num_v = 2
    num_line = 0
    for i in range(num_layer):
        num_line += (num_v ** 2 + num_v) * num_head
        num_v += num_v * num_head

        if split_mlps:
            num_line += num_v
            num_v += num_v
        else:
            num_line += 1
            num_v += 1

    num_line += num_v
    num_line += 1 # bias term
    num_line += 1 # prediction
    return num_line
```

Here, split-mlps indicates whether the element-wise operations are split into single-input operations, which further increases the line count before pruning.

# C. Expressivity of D-RASP and C-RASP

**Theorem C.1** (Correspondence of D-RASP and C-RASP, repeated from Theorem 2.1). *Let $\mathcal{D}$ be the class of D-RASP programs not using* `pos` *where all tensor entries $A_{ij}, b_i$ (for the parameters of* `select`, `project`) *are in $\{-\infty, \} \cup \{\log q : q \in \mathbb{Q}_+\}$. Then the programs in $\mathcal{D}$, using the rounded semantics, define exactly the same functions as C-RASP.*

*Remark* C.2. For the rounded semantics, we assume that outputs are rounded to $p$-bit precision for some $p \in \mathbb{N}$. We assume that numbers are rounded downwards (as a consequence, negative numbers are rounded to negative numbers).

By "define the same functions" we mean that both classes define the same maps from strings $x \in \Sigma^*$ to next-token predictions $\in \Sigma^{|x|}$, assigning each token a predicted next token.[3]

We note that, in the absence of `token`, D-RASP programs can be in principle applied at unbounded input lengths, making rigorous comparison to C-RASP feasible. Regarding `token`, the treatment of positional encodings in extensions of C-RASP is nontrivial (Huang et al., 2025). Many of the operations involving `token` that we recover resemble the local (such as in Figure 3) or periodic (such as in Figure 18) functions assumed in Huang et al. (2025), supporting their treatment.

*Proof of Theorem 2.1.* We refer to (Yang & Chiang, 2024; Huang et al., 2025; Yang et al., 2025) for expositions of C-RASP.

We begin with a D-RASP program satisfying the described constraints. By the fixed precision assumption, there is some $C \in \mathbb{Q}_+$ such that all possible values in activation variables are in $\mathfrak{V} := \{0, 2^{-p}, -2^{-p}, 2 \cdot 2^{-p}, -2 \cdot 2^{-p}, \ldots, C, -C\}$.[4] In particular, each activation variable can only take a finite number of values, which will be hard-coded in the C-RASP program (similar to the argument translating between C-RASP and transformers in (Yang et al., 2025)). For each activation variable $v$, each $z \in \mathfrak{V}$, and each $j = 1, \ldots, d(v)$, we want to define a C-RASP predicate $P_{v,z,j}$ that is true at position $i$ if and only if $v(i)_j = z$. We show this by induction; it is clearly true at `token`. For the inductive step, it is also clearly true that element-wise transformations preserve this property, because they can be directly hard-coded. We need to consider aggregation. By assumption, each softmax entry is rational. Thus, determining the rounded value of the aggregation outcome boils down to checking linear inequalities with integer coefficients over the frequencies of different activation vectors over $\mathcal{D}$, which can be performed in C-RASP. By the same argument, we can express in C-RASP which token receives largest softmax logits (or positive sigmoid logits) and is thus the predicted next token.

For the other direction, consider a C-RASP program. For each Boolean predicate $P$, we will design a variable $v$ such that $v(i) \in \{0, 1\}$ and $v(i) = 1$ iff $P(i)$ holds. Rather than simulating individual count-valued variables from C-RASP in D-RASP, it will be advantageous to directly simulate Boolean-valued terms created as inequalities between linear combinations of counts:

$$\sum_{k=1}^{A} \lambda_k \#[j \leq i] P_k(j) \geq 0 \tag{18}$$

where $A$ is some constant, $\lambda_1, \ldots, \lambda_A \in \mathbb{Z}$ are constant coefficients, $P_1, \ldots, P_A$ are Boolean-valued predicates (already defined and translated to D-RASP variables by induction). We use an element-wise operation to prepare an integer-valued (one-dimensional) D-RASP variable $v(i) = \lambda_1 1_{P_1(i)} + \cdots + \lambda_i 1_{P_A(i)}$. We then use uniform aggregation (with a zero selector) to obtain an aggregate one-dimensional variable $a(i) = \text{round}\left(\frac{\sum_{k=1}^{A} \lambda_k \#[j \leq i] P_k(j)}{i}\right)$. By checking if the rounded value is $\geq 0$ or not, we can reconstruct the truth value of the linear inequality (18). Any Boolean-valued formula in C-RASP is a Boolean combination of such ienqualities; an elementwise operation can perform such combinations. Overall, we have shown that C-RASP programs can be translated to the described D-RASP fragment. □

# D. Tasks and Model Training Details

In this section we describe how we obtain the models before interpreting them.

## D.1. Task Definitions and Data generation

Here, we define the tasks used in this paper formally. BOS, SEP, and EOS are special tokens that represent beginning of the sequence, separator, and end of sequence. All the tasks below except for those under "Formal Languages" referred to as

---

[3]In the case of a sigmoid output as we use for formal languages, we can analogously formalize this as assigning each token a *set* of possible next tokens.

[4]By induction, the set of possible entries in the vectors is bounded, ensuring the existence of such a $C$.

algorithmic tasks.

We use two different training objectives for algorithmic tasks and formal language tasks. **For algorithmic tasks**, we use language modeling as training objective, the language modeling loss only includes loss over the subsequence starting from SEP. In other words, models are trained to predict the next tokens over all tokens after SEP. For single-token answer, namely Majority, Binary Majority, and Parity, Special tokens only include BOS and SEP. For other algorithmic tasks, BOS, SEP and EOS always occur in each sequence. The models performance is measured by **Task Accuracy**, which is the proportion of the inputs on which the model makes correct prediction on *all* token positions that are included in the training loss. **For formal language tasks**, a different paradigm is used. Because formal language tasks are essentially recognition tasks, which means we need to find negative examples that do not belong to the language. However, generating such negative examples can be quite difficult empirically if they are generated purely at random, most of the time they can be quite trivial, the model usually end up with learning shortcuts because "hard" examples are not enough. In other words negative examples should cover all failure mode with enough probability such that model can learn the real rule of the formal language. Butoi et al. (2025) introduced perturbation-based method to generate non-trivial negative samples. In our preliminary experiments, we found that their method, while effectively mitigating the problem, still cannot completely solve the problem. Models learn shortcut on, for example, $D_n$. Therefore, like previous work (Bhattamishra et al., 2020; Sarrof et al., 2024; Huang et al., 2025), we use predictive modeling as the training objective. Specifically, the inputs are only valid examples, but on each of the token in the input, the model needs to predict all the possible valid next tokens, including the EOS token. For example, given the language $(aa)^*$. The input $\boxed{\texttt{BOS}\,|\,\texttt{a}\,|\,\texttt{a}\,|\,\texttt{a}\,|\,\texttt{a}}$ will have the target $\boxed{\texttt{(a, EOS)}\,|\,\texttt{(a,)}\,|\,\texttt{(a, EOS)}\,|\,\texttt{(a,)}\,|\,\texttt{(a, EOS)}}$ where "(a, EOS)" means the next token can be either "a" or "EOS". At the same time we replace the softmax after the last linear layer (so-called language modeling head) in the transformer with a sigmoid function, so that the model can predict whether each token in the vocabulary can be the valid next token. And we use binary cross entropy (BCE) loss as the training objective. Correspondingly the **Task Accuracy** is defined as the proportion of inputs on which the model can make correct prediction of *all* valid and invalid next tokens on *all* positions. Similarity, the definition of **Match Accuracy** is also modified in this way. Specifically, it becomes the proportion of inputs on which the pruned model's predictions about the validity of *all* tokens in vocabulary are all the same as the original model's predictions, on *all* positions.

We now define the tasks individually. The definitions closely follow Huang et al. (2025).

**Binary Majority.**  This problem is to find the most frequent bit in a sequence of random bits. An example is $\boxed{\texttt{BOS}\,|\,\texttt{1}\,|\,\texttt{0}\,|\,\texttt{...}\,|\,\texttt{1}\,|\,\texttt{SEP}\,|\,\texttt{1}}$ The part between BOS and SEP is the sequence of random bits. We constrain the sequences such that the number of 0s and 1s are always not equal. The bit after SEP is the label, i.e., the most frequent token.

**Binary Majority Interleave.**  The inputs in this problem are created by interleaving multiple binary majority (see above) inputs while avoiding repeating special tokens (e.g., BOS). We use 3 binary majority sequence to compose one sequence in this task. Formally speaking, given 3 binary sequences of the same length, $x_1^1, \cdots x_n^1, x_1^2, \cdots x_n^2$, and $x_1^3, \cdots x_n^3$, and their corresponding labels $y^1, y^2, y^3$, the interleaved input is BOS $x_1^1, x_1^2, x_1^3, x_2^1, x_2^2, x_2^3, \cdots, x_n^1, x_n^2, x_n^3$. SEP $y^1, y^2, y^3$ EOS. An example is $\boxed{\texttt{BOS}\,|\,\texttt{1}\,|\,\texttt{0}\,|\,\texttt{1}\,|\,\texttt{1}\,|\,\texttt{0}\,|\,\texttt{0}\,|\,\texttt{...}\,|\,\texttt{SEP}\,|\,\texttt{1}\,|\,\texttt{0}\,|\,\texttt{0}\,|\,\texttt{EOS}}$ The task is from Huang et al. (2025).

**Most Frequent.** [5]  This is similar to the binary majority problem, except the vocabulary is larger. An example is $\boxed{\texttt{BOS}\,|\,\texttt{c}\,|\,\texttt{b}\,|\,\texttt{a}\,|\,\texttt{b}\,|\,\texttt{SEP}\,|\,\texttt{b}}$, where the part between BOS and SEP is a sequence of random tokens, each of which is sampled independently from an alphabet of 26 symbols. We constrain the sequences such that there is always a unique answer.

**Sort.**  In sort problem, the model outputs a sorted version of the given sequence. An example is $\boxed{\texttt{BOS}\,|\,\texttt{14}\,|\,\texttt{23}\,|\,\texttt{6}\,|\,\texttt{9}\,|\,\texttt{SEP}\,|\,\texttt{6}\,|\,\texttt{9}\,|\,\texttt{14}\,|\,\texttt{23}\,|\,\texttt{EOS}}$ where the part between BOS and SEP is a sequence of unique numbers, and the part between SEP and EOS is the sorted version of it. The total vocabulary size of tokens except for special tokens is equal to the maximum testing length, i.e., 150. The task is from (Zhou et al., 2024; Huang et al., 2025).

**Count.**  In this problem, the model outputs the same sequence as the given sequence, which consists of unique tokens. An example is $\boxed{\texttt{BOS}\,|\,\texttt{14}\,|\,\texttt{19}\,|\,\texttt{SEP}\,|\,\texttt{14}\,|\,\texttt{15}\,|\,\texttt{16}\,|\,\texttt{17}\,|\,\texttt{18}\,|\,\texttt{19}\,|\,\texttt{EOS}}$ where the two numbers between BOS and SEP are the

---

[5]This task is called Majority in Huang et al. (2025). We feel that "Most Frequent" is more accurate.

start and end of counting, such counting generates numbers in the part between SEP and EOS. The total vocabulary size of tokens except for special tokens is equal to the maximum testing length, i.e., 150. Note that this is not trivial since the model needs to predict where to start counting and when to end counting. The task is from (Zhou et al., 2024).

**Unique Copy**    In this problem, the model outputs the same sequence as the given sequence. Tokens in the given sequence occur at most once (thus unique). An example is     `BOS | 14 | 23 | 6 | 9 | SEP | 14 | 23 | 6 | 9 | EOS`     where the part between BOS and SEP is a sequence of unique numbers, and the part between SEP and EOS is a copy of it. The total vocabulary size of tokens except for special tokens is equal to the maximum testing length, i.e., 150. The task is from (Zhou et al., 2024; Huang et al., 2025).

**Unique Bigram Copy**    In this problem, the model outputs the same sequence as the given sequence, which consists of unique bigrams. An example is     `BOS | 14 | 23 | 6 | 14 | SEP | 14 | 23 | 6 | 14 | EOS`     where the part between BOS and SEP is a sequence of unique bigrams (i.e. (14, 23), (23, 6), (56, 14)), we guarantee that each bigram only occurs once in a sequence. The part between SEP and EOS is a copy of it. The total vocabulary size of tokens except for special tokens is 16, since we allow for repetition of the same tokens in one sequence.

**Unique Reverse Copy.**    In this problem, the model copies the given sequence in reverse order, the given sequence consists of unique tokens. An example is     `BOS | 14 | 23 | 6 | 9 | SEP | 9 | 6 | 23 | 14 | EOS`     where the part between BOS and SEP is a sequence of unique numbers, and the part between SEP and EOS is a copy of it in reverse order. The total vocabulary size of tokens except for special tokens is equal to the maximum testing length, i.e., 150. The task is from (Jobanputra et al., 2025).

**Repeat Copy.**    The model outputs the same sequence as the given sequence. The given sequence can contain repeated tokens. An example is     `BOS | a | b | a | b | SEP | a | b | a | b | EOS`     where the part between BOS and SEP is a sequence of random symbols. We use an alphabet of only 2 symbols, so that we avoid n-gram being always unique for certain small n. Each symbols is sampled independently and uniformly. The task is from (Zhou et al., 2024; Huang et al., 2025).

**Parity.**    In the parity problem, the model recognizes whether the given sequence contains even number of 1s. An example is     `BOS | 1 | 0 | 0 | 1 | 0 | SEP | 0`     The bits between BOS and SEP is a random sequence of bits. The bit after SEP is the number of ones modulo 2, i.e., 0 when number of 1s is even, and 1 otherwise. The bits are randomly sampled in a way such that the number of 1s is distributed uniformly given a fixed sequence length.

**Addition.**    In    this    problem,    the    model    need    to    do    binary    addition.    An    example    is `BOS | 1 | 0 | 1 | + | 1 | 0 | SEP | 1 | 1 | 1 | EOS`     The bits between BOS and SEP are separated by "+", producing two operands. The bits between SEP and EOS are the answer of the addition. The two operands are sampled randomly. Note that we do not pad zeros in the front of operands to make them of equal length.

**Formal Languages.**    We also include 17 finite-state formal languages that are also used in (Bhattamishra et al., 2020; Sarrof et al., 2024; Huang et al., 2025).

- Tomita Grammars. Alphabet $\Sigma = \{0, 1\}$

    - Tomita 1: $1^*$
    - Tomita 2: $(10)^*$
    - Tomita 3: strings without odd-length strings of ones followed by odd-length strings of zeros (i.e., no $01^{2n+1}0^{2m+1}1$ substrings)
    - Tomita 4: strings without any 000's substrings
    - Tomita 5: strings of even length with an even number of 1's
    - Tomita 6: strings where number of 0's - number of 1's is divisible by 3
    - Tomita 7: $0^*1^*0^*1^*$

- $D_n$. Alphabet $\Sigma = \{a, b\}$. The general definition $D_n = (aD_{n-1}b)^*$. In other words, Dyck-1 language with depth bounded by n.

    – $D_2$

    – $D_3$

    – $D_4$

    – $D_{12}$

- Others. Alphabet is all the letters occurring in the expression.

    – $(aa)^*$

    – $(aaaa)^*$

    – $(abab)^*$

    – $aa^*bb^*cc^*dd^*ee^*$

    – $\{ab\}^*d\{b,c\}^*$

    – $\{0,1,2\}^*02^*$

For most of the tasks, we keep the lengths of the inputs uniformly distributed in the desired range, that is, $\leq 50$ during training (the minimum length depends on the task, e.g., minimum number of random bits in Binary Majority is 1 and in the interleaved version is 3) and $\leq 150$ during pruning. The exception includes Binary Majority Interleave, $D_n$, Tomita 2, Tomita 5, Tomita 6, $(abab)^*$ $(aa)^*$ $(aaaa)^*$, because some lengths are not possible.

### D.2. Model Training

We are interested in testing whether models can length-generalize, and see how it relates to interpretability of the model. Therefore, like Huang et al. (2025) (and similar to Zhou et al. (2024)), at train time, we add random offsets to position indices so that all position embeddings are trained. For example, for the input $\boxed{\text{BOS}\,|\,\text{c}\,|\,\text{b}\,|\,\text{a}\,|\,...}$ , the position indices are $\boxed{0\,|\,1\,|\,2\,|\,3\,|\,...}$ , after adding an offset of 7, new position indices are $\boxed{7\,|\,8\,|\,9\,|\,10\,|\,...}$ . We use the same training and testing lengths as Huang et al. (2025), i.e., train models on inputs of length $\leq 50$ and test them on $\leq 50$, $[51, 100]$ and $[101, 150]$. The offsets are sampled uniformly at random in the range of 0 to number of position embeddings subtracted by input length. Like Zhou et al. (2024), we sample independent training batches on the fly instead of using a finite-size training set. There are 3 test sets for each length range, each test set contains 2000 samples that are sampled at the beginning of each experiment, given the same random seed. We do not exclude the test samples in the test set of $\leq 50$ during training.

We train decoder-only transformers from scratch, using implementations from Hugging Face Transformers[6]. Same as (Huang et al., 2025), we train models for maximum 30k steps with a batch size of 64. We stop training early once the model's accuracy reaches 100% on the in-distribution test set (length $\leq 50$). We use AdamW, with a weight decay rate of 0.01.

We train various models with hyperparameters in the following domain: number of layers: $\{1, 2, 4\}$, number of heads $\{1, 2, 4\}$, model dimension: $\{16, 64, 256\}$, and learning rate: $\{0.001, 0.0001\}$, dropout rate: $\{0.0, 0.1\}$ (same rate for attention dropout, residual stream dropout, and embedding layer dropout).

**Representative models for each task**    We keep at least one model for each task, use this as a representative model for the task. Similar to (Huang et al., 2025), we train models on all combination of hyperparameters above, and select the one that length-generalizes best. As we mentioned in main paper, length-generalizing models tend to be decompilable. So for each task we try to get an interpretable model.

**Checkpoints at different training steps**    We also interested in seeing how model algorithms evolve throughout the training process. We save checkpoints for a few representative tasks: Copy (unique), Copy (unique bigram). The timing for saving is determined as follows: we evaluate model performance every 100 steps. In each time if we observe an increment in accuracy that is greater than 0.25 on any of the 3 test bins (i.e., input of lengths $\leq 50$, $[51, 100]$, $[101, 150]$), compared to the last saved checkpoint (compare to 0 when no previous saved checkpoint), we save the checkpoint.

---

[6] https://huggingface.co/docs/transformers/en/model_doc/gpt2#transformers.GPT2LMHeadModel

**Different model architectures**   For many tasks, we also keep additional models of different the architectures (e.g., number of layer and head), such that we have (1) both length-generalizing and non-length-generalizing models for the same task; (2) models of very different architectures but length-generalizing on the same task. This allows us to make interesting comparison to support our claims.

**Model naming scheme**   In the paper and our web application, the models we are interpreting are named as `[task name]-[number of layer]l[number of head]h[model dimension]d[`$log_{0.1}$`(learning rate)]lr[dropout rate * 10]drop`. For example, `unique_reverse-2l1h64d3lr01drop` is a model with 2 layers, 1 head per layer and model dimension of 64, trained on Reverse Copy (unique) with learning rate of $1 \times 10^{-3}$ and dropout rate of 0.1. For intermediate checkpoint models, their names also include the trained steps at the end.

## E. Complete Results on Generalization and Decompilability

| Task | Model | Task Acc in $< 50$ | Task Acc in [51-100] (OOD) | Task Acc in [101-150] (OOD) | LLNA holds? | Can split MLPs? |
|---|---|---|---|---|---|---|
| Most Frequent | 1l4h256d3lr01drop | 100.0 | 99.7 | 98.4 | ✓ | ✓ |
| | 4l4h256d3lr00drop | 100.0 | 100.0 | 99.7 | ✓ | ✓ |
| Binary Majority | 1l1h16d3lr01drop | 100.0 | 100.0 | 100.0 | ✓ | ✓ |
| Binary Majority Interleave | 2l2h16d3lr00drop | 100.0 | 99.6 | 91.7 | ✓ | ✓ |
| | 2l2h64d4lr00drop | 100.0 | 74.0 | 19.5 | × | - |
| | 4l1h64d3lr01drop | 100.0 | 98.5 | 97.9 | ✓ | ✓ |
| Sort | 1l1h256d3lr01drop | 97.8 | 100.0 | 100.0 | ✓ | ✓ |
| Count | 1l4h256d4lr01drop | 100.0 | 100.0 | 94.8 | ✓ | ✓ |
| | 1l4h256d3lr01drop | 100.0 | 24.0 | 0.0 | × | - |
| | 2l4h256d4lr01drop | 100.0 | 100.0 | 99.3 | ✓ | ✓ |
| Unique Copy | 2l1h64d3lr01drop | 100.0 | 100.0 | 99.1 | ✓ | ✓ |
| | 4l4h256d4lr00drop | 100.0 | 8.3 | 0.0 | × | - |
| | 2l1h64d3lr01drop-1100 | 26.3 | 0.0 | 0.0 | × | - |
| | 2l1h64d3lr01drop-1200 | 92.5 | 44.2 | 0.4 | × | - |
| | 2l1h64d3lr01drop-1300 | 99.4 | 93.7 | 50.8 | × | - |
| | 2l1h64d3lr01drop-1400 | 99.9 | 99.2 | 81.0 | × | - |
| Unique Reverse | 2l1h64d3lr01drop | 100.0 | 100.0 | 99.2 | ✓ | ✓ |
| | 2l4h256d4lr01drop | 100.0 | 40.7 | 0.0 | × | - |
| | 4l1h256d4lr01drop | 100.0 | 100.0 | 99.6 | ✓ | ✓ |
| Unique Bigram Copy | 2l4h256d3lr01drop | 100.0 | 100.0 | 99.7 | ✓ | × |
| | 4l2h64d3lr00drop | 100.0 | 46.7 | 0.0 | × | - |
| | 2l4h256d3lr01drop-500 | 34.3 | 0.2 | 0.0 | × | - |
| | 2l4h256d3lr01drop-700 | 61.0 | 5.8 | 0.0 | × | - |
| | 2l4h256d3lr01drop-1400 | 87.0 | 27.4 | 0.8 | × | - |
| | 2l4h256d3lr01drop-1900 | 91.4 | 53.3 | 7.8 | × | - |
| | 2l4h256d3lr01drop-2100 | 96.9 | 85.3 | 44.3 | × | - |
| | 2l4h256d3lr01drop-2200 | 99.2 | 95.2 | 74.2 | × | - |
| | 2l4h256d3lr01drop-3300 | 100.0 | 99.9 | 99.8 | ✓ | ✓ |
| Repeat Copy | 4l4h256d3lr00drop | 100.0 | 26.6 | 0.0 | × | - |
| Parity | 4l4h256d4lr00drop | 93.9 | 67.5 | 63.2 | × | - |
| Addition | 4l2h64d3lr00drop | 50.0 | 1.1 | 0.0 | × | - |

*Table 1.* Length-generalization Performance and Decompilability (whether LLNA holds) of all models trained on algorithmic tasks. The model names are based on their architecture (see the naming scheme in Appendix D.2). We highlight the task accuracy in the longest length range with green, if $\geq 90$, and red otherwise. We say LLNA holds if there exists a model with linearized layer norm achieving match accuracy of $\geq 90$ (stage 1, Appendix F.1). Our decompilation pipeline works if LLNA holds, so the LLNA column shows decompilability. Across all the models here, we see that the green and red color in two highlighted columns exactly match. In other words, in all models in this table, our method work on models that length-generalize.

As we mentioned in main paper, for each task, we keep the model that length-generalize the best among all models trained with different hyperparameters. So if a task has only one model, that's the model with the best length-generalizing result. We show the complete results in Table 1 and Table 2.

| Task | Model | Task Acc in $< 50$ | Task Acc in $[51-100]$ (**OOD**) | Task Acc in $[101-150]$ (**OOD**) | LLNA holds? | Can split MLPs? |
|---|---|---|---|---|---|---|
| Tomita1 | 1l1h16d3lr00drop | 100.0 | 100.0 | 100.0 | ✓ | ✓ |
| Tomita2 | 1l1h16d3lr00drop | 100.0 | 100.0 | 100.0 | ✓ | ✓ |
| Tomita3 | 4l4h64d3lr00drop | 100.0 | 97.6 | 88.3 | ✗ | - |
| Tomita4 | 2l2h16d3lr01drop | 100.0 | 100.0 | 100.0 | ✗ | - |
| Tomita5 | 2l1h64d3lr00drop | 98.9 | 70.5 | 21.6 | ✗ | - |
| Tomita6 | 2l2h256d4lr00drop | 68.1 | 9.5 | 2.3 | ✗ | - |
| Tomita7 | 2l1h16d3lr01drop | 100.0 | 100.0 | 100.0 | ✓ | ✓ |
| $D_2$ | 1l1h16d3lr00drop | 100.0 | 100.0 | 100.0 | ✓ | ✗* |
| $D_3$ | 1l1h16d4lr00drop | 100.0 | 100.0 | 100.0 | ✗ | - |
| $D_4$ | 1l2h256d4lr00drop | 100.0 | 100.0 | 98.3 | ✓ | ✓ |
| $D_{12}$ | 4l2h256d3lr00drop | 100.0 | 100.0 | 97.4 | ✗* | ✓ |
| $(aa)^*$ | 1l2h16d3lr01drop | 100.0 | 100.0 | 100.0 | ✓ | ✗ |
| $(aaaa)^*$ | 2l1h64d4lr01drop | 100.0 | 100.0 | 100.0 | ✗ | ✓ |
| $(abab)^*$ | 1l1h64d3lr00drop | 100.0 | 100.0 | 100.0 | ✗ | - |
| $aa^*bb^*cc^*dd^*ee^*$ | 1l1h16d3lr00drop | 100.0 | 100.0 | 100.0 | ✓ | ✓ |
| $\{a,b\}^*d\{b,c\}^*$ | 1l1h16d3lr01drop | 100.0 | 100.0 | 100.0 | ✓ | ✓ |
| $\{0,1,2\}^*02^*$ | 1l1h64d4lr00drop | 100.0 | 100.0 | 100.0 | ✗ | - |

*Table 2.* Length-generalization Performance and Decompilability (whether LLNA holds) of all models trained on formal language tasks. Same as Table 1, the model names are based on their architecture (see the naming scheme in Appendix D.2). We highlight the task accuracy in the longest length range with green, if $\geq 90$, and red otherwise. We say LLNA holds if there exists a model with linearized layer norm achieving match accuracy of $\geq 90$ (stage 1, Appendix F.1). Our decompilation pipeline works if LLNA holds, so the LLNA column shows decompilability. There are 2 special cases: the ×* in LLNA column of $D_{12}$ means that although the highest match accuracy is (only slightly) below 90 in stage 1, it becomes higher than 90 in later pruning stage, thus the resulting program still match the original model and we treat this result as a success of decompilation; the ×* in last column for $D_2$ means that although match accuracy is below 90 for stage 2, it becomes higher than 90 in stage 3, thus we can still split MLPs. We can see that there are 5 cases where non-linear layer norm plays an important role in length-generalizing models: Tomita4, $D_3$, $(aaaa)^*$, $(abab)^*$, and $\{0,1,2\}^*02^*$. Nonetheless, for models trained on formal languages, the overall trend still exists, the majority of length-generalizing models are decompilable.

**Does architecture choice affect decompilability?** We show pareto frontiers when layer norm is linearized in Figure 10. For each task, we show 2-3 models different architectures. We can see that very different architectures can all be decompilable (e.g., two green lines in Figure 10(a) correspond to 4 layer vs. 2 layer models), while similar architectures can exhibit different decompilability (e.g., two 2-layer models in Figure 10(a)).

**How does decompilability evolve throughout training?** We save checkpoints during training for two models: the length-generalizing one for Unique Copy (2l1h64d3lr01drop), and for Unique Bigram Copy (2l4h256d3lr01drop). We would like to see how the algorithm evolves during training. But we find that most checkpoints before the final one cannot be decompiled, as shown in Figure 11. We can only conclude that for these two models, layer norm plays an non-trivial role until the model finally reaches perfect performance on training length ($\leq 50$) (recall that we evaluate every 3k step and stop training once accuracy is 100% on training length).

**Can we get sparse graphs if original layer norm is allowed?** In Figure 4 we see decompilable models can usually be pruned to a sparse form, resulting simple program. One might wonder if this is true in general for all models trained on these synthetic tasks. Thus in Figure 12 we allow models to use original layer norm and do pruning over component-level circuit. We can see that for those models where LLNA does not hold, their inner computational graph is usually relatively dense.

**What if we do decompilation on training length?** As mentioned before, we run decompilation using all input lengths, i.e., $\leq 150$, in order to match models' behavior on both in- and out-of-distribution data. One might wonder what would happen if we remove the all the settings about length-generalization, so models are both trained and decompiled on the same length (e.g, $\leq 50$)? The reason that LLNA does not hold and circuits are not sparse might be because the pruned models are matching the original models' messy and unpredictable behavior on OOD data, they might be decompilable if we only care about behavior on training data. To investigate this, we select all models whose performance is nearly perfect on training length, but bad on OOD testing length. There are 6 this kind of models. The lowest task accuracy on training length is 99.4%, and on test length of $[101-150]$, 5 models' accuracy is 0.0% accuracy and one model's is 50.8%.

From the result we can see that although for 3 of them, focusing on lengths where models perform perfectly would enable

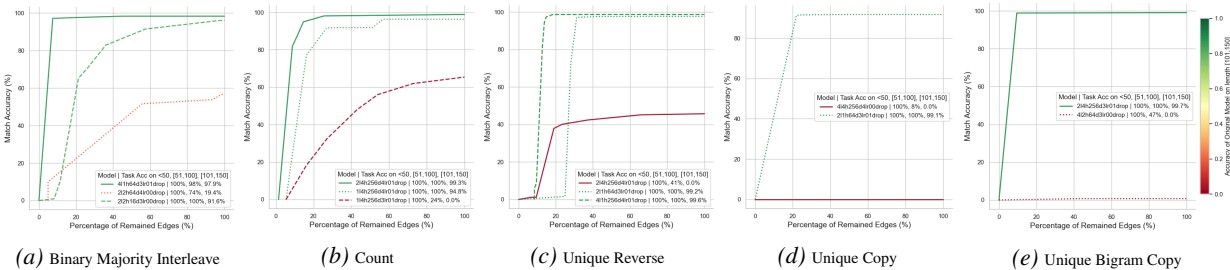

| (a) Binary Majority Interleave | (b) Count | (c) Unique Reverse | (d) Unique Copy | (e) Unique Bigram Copy |

*Figure 10.* Each subplot shows pareto frontiers for multiple models trained on the same task. They are trained with different hyperparameters and architectures, resulting in different length-generalization and decompilability. Results are from stage 1 of causal pruning, where layer norm is linearized. In the tasks shown here, we see in LLNA does not hold on models that do not generalize, so the decompilability correlates strongly with length-generalization, and much stronger than with model architectures (compare models of the same layers and different layers in the figure). Moreover, when LLNA holds, there are a large proportion of edges can be pruned.

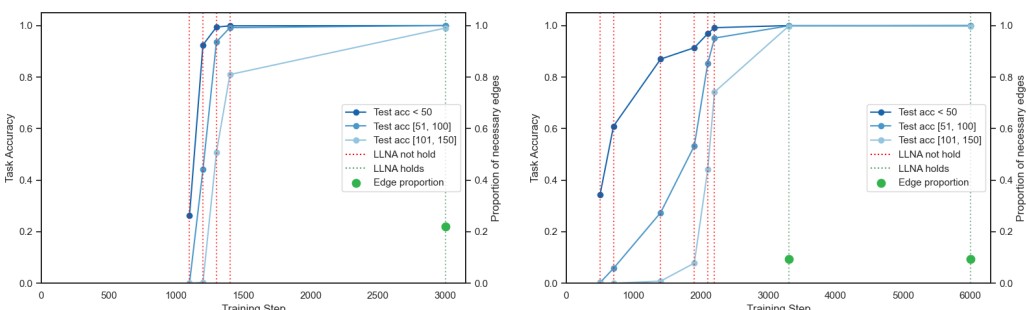

*Figure 11.* The checkpoints saved when training model for Unique Copy task (Left) and Unique Bigram Copy task (right). We show both task accuracy on 3 input ranges and decompilability. LLNA holds if the model can achieve $> 90\%$ match accuracy after linearizing Layer Norm. Green dot shows the minimum proportion of edges needed to achieve $> 90\%$ match accuracy.

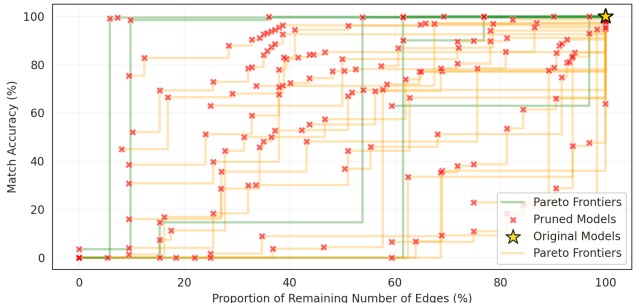

*Figure 12.* We show the pareto frontiers in stage 0 of causal pruning (i.e., component-level circuits with original layer norm enabled, see F.4 ) for all models where LLNA does not hold. The green lines correspond to the 6 models that length-generalize but requires original layer norm (see Table 2). We can see that most models cannot be pruned to be very sparse (consider how many edges they need in order to make match accuracy $\geq 90$.)

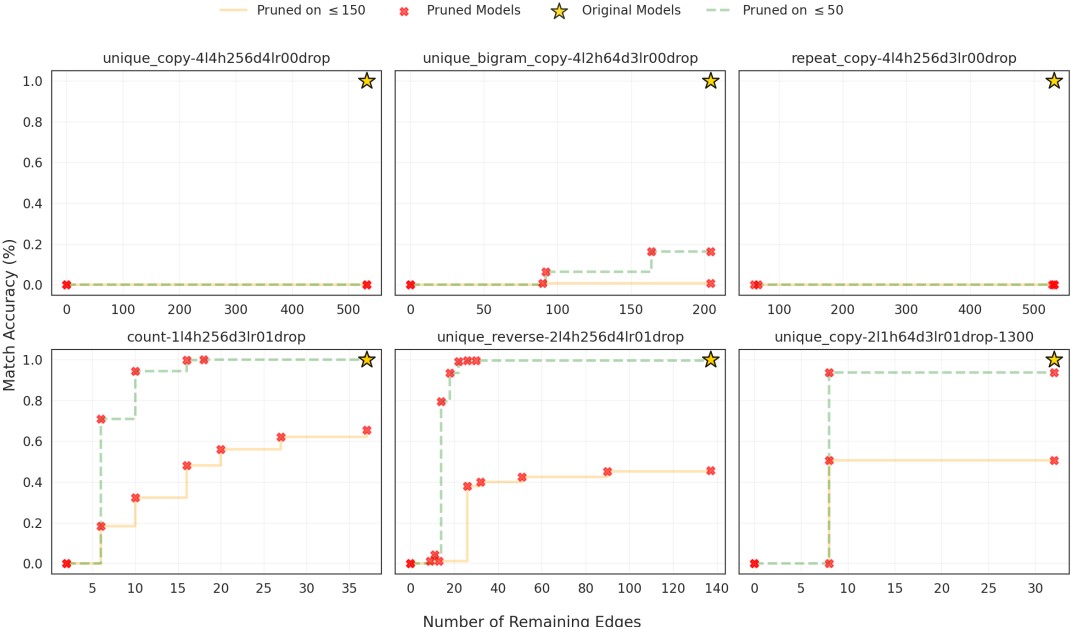

*Figure 13.* Pareto frontiers in stage 1 of causal pruning (i.e., component-level circuits with linearized layer norm.) We find all models that achieves nearly perfect task accuracy in training length (specifically, all are $\geq 99.4$), and very low accuracy in test length of $[101 - 150]$ (only one is 50.8, others are 0.0). For each model we show the frontier when pruning is done and match accuracy is measured on $\leq 150$ and $\leq 50$. The purpose is to see whether we can decompile the models by restricting output matching only on inputs where model's performance is perfect. We can see positive results on bottom row, but still negative results on the top row.

decompilation, but for the other 3, it does not help at all. Linearizing layer norm would still result in nearly 0 match accuracy. For these models, their circuits that produce perfect in-distribution prediction are still complex.

Since we can decompile some of them by restricting length to $\leq 50$, one might wonder if we can get interpretable programs and thus know why they fail on longer length. The answer is no, the layer norm starts to play an important role at longer length. We can probably get interpretable programs but the output of the programs do not match the models' output on longer length, thus cannot be used to explain model behavior in that scenario.

**Does training with different random seeds affect decompilability?** For each length-generalizable algorithmic task, we train 10 models with the same architecture and hyperparameters but different random seeds. As shown in Table 3, the choice of random seed affects the model's length-generalization performance and, consequently, its decompilability. However, we observe that, for decompilable models, our method tends to recover similar programs. Interestingly, in 4 out of 8 tasks, our method discovers fewer unique programs than unique pruned computation graphs. This indicates that the method can sometimes recover the same program for two models even when their circuits after the pruning step differ. This is possible because the extracted programs are agnostic to the specific locations in the model where a mechanism is implemented (e.g., particular layers or heads), and because the additional primitive simplification steps in our pipeline can further increase similarity between programs.

**How stable are the programs resulting from different pruning runs?** For each model, we run the pruning algorithm multiple times with different sparsity penalties to obtain Pareto frontiers of pruning runs (e.g., Figure 4). Since there is a tradeoff between faithfulness to the original model and program length, different pruning runs are expected to yield different programs. A natural question, however, is whether these programs remain similar across runs. Empirically, the extracted programs are quite stable. For all 21 models in our evaluation set (excluding Tomita-1, which has an empty computation graph), we consider all pruning runs that achieve more than 90% match accuracy. Pruning is structurally consistent: for 17/21 models, all selected pruning runs produce a computation graph that is either a subgraph or a supergraph of the computation graph selected for the paper. In many cases, pruning recovers exactly the same computation graph regardless of hyperparameters: for 10/21 models, all runs resulted in the same programs as those shown in the paper.

| Task | Architecture | Successfully Decompiled | Length-Generalized | Unique Graphs | Unique Programs | Avg. Corr. |
|---|---|---|---|---|---|---|
| Most Frequent | 1l4h256d3lr01drop | 10 | 10 | 7 | 1 | |
| Binary Majority | 1l1h16d3lr01drop | 10 | 10 | 6 | 1 | 1.00 |
| Binary Majority Interleave | 2l2h16d3lr00drop | 2 | 0 | 2 | 2 | |
| Count | 1l4h256d4lr01drop | 1 | 1 | 1 | 1 | |
| Sort | 1l1h256d3lr01drop | 10 | 10 | 1 | 2 | 0.96 |
| Unique Bigram Copy | 2l4h256d3lr01drop | 7 | 10 | 7 | 4 | |
| Unique Copy | 2l1h64d3lr01drop | 7 | 7 | 3 | 3 | 0.94 |
| Unique Reverse | 2l1h64d3lr01drop | 9 | 8 | 5 | 4 | 0.67 |

*Table 3.* Similarity of programs extracted from models trained with different random seeds. For each length-generalizable algorithmic task, we train 10 models with the same architecture and hyperparameters but different random seeds. {A}l{B}h{C}d{D}lr{E}drop denotes a model with A layers, B heads, hidden dimension C, learning rate $10^{-D}$, and dropout rate either 0.1 or 0. Out of the 10 trained models, we report the number that were successfully decompiled, i.e., those that satisfy LLNA and were successfully pruned while retaining program-to-model match accuracy above 0.9. We also report the number of models that length-generalized, defined as achieving accuracy above 0.9 on inputs of lengths [101, 150].For each decompiled model, we extract its pruned computation graph and the program, and report the number of unique pruned graphs and programs. For each pair of models with the same computation graph, we compute the Pearson correlation between the corresponding $A$ and $b$ tensors in their `select` and `project` operations, as well as the biases in the final softmax, averaging across all such operations and model pairs. Correlations are not reported for tasks with no pairs of decompiled models sharing the same pruned computation graph. The Most Frequent task is also excluded, as it does not contain tensors in `select` or `project`.

**Ablation of pipeline stages and statistics on the use of human-designed primitives**   We look at the gains of each stage of the pipeline to match accuracy and task accuracy. After linearizing layer norm and the first stage of pruning, the average match accuracy across all decompilable models in our evaluation set is 93%; after splitting MLPs and path-based pruning, it remains 93%; after pruning QK products, it increases to 95%; and after replacing operations with primitives, it remains at 95% (95% task accuracy). None of the pipeline stages significantly reduce faithfulness.

We also report statistics on the use of human-designed primitives. Among the programs provided in Appendices J and K, 34% of all `select` operations, 30% of `project` operations, and 19% of MLP operations are replaced with primitives. Additionally, 88% of all MLP operations have a single path as input, indicating that they were successfully split.

**Remark on FlipFlop**   We remark that $\{0, 1, 2\}^{*}02^{*}$, equivalent to FlipFlop and challenging for transformers to generalize perfectly on (Liu et al., 2023; Jobanputra et al., 2025), shows strong behavior in Table 2, potentially because we did not specifically design a hard test set testing out-of-distribution generalization (beyond length generalization) as done by Liu et al. (2023). Nonetheless, the model cannot be decompiled, aligning with theoretical predictions about the inability of C-RASP to represent the language (Huang et al., 2025).

# F. Details of Causal Pruning & Linearizing Layer Norm

Here, we describe the step 2.1 in Sec. 3 in detail. Conceptually, it corresponds to pruning variables and lines from the D-RASP program – equivalently, to computation graph edges between transformer components. To make it scalable, we draw on ideas from the circuit discovery literature. For the convenience of readers familiar with that literature, and to enable comparison with it, we describe the pruning not on the level of D-RASP, but rather in the language of computation graphs representing transformer calculations.

As we mentioned in main paper, because the size of D-RASP program grows exponentially with the original transformer's size, we do not unfold the transformer into full program and then do pruning. Instead, we perform multi-stage pruning, with different definition of computational graph in each stage. Simply speaking, an edge in early stage correspond to many edges in later stage, so in early stage we prune the model in a coarse-grained but efficient way, resulting in much smaller graphs for later stage. Therefore, multi-stage pruning avoids memory explosion and improves efficiency.

### F.1. Stage 1: Pruning Components

**Component-level Circuit**   In this stage, we define the same computational graph as circuit discovering research (Conmy et al., 2023; Li & Janson, 2024; Bhaskar et al., 2024). As we described in Sec. B.1, due to residual connection, the output of

each transformer layer is accumulated across layers. When a component (e.g. an attention head) takes the residual stream $\boldsymbol{y}_i^{(l)}$ as input, all previous components whose output adds to the residual stream are connected to this component, as this component depends on their outputs. Because this dependency is directed, we refer the component that receives other components' output as *receiving* vertex and refer the other side of the connection as *sending* vertex. Importantly, we treat the query, key and value of an attention head as three separate receiving vertices, while we treat the attention head as one sending node. MLP layer are both receiving vertices and sending vertices; token and position embedding are only sending nodes, final unembedding layer is only a receiving vertex.

If the connection between attention head $(l_i, h_j)$ and $(l_k, h_l)$'s value is pruned, it means all $\langle i, p \rangle \in \mathcal{V}$ where $p$ includes $\langle...(l_i, h_j), (l_k, h_l)...\rangle$ are pruned together. In this way, we first perform coarse-grained but efficient pruning.

**Pruning Algorithm**  We apply Uniform Gradient Sampling (UGS) and Optimal Ablation from Li & Janson (2024). We describe the overall idea of the algorithm and emphasize the parts in our implementation that differ from the original paper. For details of the method, please refer to Li & Janson (2024).

The method learns masks for edges. The masks are trained for an objective balancing two pressures: on one hand, (i) the masks are encouraged to be zero to completely mask out connections by replacing them with learnable vectors, known as Optimal Ablation; on the other hand, (ii) the KL divergence between the output distribution of pruned model and of original model is minimized during pruning, thus encouraging important connections to be kept. The method optimizes all masks simultaneously using gradient signal, and is thus very scalable.

Specifically, if an edge is pruned, the sending vertex (component $A$)'s output $A(x) \in \mathbb{R}^{N \times d}$ ($x$ is the input) is replaced by a learned vector $a^* \in \mathbb{R}^d$ when computing the input of the receiving vertex. In other words, the replacement is specific to the receiving vertex; other receiving nodes can still receive the original $A(x)$. Because we replace with a constant, the sending vertex's output does not depend on the input, thus carrying no information. As in Li & Janson (2024), the constant $a^*$ is specific to each vertex $A$, instead of each edge — this means that, if multiple out-edges from $A$ are pruned, the same constant is used across different receiving vertices. Importantly, unlike Li & Janson (2024), we do not learn different $a^*$ for different sequence positions, we learn a single $a^*$ that is applied to the whole sequence uniformly, so that each row of $A(x) \in \mathbb{R}^{N \times d}$ is replaced by the same $a^*$. The loss is defined as the mean Kullback-Leibler (KL) divergence between the output distribution of the original model and that of the pruned model. Therefore, these learned optimal ablations are optimized to help the pruned model faithfully preserve the behavior of the original model.

Masks are learned simultaneously with the ablation vectors. For each edge, we learn a parameter $\tilde{\theta}$, such that $\theta = \text{sigmoid}(\tilde{\theta})$ indicates the probability of the edge remaining in the computation graph (as opposed to being replaced with the learned ablation vector). The gradient of $\theta$ comes from two terms in the loss function. One is (A) the number of edges in the circuit, represented by its continuous relaxation $\sum \theta$, where the sum runs over all edges[7]. This term is differentiable. The other term is (B) the KL divergence compared to the original model, as used for learning optimal ablations. The gradient of this term with respect to $\theta$ is estimated as follows: the ablation coefficient $\alpha$ is sampled from uniform distribution $\text{Unif}(0, 1)$; this coefficient is used to mix the original $A(x)$ and the learned replacement $a^*$ as a convex combination, which is fed into the receiving vertex. The gradient with respect to $\alpha$ is thus differentiable. The average gradient with respect to $\alpha$ is used as the estimated gradient for $\theta$. Importantly, we do not always do sampling for $\alpha$. When not sampling from uniform distribution, $\alpha$ is either 0 or 1. Whether or not to do sampling is controlled by a sampling frequency, which is low when $\theta$ is close to 0 or 1 (high confidence). Optimal ablation is learned at the same time, but only on examples where $\alpha = 0$ (not sampled).

In summary, the training objective we used is as follows:

$$\min_{\boldsymbol{a}^*, \tilde{\boldsymbol{\theta}}} \left[ \mathbb{E}_{X \sim \mathcal{D}, \boldsymbol{\alpha} \sim \mathcal{P}_{\tilde{\boldsymbol{\theta}}}} D_{KL}(\mathcal{M}(X) || \mathcal{M}^{\boldsymbol{a}^*, \boldsymbol{\alpha}}(X)) + \lambda \mathcal{R}(\tilde{\boldsymbol{\theta}}) \right] \tag{19}$$

where $\boldsymbol{a}^*$ represents the collection of $a^*$ for all edges, $\tilde{\boldsymbol{\theta}}$ denotes the collection of $\tilde{\theta}$ for all edges, and similarly $\boldsymbol{\alpha}$ denotes all $\alpha$. $\mathcal{P}_{\tilde{\boldsymbol{\theta}}}$ denotes the distribution of ablation coefficients specified by $\tilde{\boldsymbol{\theta}}$. $X$ and $\mathcal{D}$ denote input data and its distribution. $\mathcal{M}$ and $\mathcal{M}^{\boldsymbol{a}^*, \boldsymbol{\alpha}}$ represent the original model's and the pruned model's output distribution respectively. $\mathcal{R}(\tilde{\boldsymbol{\theta}})$ is the sparsity regularizer representing the expected number of edges. $\lambda$ is a hyperparameter used to balance between the two terms in the training objective.

---

[7]Unlike (Li & Janson, 2024), we do not include a pressure towards vertex sparsity because, in some preliminary experiments, we do not find it important.

**Notes on Layer Normalization**    Our translation assumes LLNA; we thus replace Layer Normalization (LN) with a linear operation. Since the gradient signal is available during pruning, we also learn scalar constants to replace the variance term in LN in the same way as we learn optimal ablation. For each receiving vertex, a scalar constant is learned.

**Hyperparameters**    We use batch size of 120 with only 10 unique inputs (each copied 12 times), which is same as (Li & Janson, 2024). We use learning rate of 0.002, 0.1 and 0.1 for optimal ablation $a^*$, masking parameter $\tilde{\theta}$, and scalar constants in linearized LN respectively. We clip the gradient for $\tilde{\theta}$ to maximum norm of 5. We stop training once no $\tilde{\theta}$ is between -1 and 1 (no ambivalent masks) for 1000 steps, or the maximum step of 5000 is reached.

### F.2. Stage 2: Pruning Paths

**Path-level Circuit**    We first convert the resulting graph from step 1 to a new graph where the sending vertices are paths (multiple edges from previous stage connected). Suppose the **Value** of attention head $(l, h)$ receives inputs from remaining vertices $p_1, ..., p_n$, and the same head $(l, h)$ is sending to downstream vertices $c_1, ..., c_m$. For each downstream vertices, we replace its parent, $(l, h)$, with multiple new parents $p_1 \oplus (l, h), .., p_n \oplus (l, h)$. We do this from earlier layer to later layers. The process implements the definition B.1. The new graph is then used as the initial base graph for further pruning in this step.

For example, suppose the following graph is remained after stage 1. There is a list for each vertex, showing the upstream vertices it connects to (it receives inputs from them).

```
layer 0:
  head 0:
    q: [pos]
    k: [pos]
    v: [token]
  mlp: [token, head₀₀]

layer 1:
  head 0:
    q: [mlp₀]
    k: [mlp₀]
    v: [head₀₀, mlp₀]
  mlp: []

unembedding:
  [head₁₀]
```

At the beginning of stage 2, it is converted to the following new graph:

```
layer 0:
  head 0:
    q: [pos]
    k: [pos]
    v: [token]
  mlp: [token, head₀₀-token]

layer 1:
  head 0:
    q: [mlp₀]
    k: [mlp₀]
    v: [head₀₀-token, mlp₀]
  mlp: []

unembedding:
  [head₁₀-head₀₀-token, head₁₀-mlp₀]
```

**Pruning Details**   In this step, new optimal ablations are learned for each sending vertex. We need to capture the output of the new vertices, which are "paths". In one forward pass of the model, we run forward pass of each attention head multiple times, iterating over each of the inputs received by its **Value**. Because of LLNA, running an attention head with its *Value* taking as input the sum of multiple terms is equivalent to the sum of individual outputs when running it with its *Value* taking each of one these terms as input (if bias terms are not considered).

$$\textbf{AttnHead}(\text{key} = \sum_i p_i, \text{query} = \sum_j p'_j, \text{value} = \sum_k p''_k) =$$
$$= \sum_k \textbf{AttnHead}(\text{key} = \sum_i p_i, \text{query} = \sum_j p'_j, \text{value} = p''_k)$$

We simply capture each of these individual output of the attention head, which is the output of each path.

**Learned Bias Term for Receiving Vertices**   Previously, we mentioned that an optimal ablation vector is learned for each sending vertex. We also learn constants (bias term) for each receiving vertex. We note that this should not be viewed as an ablation because these constants are not used to replace real activations. For each receiving vertex, its bias term is always added to its input, so it represents the sum of all previous constant terms. There are many constant terms excluded from the graph, e.g., the bias term of the linear layer computing values in attention heads, it is set to zero when running attention heads over single input source because we do not want to add the bias term repeatedly. Therefore, each receiving vertex learns a bias term to account for all bias terms in previous layer normalization ($\beta$), in linear layers computing value, in linear layers computing the attention output, and most importantly, optimal ablation vectors of vertices pruned in step 1.

**MLP Splitting**   As mentioned in Sec. 3, we try to split each MLP layer into a sum of new MLPs, so that each new MLP has only a single input. Equivalently, in terms of D-RASP programs, we try to replace multi-input per-position operations with multiple single-input operations.

If MLP splitting is enabled, when converting the resulting graph from step 1, we do the same for MLP layers as we do for attention head values. More specifically, suppose an MLP layer $mlp_k$ receives values from vertices $p_1, ..., p_n$, and it sends values to vertices $c_1, ..., c_m$. For each $c_i$, we replace its connection to $mlp_k$, with multiple new connections $p_1 \oplus mlp_k, .., p_n \oplus mlp_k$. We do this from earlier layer to later layers, together with attention heads.

When replacing with new MLP layers, we keep the output dimension of the MLPs the same as the original model dimension.

For example, suppose the following graph is remained after stage 1 (Same as the previous one).

```
layer 0:
  head 0:
    q: [pos]
    k: [pos]
    v: [token]
  mlp: [token, head_00]

layer 1:
  head 0:
    q: [mlp_0]
    k: [mlp_0]
    v: [head_00, mlp_0]
  mlp: []

unembedding:
  [head_10]
```

At the beginning of stage 2, if MLP splitting is enabled, it is converted to the following new graph:

```
layer 0:
  head 0:
    q: [pos]
```

```
    k: [pos]
    v: [token]
  mlp: [token, head₀₀-token]
```

```
layer 1:
  head 0:
    q: [mlp₀-token, mlp₀-head₀₀-token]
    k: [mlp₀-token, mlp₀-head₀₀-token]
    v: [head₀₀-token, mlp₀-head₀₀-token]
  mlp: []
```

```
unembedding:
  [head₁₀-head₀₀-token, head₁₀-mlp₀-head₀₀-token]
```

Empirically we found MLP splitting is usually possible with little effect on match accuracy.

**Hyperparameters**  We use the same hyperparameters as the previous stage, except for the following: we stop training once no $\tilde{\theta}$ is between -1 and 1 (no ambivalent masks) for 500 steps, or the maximum step of 5000 is reached.

### F.3. Stage 3: Pruning QK products

**Product-level Attention Computation**  This stage focuses on the Query and Key vertices of attention heads – equivalently, on the `select` operations. We first convert the resulting graph from stage 2 into a new graph. For each head, we take the Cartesian product of vertices sending to its Query and its Key. The results are pairs of paths, which sending to a new vertex of the head, denoted as *QK vertex*. In previous stages, the attention logits had been represented as the product of sums, for example:

$$\text{AttnLogit} = (\boldsymbol{E}\texttt{token} + \boldsymbol{P}\texttt{pos})^{\top} Q^{\top} K (\boldsymbol{E}\texttt{token} + \boldsymbol{P}\texttt{pos}), \tag{20}$$

while in this stage attention logits are rewritten as sum of products:

$$\text{AttnLogit} = \texttt{token}^T \boldsymbol{E}^T Q^{\top} K \boldsymbol{E}\texttt{token} + \texttt{token}^T \boldsymbol{E}^T Q^{\top} K \boldsymbol{P}\texttt{pos} + \texttt{pos}^T \boldsymbol{P}^T Q^{\top} K \boldsymbol{E}\texttt{token} + \texttt{pos}^T \boldsymbol{P}^T Q^{\top} K \boldsymbol{P}\texttt{pos} \tag{21}$$

Each pair of paths represents a product. The Query vertex and its edges are thus deleted, while the Key vertex remains. The incoming edges of Key vertex represent the unary (key-only) `select` operations; this accounts for the fact that a constant term in Query might play a role for attention computation, i.e., some keys are always attended more regardless of the query. This conversion process follows the Equation 2.

Use our running example from last stage, and suppose MLP splitting is enabled. Suppose after the last stage's pruning, the following graph is remained:

```
layer 0:
  head 0:
    q: [pos]
    k: [pos]
    v: [token]
  mlp: [token, head₀₀-token]
```

```
layer 1:
  head 0:
    q: [mlp₀-token, mlp₀-head₀₀-token]
    k: [mlp₀-token, mlp₀-head₀₀-token]
    v: [mlp₀-head₀₀-token]
  mlp: []
```

```
unembedding:
  [head₁₀-mlp₀-head₀₀-token]
```

the beginning of this stage, the graph is converted to the following:

```
layer 0:
  head 0:
    qk: [(pos, pos)]
    k: [pos]
    v: [token]
  mlp: [token, head₀₀-token]

layer 1:
  head 0:
    qk: [(mlp₀-token, mlp₀-token), (mlp₀-token, mlp₀-head₀₀-token),
        (mlp₀-head₀₀-token, mlp₀-token), (mlp₀-head₀₀-token, mlp₀-head₀₀-token)]
    k: [mlp₀-token, mlp₀-head₀₀-token]
    v: [mlp₀-head₀₀-token]
  mlp: []

unembedding:
  [head₁₀-mlp₀-head₀₀-token]
```

Pruning in this stage is only applied to terms in QK vertices and K vertices, others remain unchanged. In other words, we are pruning away unimportant `select` operations.

**Pruning Details**   In forward passes, we compute individual products of each pair, which are then multiplied with their corresponding mask (recall the $\theta$ in previous section F.1) before summed together. The same mask is broadcasted to all token position pairs, i.e., the mask varies for different edges, but stay the same for different token positions in the sequence, just like previous stages. However, unlike previous stages, the optimal ablation is zero here. Because ablating a product means an element of the pair is replaced by a constant. When this element is key, any constant is equivalent to zero because of the softmax later. When this element is query, this product becomes essentially a key-only term, which is the reason for why we introduce key-only terms. For each key-only term, a bias term is learned to act as the query. The product of the learned bias and the key is also multiplied with mask and added to the attention logits.

$$\text{AttnLogit}(i, s) = \underbrace{\sum_{r=1}^{t} \theta_r Q_r(i)^\top K_r(s)}_{\text{binary (key+query) terms}} + \underbrace{\sum_{u=1}^{w} \theta_u \tilde{b}_u^\top K_{t+u}(s)}_{\text{unary (key-only) terms}} + \text{mask}(i, s),$$

where we use $K$ to represent the real activations (output of paths) to distinguish from the $k$ used in 2, and likewise for $Q$, $\theta$ is the mask, and $\tilde{b}$ is the learned bias term for each key-only term. Other symbols retain the same meaning as in equation 2. The logits are then fed into softmax function to compute the attention weights. Note that like learning optimal ablation, the bias terms are only trained on samples where the mask $\theta$ is 1.

### F.4. Automatic Pruning Coefficients Searching

During pruning, a coefficient $\lambda$ is used to balance between faithfulness and sparsity, as described in Eq. 19. In this section, we describe our solution for automatically adjusting the coefficients in different stages, in order to cover as many different sparsity levels as we can and obtain Pareto frontiers.

Instead of using the same $\lambda$ through out all stages, we allow different coefficients for each stages, we denote them as $\lambda_1, \lambda_2, \lambda_3$. We begin by establishing a baseline accuracy, which is the accuracy of a model without any attention and MLP layers (a bigram model). For the stage 1, we partition the accuracy range between baseline and perfect performance into 10 evenly spaced intervals. The algorithm adaptively searches for $\lambda_1$ values based on previous runs, which are (accuracy, coefficient) points. When both higher and lower accuracy points exist for a target interval, we select the geometric mean of their coefficients; when data exists on only one side, we explore new range by scaling the known coefficient by a factor of 5. This process continues until each accuracy interval contains at least one data point or a maximum number of experiments is reached (almost always latter is the case). For each experiments, we stop after stage 1 is finished.

Once this initial curve is established, we identify the Pareto-efficient runs (i.e., no other run has both fewer edges and higher match accuracy). From this frontier, we select a handful of representative runs that are evenly spaced along the curve. If even the highest accuracy achieved in stage 1 runs is below 0.7 (even with negative $\lambda_1$), we stop and do not continue to later stages.

The stage 2 searching is then built directly upon these selected runs. We resume the pruning of the selected stage 1 runs. For each stage 1 run, we run stage 2 with different $\lambda_2$ to explore the accuracy range near its original accuracy obtained in stage 1. Then similarly we select efficient and representative runs whose resulting (accuracy, number of edges) points are on Pareto frontier. We then resume these runs with different $\lambda_3$ to do stage 3 searching, in a similar way as before. Hence, the most promising (Pareto-optimal) models from stage 1 are refined in the next, systematically tracing out the full trade-off between sparsity and faithfulness across coefficient configurations.

In addition, to see the Pareto frontiers when original layer normalization is enabled, we also run similar coefficient searching before stage 1, which is referred as stage 0. In stage 0 and stage 1, the computational graphs are defined in the same way, the only difference is whether to linearize layer normalization. Importantly, results from stage 0 are not used as the base checkpoints to continue pruning. The coefficient selection in stage 1 is also not informed by stage 0. In other words, stage 0 is separate from other stages, and it is also not a necessary part of the pipeline; its purpose is just to evaluate the Pareto frontier in the presence of full layer normalization.

## G. Details of Explaining Elementwise Operations

This section describes the details of Step 2.2 in Sec. 3. After the 3-stage pruning, we obtain a sparse (if possible) computational graph for the model we are interested in, corresponding to a pruned D-RASP program. We then try to automatically replace the components in the graph (equivalently, the elementwise operations and the $A, b$ matrices in the D-RASP program) with pre-defined primitives from a library, allowing us to understand their functions and generate informative programs.

### G.1. Tracing activation variables through paths

This section supplements Sec. 3 in the main paper. We list real activations, activation variables, etc starting from bottom layer. Initial residual stream is:
$$\boldsymbol{E}\texttt{token} + \boldsymbol{P}\texttt{pos} \tag{22}$$
$\texttt{token}$ as a graph vertex (see above examples of graph) sends real activation value $\boldsymbol{E}\texttt{token} \in \mathbb{R}^{d \times N}$ to downstream vertices, where $\texttt{token} \in \mathbb{R}^{|\Sigma| \times N}$ is the activation variable. Similarly for pos.

In head 0 at layer 0, head output is
$$\boldsymbol{V}_{0,0}(\boldsymbol{E}\texttt{token} + \boldsymbol{P}\texttt{pos})a_{00}^T \tag{23}$$
$$= \boldsymbol{V}_{0,0}\boldsymbol{E}\texttt{token}a_{00}^T + \boldsymbol{V}_{0,0}\boldsymbol{P}\texttt{pos}a_{00}^T \tag{24}$$

where $a_{00} \in \mathbb{R}^{N \times N}$ is the attention weights of this head (recall $a_{i,s}$ defined in Sec. 2, $a$ is softmax over the sum of selectors $\alpha$). In general, it is query sequence length by key sequence length. Recall that $\boldsymbol{V}$ includes linearized layer norm, Value matrix, and Output matrix of an attention head. Again, $\text{head}_{00}$-$\texttt{token}$ as a graph vertex sends real activation $\boldsymbol{V}_{0,0}\boldsymbol{E}\texttt{token}a_{00}^T$, and $\texttt{token}a_{00}^T$ is the activation variable $v_{\langle tok, \langle 1, h \rangle \rangle}$ (defined in Sec. 3 as the output of $\texttt{aggregate}$).

Therefore, similarly $\text{head}_{10}$-$\text{head}_{00}$-$\texttt{token}$ as a graph vertex sends real activation $\boldsymbol{V}_{1,0}\boldsymbol{V}_{0,0}\boldsymbol{E}\texttt{token}a_{00}^Ta_{10}^T$, and $\texttt{token}a_{00}^Ta_{10}^T$ is the activation variable $v_{\langle tok, \langle 0,0 \rangle, \langle 1,0 \rangle \rangle}$

For MLP layers, as mentioned in Step 2.2 in Sec. 3, we replace MLP with primitives. If the primitive is found, we can trace activation variables through the MLP. For example, $\text{mlp}_0$-$\text{head}_{00}$-$\texttt{token}$ as a vertex sends the following activation:
$$\texttt{mlp}(\boldsymbol{V}_{00}\boldsymbol{E}\texttt{token}a_{00}^T) \tag{25}$$
$$= \boldsymbol{C}_{\texttt{mlp}-\text{head}_{00}-\texttt{token}}\texttt{element\_wise\_op}(\texttt{token}a_{00}^T, \texttt{func=}f) \tag{26}$$

where $f$ is a known primitive, and $\texttt{element\_wise\_op}(\texttt{token}a_{00}^T, \texttt{func=}f)$ is the activation variable $v_{\langle v_{\langle tok, \langle 0,0 \rangle \rangle} \rangle}$

One more example: if a pair ($\text{head}_{10}$-$\text{head}_{00}$-$\texttt{token}$, $\text{mlp}_0$-$\text{head}_{00}$-$\texttt{token}$ ) is remained as input for the QK vertex of head 0 at layer 2, it means the following selector:

$$(\boldsymbol{Q}_{2,0}\boldsymbol{V}_{1,0}\boldsymbol{V}_{0,0}\boldsymbol{E}\texttt{token}a_{00}^T a_{10}^T)^T (\boldsymbol{K}_{2,0}\boldsymbol{C}_{\texttt{mlp-head}_{00}-\texttt{token}}\texttt{element\_wise\_op}(\texttt{token}a_{00}^T,\texttt{func=}f)) \quad (27)$$

$$= \underbrace{a_{10}a_{00}\texttt{token}^T}_{\texttt{q}=v_{\langle tok,\langle 0,0\rangle,\langle 1,0\rangle\rangle}} \underbrace{\boldsymbol{E}^T\boldsymbol{V}_{0,0}^T\boldsymbol{V}_{1,0}^T\boldsymbol{Q}_{2,0}^T\boldsymbol{K}_{2,0}\boldsymbol{C}_{\texttt{mlp-head}_{00}-\texttt{token}}}_{\texttt{op}=\boldsymbol{A}} \underbrace{\texttt{element\_wise\_op}(\texttt{token}a_{00}^T,\texttt{func=}f))}_{\texttt{k}=v_{\langle v_{\langle tok,\langle 0,0\rangle\rangle}\rangle}} \quad (28)$$

In summary, activation variables can be captured by tracing from initial `token` or `pos`, and iteratively consider each unit in the path from bottom to top. If it is attention head, then multiply the attention weights. If it is MLP layer, evaluate the `element_wise_op` with the known primitive. If the MLP does not have a matched primitive, we lose track of the activation variables and use backup approach of inspecting downstream effect, which will be described later.

### G.2. Primitive Matching for Single-input MLP

We first describe the primitive matching for per-position operations in the D-RASP program – equivalently, for MLPs in the pruned transformer.

In most cases, MLPs can be split into single-input MLPs (see Appendix F.2) with little effect on match accuracy. In these cases, for each single-input MLP remaining in the pruned graph, we search among a list of predefined single-input primitives to replace it, starting from the first layer. For each MLP, the replacement is done as follows:

1. Collect output activations $v \in \mathbb{R}^{d \times N}$ (see method in Appendix F.2) of the target MLP (which is also activation variables before finding a primitive), and activation variables $x \in \mathbb{R}^{d(x) \times N}$ (see method in Appendix G.1) of its inputs. Columns of $v$ is the result of per-column operation of $x$, i.e., $v(i)$ only depends on $x(i)$. We collect $N = 20000$ such pairs after filtering out $v(i)$ whose gradient is zero. Note that capturing $x$ requires successful primitive matching of previous MLP in the path, if the previous MLP does not have a matched primitive, we stop the primitive matching process for the current MLP as we do not know the input activation variables. So the current MLP does not have a matched primitive either.

2. Test primitives. For each primitive $f_j$ in the single input primitive list, do the following:
   (a) Obtain the transformed activation variables $w \in \mathbb{R}^{d(w) \times N}$ by applying $f_j$ to $x$ (i.e., $w(i) = $ `element_wise_op`$(x(i),\texttt{func=}f_s)$). $d(w)$ depends on the output dimension of $f_j$.
   (b) Obtain a linear mapping $\boldsymbol{C}$ by solving the problem $\min_{\boldsymbol{C}} \|v - \boldsymbol{C}w\|_F^2$, with solution $\boldsymbol{C} = vw^+$, where $w^+$ is the pseudo-inverse of $w$.
   (c) Evaluate the replaced MLP with match accuracy. In each forward pass, replace MLP output $v$ with $\boldsymbol{C}$`element_wise_op`$(x,\texttt{func=}f_j)$ and compute match accuracy.

3. Select the best primitive. Importantly, to simplify the generated program, we favor replacing MLP with linear operation, i.e., $f_{no-op}$. We first try $f_{no-op}$, if match accuracy is above a threshold then we match $f_{no-op}$ with the MLP and skip all other primitives. In our experiments we use 0.92 as the threshold. Otherwise, we test all primitives and match the one with the highest accuracy[8] with the MLP. Meanwhile, save the $\boldsymbol{C}$ corresponding to the selected primitive. Afterwards for downstream layers, $w$ computed from the selected primitive will be the activation variable for the MLP, and $\boldsymbol{C}$ will be used to compose the `op=` matrix in `select` and `project`

4. However, if all primitives fail to match original model's output — that is, the match accuracy of all primitives is lower than a threshold (we use 0.9) — we conclude that no primitive in our library matches the MLP. In this case, we store the variables and visualize them so that one can still interpret the MLP manually (Section G.4). One may also add new primitive once they get insights from manual inspection, so that the MLP is matched to a primitive.

In our experiments, we search among the library below:

---

**Library of Single-input primitives**

1. $f_{no-op}(x) := x$

---

[8] we favor $f_{no-op}$ again by increasing its accuracy by 0.01 when comparing with others

2. $f_{sharpen}(x)_i := \frac{x_i^n}{\sum_i x_i^n}$, where hyperparameter $n \in \{2, 3, 5\}$.

3. $f_{harden}(x) := e_{\arg\max_j} x$, i.e., $n \to \infty$ version of $f_{sharpen}$

4. $f_{01-balanced} := \left[\max(x_1 - x_0, 0)^n, \max(x_0 - x_1, 0)^n, 1 - \max(x_1 - x_0, 0)^n - \max(x_0 - x_1, 0)^n\right]$, where hyperparameter $n \in \{0.5, 0.05, 0.01\}$. Only test this when input activation variable is `token` and the alphabet $\Sigma$ only contains "0" "1" besides special tokens.

5. $f_{is-pure} := \left[\mathbb{I}(x_1 > \tau), \cdots, \mathbb{I}(x_{d(x)} > \tau), \ 1 - \sum_{i=1}^d \mathbb{I}(x_i > \tau)\right]$, hyperparameter $\tau \in \{0.95, 0.9, 0.85, 0.8, 0.75, 0.7\}$.

where we abuse notation a bit, use $x$ as a vector in this list (so is the $x(i)$ outside of the list), $x_i$ is an entry of it, $x_a$ denotes the entry correspond to token "a", and similarly $x_b, x_0, x_i$.

## G.3. Primitive Matching for Multi-input MLP

For some models, we observe that if we split MLPs, the resulting model often do not make the same prediction as the original model, no matter how many edges are enabled. Therefore, we do not split MLP for these models. Our idea of replacing MLP with primitives can still work in this case, though the search space of possible primitives becomes much larger.

We do the same as we do for single-input MLPs, with the following small changes:

1. There are multiple input activation variables, $x_1, \ldots, x_s$, we collect them all.

2. The tested $f_i$ operates over multiple inputs $w = $ `element_wise_op`$(x_1, \ldots, x_s, $ `func`$=f_i)$, and is from a different list (see below).

---

**Library of Multiple-input primitives**

1. $f_{keep-i}(x^{(1)}, \ldots, x^{(i)}, \ldots, x^{(s)}) := x^{(i)}$, where hyperparameter $i \in \{1, \ldots, s\}$, $x^{(i)}$ is a vector corresponding to the i-th operand.

2. $f_{cartesian}(x^{(1)}, \ldots, x^{(s)})_{j_1, \ldots, j_s} := \frac{\min\{x_{j_1}^{(1)}, \ldots, x_{j_s}^{(s)}\}}{\sum_{j_1, \ldots, j_s} \min\{x_{j_1}^{(1)}, \ldots, x_{j_s}^{(s)}\}}$, where output $f_{cartesian}(x^{(1)}, \ldots, x^{(s)}) \in \mathbb{R}^{d(x^{(1)}) \times \cdots \times d(x^{(s)})}$, and $j_1, \ldots, j_s$ are indices for each variable. The output is then flatten to a vector $\in \mathbb{R}^{d(x^{(1)}) \ldots d(x^{(s)})}$.

---

## G.4. Backup Approach (No Matching Primitive): Inspecting Downstream Effect

In case no primitive is found for the MLP, one can still interpret it manually by observing the downstream effects caused by the MLP receiving different inputs. There are two types of downstream effects: one is how MLPs affect the QK product and thus attention weights (or information flow between residual streams); the other is, given fixed attention weights, how MLPs affect model output. The former corresponds to MLPs in the paths ends with QK product in the final pruned graph (i.e., a `select` operator), the latter corresponds to MLPs in the paths ends with unembedding (i.e., a `project` operator). We discuss these two cases separately.

**Paths ending with unembedding (i.e., `project` operator)** Our method for collecting output effect is as follows:

1. Given a path containing unexplained MLP (i.e., MLP not replaced with a primitive from the library) and ending with unembedding in the pruned graph, we first identify the first unexplained MLP in the path (in earliest layer). Collect its input activation variable samples $v_{inp}$ and corresponding output of this MLP $v_{out}$. We aim to make $v_{out}$ interpretable by tracing downstream effect until output vocabulary space.

2. We absorb all matrices acting on the MLP output throughout the downstream path into the MLP output. Formally, we iterate over downstream components afterwards on the path. For each attention head in the path, we update it

by multiplying with OV matrix $v_{out} \leftarrow \boldsymbol{V} v_{out}$. If it is an MLP layer, we update it by applying the MLP layer at a black-box function $v_{out} \leftarrow \mathtt{mlp}(v_{out})$. If it is the final logit, we update it by multiplying with unembedding matrix $v_{out} \leftarrow \boldsymbol{U} v_{out}$. So finally $v_{out}$ is projected to vocabulary space. Therefore, we obtain pairs of interpretable vectors, which are columns of $(v_{inp}, v_{out})$.

3. In terms of the D-RASP program, we add a per-position operation mapping $v_{inp}$ to an output of dimension $d(v_{out}) = |\Sigma|$. We replace the original $\mathtt{project}$ operation by $\mathtt{project}(v_{out}, I_{|\Sigma| \times |\Sigma|})$, that is, the $\mathtt{project}$ operation uses the identity matrix.

4. To aid interpretation, we subtract each column of $v_{out}$ with its second largest value, since this is equivalent under softmax. We emphasize the token promoted most. When using BCE loss, we do not do this.

**Example**  For example, in the pruned model, if an unexplained MLP is in a path of the form:

$$\mathtt{unembedding} : \mathrm{head}_{10}\text{-}\mathrm{mlp}\text{-}\mathrm{head}_{01}\text{-}\mathrm{token}$$

then we inspect the following pairs: instances of its input activation variable $v_{\langle tok, \langle (0,1) \rangle \rangle}$, and their corresponding MLP output $v_{mlp}$ after multiplied with $\boldsymbol{V}_{1,0}$ and unembedding matrix $\boldsymbol{U}$. Each pair is two vectors $\in \mathbb{R}^{|\Sigma|}$, one shows an aggregation of input tokens from the context, another shows which tokens would be promoted or suppressed if the MLP's output for such aggregation is attended or routed to the prediction by downstream heads.

For another example, the path can be:

$$\mathtt{unembedding} : \mathrm{head}_{20}\text{-}\mathrm{mlp}'\text{-}\mathrm{head}_{10}\text{-}\mathrm{mlp}\text{-}\mathrm{head}_{01}\text{-}\mathrm{token}$$

Assuming $\mathrm{mlp}$ can't find a matched primitive, we cannot apply primitive matching to $\mathrm{mlp}'$ as well because we cannot get the input activation variable. Our backup approach explain the two MLPs together, by treating the part $\mathrm{mlp}'$-$\mathrm{head}_{10}$-$\mathrm{mlp}$ as a black-box and inspecting its input and output.

However, if there are attention layers between the first unexplained MLP to the last MLP on the path, and any of them produces non-one-hot attention weights, then the final transformed $v_{out}$ is not correct. Because when tracing through MLPs, we actually assume each representation vector is fully attended, instead of being mixed with others.

In the last example, the final $v_{out}$ represents what output token would be promoted/suppressed when the output of $\mathrm{mlp}$ is fully routed to the prediction. Because the MLP output on a weighted sum of representations is not equal to the weighted sum of MLP output on the same representations, so we do not know the true effect of each individual $\mathrm{mlp}$ output vector when they are mixed with others by the $\mathrm{head}_{10}$.

Therefore, we rigorously identify the condition when the backup approach faithfully represents the downstream effect and when it does not. We raise a warning in our pipeline if such condition is not satisfied.

**Paths ending with QK product**  Similar to the previous approach, we do the following:

1. Given a QK product whose query or key contains at least one unexplained MLP in the pruned graph, we first identify the first unexplained MLP in the path (in earliest layer) for both query and key. Collect its input activation variable samples $v_{inp}^{(q)}$ and corresponding output of this MLP $v_{out}^{(q)}$. Similarly for $v_{inp}^{(k)}$ and $v_{out}^{(k)}$.

2. Similar to previous method, we iterate over downstream components afterwards on the path to get final $v_{out}^{(q)}$ and $v_{out}^{(k)}$. When only one of query and key contains unexplained MLP, we apply the method in Appendix G.1 for the other to get these two variables easily.

3. We do $(v_{out}^{(q)})^T \cdot \boldsymbol{Q}_{l,h}^T \cdot \boldsymbol{K}_{l,h} \cdot v_{out}^{(k)}$. Each entry in this matrix describes how each pair of columns in $v_{inp}^{(k)}$ and $v_{inp}^{(q)}$ is associated. So We make $v_{out}$ interpretable by tracing downstream effect on attention. In other words, we inspect what are the input variable pairs that will be associated in the QK product. The unexplained MLP plays a role in this association, and it is understood together with other components.

4. In terms of the D-RASP program, we add a per-position operation mapping to this output, and replace the $\mathtt{select}$ operation by one where the $\boldsymbol{A}$ matrix is the identity matrix.

5. To aid interpretation, we center the products of the same query, since this is equivalent under softmax. Moreover, we apply K-means clustering to query's activation variables, and show what are the keys associated with each "type" (cluster) of query.

### G.5. Remark on Logit Lens

Our backup approach uses an idea somewhat similar to logit lens (nostalgebraist, 2020) or Direct Logit Attribution, but our method allows us to determine the **correct** way to transform the MLP output. For example, when explaining the MLP in this path: `unembedding` : $\text{head}_{10}\text{-mlp-head}_{01}\text{-token}$, the activation should first be multiplied by OV matrix of head 0 at layer 1 before multiplying by the unembedding matrix, because this is how it contributes to the model prediction according to the pruned graph. This is a very important distinction from naively applying the unembedding matrix, which assumes that a components always contributes mainly via a direct connection to the output. Thus, for components which contribute to the output mainly by indirect connection (there are other components along the path between it and the final logit), such naive projecting to vocabulary space is reading the "side-effect" in the direct path and causing **illusory** interpretation. Unfortunately, many studies have done (and are still doing) this with justification only based on plausibility of the results. Moreover, many components contribute by forming the correct QK circuits (i.e., which tokens should be attended) instead of via OV circuits (what's the effect on output if attended). Our backup approach successfully accounts for this (i.e., paths ending with QK product).

## H. Replacing attention and unembedding matrices with primitives

We try to replace each parameter matrix in `select` and `project` with a subset of primitives from Table 4. We use only a subset of primitives for each type of parameter matrices, since some of the primitives are not applicable to some of the types of parameter matrices. Specifically, "Diagonal", "$k$th Diagonal" and "Every $k$th" primitives are designed for replacing matrices and are not applicable to replacing vectors. Apart from that, unembedding matrices always project the activation to the vocabulary, which makes "Decreasing", "Increasing", "$k$th Diagonal" and "Every $k$th" primitives not interpretable when applied to unembedding projections, since vocabulary tokens do not have intrinsic ordering.

We then notice that in many cases the dimension of the matrix that needs to be replaced corresponds to vocabulary size (the variable which is being multiplied by the primitive corresponds to a path $\langle tok, x_1 \ldots x_k \rangle$). In these cases, it is beneficial to separate the entries in the primitives corresponding to special tokens and non-special tokens. We then separate the primitive matrix into two pieces: when query is a special token and when it is a non-special token, and select a separate primitive for each. In this case, we use separate arguments `op=` and `special_op=` for those two primitives.

**Selecting Primitives**     When selecting the primitives, we replace primitives one by one, starting from the lowest layers. When replacing the primitive, we try all possible combinations of tuples of primitives (attend-to-special, attend-to-non-special) in the order:

- For attention bias (vector): "Zeros", "to BOS", "to EOS", "to SEP", "Decreasing", "Increasing".

- For attention matrix: "Zeros", "to BOS", "to SEP", "Diagonal", "Second Diagonal", "Third Diagonal", "Every Second", "Every Third", "Decreasing", "Increasing".

- For logits bias (vector): "Zeros", "to EOS".

- For logit matrix: "Zeros", "to EOS", "Diagonal".

When the accuracy of replacement is above the threshold ($0.95\times$ match accuracy of the model after pruning), we accept it and do not try other primitives.

**Simplifying Remaining Matrices**     We then simplify the parameter matrices that were not replaced with predefined primitives. The goal here is especially to remove information that is not causally relevant to the model's predictions, to aid in interpretation. We initialize each of the parameter matrices not replaced with predefined primitives $A_i$ as the original parameter matrices, and optimize them all together in two steps. We denote all the optimized parameters as $A = [A_1, A_2, \ldots]$. First, we incentivize the parameters to be close to zero (penalty $p := \|\boldsymbol{A}\|_1$), and then we incentivize them to be integers (penalty $p := \|\boldsymbol{A} - round(\boldsymbol{A})\|_1$). We train the matrix for 2,000 steps with the loss $l + \lambda p$, where $p$ is

| Primitive Name | Matrix Definition $(A = \{a_{ij}\})$ | Matrix example | Vector Definition $(a = \{a_i\})$ | Vector example |
|---|---|---|---|---|
| Diagonal `op=(k==q)` `op=(inp==out)` | $a_{ij} = \delta_{i=j}$ | | — | — |
| $k$th Diagonal `op=(k==q-k)` | $a_{ij} = \delta_{i=j+k}$ | | — | — |
| Every $k$th `op=(k%k==q%k==0)` | $a_{ij} = \delta_{i \bmod k = 0\ \&\ j \bmod k = 0}$ | | — | — |
| to BOS/SEP/EOS `op=(k==BOS/SEP/EOS)` `op=(out==BOS/SEP/EOS)` | $a_{ij} = \delta_{j\text{ is BOS/SEP/EOS}}$ | | $a_i = \delta_{i\text{ is BOS/SEP/EOS}}$ | |
| Decreasing `op=(k is first)` | $a_{ij} = \frac{j}{m}, \quad (A \in \mathbb{R}^{n \times m})$ | | $a_i = \frac{j}{n}, \quad (a \in \mathbb{R}^n)$ | |
| Increasing `op=(k is last)` | $a_{ij} = \frac{m-j+1}{m}, \quad (A \in \mathbb{R}^{n \times m})$ | | $\frac{n-j+1}{n}, \quad (a \in \mathbb{R}^n)$ | |
| Zeros `op=(uniform selection)` | $a_{ij} = 0$ | | $a_i = 0$ | |

*Table 4.* Our library of primitives for $\boldsymbol{A}, \boldsymbol{b}$. We scale all matrices with a large number (we take $10{,}000$). This allows us to simulate near-hard outputs after softmax. This is both in line with the learned solutions, which often hold large magnitudes (cf. Figure 15) and with theoretical arguments showing that large attention logits are needed to produce discrete outputs (e.g. Chiang & Cholak, 2022; Zhou et al., 2024). In the case of attention, we note that rows denote the query dimensions and columns denote key dimensions. In the case of prediction logits, we note that rows denote the input dimensions and columns denote output dimensions. In the case of selector matrices $A$, rows denote queries; columns denote keys. In the case of `project` matrices $A$, rows denote inputs; columns denote outputs. When row and column counts differ, matrices are not square; in this case, we start from the top left corner and truncate the matrix accordingly.

the penalty and $l$ is the original task loss. We stop training if average match accuracy with the original model drops below the threshold (defined as 0.91 * match accuracy of the model after pruning) over the interval of 10 steps. If this happens, we revert to the checkpoint when the match accuracy is still above the threshold. We use batch size 120, lr $10^{-4}$, and sweep $\lambda$ over $10^{-1}, 10^{-2}, 10^{-4}$.

We then forcibly round each primitive matrix if the match accuracy of the model stays above the threshold, to account for potential imperfections in rounding after the optimization procedure .

# I. Example of Primitive Replacement

Here, we show examples of the matrices in Unique Copy (Figure 3), after replacement with primitives (Figure 14) and for comparison when performing regularization towards zero and integers, but no primitive replacement (Figure 15). Comparison shows that the basic patterns are similar (e.g., diagonal and off-diagonal matrices, with special behavior for special tokens), but the primitives bring the behavior out more clearly.

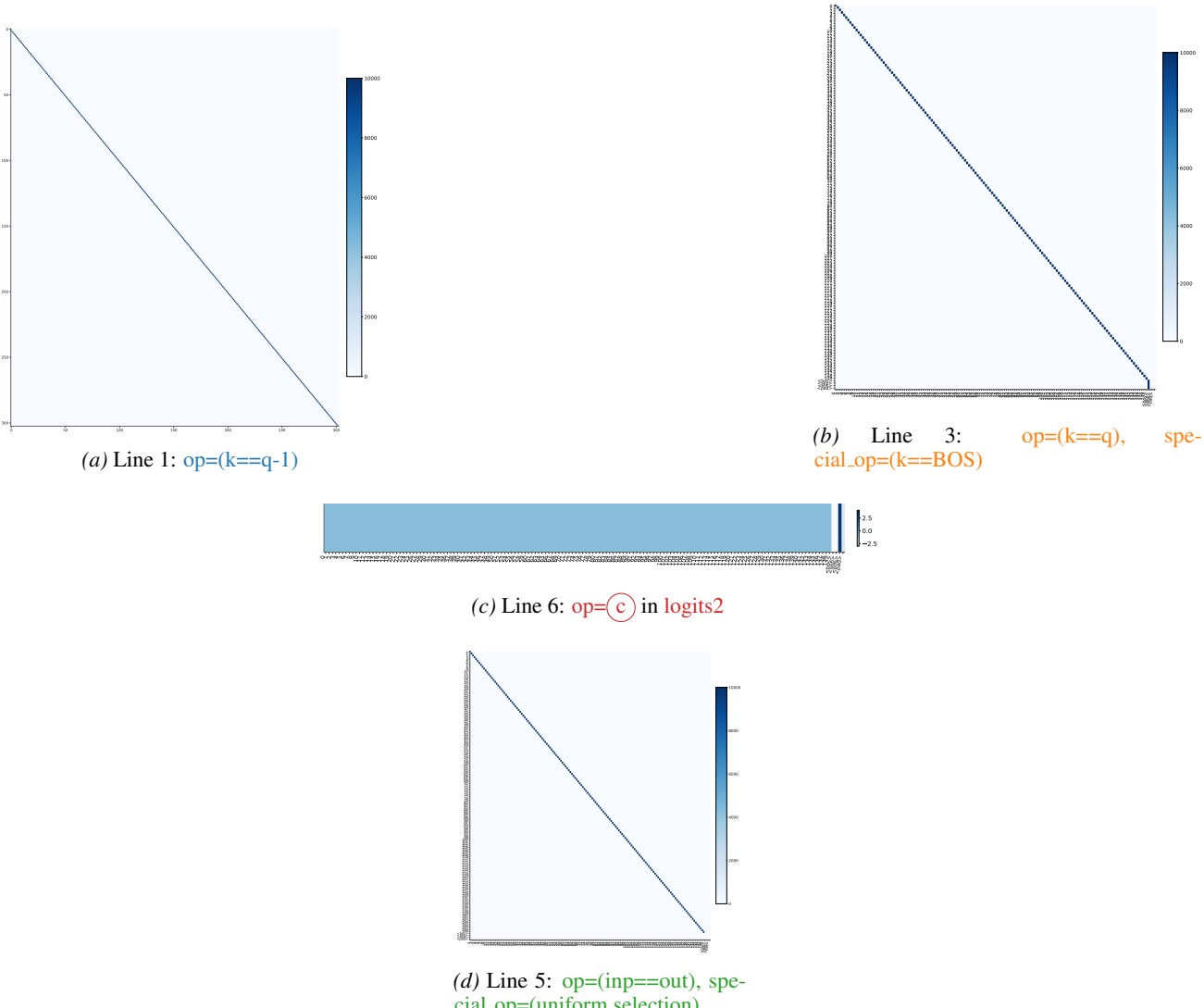

*(a)* Line 1: op=(k==q-1)

*(b)* Line 3: op=(k==q), special_op=(k==BOS)

*(c)* Line 6: op=(c) in logits2

*(d)* Line 5: op=(inp==out), special_op=(uniform selection)

*Figure 14.* Heatmaps supporting the program for unique_copy model. This is discussed in Main paper, Figure 3.

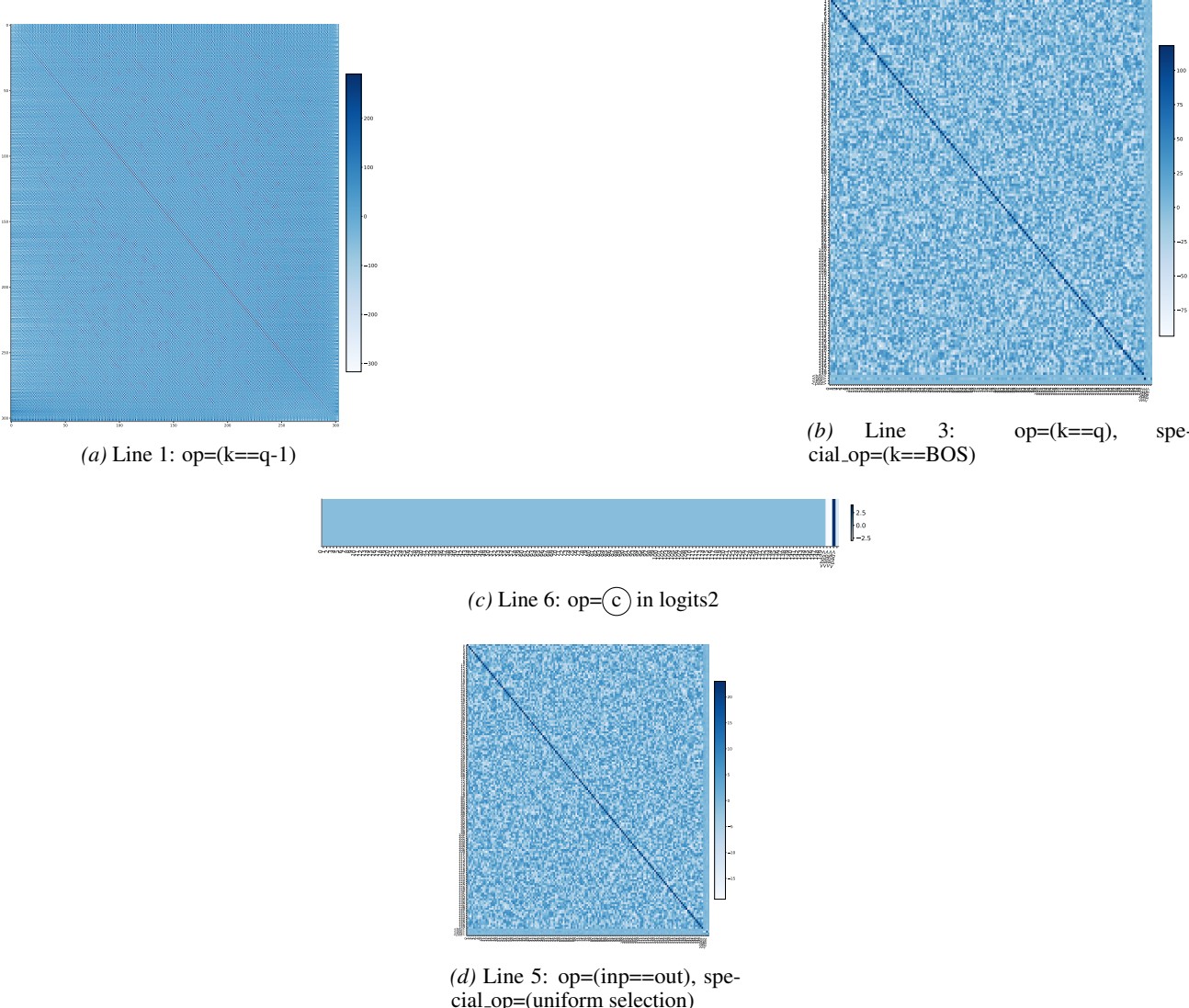

*(a)* Line 1: op=(k==q-1)

*(b)* Line 3: op=(k==q), spe-cial_op=(k==BOS)

*(c)* Line 6: op=©c in logits2

*(d)* Line 5: op=(inp==out), spe-cial_op=(uniform selection)

*Figure 15.* Heatmaps supporting the program for unique_copy model. Heatmaps are only rounded, not replaced with primitives.

## J. Decompiled Programs on Algorithmic Tasks

**How to read this section** Each subsection below shows a D-RASP decompilation of a trained transformer. We report the target task, description of the model's architecture, and two performance metrics: task accuracy and match accuracy with the original model. We report accuracies after reparametrization and simplification (pruning) steps, but before replacement of attention, unembedding projections, or MLPs with primitives, as well as accuracies after all steps of the pipeline.

D-RASP code for each model consists of code and supporting heatmaps that specify operations. For ease of understanding, we also report visualizations of variables computed on example input.

Throughout, for selector matrices $A$, rows denote query dimensions and columns denote key dimensions. For projection matrices $A$, rows denote input dimensions and columns denote output dimensions. For *variables heatmaps*, in the case of activation variables, the x-axis denotes the positions in a sample input string; the y-axis denotes the dimensions in the variable. In the case of selector variables, the rows denote query positions, the columns denote key positions. We show causal masking by graying out of cells.

In the programs where MLP operations cannot be replaced with primitives, we show visualizations inspecting their downstream affect, computed as described in Section G.4. When interpreting the effect on unembedding projection, we

| Task | Program |
|------|---------|
| Binary Majority | Sec. J.1 |
| Binary Majority Interleave | Sec. J.2  J.3 |
| Count | Sec. J.4  J.5 |
| Most Frequent | Sec. J.6  J.7 |
| Sort | Sec. J.8 |
| Unique Bigram Copy | Sec. J.9  J.10 |
| Unique Copy | Sec. J.11 |
| Unique Reverse | Sec. J.12  J.13 |
| $D_{12}$ | Sec. K.1 |
| $D_2$ | Sec. K.2 |
| $D_4$ | Sec. K.3 |
| $(aa)^*$ | Sec. K.4 |
| $aa^*bb^*cc^*dd^*ee^*$ | Sec. K.5 |
| $\{a,b\}^*d\{b,c\}^*$ | Sec. K.6 |
| Tomita1 | Sec. K.7 |
| Tomita2 | Sec. K.8 |
| Tomita7 | Sec. K.9 |

visualize inputs to MLP that give rise to high and low logits of a few representative tokens in separate plots. When interpreting the effect on attention logits, we visualize pairs of keys and queries together that give rise to high and low attention logits.

We note that, when interpreting these operations, we use the terms "elementwise operation" (which is the formal term in D-RASP as in the original RASP, Weiss et al. (2021)), "per-position operation", and "MLP" interchangeably.

### J.1. Majority

**Task Description:**

$$\langle\text{bos}\rangle\ s\ \langle\text{sep}\rangle\ \text{Maj}(s)\ \text{where}\ s \in \{0,1\}^*$$

**Architecture:** Layers: 1    Heads: 1    Hidden Dim: 16 LR: 0.001    Dropout: 0.1
**Performance (w/Pruning → w/Primitives):** Task Accuracy: $1.00 \rightarrow 1.00$; Match Accuracy: $1.00 \rightarrow 1.00$

**Code**

```
1. a1 = aggregate(s=[], v=token)           # layer 0 head 0
2. logits1 = project(inp=a1, op=(inp==out),
              special_op=(uniform selection))
3. prediction = softmax(logits1)
```

**Interpretation**  The program is essentially equivalent to the program shown for the non-binary version in Figure 1 of the main paper. `a1` holds counts of the relative frequencies of 0, 1, BOS, SEP (Figure 17a). Line 2 then projects the counts of 0, 1 onto output logits, disregarding the special tokens (Figure 17b). As a result, at SEP (where the prediction relevant to solving the task is made), the more common symbol receives the higher logit.

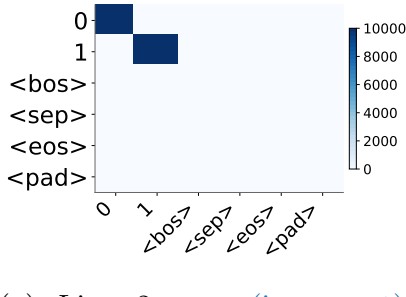

(a) Line 2:   op=(inp==out),
special_op=(uniform selection)

*Figure 16.* Heatmaps supporting the program for Majority model. (a) The identity matrix (temperature-scaled to create effectively hard attention) for normal tokens, uniform on special tokens.

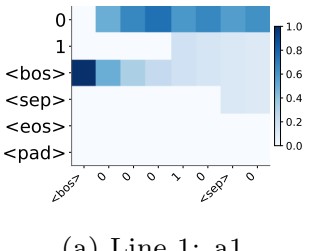
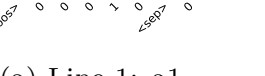

(a) Line 1: a1

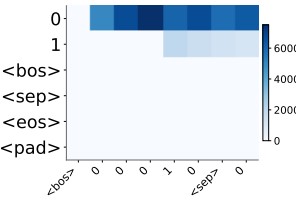

(b) Line 2: logits1

*Figure 17.* Variables Heatmaps for Majority model on an example input. (a) Aggregation computes a histogram of symbols seen so far (x-axis is the input, y-axis are the dimensions of the activation). (b) Output logits are directly obtained from (a) in the case of normal tokens, and erased for special tokens, reflecting the matrix in Figure 16.

## J.2. Majority Interleave

**Task Description:**

$$\langle\text{bos}\rangle\ s\ \langle\text{sep}\rangle\ \text{Maj}(\{s_i|i\%3 = 0\})\text{Maj}(\{s_i|i\%3 = 1\})\text{Maj}(\{s_i|i\%3 = 2\})\ \langle\text{eos}\rangle\text{where}\ s \in \{0, 1\}^*$$

**Architecture:** Layers: 2    Heads: 2    Hidden Dim: 16 LR: 0.001    Dropout: 0
**Performance (w/Pruning → w/Primitives):** Task Accuracy: $0.89 \to 0.77$; Match Accuracy: $0.92 \to 0.80$

**Code**

```
1. s1 = select(q=token, k=token, op=(a))          # layer 0 head 0

2. s2 = select(q=pos, k=pos, op=(b))          # layer 0 head 0
3. a1 = aggregate(s=s1+s2, v=token)          # layer 0 head 0
4. s3 = select(q=pos, k=pos, op=(c))          # layer 0 head 1
5. a2 = aggregate(s=s3, v=token)          # layer 0 head 1
6. s4 = select(q=pos, k=token, op=(j))          # layer 1 head 0
7. s5 = select(q=pos, k=pos, op=(k==q-2))          # layer 1 head 0
8. a3 = aggregate(s=s4+s5, v=token)          # layer 1 head 0
9. a4 = aggregate(s=s4+s5, v=pos)          # layer 1 head 0
10. is_pure_token = is_pure(token)          # layer 1 mlp
11. is_01_balance_a3 = is_01_balance(a3)          # layer 1 mlp
12. logits1 = project(inp=a4, op=(k))
13. logits2 = project(inp=token, op=(e))
```

```
14. logits3 = project(inp=pos, op=(l))
15. logits4 = project(inp=is_pure_token, op=(f))
16. logits5 = project(inp=a1, op=(g))
17. logits6 = project(inp=a2, op=(h))
18. logits7 = project(inp=is_01_balance_a3, op=(i))
19. logits8 = project(inp=pos, op=(m))
20. prediction = softmax(logits1+
                         logits2+
                         logits3+
                         logits4+
                         logits5+
                         logits6+
                         logits7+
                         logits8)
```

**Interpretation**   In this task, the model is tasked with solving three binary majority tasks in succession, with the three input strings presented in interleaved order. The main algorithm is similar to the one of the majority model (Figure 1): a1 roughly holds counts of frequencies of tokens at every third position (Figure 17c), which then get projected with the diagonal projection matrix in Line 16 (Figure 18g). a2 also shows a variation of position-driven aggregation (Figure 18c), however, it aggregates the values in all positions, except for current one (Figure 17e). It then gets projected in Line 17 (Figure 18h) by a anti-diagonal matrix, with amplitude of weights twice as low as of the projection matrix on Line 16, which suggests that this might act as a kind of correction mechanism. Logit projections in Lines 13, 15, 18 implement special behavior to predict EOS. Lines 14 and 19 implement position-based logic that promotes generation of $\langle eos \rangle$ and cancels each other's effect in 0 or 1 tokens (Figure 19k,p). Interestingly, a simpler and more accurate program is extracted from a 4-layer 1-head model (App. J.3).

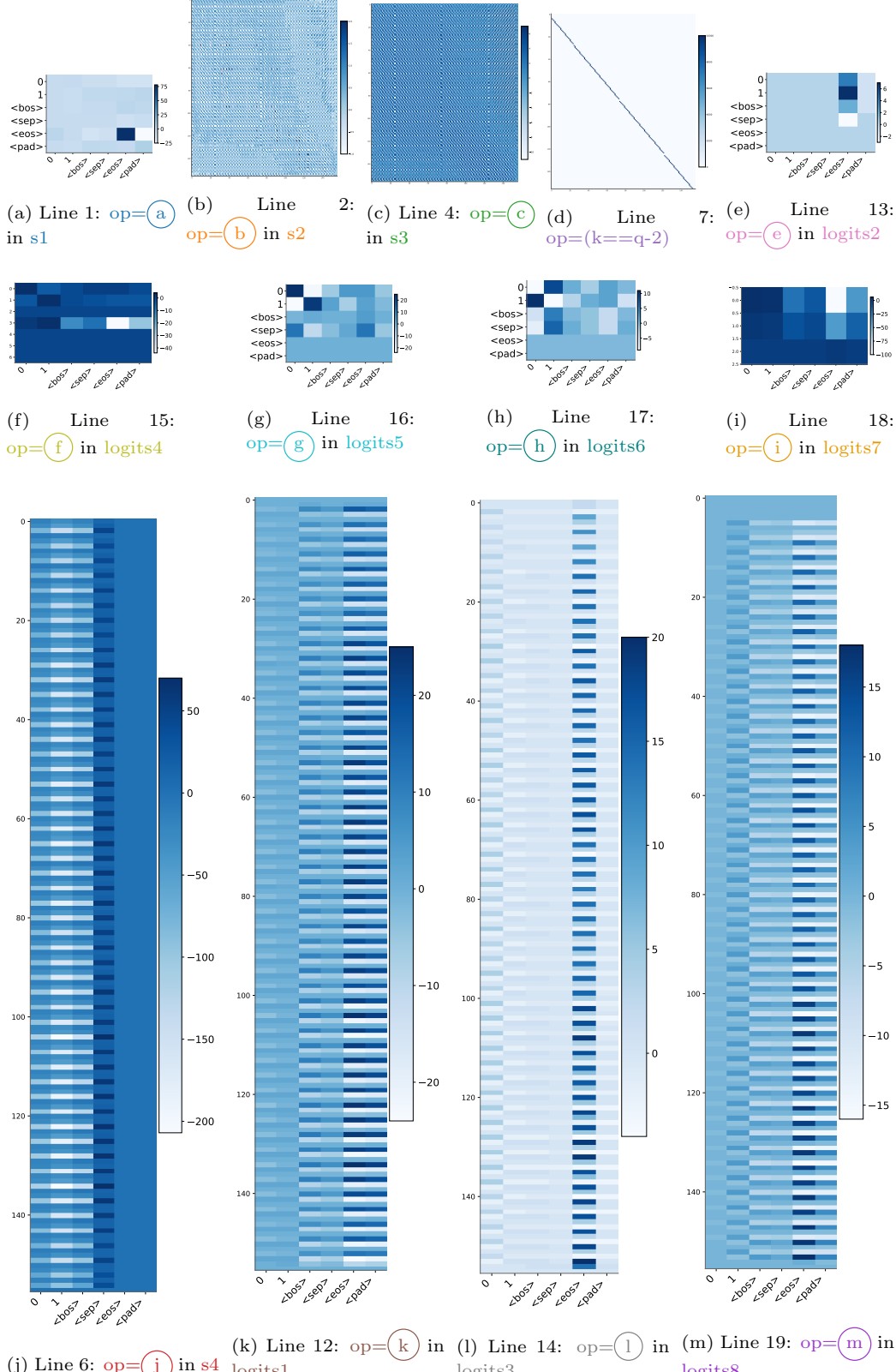

Figure 18. Heatmaps supporting the program for Majority Interleave model.

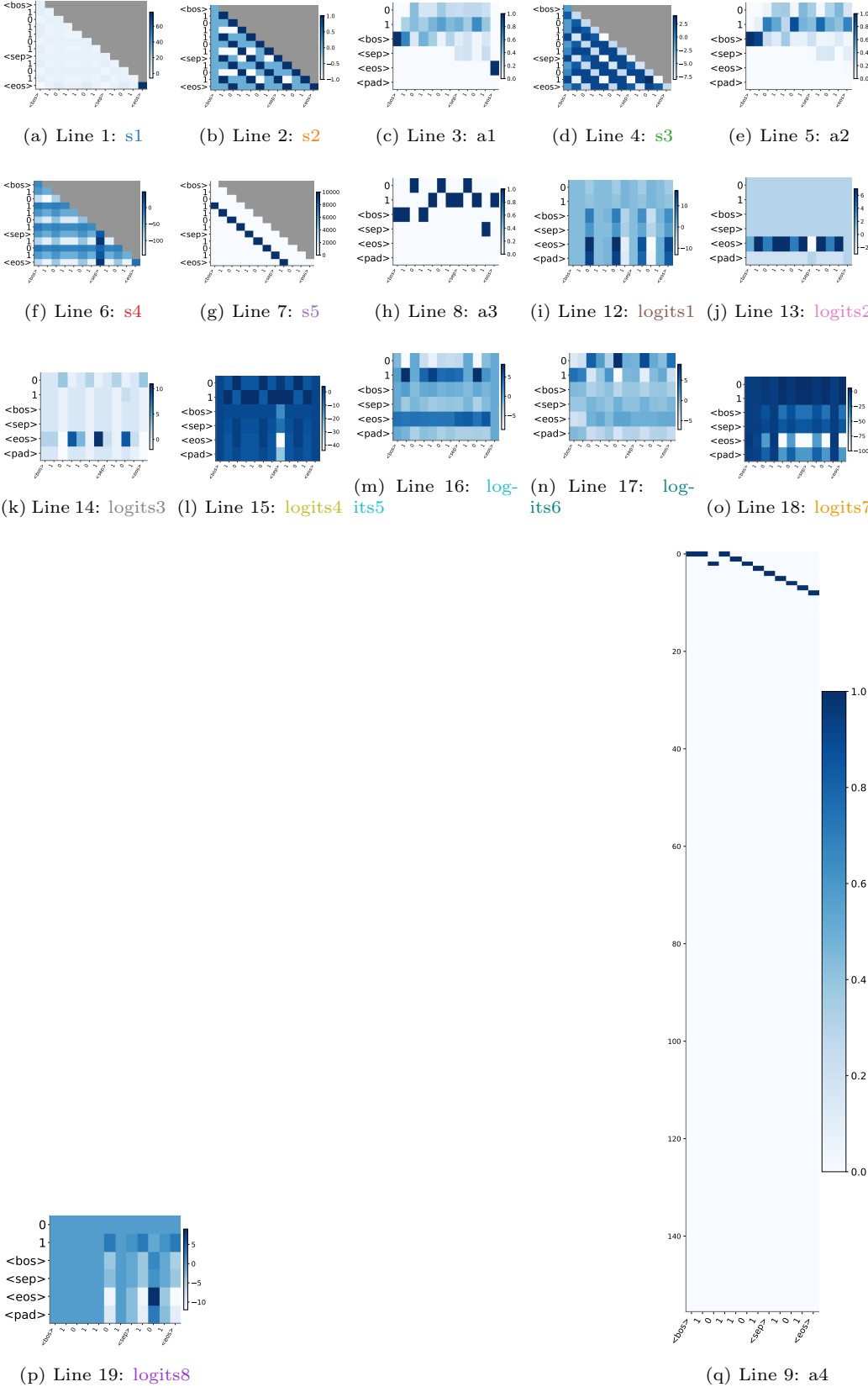

(a) Line 1: s1 (b) Line 2: s2 (c) Line 3: a1 (d) Line 4: s3 (e) Line 5: a2

(f) Line 6: s4 (g) Line 7: s5 (h) Line 8: a3 (i) Line 12: logits1 (j) Line 13: logits2

(k) Line 14: logits3 (l) Line 15: logits4 (m) Line 16: logits5 (n) Line 17: logits6 (o) Line 18: logits7

(p) Line 19: logits8

(q) Line 9: a4

*Figure 19.* Variables Heatmaps for Majority Interleave model on an example input.

## J.3. Majority Interleave : different arch with same length generalization performance

**Task Description:**

$$\langle\text{bos}\rangle \, s \, \langle\text{sep}\rangle \, \text{Maj}(\{s_i|i\%3 = 0\})\text{Maj}(\{s_i|i\%3 = 1\})\text{Maj}(\{s_i|i\%3 = 2\}) \, \langle\text{eos}\rangle \text{where } s \in \{0, 1\}^*$$

**Performance (w/Pruning $\rightarrow$ w/Primitives):** Task Accuracy: $0.97 \rightarrow 0.97$; Match Accuracy: $0.97 \rightarrow 0.97$

**Code**

```
1. s1 = select(q=token, k=token, op=(a))        # layer 0 head 0
2. s2 = select(q=pos, k=pos, op=(k%3==q%3==0))        # layer 0 head 0
3. a1 = aggregate(s=s1+s2, v=token)        # layer 0 head 0
4. new_a1 = element_wise_op(a1)        # layer 0 mlp
5. logits1 = project(inp=new_a1, op=(inp==out))
6. logits2 = project(op=(c))
7. prediction = softmax(logits1+
                         logits2)
```

**Interpretation**   In this task, the model is tasked with solving three binary majority tasks in succession, with the three input strinfs presented in interleaved order. Line 1 defines a selector ensuring that, at SEP, attention specifically goes to the non-special tokens (Figure 21a). Line 2 defines a selector putting attention on the basis of positions modulo 3 (Figure 21b). For instance, on SEP, this leads to attention to the positions 1, 4, 7, ... of the input string – which consists to the first of the three interleaved inputs. Line 3 then aggregates across these positions. Similarly, after outputting the first and second result, attention will go onto the second and third of the three interleaved input string (Figure 21c) to obtain relative counts of 0, 1 throughout the relevant slice of the input. Line 4 then performs an element-wise operation. Inspecting it shows that it provides a high logit for 1 if 1 is more common than 0 (Figure 23), a high logit for 0 if 0 is more common (Figure 22), and EOS if SEP occupies a substantial amount of activation (Figure 24). The last case happens when all three results have been output, causing the model to output EOS and stop. Line 6 boosts EOS further. Overall, the prediction is first three times a 0/1 label, one for each of the three interleaved strings, and then EOS.

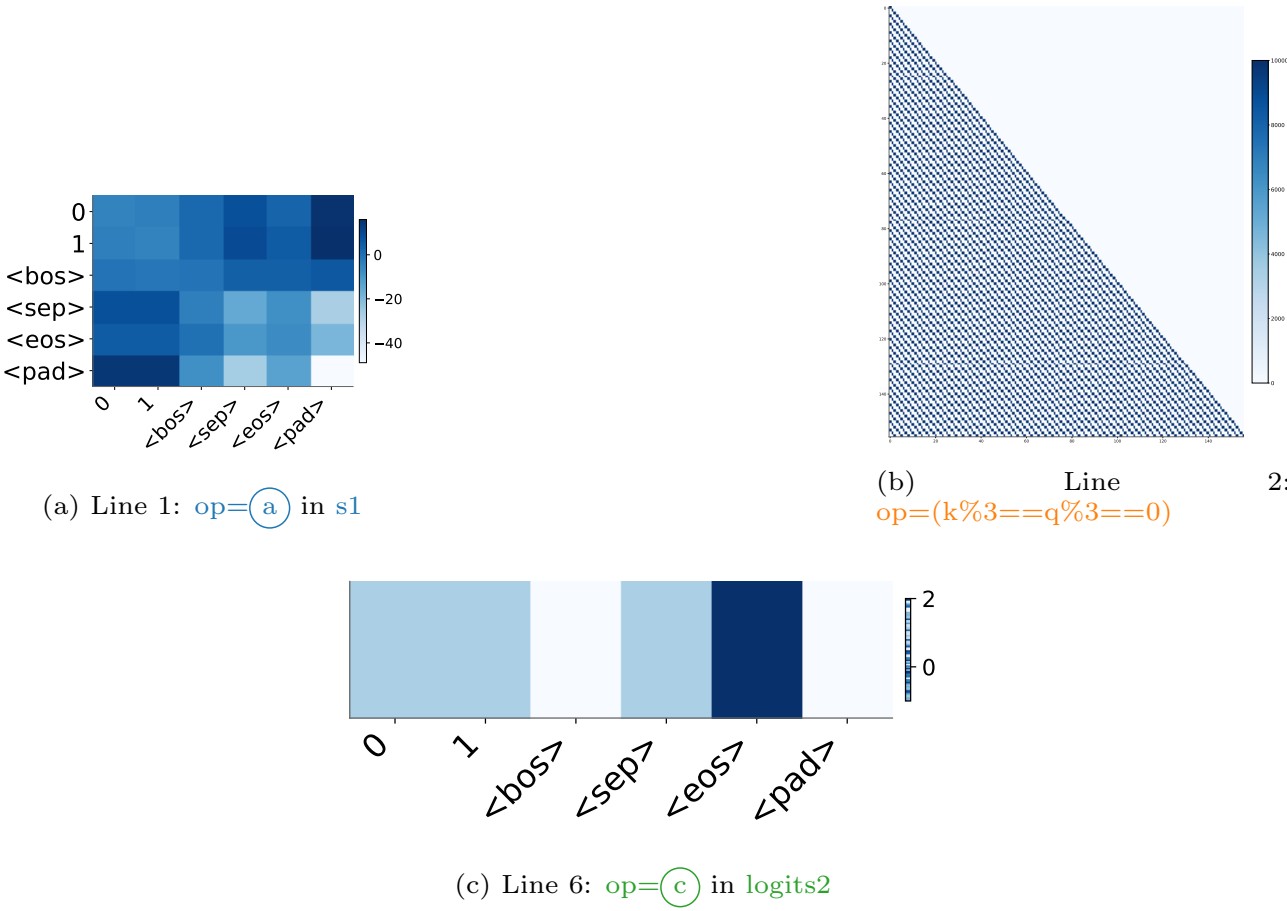

Figure 20. Heatmaps supporting the program for Majority Interleave : different arch with same length generalization performance model.

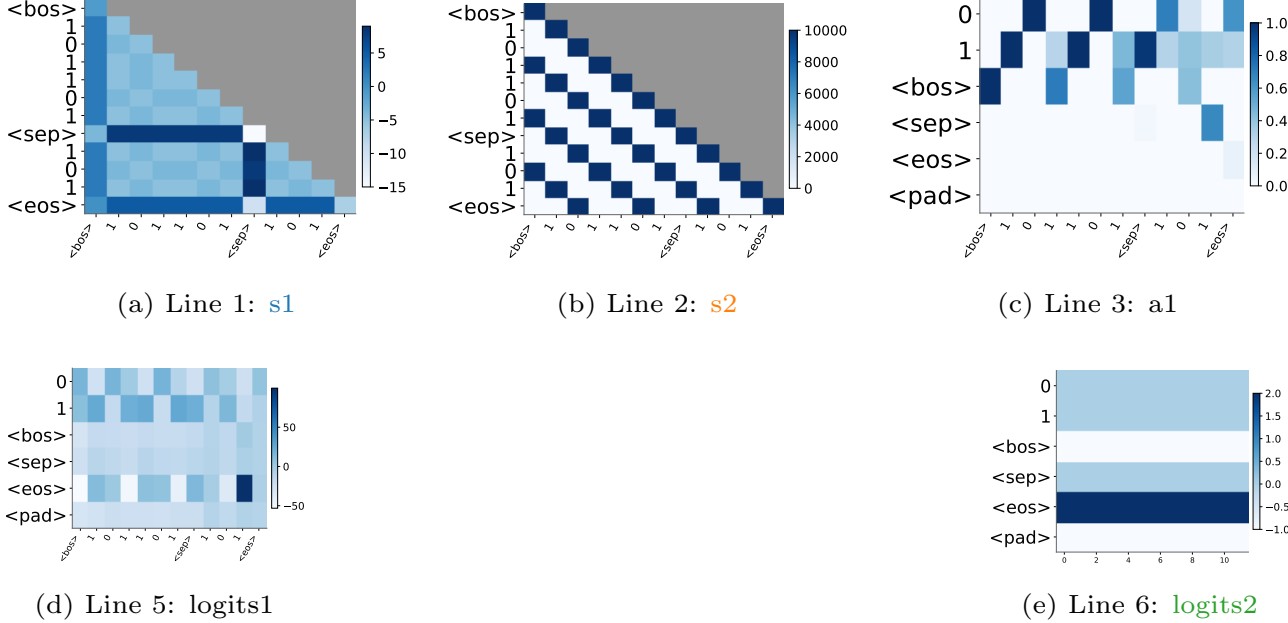

Figure 21. Variables Heatmaps for Majority Interleave : different arch with same length generalization performance model on an example input.

**MLP Input-Output Distributions**   Explaining per-position operation in Line 4 via its effect on Output Logits in Line 5

**Output Token: 0**

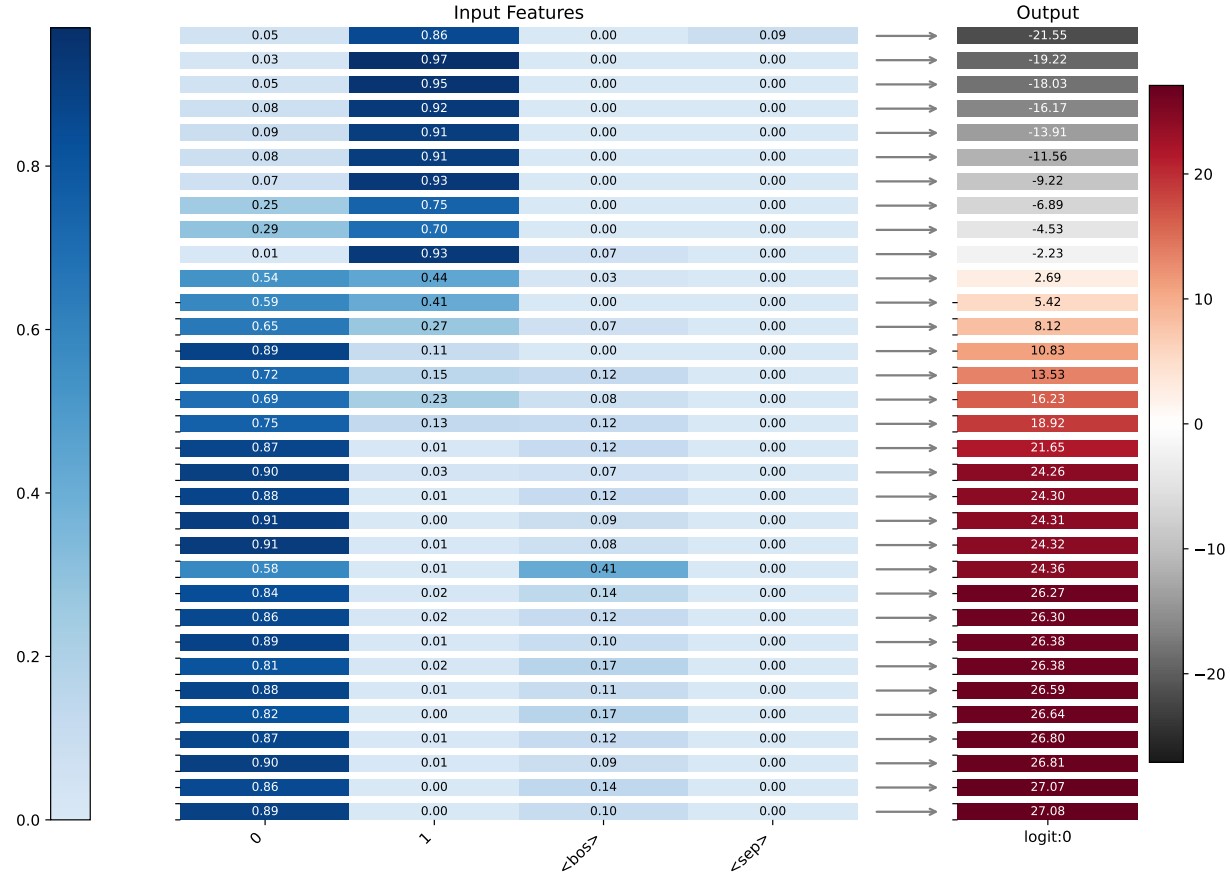

*Figure 22.* MLP Input-Output for token: 0 (sorted by logits)

**Output Token: 1**

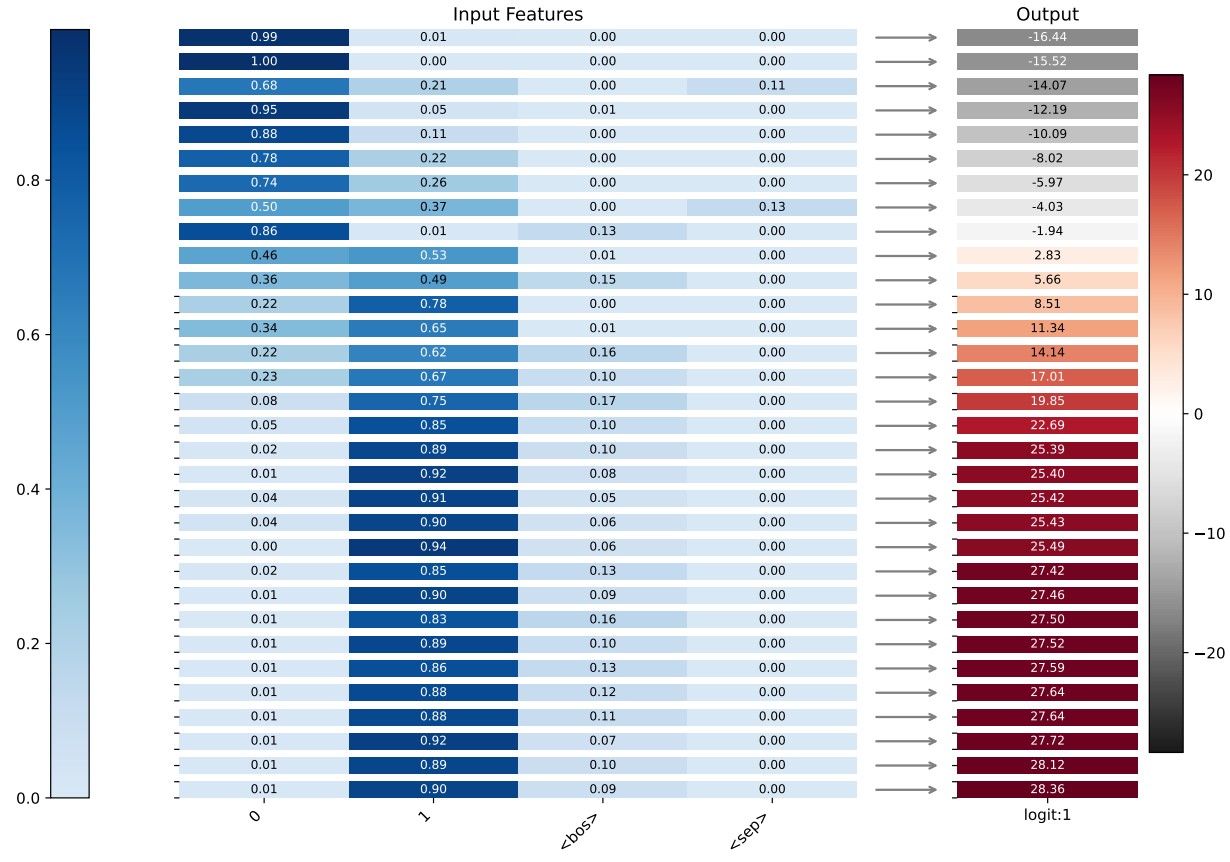

*Figure 23.* MLP Input-Output for token: 1 (sorted by logits)

**Output Token: EOS**

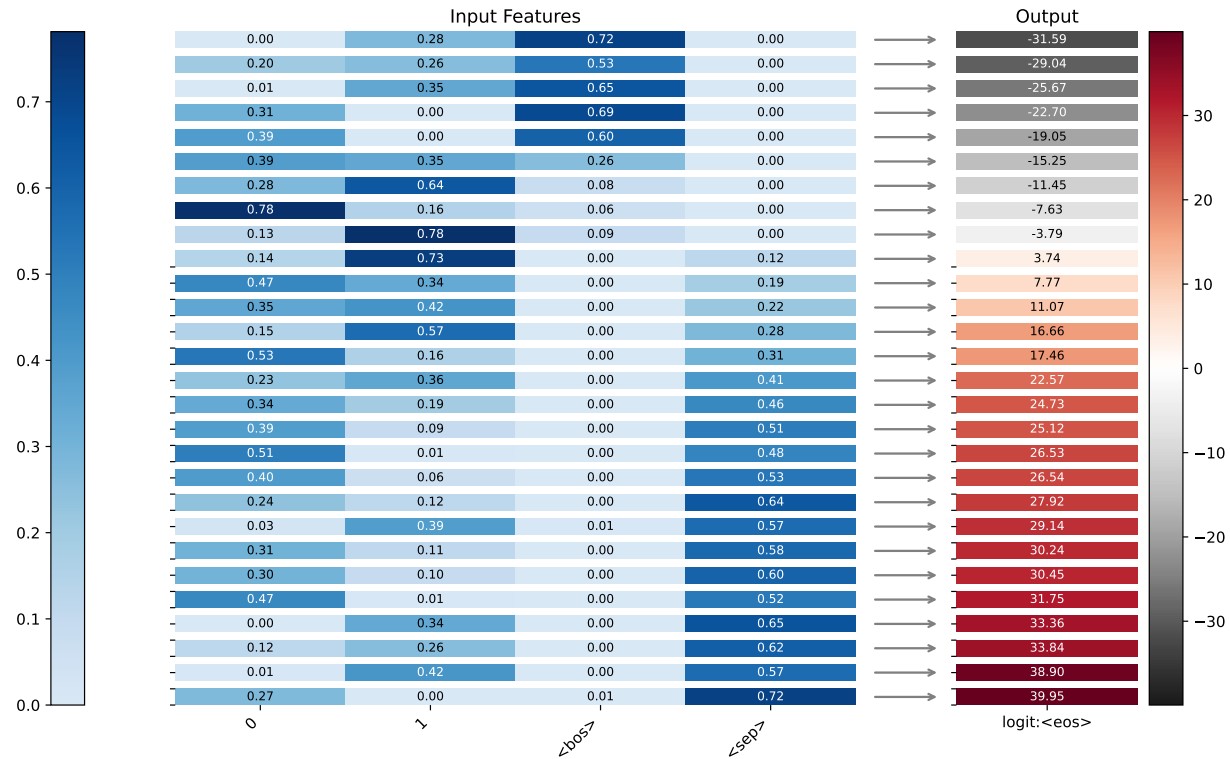

*Figure 24.* MLP Input-Output for token: EOS (sorted by logits)

## J.4. Count

**Task Description:**

$$\langle\text{bos}\rangle \, s_0, s_n \, \langle\text{sep}\rangle \, s_0 s_1 \ldots s_n \, \langle\text{eos}\rangle \text{where } s_0, s_n \in \{0, 1, \ldots, 150\}, s_n > s_0, s_{i+1} = s_i + 1$$

**Architecture:** Layers: 1     Heads: 4     Hidden Dim: 256 LR: 0.0001     Dropout: 0.1
**Performance (w/Pruning → w/Primitives):** Task Accuracy: $0.92 \rightarrow 0.92$; Match Accuracy: $0.90 \rightarrow 0.92$

**Code**

```
1.  s1 = select(q=token, k=token, op=(a))         # layer 0 head 0
2.  a1 = aggregate(s=s1, v=token)              # layer 0 head 0
3.  s2 = select(q=token, k=token, op=(b))          # layer 0 head 1
4.  s3 = select(k=token, op=(c))
5.  a2 = aggregate(s=s2+s3, v=token)             # layer 0 head 1
6.  new_a2 = element_wise_op(a2)            # layer 0 mlp
7.  logits1 = project(inp=token, op=(d))
8.  logits2 = project(inp=a1, op=(e))
9.  logits3 = project(inp=new_a2, op=(inp==out))
10. prediction = softmax(logits1+
                         logits2+
                         logits3)
```

**Interpretation**   Line 1 defines a selector favoring weight to numbers larger than the present one. So each number would attend to the previous largest number, which the terminating number. In contrast, we also notice this pattern is reversed

for SEP (please zoom in on the op matrix), so SEP always attends to the smaller number, which is the starting number. Line 2 moves the attended token, so at this point the model already get the needed information. Lines 3–5 perform a very similar operation. Interestingly, whereas Lines 1–2 used only a single $A$ matrix, these lines 3–5 achieve this effect using both a $A$ matrix and a $b$ vector giving higher weight to large numbers. We can also see the patterns in $A$ for both heads are not perfect, so they probably supplementing each other. `a1` and `a2` contain similar information, but feed into different downstream paths in the program: `a1` feeds into `logits2` (discussed below), whereas `a2` does so only after going through a per-position operation (line 6). Inspecting the operation shows that it performs a hardening operation (strongly visible at 1 and 10, Figures 27, 28), but does so mostly only for smaller numbers (no clear effect at 100, Figure 29). We attribute this to the fact that the input is heavily skewed towards larger numbers due to the $b$ vector. We are now ready to discuss how the output logits are generated. First, as also discussed in the main paper, line 7 directly maps `token` to the increment of the current number via an off-diagonal matrix (Figure 25d); this accounts for the bulk of the counting behavior, but is not sufficient for starting counting after SEP and ending it with EOS when reaching the upper limit. First, line 8 is responsible for starting counting after SEP, outputting the value retrieved from `a1` at SEP (faintly visible in Figure 26g), but this effect is not as strong as line 7, so except at SEP, model would still predict the increment-by-one token. Line 9 forwards the output from line 6. At the end, no larger number is found in the context, triggering EOS.

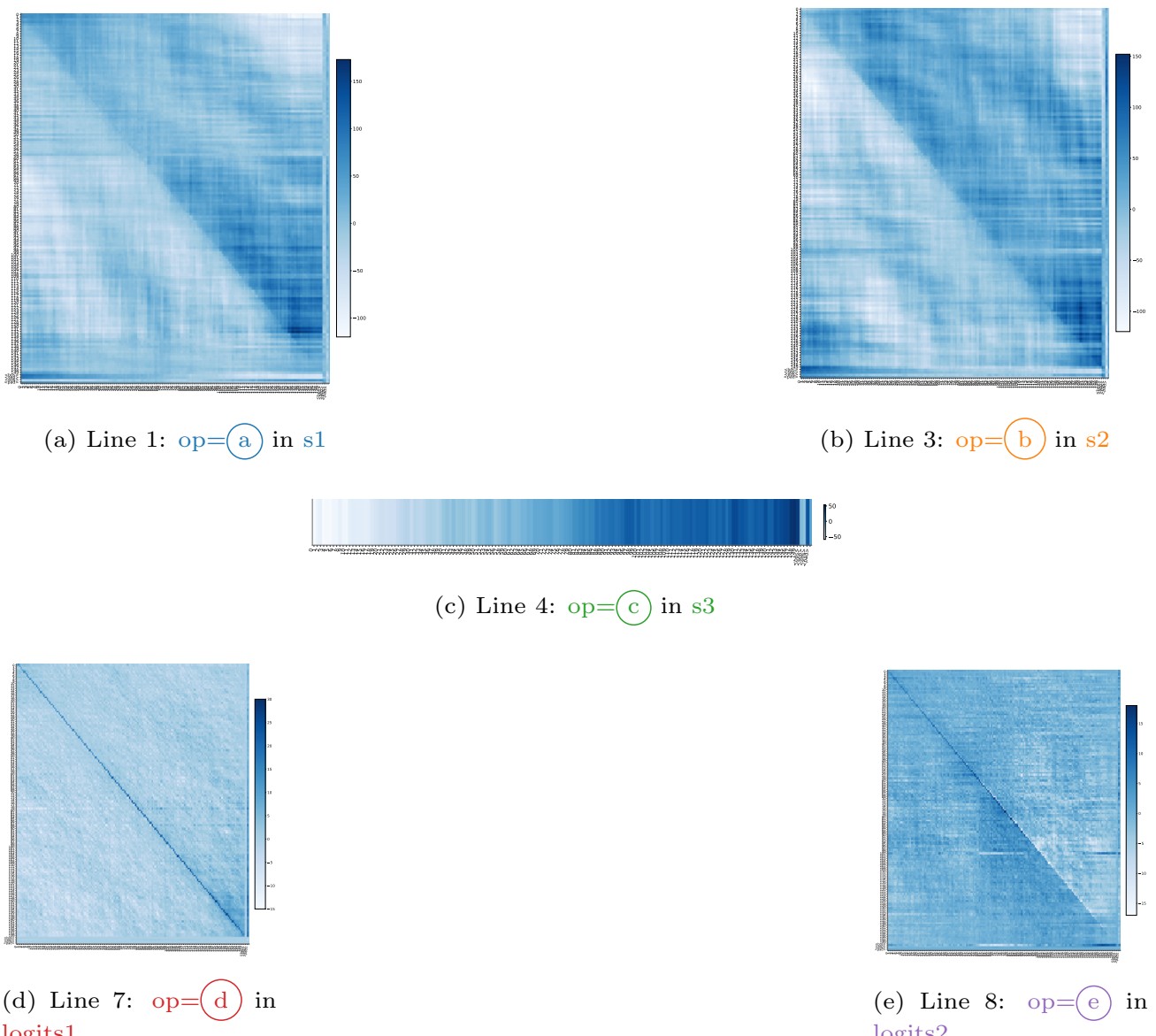

(a) Line 1: op=(a) in s1

(b) Line 3: op=(b) in s2

(c) Line 4: op=(c) in s3

(d) Line 7: op=(d) in logits1

(e) Line 8: op=(e) in logits2

*Figure 25.* Heatmaps supporting the program for Count model.

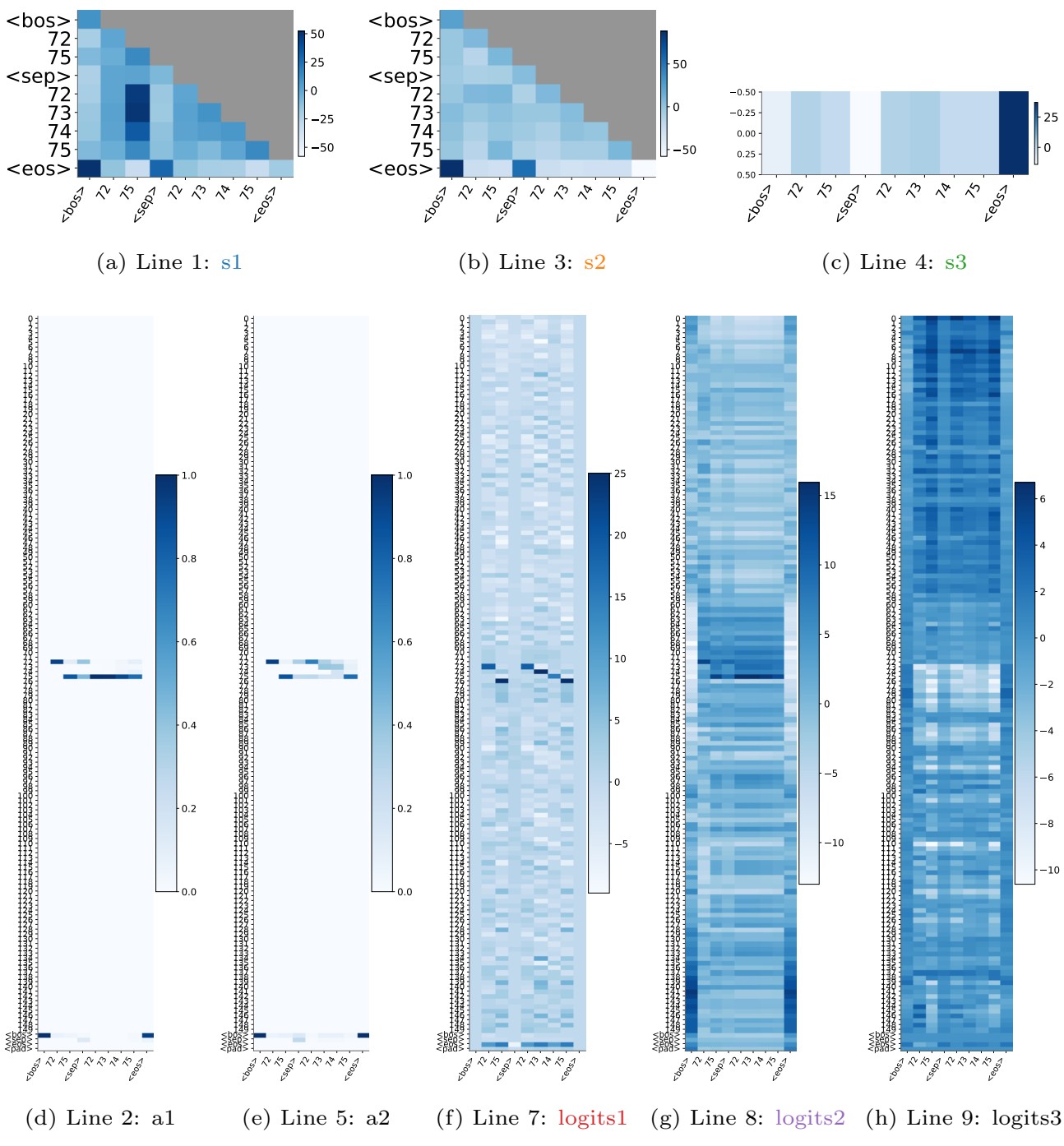

*Figure 26.* Variables Heatmaps for Count model on an example input.

**MLP Input-Output Distributions**    Explaining per-position operation in Line 6 via its effect on Output Logits in Line 9

**Output Token: 1**

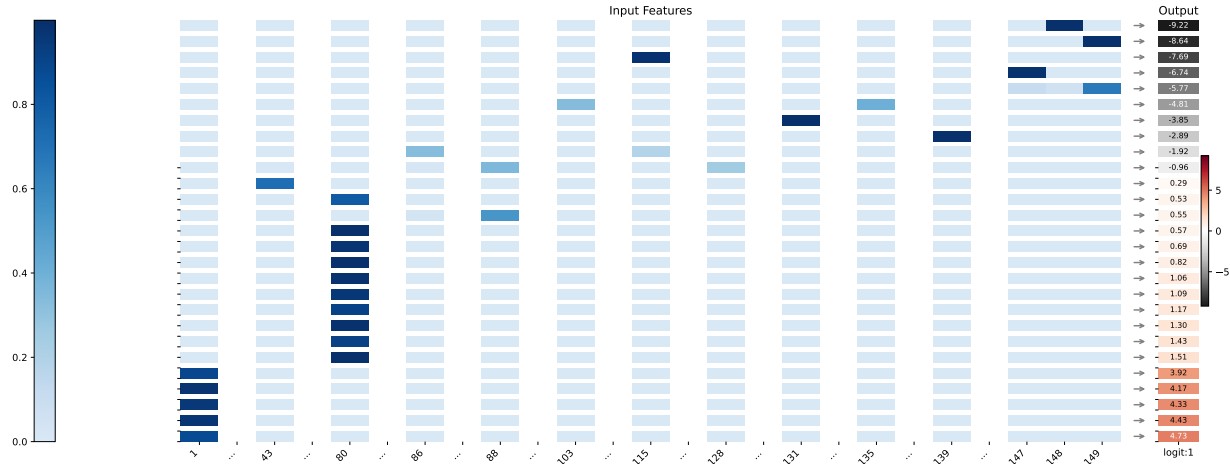

*Figure 27.* MLP Input-Output for token: 1 (sorted by logits)

**Output Token: 10**

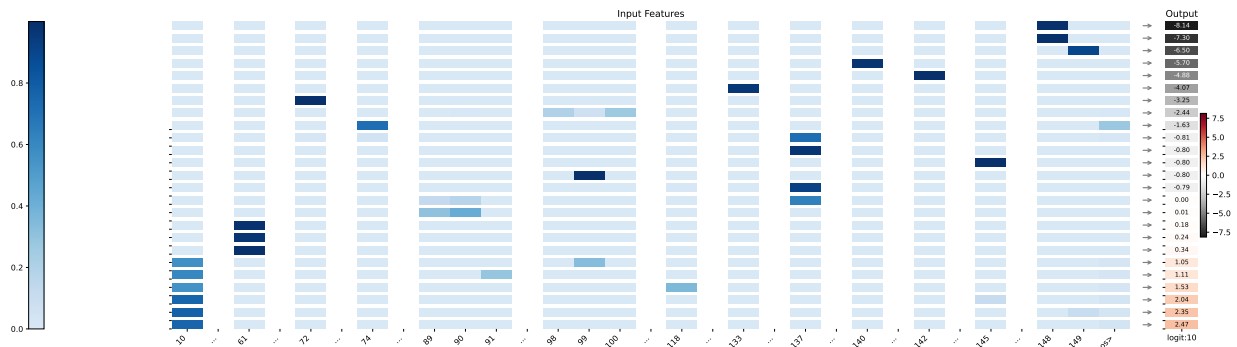

*Figure 28.* MLP Input-Output for token: 10 (sorted by logits)

**Output Token: 100**

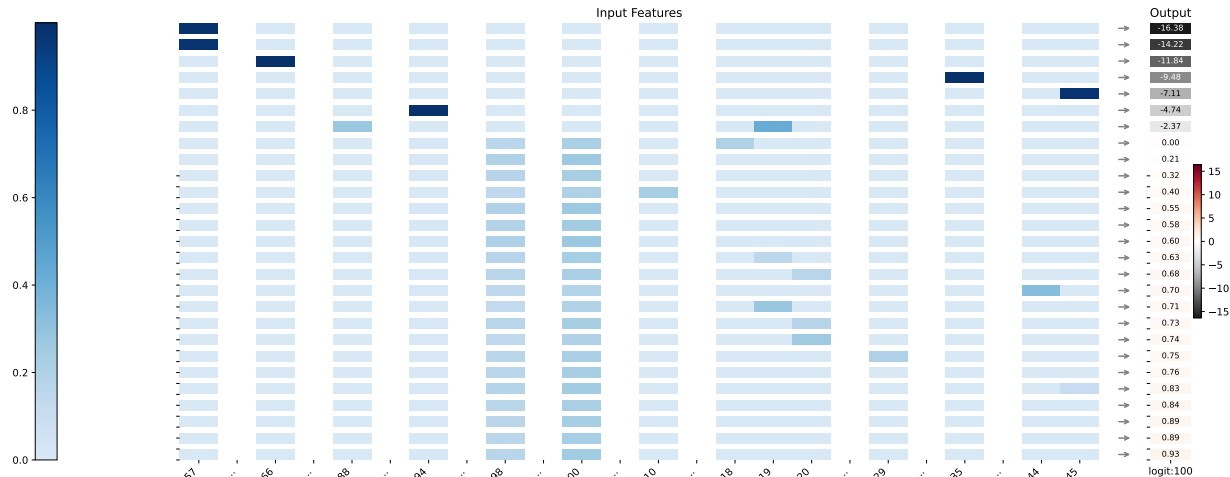

*Figure 29.* MLP Input-Output for token: 100 (sorted by logits)

**Output Token: EOS**

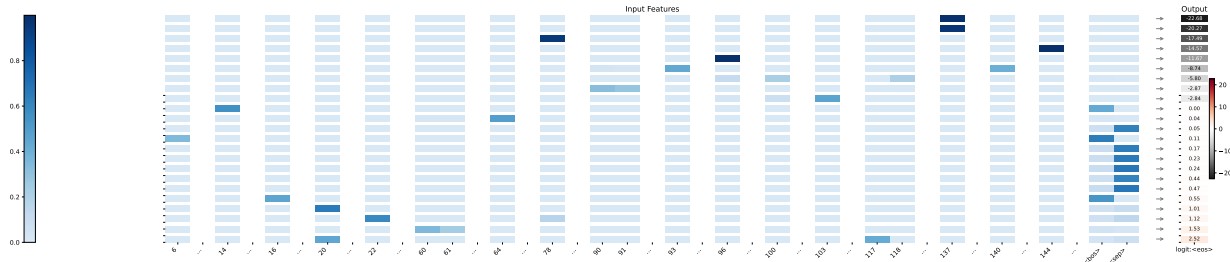

*Figure 30.* MLP Input-Output for token: EOS (sorted by logits)

## J.5. Count : different arch with same length generalization performance

**Task Description:**

$$\langle\text{bos}\rangle\, s_0, s_n\, \langle\text{sep}\rangle\, s_0 s_1 \ldots s_n\, \langle\text{eos}\rangle \text{where } s_0, s_n \in \{0, 1, \ldots, 150\}, s_n > s_0, s_{i+1} = s_i + 1$$

**Performance (w/Pruning $\rightarrow$ w/Primitives):** Task Accuracy: $0.94 \rightarrow 0.86$; Match Accuracy: $0.94 \rightarrow 0.86$

**Code**

```
1. s1 = select(q=token, k=token, op=(uniform selection),
                special_op=(k is last))          # layer 0 head 2
2. a1 = aggregate(s=s1, v=token)                 # layer 0 head 2
3. new_a1 = element_wise_op(a1)           # layer 0 mlp
4. s2 = select(q=token, k=token, op=(k==q),
                special_op=(uniform selection))       # layer 1 head 3
5. s3 = select(q=token, k=new_a1, op=(q==k))          # layer 1 head 3
6. a2 = aggregate(s=s2+s3, v=new_a1)          # layer 1 head 3
7. logits1 = project(inp=a2, op=(inp==out))
8. logits2 = project(inp=token, op=(c))
9. logits3 = project(inp=new_a1, op=(inp==out))
10. logits4 = project(inp=token, op=(d))
11. prediction = softmax(logits1+
                         logits2+
                         logits3+
                         logits4)
```

**Interpretation**  This program uses a similar strategy to the other version, but with a few differences. In line 1, SEP attends to the smallest number, which is the starting number. Line 1, 2, 3, and 9 forms the mechanism to predict starting number (e.g., Figure 37 shows the obtained number is predicted). Meanwhile, line 8 and 10 form the mechanism for incrementing the number by 1. This mechanism is activated on normal tokens but is not activated on special tokens (magnitude is nearly 0 on rows correspond to special tokens in $\boldsymbol{A}$). The mechanism for predicting EOS token is a bit more complex. Figure 36 shows that line 3-7 is responsible for regulating when EOS is output, as logits1 specifically holds predictions for EOS.

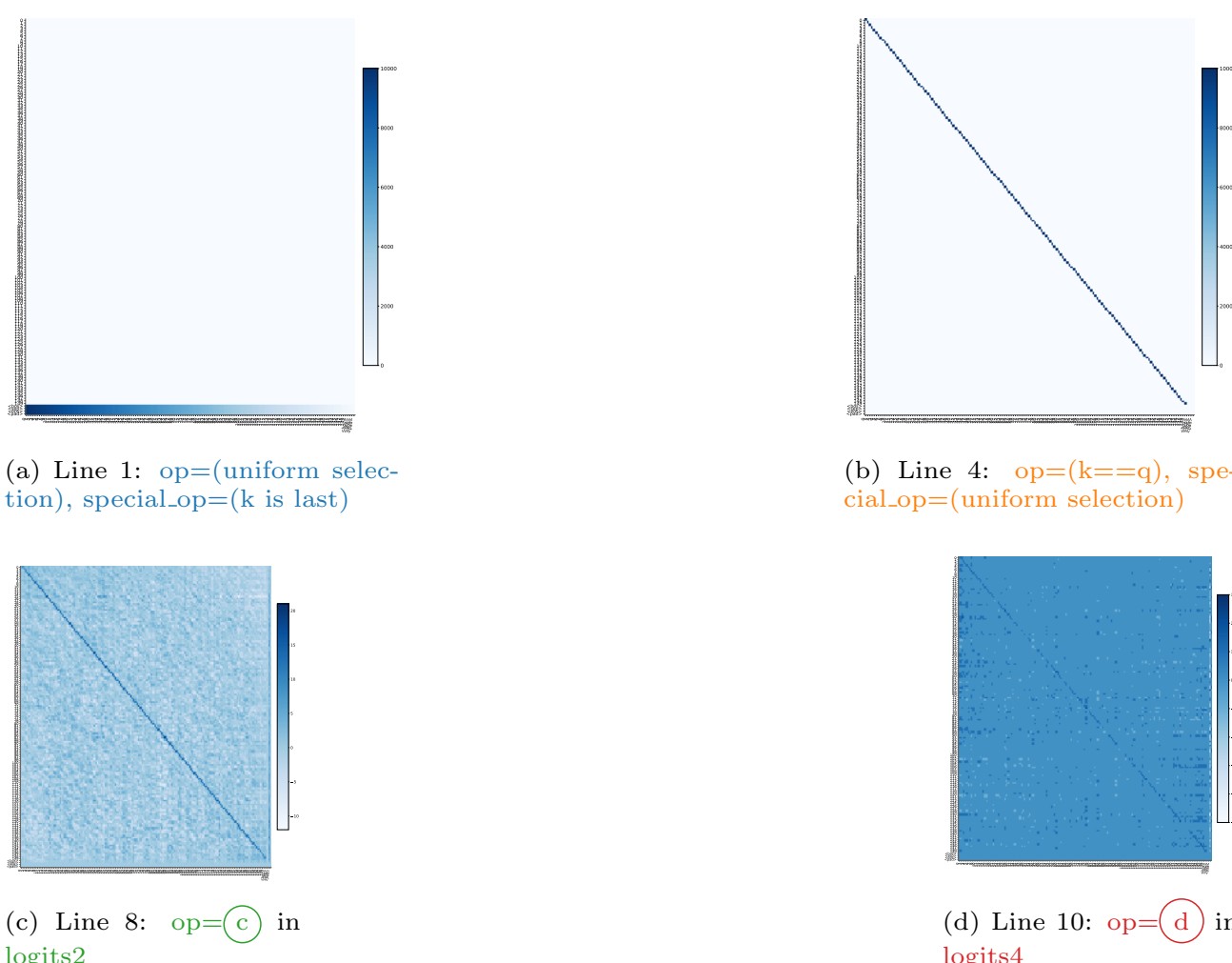

(a) Line 1: op=(uniform selection), special_op=(k is last)

(b) Line 4: op=(k==q), special_op=(uniform selection)

(c) Line 8: op=ⓒ in logits2

(d) Line 10: op=ⓓ in logits4

*Figure 31.* Heatmaps supporting the program for Count : different arch with same length generalization performance model.

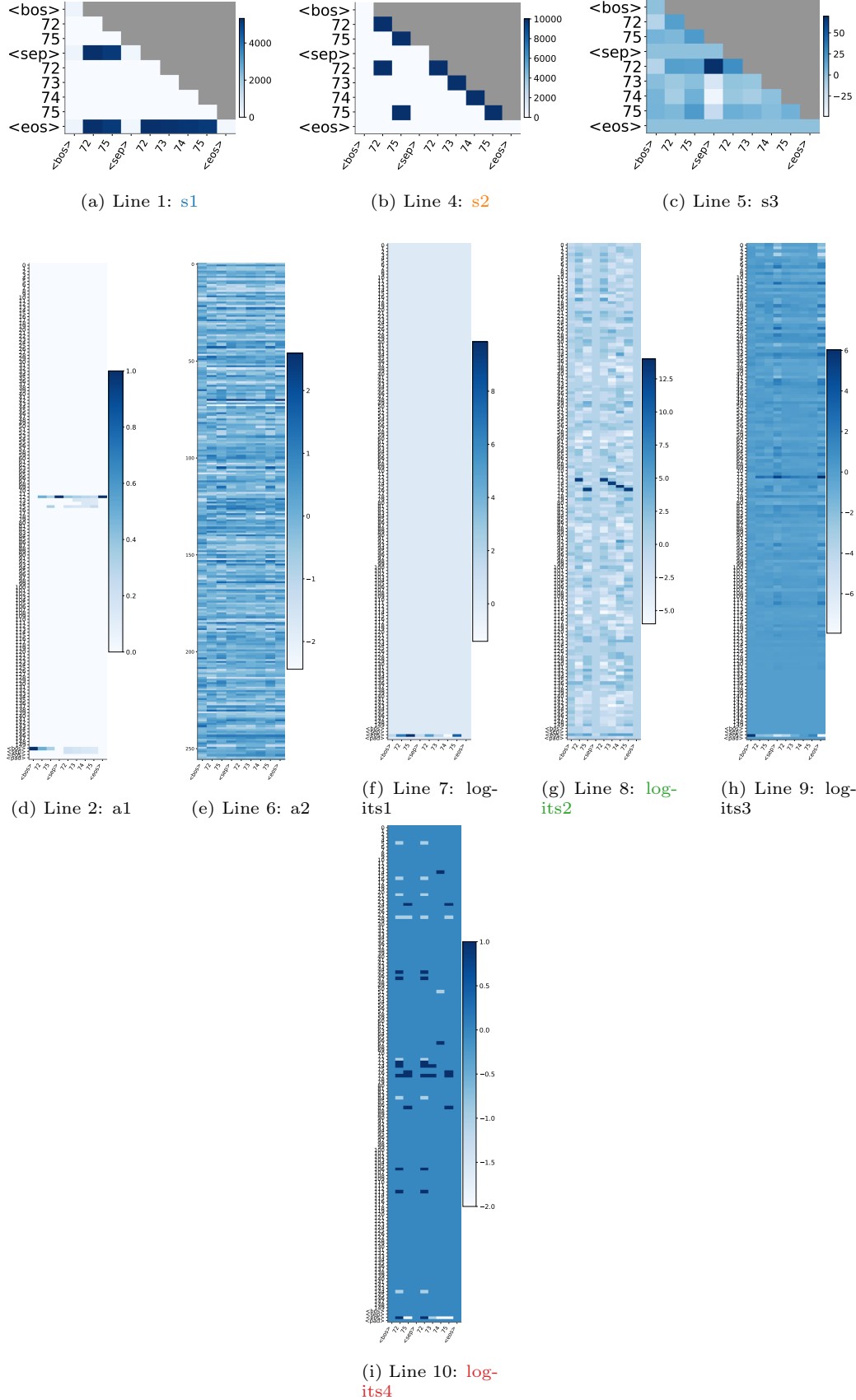

*Figure 32.* Variables Heatmaps for Count : different arch with same length generalization performance model on an example input.

**MLP Input-Output Distributions**   Explaining per-position operation in Line 3 via its effect on Output Logits in Line 7

**Output Token: 1**

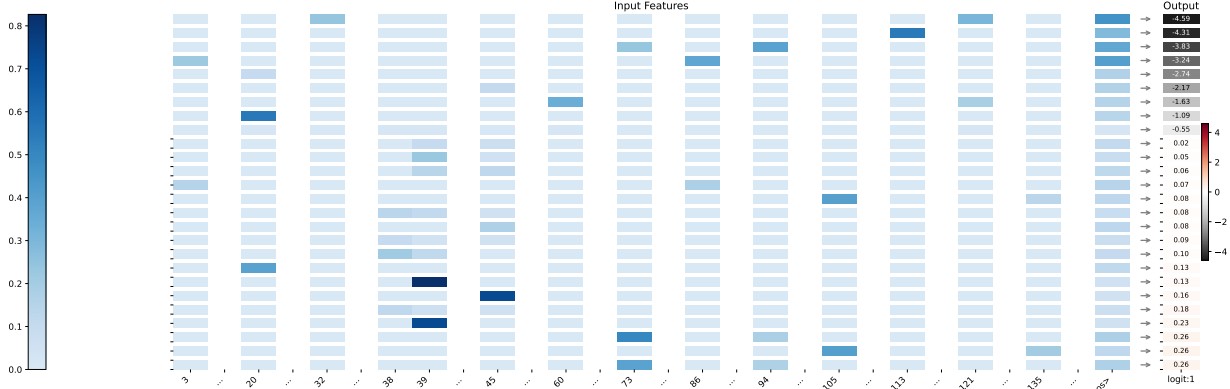

*Figure 33.* MLP Input-Output for token: 1 (sorted by logits)

**Output Token: 10**

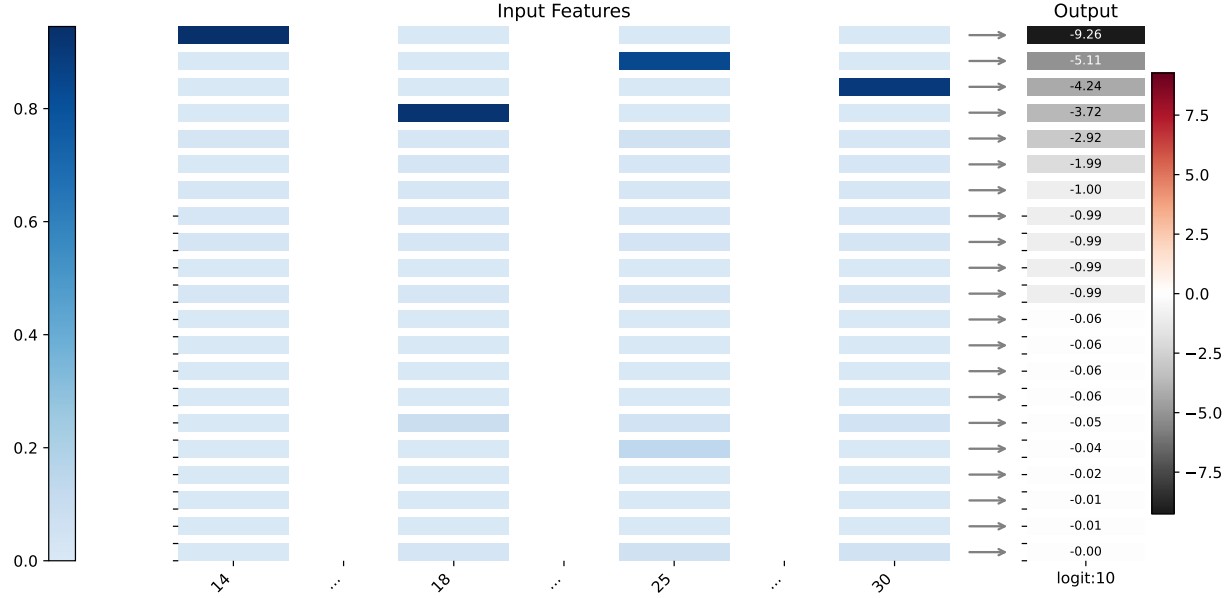

*Figure 34.* MLP Input-Output for token: 10 (sorted by logits)

**Output Token: 100**

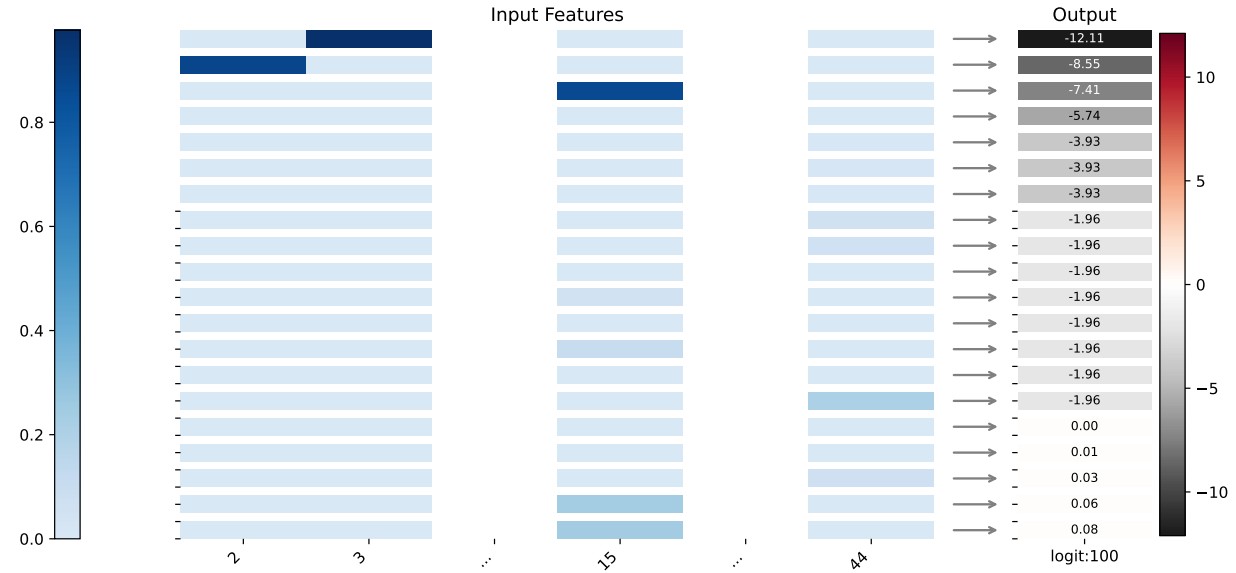

*Figure 35.* MLP Input-Output for token: 100 (sorted by logits)

**Output Token: EOS**

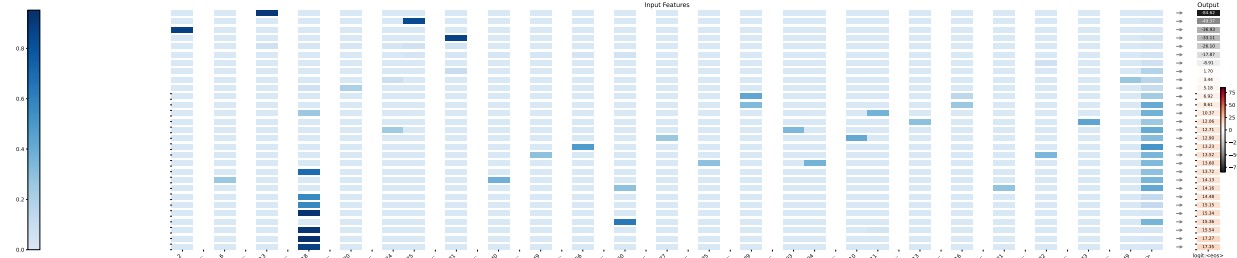

*Figure 36.* MLP Input-Output for token: EOS (sorted by logits)

Explaining per-position operation in Line 3 via its effect on Output Logits in Line 9

**Output Token: 1**

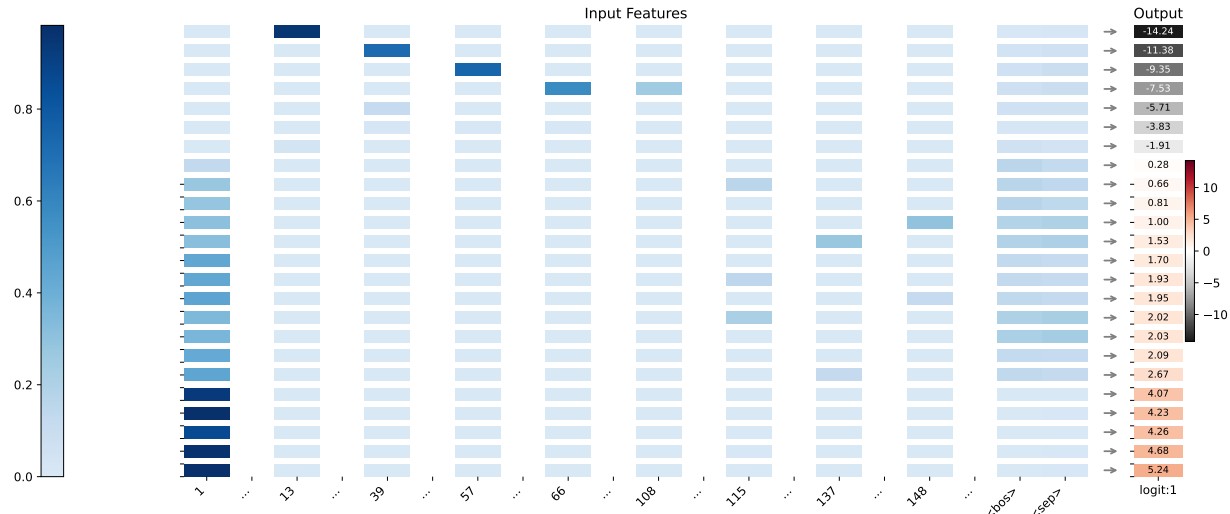

*Figure 37.* MLP Input-Output for token: 1 (sorted by logits)

**Output Token: 10**

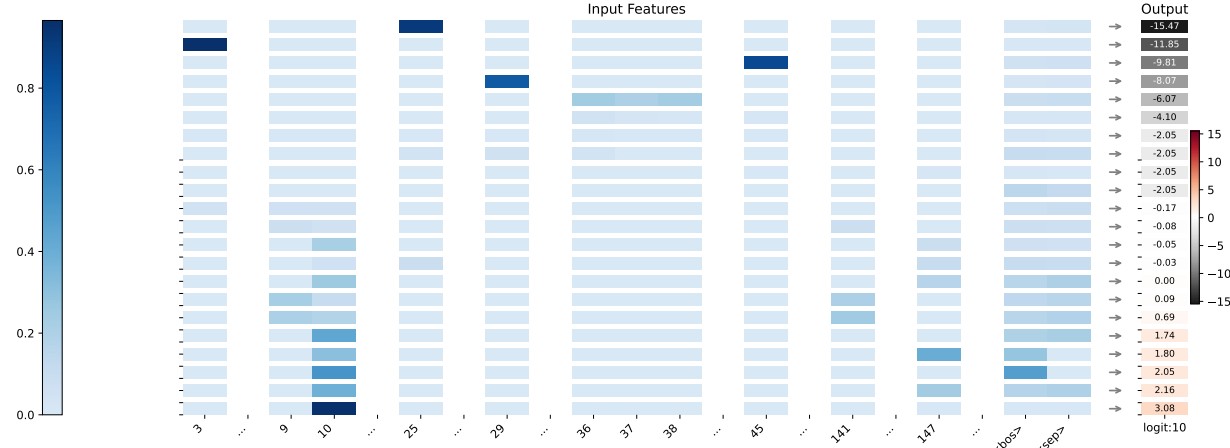

*Figure 38.* MLP Input-Output for token: 10 (sorted by logits)

**Output Token: 100**

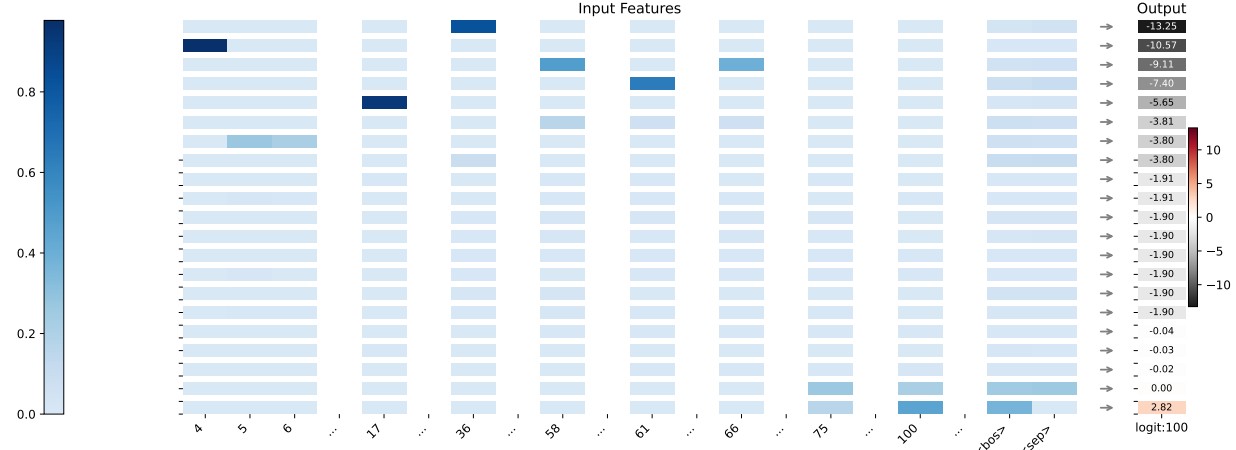

*Figure 39.* MLP Input-Output for token: 100 (sorted by logits)

**Output Token: EOS**

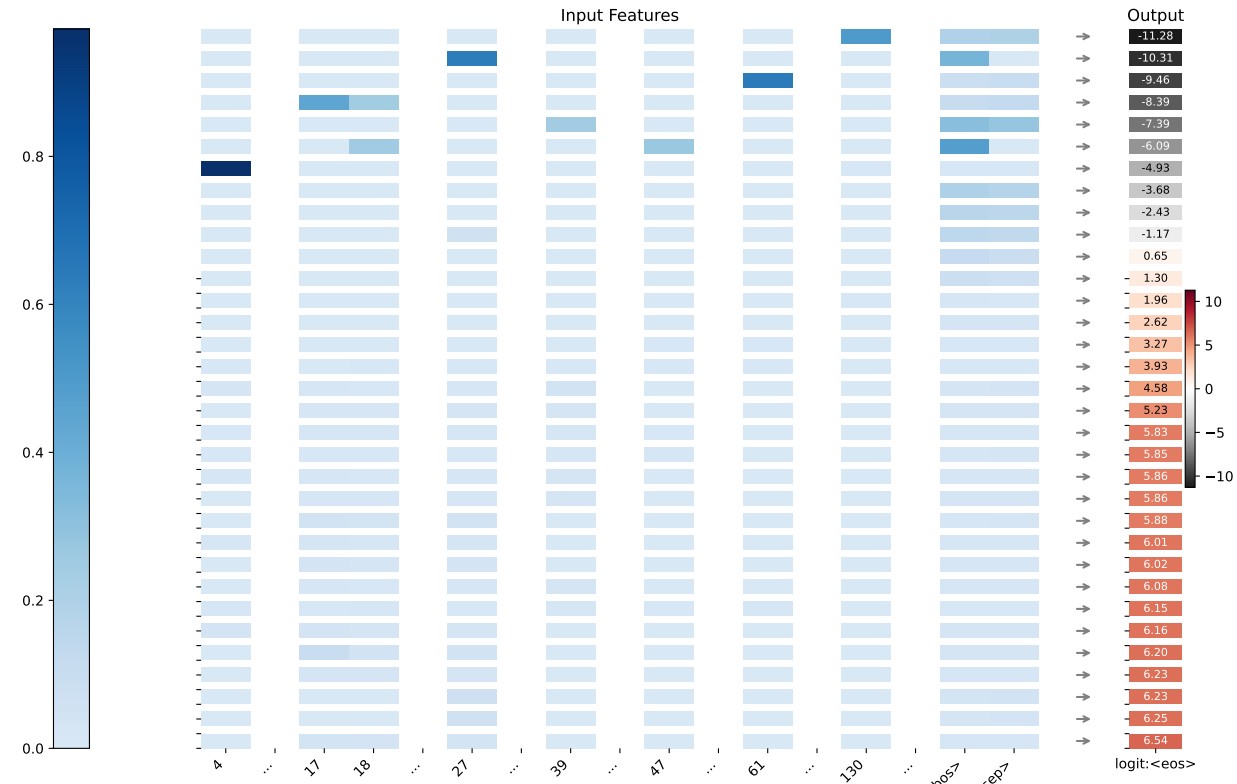

*Figure 40.* MLP Input-Output for token: EOS (sorted by logits)

**Explaining per-position operation in Line 3 via its effect on Output Key in Line 5**   See Figure 41.

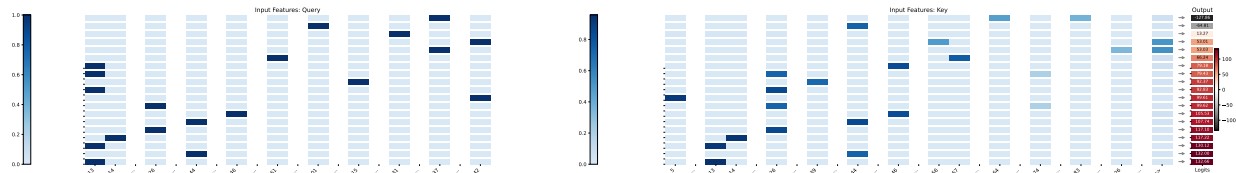

*Figure 41.* MLP Input-Output (sorted by logits)

## J.6. Most Frequent

**Task Description:**
$$\langle \text{bos} \rangle \; s \; \langle \text{sep} \rangle \; \text{Maj}(s) \text{ where } s \in \{a, b, c, \ldots, y, z\}^*$$

**Architecture:** Layers: 1    Heads: 4    Hidden Dim: 256 LR: 0.001    Dropout: 0.1
**Performance (w/Pruning → w/Primitives):** Task Accuracy: $0.99 \to 1.00$; Match Accuracy: $0.98 \to 0.99$

**Code**

```
1. a1 = aggregate(s=[], v=token)          # layer 0 head 1
2. logits1 = project(inp=a1, op=(inp==out),
              special_op=(uniform selection))
3. prediction = softmax(logits1)
```

**Interpretation**    This program is interpreted in Main Paper, Figure 1.

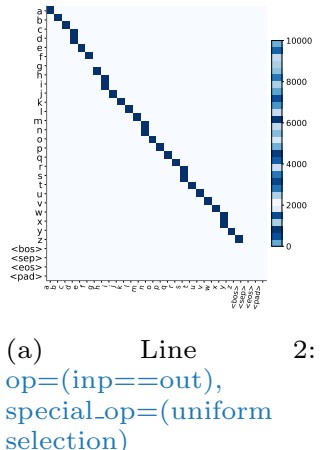

(a)        Line        2:
op=(inp==out),
special_op=(uniform
selection)

*Figure 42.* Heatmaps supporting the program for Most Frequent model. The identity matrix (temperature-scaled to create effectively hard attention) for normal tokens, uniform on special tokens.

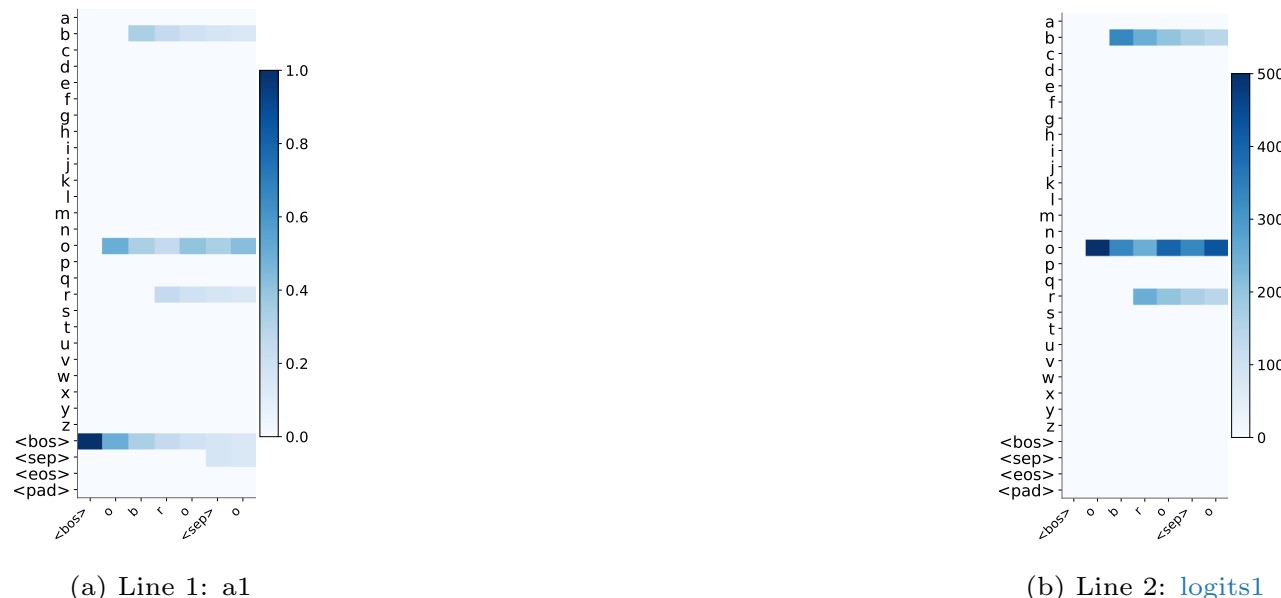

(a) Line 1: a1                     (b) Line 2: logits1

*Figure 43.* Variables Heatmaps for Most Frequent model on an example input. (a) Aggregation computes a histogram of symbols seen so far (x-axis is the input, y-axis are the dimensions of the activation). (b) Output logits are directly obtained from (a) in the case of normal tokens, and erased for special tokens, reflecting the matrix in Figure 42a.

## J.7. Most Frequent : different architecture

**Task Description:**

$$\langle bos \rangle \, s \, \langle sep \rangle \, \text{Maj}(s) \text{ where } s \in \{a, b, c, \ldots, y, z\}^*$$

**Architecture:** Layers: 4    Heads: 4    Hidden Dim: 256 LR: 0.001    Dropout: 0
**Performance (w/Pruning → w/Primitives):** Task Accuracy: $0.98 \to 1.00$; Match Accuracy: $0.98 \to 1.00$

**Code**

```
1. a1 = aggregate(s=[], v=token)          # layer 0 head 0
2. logits1 = project(inp=a1, op=(inp==out),
                special_op=(uniform selection))
3. prediction = softmax(logits1)
```

**Interpretation**    The program is essentially the same as in App. J.6.

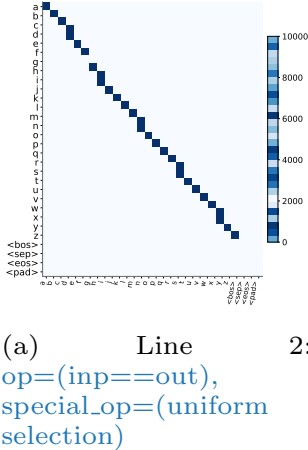

(a) Line 2: op=(inp==out), special_op=(uniform selection)

*Figure 44.* Heatmaps supporting the program for Most Frequent : different architecture model.

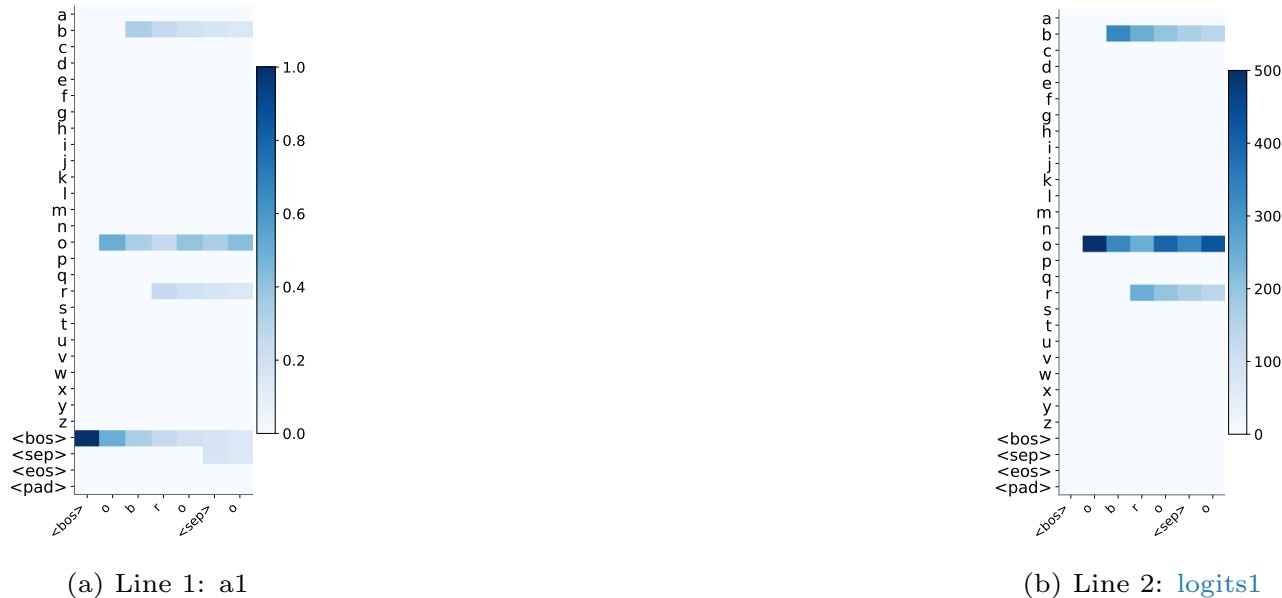

(a) Line 1: a1

(b) Line 2: logits1

*Figure 45.* Variables Heatmaps for Most Frequent : different architecture model on an example input.

## J.8. Sort

**Task Description:**

$$\langle\text{bos}\rangle \, s \, \langle\text{sep}\rangle \, s_{\sigma(0)}, s_{\sigma(0)} \dots s_{\sigma(n)} \, \langle\text{eos}\rangle \text{where } s_0 \in \{0, 1, \dots, 150\}, s_{i+1} = s_i + 1, \sigma \, sorts \, s$$

**Architecture:** Layers: 1    Heads: 1    Hidden Dim: 256 LR: 0.001    Dropout: 0.1
**Performance (w/Pruning → w/Primitives):** Task Accuracy: $0.95 \to 0.90$; Match Accuracy: $0.95 \to 0.90$

**Code**

```
1. s1 = select(q=token, k=token, op=(a))       # layer 0 head 0
2. a1 = aggregate(s=s1, v=token)               # layer 0 head 0
```

```
3. new_a1 = element_wise_op(a1)              # layer 0 mlp
4. logits1 = project(inp=new_a1, op=(inp==out))
5. prediction = softmax(logits1)
```

**Interpretation**   This is discussed in Main paper, Figure 5. Line 1 assigns weight to input numbers that are a little larger than the current token. Line 2 then creates a histogram of larger numbers, with the biggest weight given to the smallest one. The operation in line 3 essentially performs a hardening operation and produces the output logits.

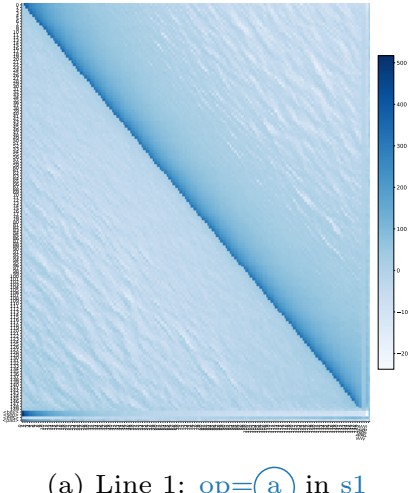

(a) Line 1: op=( a ) in s1

*Figure 46.* Heatmaps supporting the program for Sort model. Part of this matrix is shown in Figure 5.

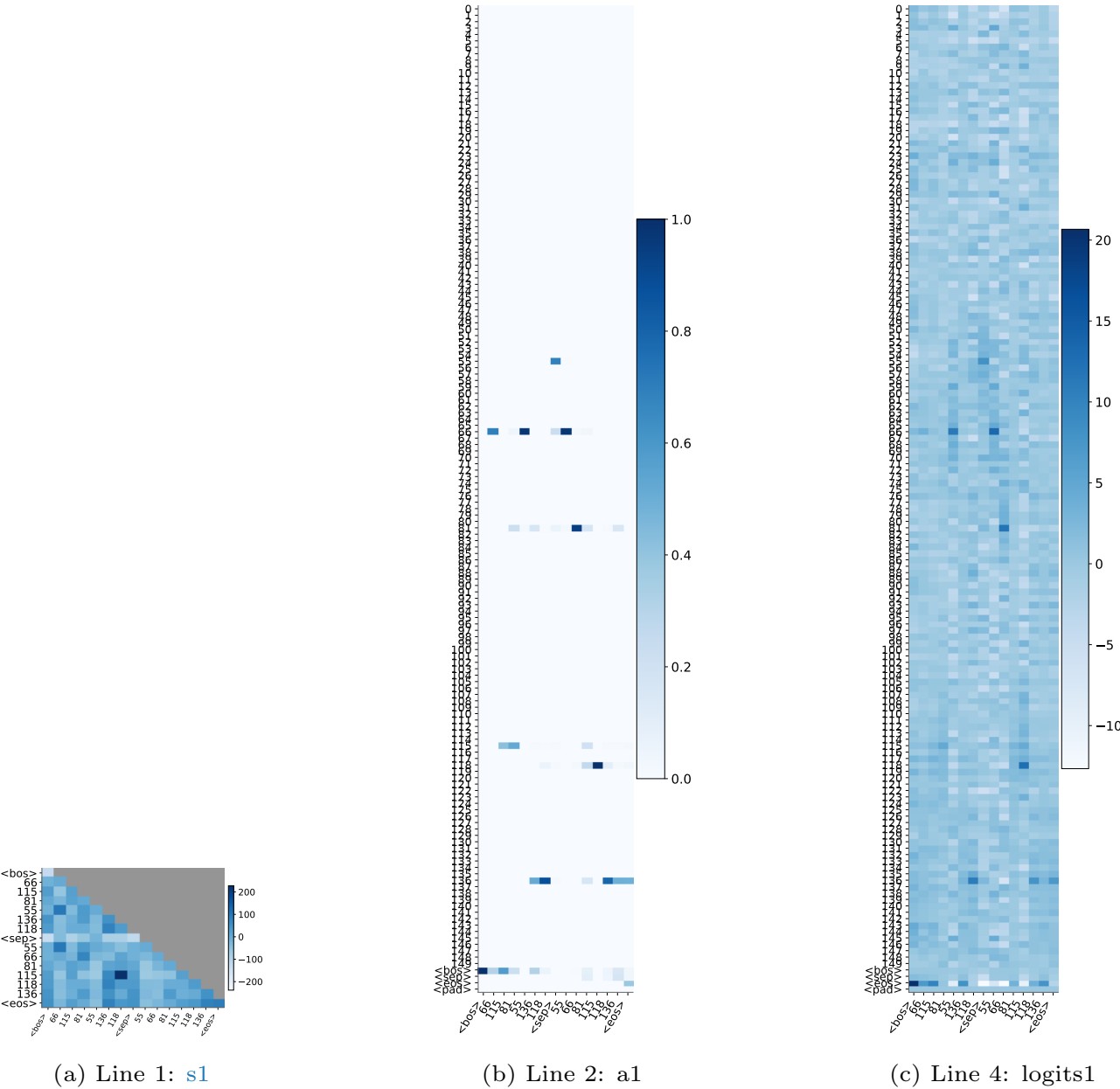

(a) Line 1: s1  (b) Line 2: a1  (c) Line 4: logits1

*Figure 47.* Variables Heatmaps for Sort model on an example input. (a) The selector (rows index query positions, columns index key positions) favors numbers slightly larger than the current one. (b) The resulting weighted histogram. (c) Next-token predictions result from applying the elementwise operation on a1 and projecting via the identity matrix. At each generation step (after SEP), logit is highest on the next symbol; and on EOS at the end.

**MLP Input-Output Distributions**  Explaining per-position operation in Line 3 via its effect on Output Logits in Line 4

**Output Token: 0**

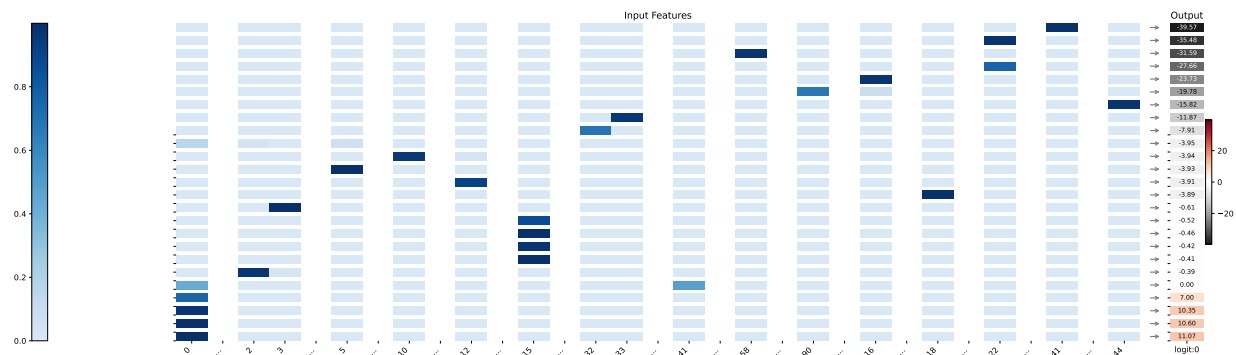

*Figure 48.* MLP Input-Output for token: 0 (sorted by logits). The operation hardens the input by promoting the output dimension for "0" when the input has a high entry in this dimension; similarly for the other dimensions.

## Output Token: 10

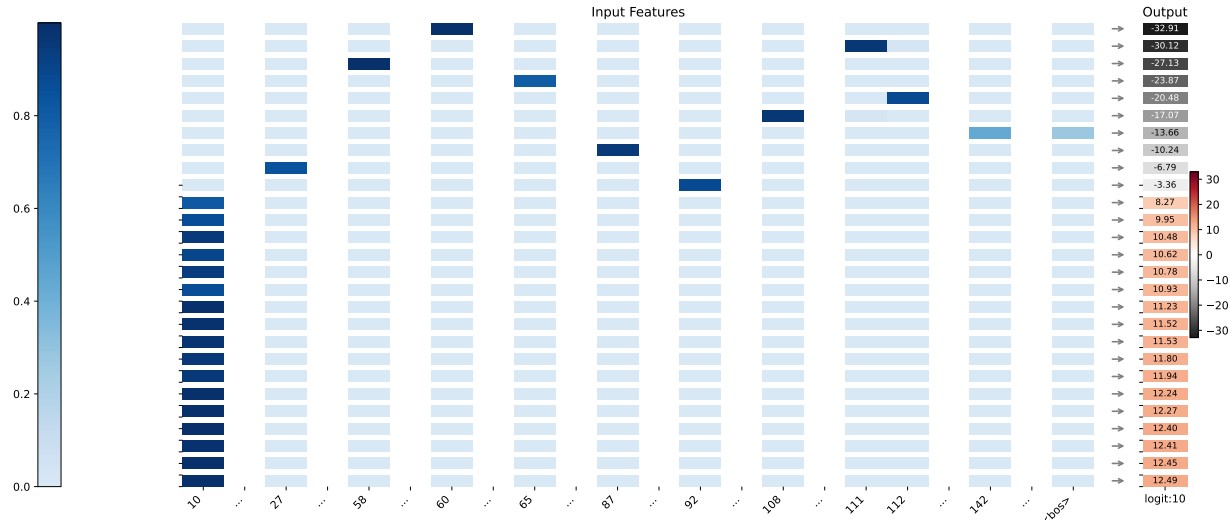

*Figure 49.* MLP Input-Output for token: 10 (sorted by logits)

## Output Token: 100

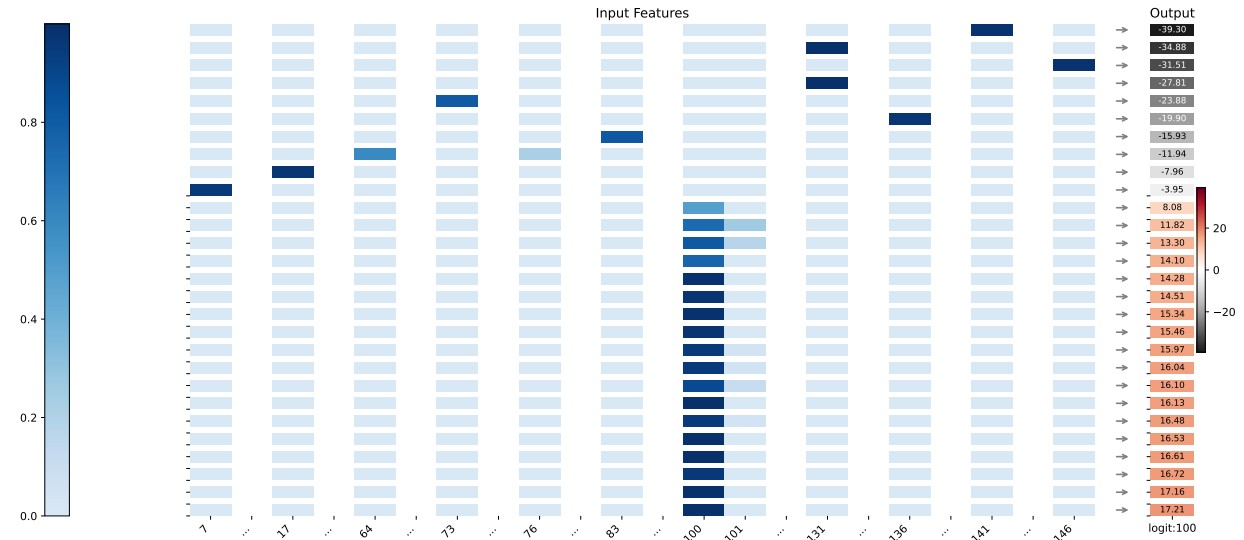

*Figure 50.* MLP Input-Output for token: 100 (sorted by logits)

## Output Token: EOS

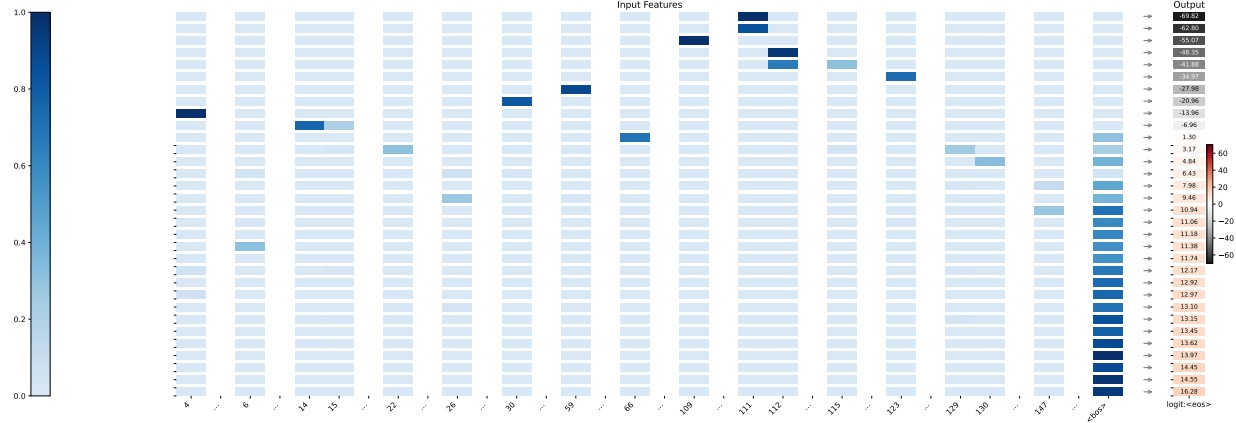

*Figure 51.* MLP Input-Output for token: EOS (sorted by logits). The operation generates a high logit for EOS when the input activation has a high entry for BOS.

## J.9. Unique Bigram Copy

**Task Description:**

$$\langle\text{bos}\rangle \, s_0, s_1, \ldots, s_n \, \langle\text{sep}\rangle \, s_0, s_1, \ldots, s_n \, \langle\text{eos}\rangle \text{where } s_i \in \{0, 1, \ldots, 15\}, (s_i, s_{i+1}) \neq (s_j, s_{j+1}) \text{ iff } i \neq j$$

**Architecture:** Layers: 2    Heads: 4    Hidden Dim: 256 LR: 0.001    Dropout: 0.1
**Performance (w/Pruning → w/Primitives):** Task Accuracy: $0.95 \to 0.92$; Match Accuracy: $0.94 \to 0.91$

## Code

```
1. s1 = select(q=pos, k=pos, op=(a))      # layer 0 head 0
2. a1 = aggregate(s=s1, v=token)          # layer 0 head 0
```

```
3. s2 = select(q=pos, k=pos, op=(k==q-1))          # layer 0 head 2
4. a2 = aggregate(s=s2, v=token)              # layer 0 head 2
5. token_x_a1_x_a2 = Cartesian_product(token, a1, a2)            # layer 0 mlp
6. s3 = select(q=token_x_a1_x_a2, k=token_x_a1_x_a2, op=(c))         # layer 1 head 2
7. a3 = aggregate(s=s3, v=token_x_a1_x_a2)              # layer 1 head 2
8. logits1 = project(inp=a3, op=(d))
9. prediction = softmax(logits1)
```

**Interpretation** This is an extension of the induction head program discussed in Main Paper Figure 3; it crucially involves a joint nonlinear representation of trigrams. Lines 1–4 retrieve the two preceding symbols; Line 5 creates a joint (nonlinear) representation of the trigram ending with the current symbol. In the original transformer, this had been provided by the MLP in the first layer. Each dimension in variable token_x_a1_x_a2 corresponds to a combination of token $t_0$-$t_{-2}$-$t_{-1}$. Line 6 then defines a selector where a query representing a trigram X-...-Y matches a key representing any trigram ...-Y-X (Figure 52c, e.g., (0-0/1/2-2, 0-2-0), (0-0/1/2-1, 0-1-0) have high values.). That is, matching is done based only on two tokens $t_0^q$ $t_{-1}^q$ of the query and $t_{-1}^k$ $t_{-2}^k$ of the key. Because bigrams are unique in the string, there will be a unique (if any) match. The resulting aggregate a3 now holds this key trigram. The current token of the key trigam $t_0^k$ is forwarded to the output logit in line 8.

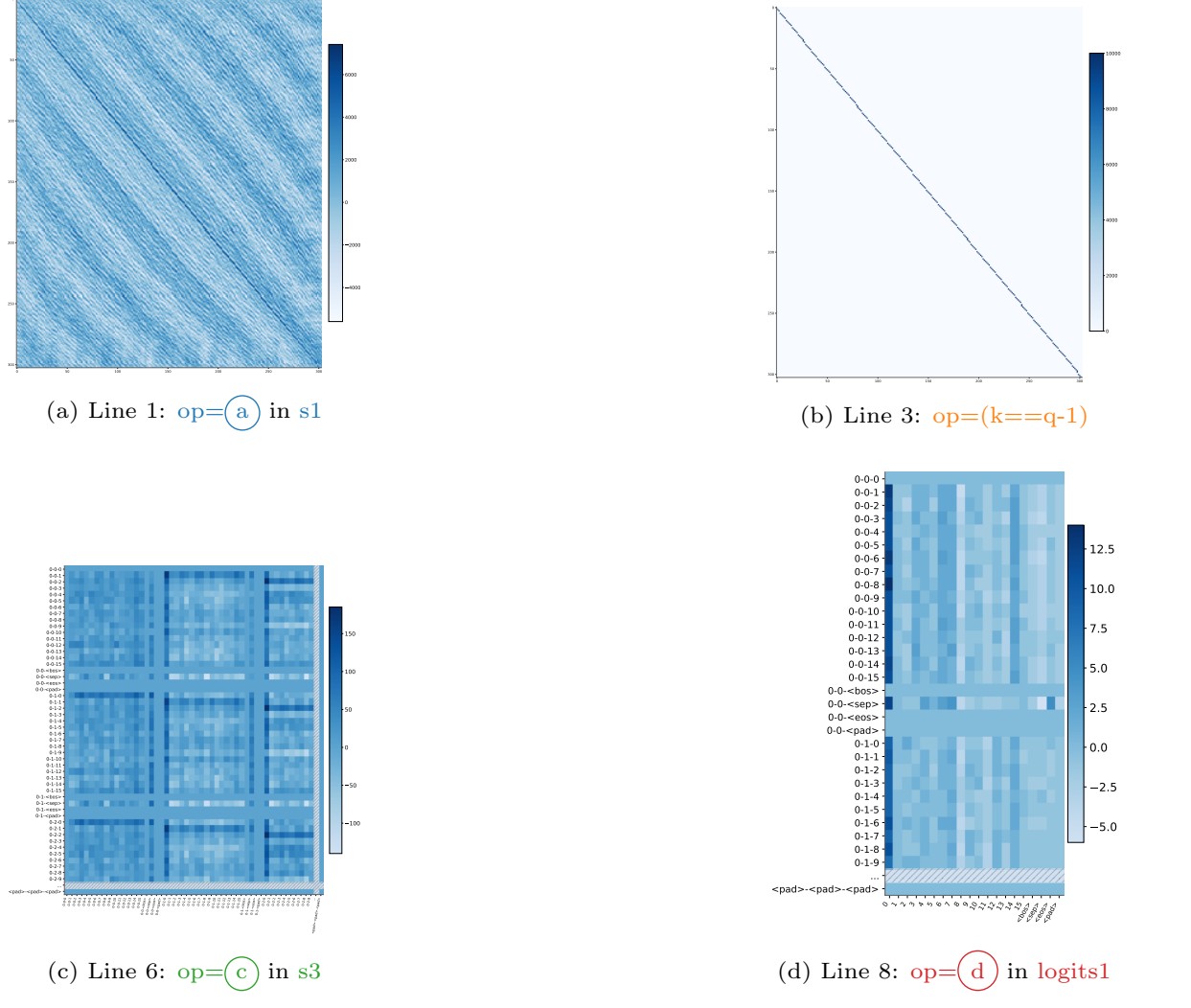

(a) Line 1: op=(a) in s1

(b) Line 3: op=(k==q-1)

(c) Line 6: op=(c) in s3

(d) Line 8: op=(d) in logits1

*Figure 52.* Heatmaps supporting the program for Unique Bigram Copy model. In (c) and (d), the input to the operation is a result of Cartesian product of three variables, all of which have dimension $|\Sigma|$, resulting in a total input size of $|\Sigma| \times |\Sigma| \times |\Sigma|$. We visualize only the top left part of the full heatmap for presentation.

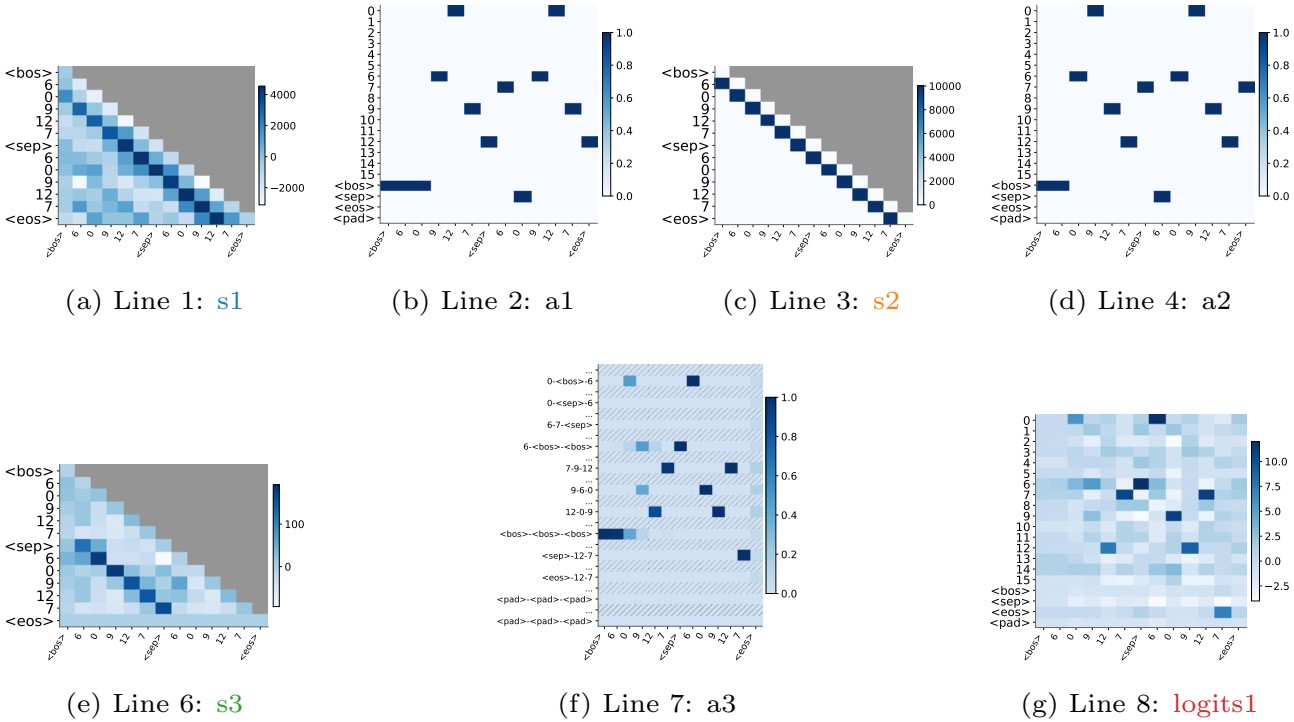

(a) Line 1: s1      (b) Line 2: a1      (c) Line 3: s2      (d) Line 4: a2

(e) Line 6: s3      (f) Line 7: a3      (g) Line 8: logits1

*Figure 53.* Variables Heatmaps for Unique Bigram Copy model on an example input. In (f), similarly to Figure 52, as a result of Cartesian product, the variables have dimension $|\Sigma| \times |\Sigma| \times |\Sigma|$. We hide the rows which have only zero values.

## J.10. Unique Bigram Copy : checkpoint at step 3300

**Task Description:**

$$\langle bos \rangle \, s_0, s_1, \ldots, s_n \, \langle sep \rangle \, s_0, s_1, \ldots, s_n \, \langle eos \rangle \text{where } s_i \in \{0, 1, \ldots, 15\}, (s_i, s_{i+1}) \neq (s_j, s_{j+1}) \text{ iff } i \neq j$$

**Architecture:** Layers: 2    Heads: 4    Hidden Dim: 256 LR: 0.001    Dropout: 0.1
**Performance (w/Pruning → w/Primitives):** Task Accuracy: $0.99 \rightarrow 0.99$; Match Accuracy: $0.99 \rightarrow 0.99$

**Code**

```
1. s1 = select(q=pos, k=pos, op=(a))          # layer 0 head 1
2. a1 = aggregate(s=s1, v=token)             # layer 0 head 1
3. s2 = select(q=pos, k=pos, op=(k==q-1))         # layer 0 head 2
4. a2 = aggregate(s=s2, v=token)             # layer 0 head 2
5. s3 = select(q=token, k=a2, op=(k==q),
               special_op=(k==BOS))           # layer 1 head 1
6. s4 = select(q=a2, k=a1, op=(d))          # layer 1 head 1
7. a3 = aggregate(s=s3+s4, v=token)               # layer 1 head 1
8. logits1 = project(inp=a3, op=(e))
9. prediction = softmax(logits1)
```

**Interpretation**    This is a different extension of the induction head program discussed in Main Paper, Figure 3; it relies only on select and aggregate operations. Similar to the previous Unique Bigram program, the model takes the previous previous token $t_{-2}$ (s1 and a1), and the previous token $t_{-1}$ (s2 and a2). But unlike previous preogram, they are not combined together with $t_0$ (token). They are used to form two separate selectors s3 (match $t_0^q$ and $t_{-1}^k$) and s4 (match

$t^q_{-1}$ and $t^k_{-2}$)). These two selectors are added together in line 7 such that the intersection would stand out (i.e., performing bigram matching). Only `token` is aggregated (which simpler than the previous program), so the model obtains the correct continuation of the bigram $t^k_0$, which is `a3`. Line 8 outputs `a3`, with special behavior to correctly predict EOS.

In sum, we see a simpler program performed by this earlier checkpoint, compared to the previous Unique Bigram Copy model. This model has separate matching mechanisms and results are simply added. So we can see that in this case, continuous training after this checkpoint makes the inner mechanism more nonlinear and strengthens the role played by MLP layers. Even though the performance stay the same, the inner mechanism does not stop evolving.

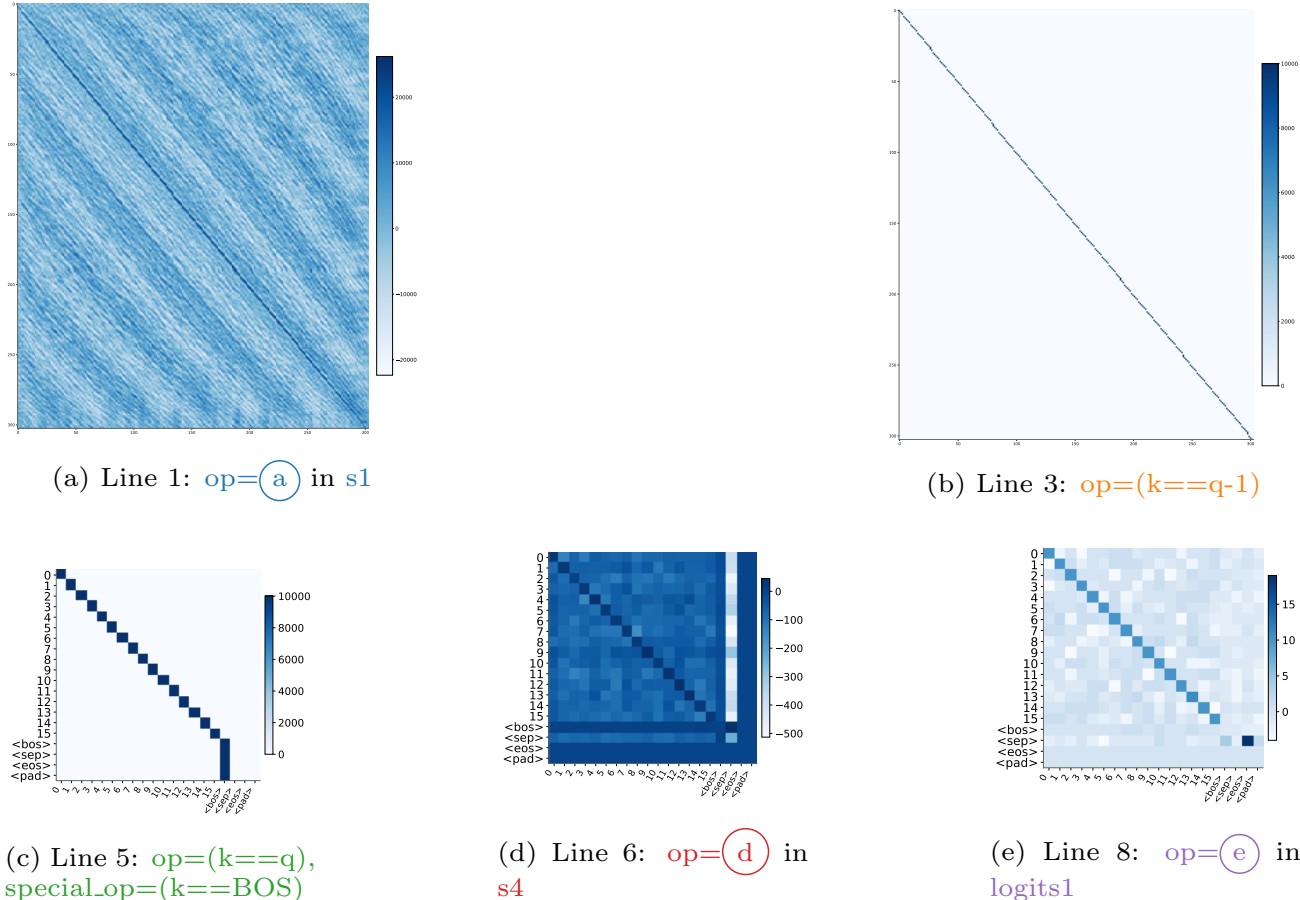

(a) Line 1: op=(a) in s1

(b) Line 3: op=(k==q-1)

(c) Line 5: op=(k==q), special_op=(k==BOS)

(d) Line 6: op=(d) in s4

(e) Line 8: op=(e) in logits1

*Figure 54.* Heatmaps supporting the program for Unique Bigram Copy : different checkpoints model. We note that replacement with a primitive was successful in (b,c), and not in (a,d,e).

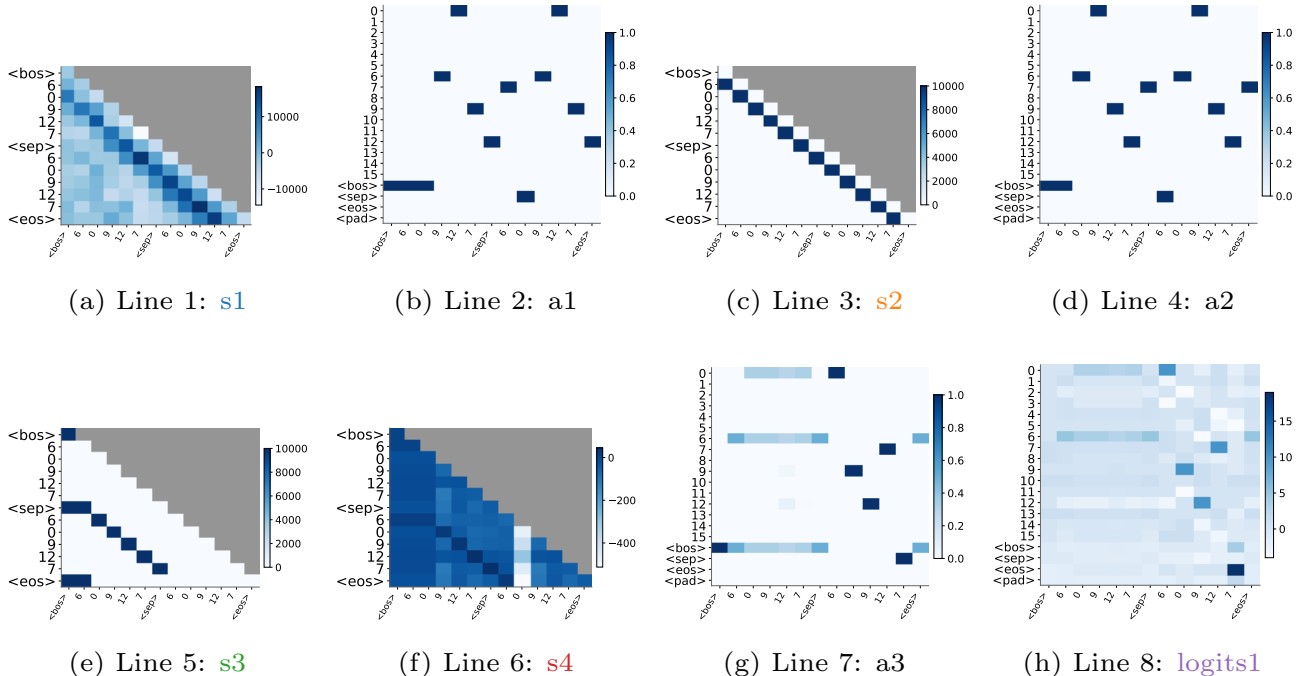

(a) Line 1: s1      (b) Line 2: a1      (c) Line 3: s2      (d) Line 4: a2

(e) Line 5: s3      (f) Line 6: s4      (g) Line 7: a3      (h) Line 8: logits1

*Figure 55.* Variables Heatmaps for Unique Bigram Copy : different checkpoints model on an example input.

## J.11. Unique Copy

**Task Description:**

$$\langle\text{bos}\rangle\, s_0, s_1, \ldots, s_n\, \langle\text{sep}\rangle\, s_0, s_1, \ldots, s_n\, \langle\text{eos}\rangle \text{where } s_i \in \{0, 1, \ldots, 150\}, s_i \neq s_j \text{ iff } i \neq j$$

**Architecture:** Layers: 2    Heads: 1    Hidden Dim: 64 LR: 0.001    Dropout: 0.1
**Performance (w/Pruning → w/Primitives):** Task Accuracy: $0.93 \rightarrow 1.00$; Match Accuracy: $0.92 \rightarrow 0.94$

## Code

```
1. s1 = select(q=pos, k=pos, op=(k==q-1))        # layer 0 head 0
2. a1 = aggregate(s=s1, v=token)          # layer 0 head 0
3. s2 = select(q=token, k=a1, op=(k==q),
                special_op=(k==BOS))       # layer 1 head 0
4. a2 = aggregate(s=s2, v=token)          # layer 1 head 0
5. logits1 = project(inp=a2, op=(inp==out),
                special_op=(uniform selection))
6. logits2 = project(op=(c))
7. prediction = softmax(logits1+
                         logits2)
```

**Interpretation**    This is discussed in Main paper, Figure 3.

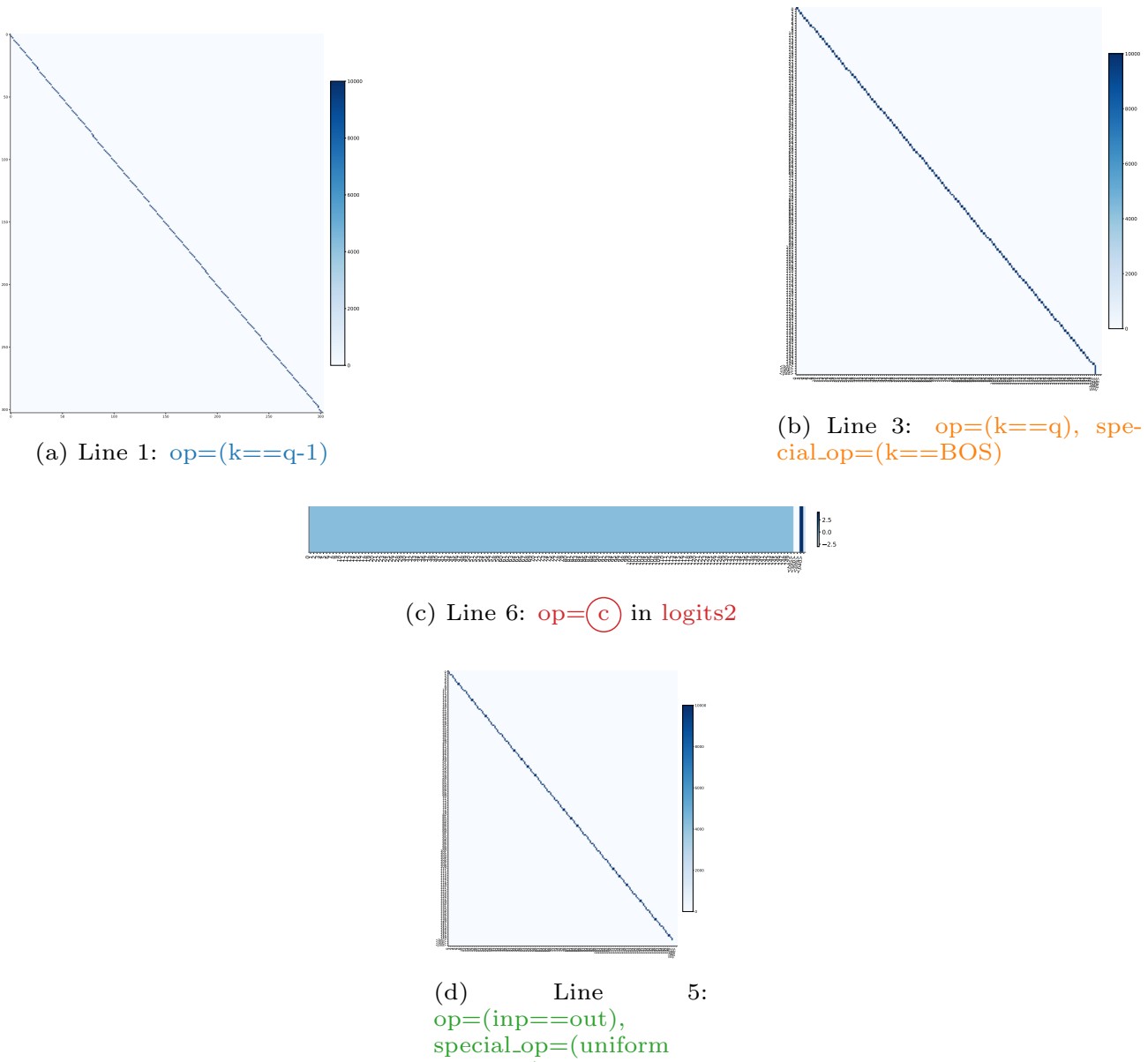

(a) Line 1: op=(k==q-1)

(b) Line 3: op=(k==q), special_op=(k==BOS)

(c) Line 6: op=(c) in logits2

(d) Line 5: op=(inp==out), special_op=(uniform selection)

*Figure 56.* Heatmaps supporting the program for Unique Copy model.

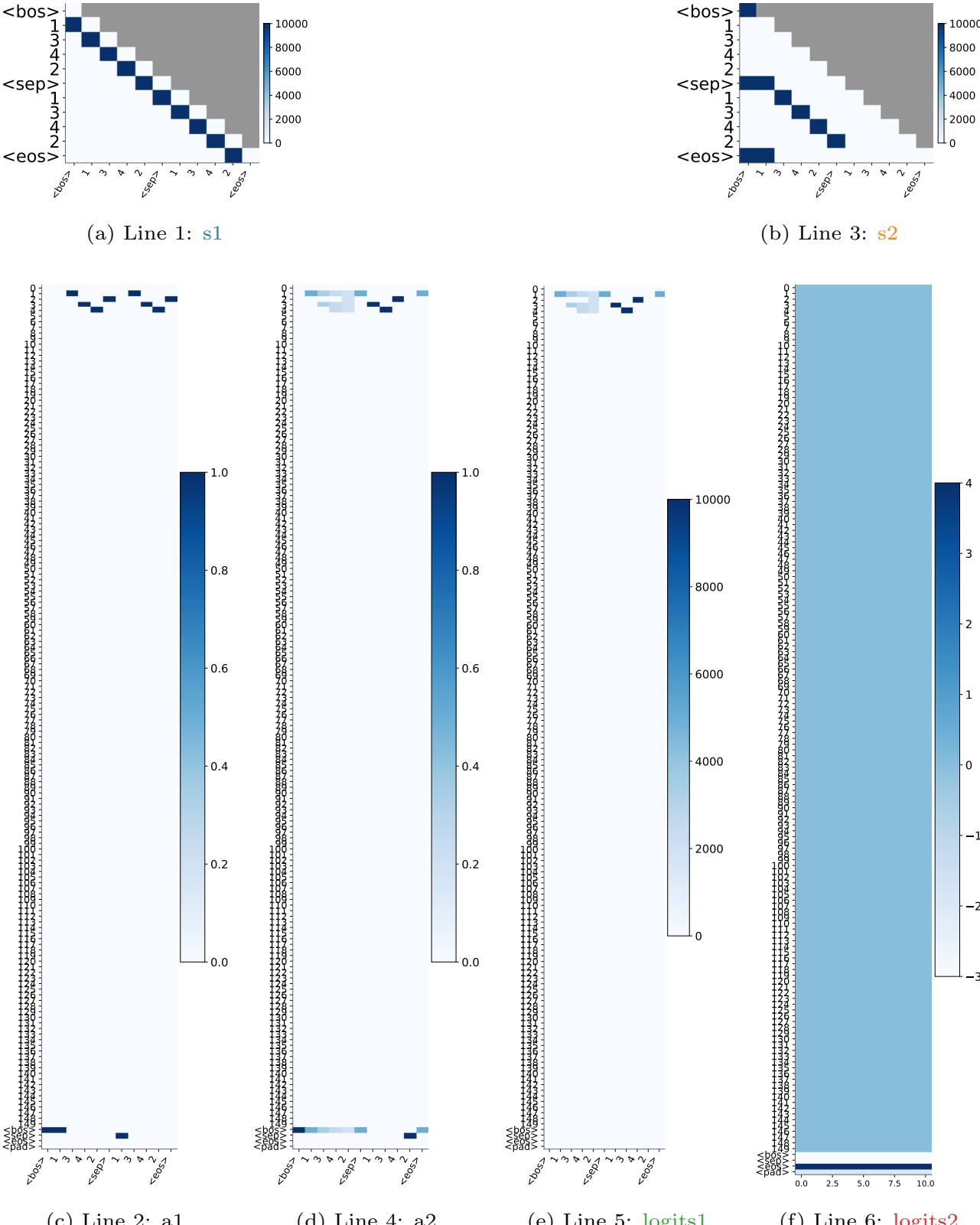

*Figure 57.* Variables Heatmaps for Unique Copy model on an example input.

## J.12. Unique Reverse

**Task Description:**

$$\langle\text{bos}\rangle\, s_0, s_1, \ldots, s_n\, \langle\text{sep}\rangle\, s_n, s_{n-1}, \ldots, s_0\, \langle\text{eos}\rangle \text{where } s_i \in \{0, 1, \ldots, 150\}, s_i \neq s_j \text{ iff } i \neq j$$

**Architecture:** Layers: 2   Heads: 1   Hidden Dim: 64 LR: 0.001   Dropout: 0.1
**Performance (w/Pruning $\rightarrow$ w/Primitives):** Task Accuracy: $0.98 \rightarrow 0.97$; Match Accuracy: $0.96 \rightarrow 0.95$

**Code**

```
1. s1 = select(q=token, k=token, op=(a))          # layer 0 head 0
2. s2 = select(q=pos, k=pos, op=(b))          # layer 0 head 0
3. s3 = select(k=token, op=(d))
4. s4 = select(k=pos, op=(e))
5. a1 = aggregate(s=s1+s2+s3+s4, v=token)          # layer 0 head 0
6. s5 = select(q=token, k=token, op=(k==q),
                special_op=(k==SEP))          # layer 1 head 0
7. a2 = aggregate(s=s5, v=a1)          # layer 1 head 0
8. logits1 = project(inp=a2, op=(f))
9. prediction = softmax(logits1)
```

**Interpretation**   Line 1 defines a selector that assigns increased weight to SEP, with strength varying with the query. (Figure 58a). Line 2 defines a selector assigning increased weight to the immediately preceding position (Figure 58b). Line 3 defines a selector assigning high weight to SEP. Line 4 defines a selector that tends to assign increased weight to later positions. These four selectors jointly come together in line 5, producing a1. Inspecting a1 (Figure 59f) shows that (i) up until SEP, it collects the immediately preceding symbol; (ii) after SEP, it just holds SEP. Line 6 defines a selector assigning weight to occurrences of the current token, or (in the case of a special token), SEP. The resulting variable a2 holds the token that immediately precedes prior (pre-SEP) occurrences of the current token. Here, we understand why a1 just holds SEP in the post-SEP tokens: because the strings before and after SEP have opposite orders, this is needed to find the predecessor of the pre-SEP occurrence of the current token. This is additional complexity introduced by the fact that the model here copies *backwards*. This resulting token is then forwarded to output logits. By and large, the output logits are a "noisy" version of a2 (compare Figures 59 g and h), with one exception: at the last token, 75, a2 has retrieved BOS and SEP, which indicates that generation should stop, i.e., the next token should actually be EOS. The `project` operation converts the BOS/SEP entries into EOS entries. We note that the matrices in Figure 58 are similar to various primitives (e.g., (b) is similar to the identity matrix), but only (e) was successfully replaced. One possible reason is that the program has relatively specific behavior on special tokens, which was not captured by any of the primitives.

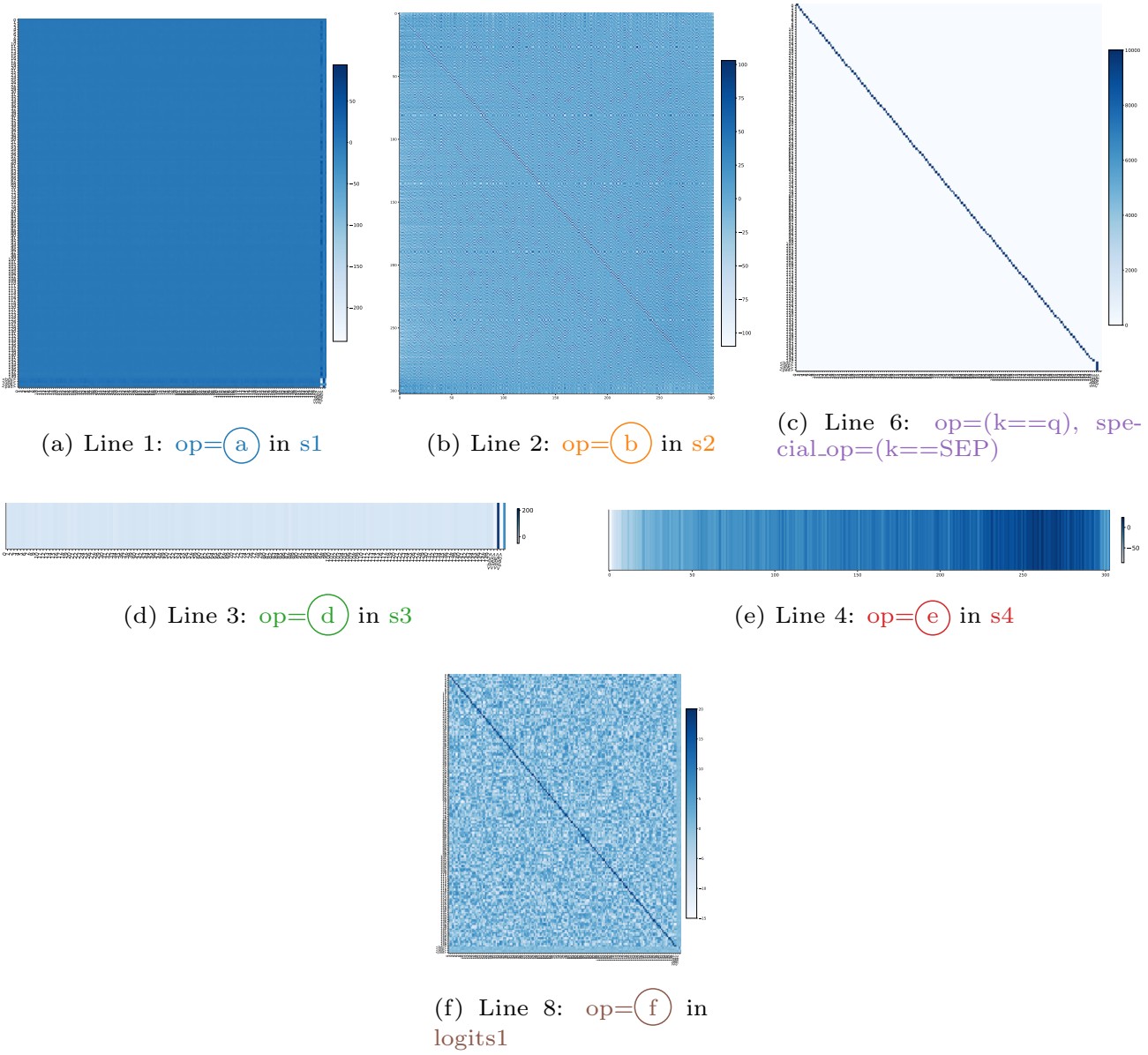

(a) Line 1: op=(a) in s1

(b) Line 2: op=(b) in s2

(c) Line 6: op=(k==q), special_op=(k==SEP)

(d) Line 3: op=(d) in s3

(e) Line 4: op=(e) in s4

(f) Line 8: op=(f) in logits1

*Figure 58.* Heatmaps supporting the program for Unique Reverse model.

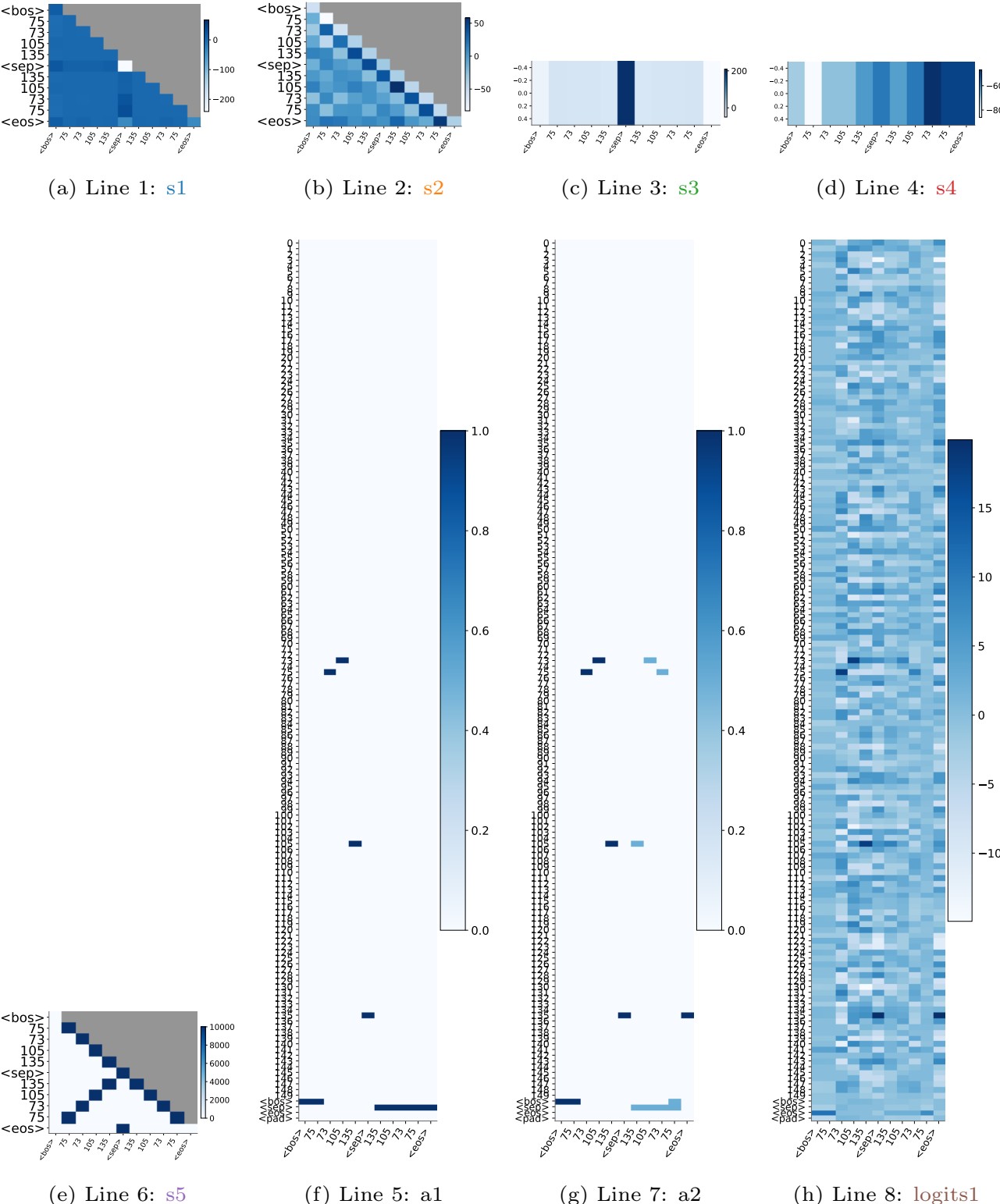

(a) Line 1: s1   (b) Line 2: s2   (c) Line 3: s3   (d) Line 4: s4

(e) Line 6: s5   (f) Line 5: a1   (g) Line 7: a2   (h) Line 8: logits1

*Figure 59.* Variables Heatmaps for Unique Reverse model on an example input.

## J.13. Unique Reverse : different architecture with same length generalization performance

**Task Description:**

$$\langle\text{bos}\rangle\, s_0, s_1, \ldots, s_n\, \langle\text{sep}\rangle\, s_n, s_{n-1}, \ldots, s_0\, \langle\text{eos}\rangle \text{where } s_i \in \{0, 1, \ldots, 150\}, s_i \neq s_j \text{ iff } i \neq j$$

**Architecture:** Layers: 4    Heads: 1    Hidden Dim: 256 LR: 0.0001    Dropout: 0.1
**Performance (w/Pruning → w/Primitives):** Task Accuracy: $0.98 \to 0.99$; Match Accuracy: $0.97 \to 0.98$

**Code**

```
1. s1 = select(q=pos, k=pos, op=(a))          # layer 0 head 0

2. s2 = select(k=token, op=(d))
3. a1 = aggregate(s=s1+s2, v=token)           # layer 0 head 0
4. s3 = select(q=token, k=token, op=(k==q),
               special_op=(uniform selection))       # layer 2 head 0
5. s4 = select(q=token, k=a1, op=(c))         # layer 2 head 0
6. a2 = aggregate(s=s3+s4, v=a1)              # layer 2 head 0
7. new_a2 = element_wise_op(a2)               # layer 3 mlp
8. logits1 = project(inp=new_a2, op=(inp==out))
9. prediction = softmax(logits1)
```

**Interpretation** Comparing the activations in Figure 61 with those of the other program on the same task (Figure 59) shows some commonalities, and indeed the algorithm is similar. s1 defines a selector that looks at the previous token in the input string, and s2 puts increased attention on SEP (Figure 60a,d). Aggregating information based on both these selectors, a1 then for each token holds the identity of the previous token, and for the tokens after SEP it also holds SEP. a2 copies the information from a1 into the residual stream of the same token later in the input, essentially retrieving the output. MLP in Line 7 sharpens a2 (Figures 62, 63, 64), and forces the model to generate EOS instead of BOS (Figure 65).

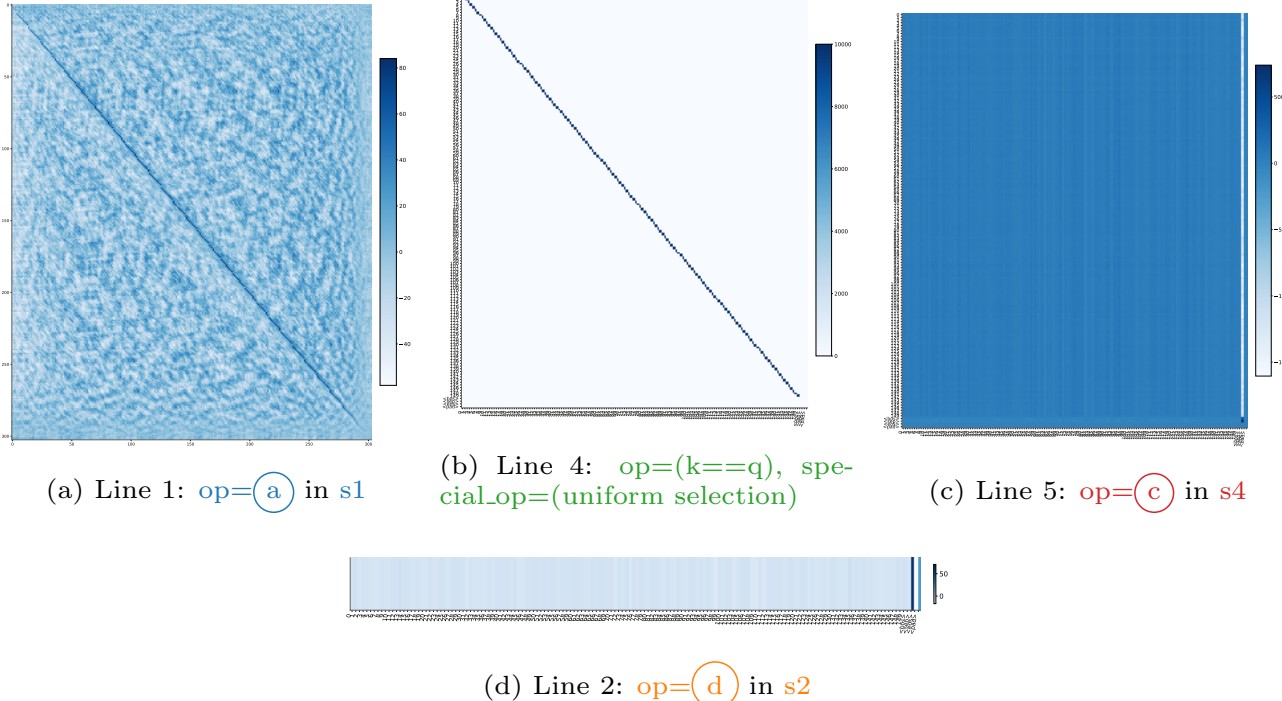

(a) Line 1: op=(a) in s1

(b) Line 4: op=(k==q), special_op=(uniform selection)

(c) Line 5: op=(c) in s4

(d) Line 2: op=(d) in s2

*Figure 60.* Heatmaps supporting the program for Unique Reverse : different architecture with same length generalization performance model.

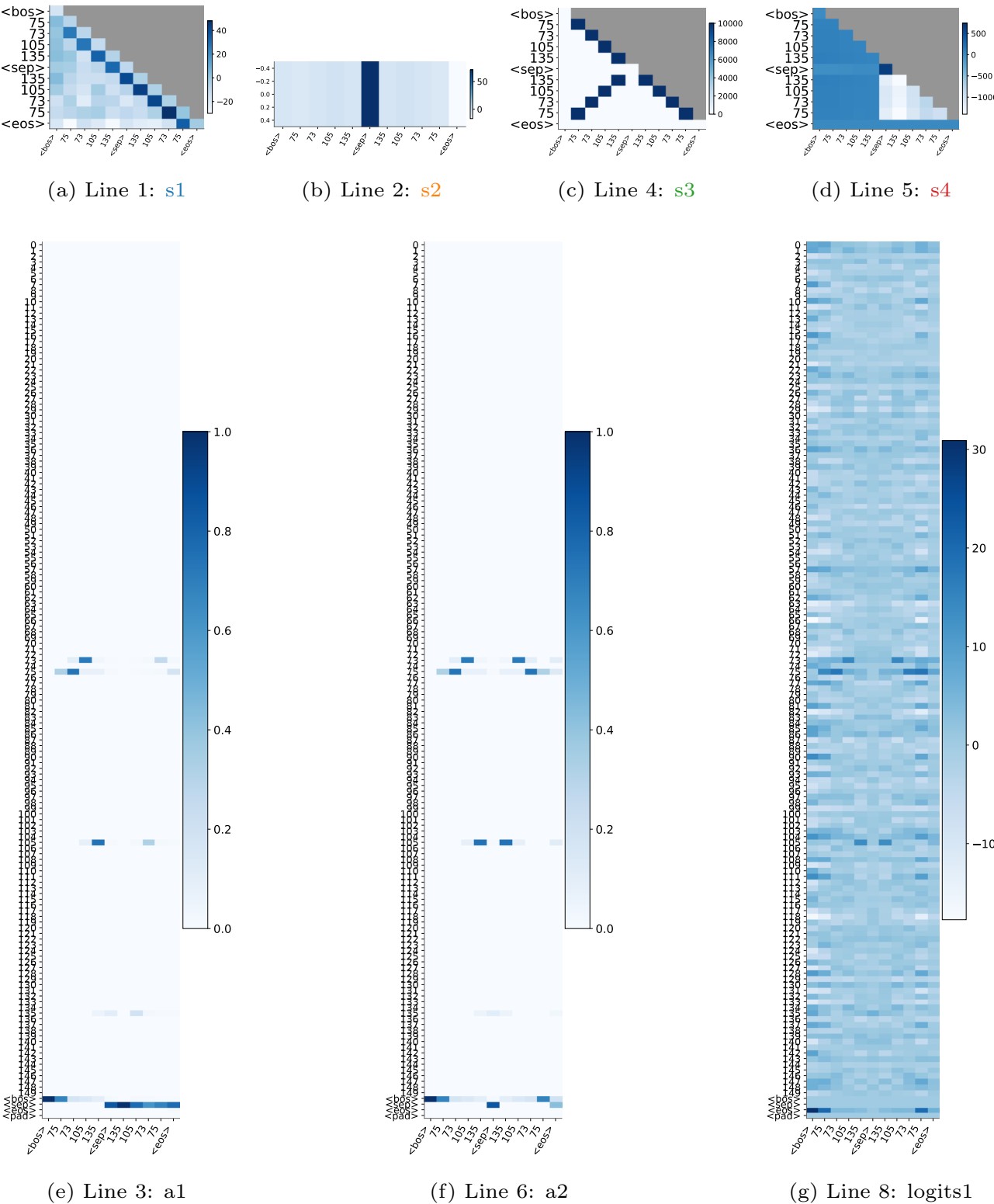

*Figure 61.* Variables Heatmaps for Unique Reverse : different architecture with same length generalization performance model on an example input.

**MLP Input-Output Distributions**    Explaining per-position operation in Line 7 via its effect on Output Logits in Line 8

**Output Token: 0**

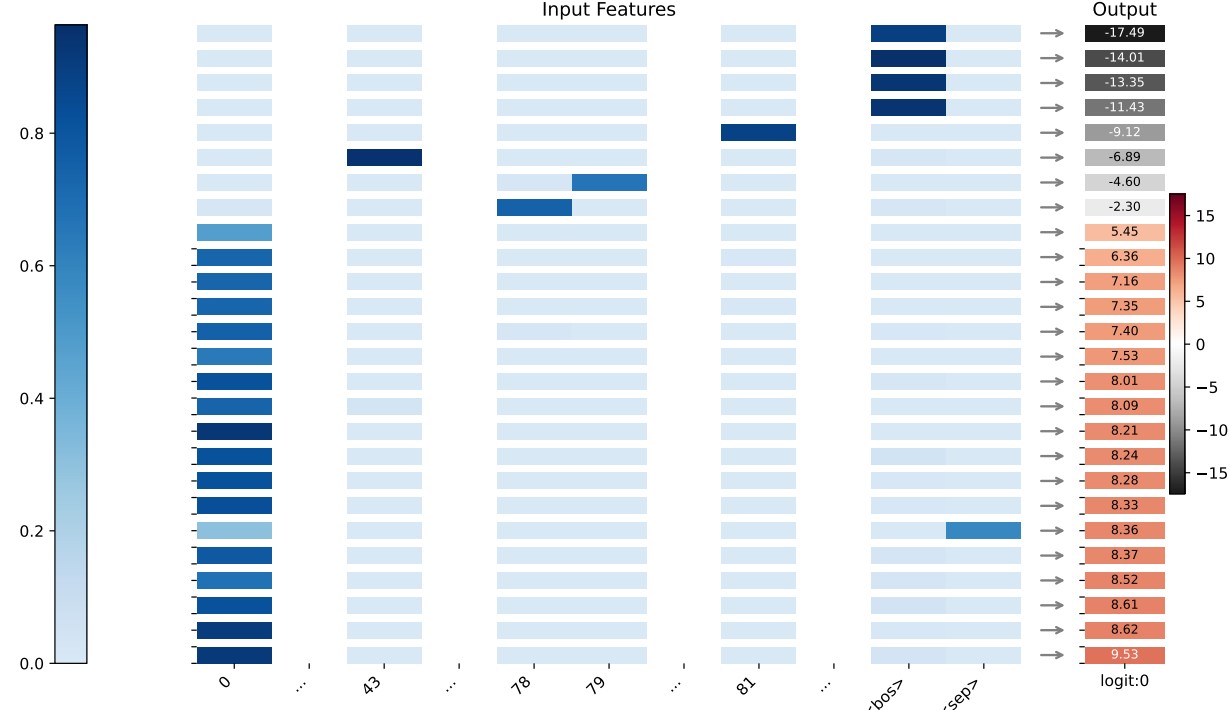

*Figure 62.* MLP Input-Output for token: 0 (sorted by logits)

**Output Token: 10**

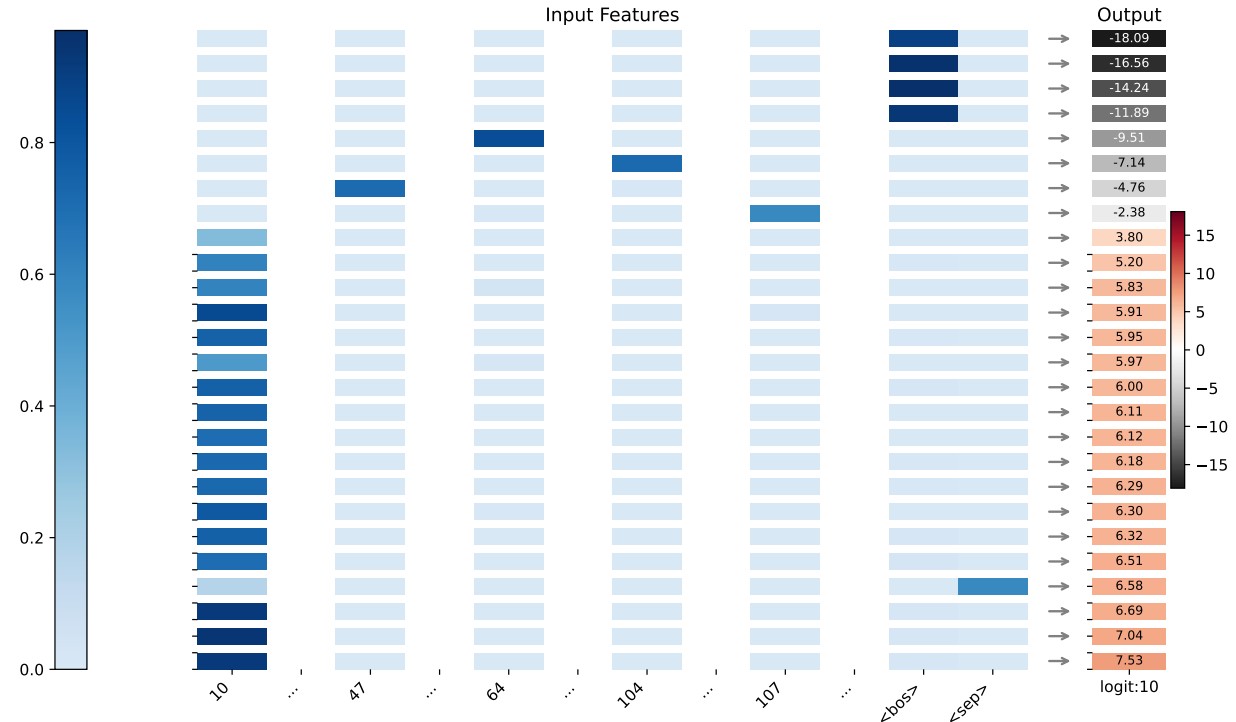

*Figure 63.* MLP Input-Output for token: 10 (sorted by logits)

**Output Token: 100**

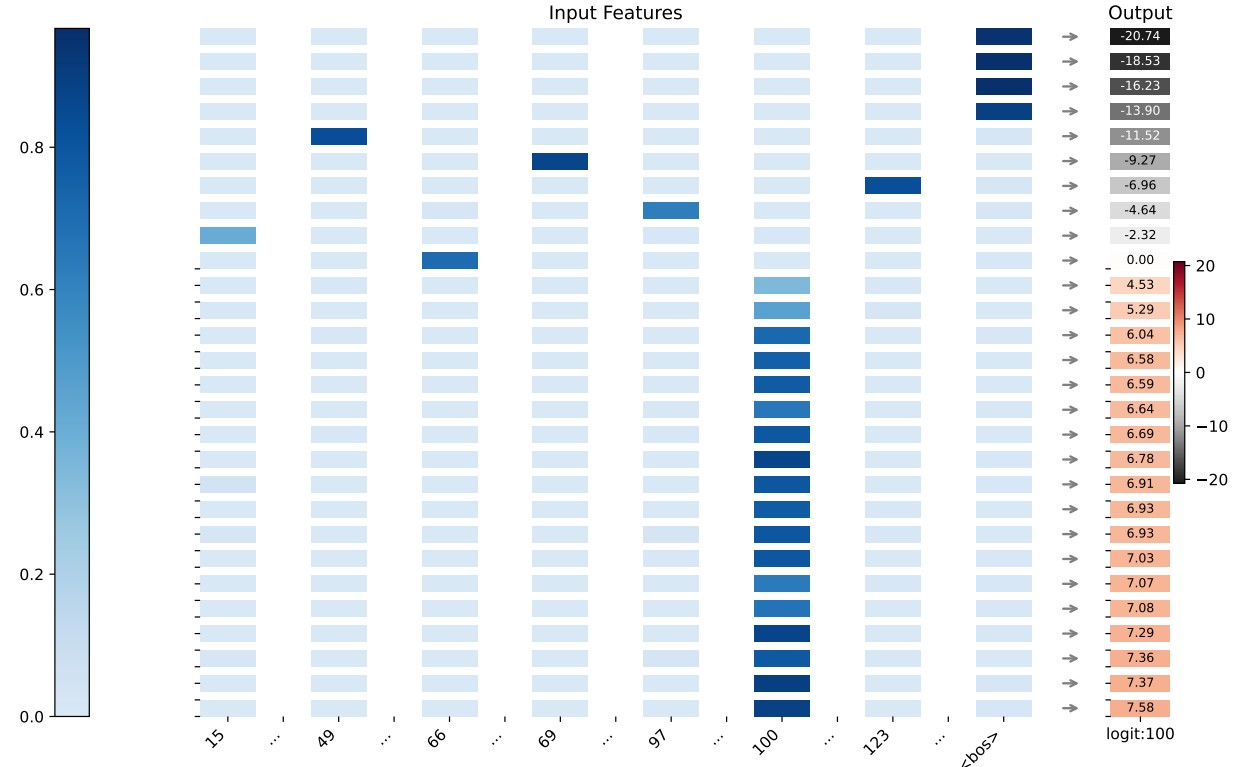

*Figure 64.* MLP Input-Output for token: 100 (sorted by logits)

**Output Token: EOS**

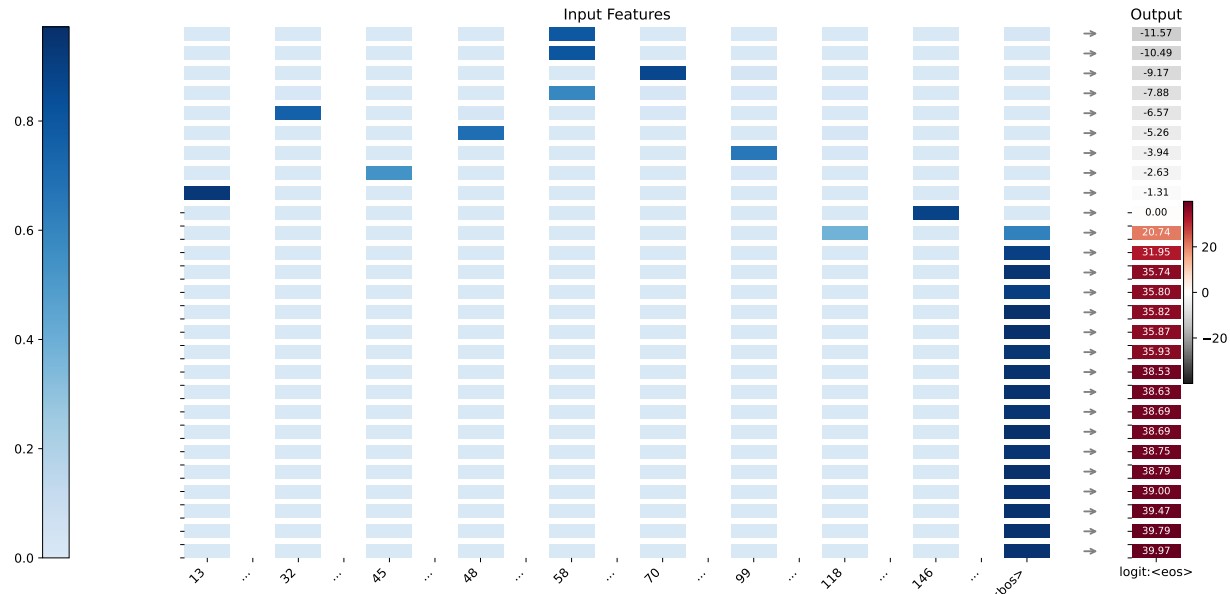

*Figure 65.* MLP Input-Output for token: EOS (sorted by logits)

# K. Decompiled Programs on Formal Languages

## K.1. Dyck-12

**Task Description:**

$$\langle \text{bos} \rangle \underbrace{(a \dots (a(a\,b)^*b)^*}_{12 times} \underbrace{\dots b)^*}_{12 times} \langle \text{eos} \rangle$$

**Architecture:** Layers: 4    Heads: 2    Hidden Dim: 256 LR: 0.001    Dropout: 0
**Performance (w/Pruning $\rightarrow$ w/Primitives):** Task Accuracy: $0.92 \rightarrow 1.00$; Match Accuracy: $0.92 \rightarrow 0.99$

**Code**

```
1. a1 = aggregate(s=[], v=token)          # layer 0 head 0
2. a2 = aggregate(s=[], v=pos)          # layer 0 head 0
3. s1 = select(q=token, k=token, op=(a))        # layer 0 head 1
4. s2 = select(q=pos, k=token, op=(c))         # layer 0 head 1
5. s3 = select(k=token, op=(b))
6. a3 = aggregate(s=s1+s2+s3, v=token)            # layer 0 head 1
7. a4 = aggregate(s=s1+s2+s3, v=pos)          # layer 0 head 1
8. new_a1 = element_wise_op(a1)          # layer 0 mlp
9. new_a3 = element_wise_op(a3)          # layer 0 mlp
10. logits1 = project(inp=new_a1, op=(inp==out))
11. logits2 = project(inp=a2, op=(d))
12. logits3 = project(inp=new_a3, op=(inp==out))
13. logits4 = project(inp=a4, op=(e))
14. prediction = sigmoid(logits1+
                         logits2+
                         logits3+
                         logits4)
```

**Interpretation**   This program is a more complex version of the D4 program discussed in the main paper Figure 8. Similar to Figure 8, `a1` aggregates all the tokens in the input string, and then feeds into the MLP in Line 8, which roughly checks the balance between '$a$' and '$b$' symbols. It promotes the '$b$' only when the string is imbalanced (Figure 69), and EOS only when it is balanced (Figure 70). Some "imperfections" in this MLP are counteracted by the MLP in Line 9. For example, when on the input '$\langle bos \rangle a$' the MLP on Line 8 downweights the logit of '$a$' (Figure 68), while the MLP on Line 9 gives it a much higher logit (Figure 71), mitigating the effect of MLP on Line 8. `s1`, `s2`, `s3` act together to put attention on BOS token, while also aggregating information about '$a$'s and '$b$'s (Figure 66a,b,c). The MLP on Line 9 then encourages the model to output EOS if the value of BOS token after aggregation is low (Figure 73), which happens when the string is long. It additionally checks the balance of '$a$'s and '$b$'s, similarly to the MLP on Line 8. `a4` and `a2` perform aggregation of positional information, which then gets projected in Lines 11 and 13, acting similar to a bias term, prohibiting the model from promoting SEP, PAD or BOS (Figure 66d,e).

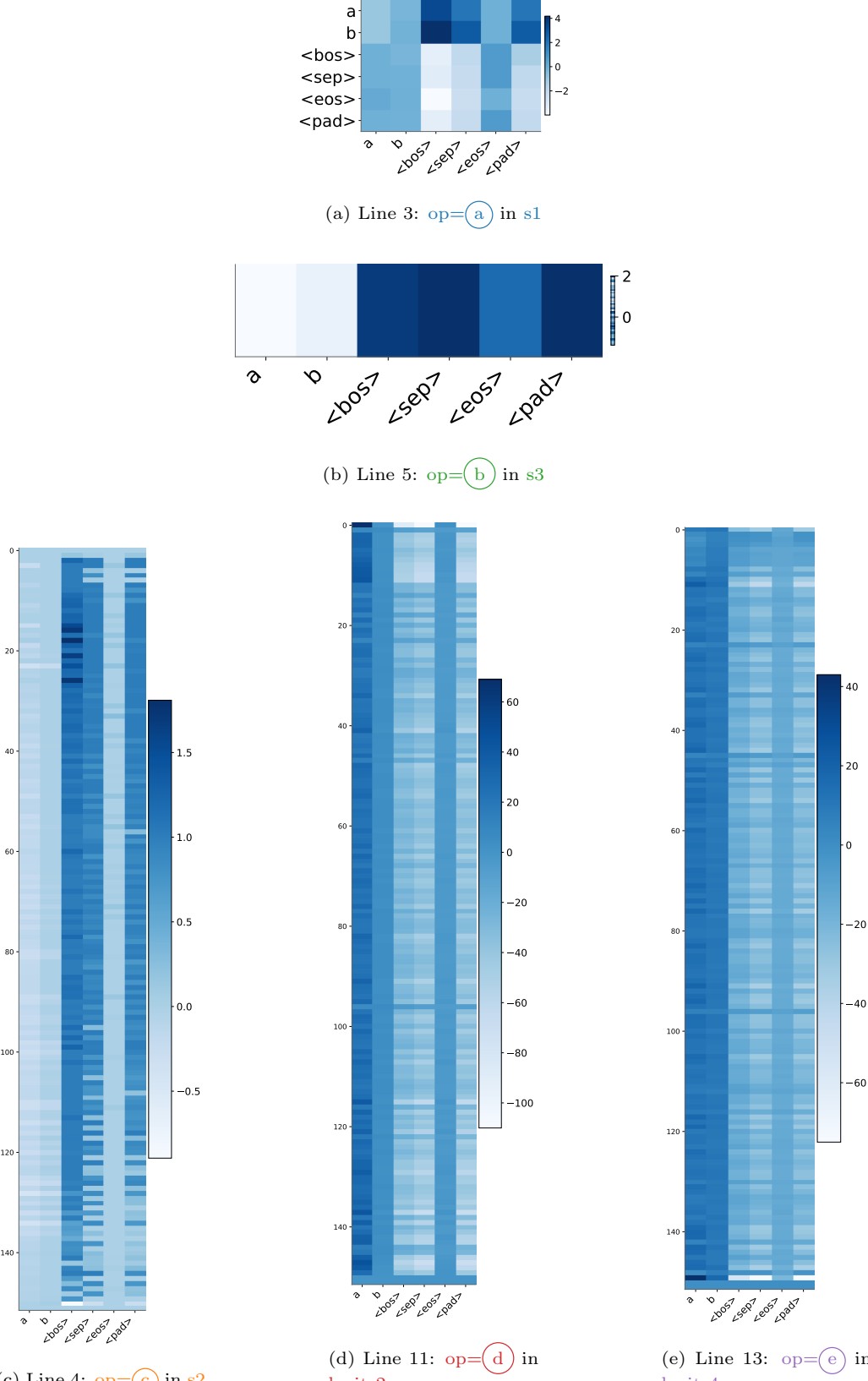

*Figure 66.* Heatmaps supporting the program for Dyck-12 model.

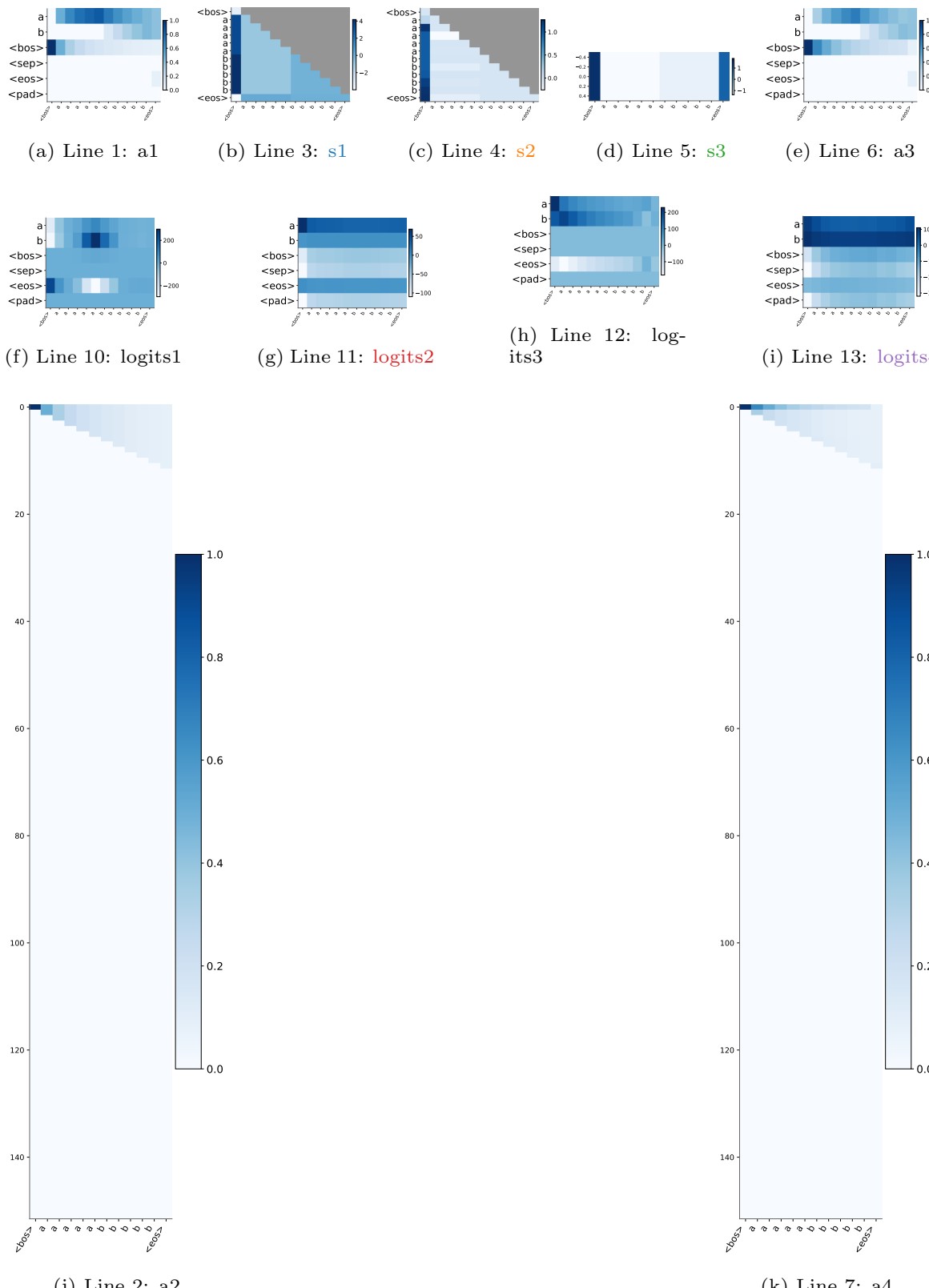

(a) Line 1: a1

(b) Line 3: s1

(c) Line 4: s2

(d) Line 5: s3

(e) Line 6: a3

(f) Line 10: logits1

(g) Line 11: logits2

(h) Line 12: logits3

(i) Line 13: logits4

(j) Line 2: a2

(k) Line 7: a4

*Figure 67.* Variables Heatmaps for Dyck-12 model on an example input.

**MLP Input-Output Distributions**   Explaining per-position operation in Line 8 via its effect on Output Logits in Line 10

**Output Token: a**

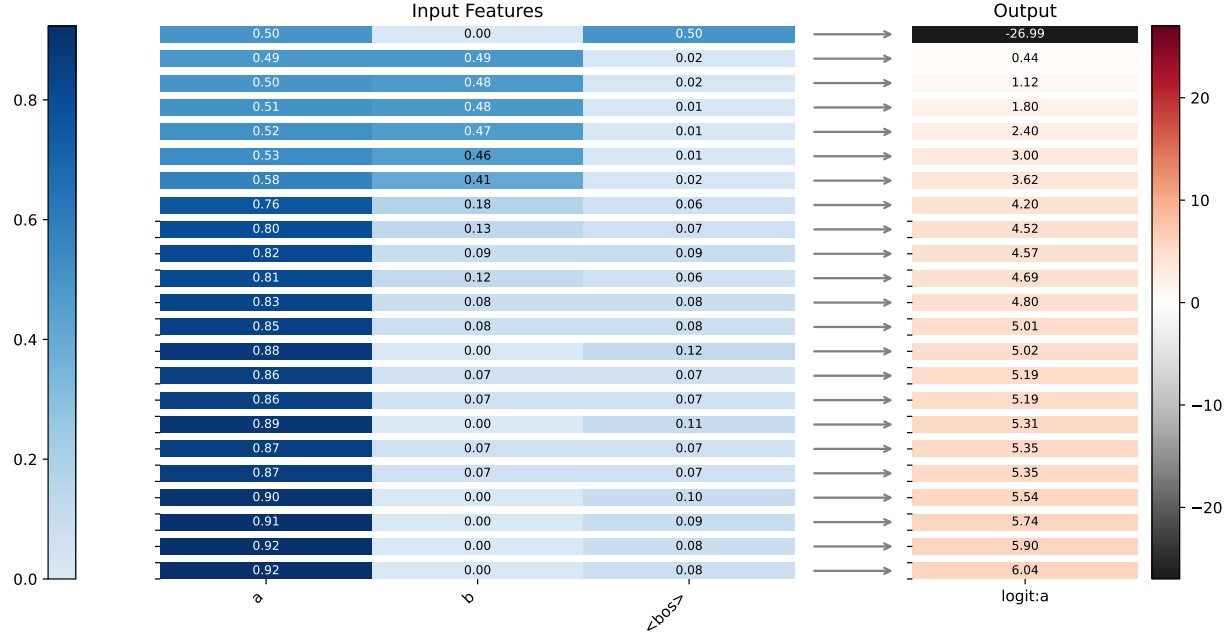

*Figure 68.* MLP Input-Output for token: a (sorted by logits)

**Output Token: b**

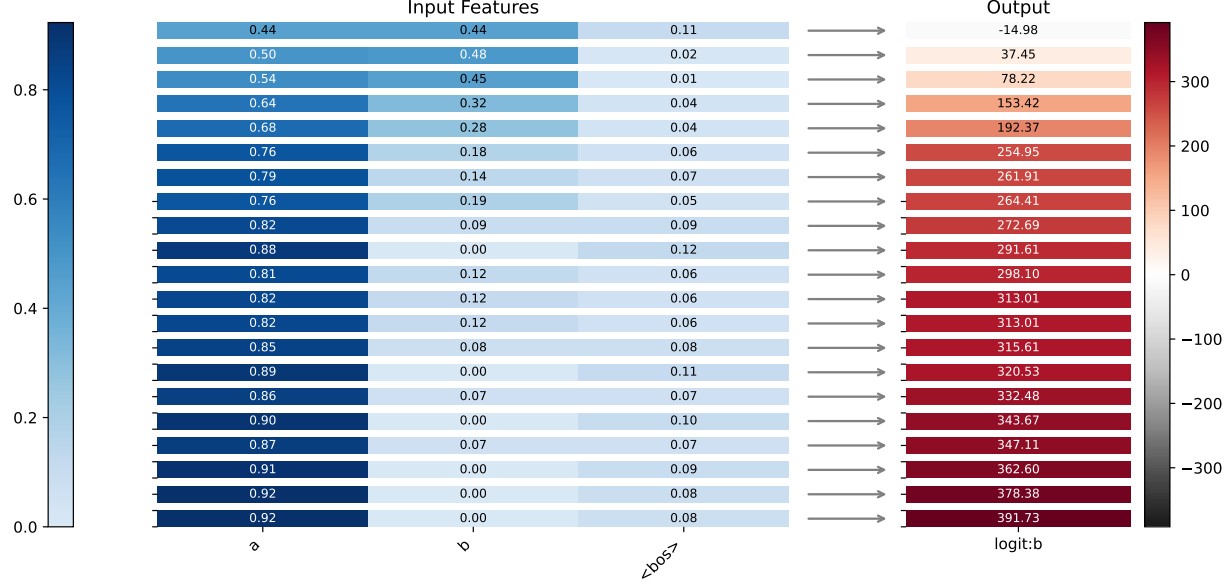

*Figure 69.* MLP Input-Output for token: b (sorted by logits)

**Output Token: EOS**

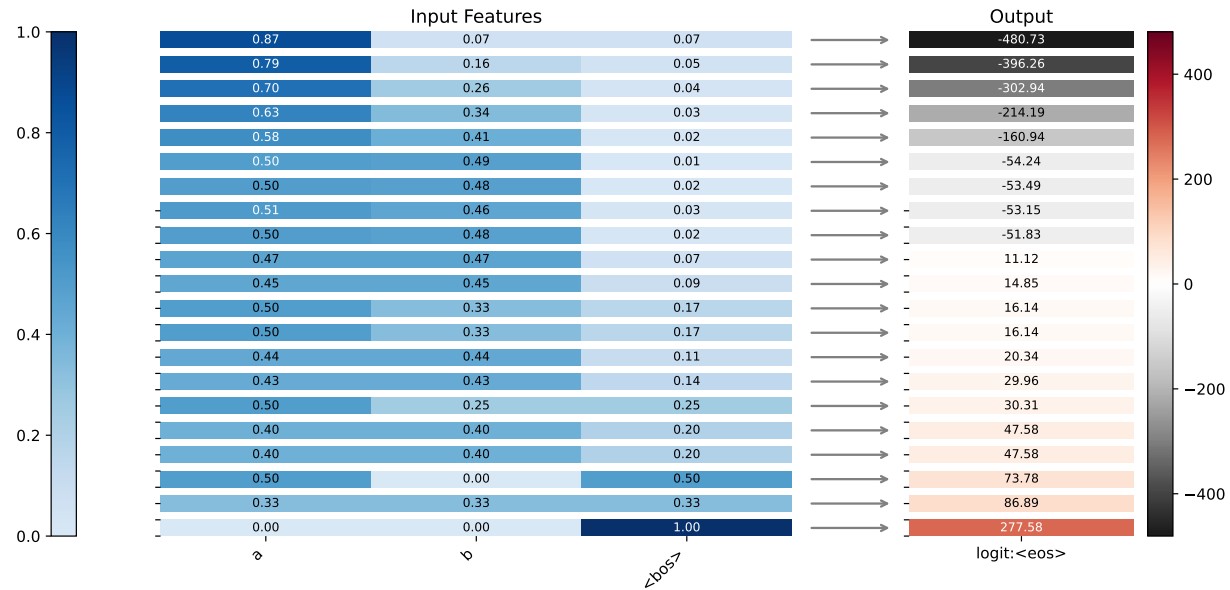

*Figure 70.* MLP Input-Output for token: EOS (sorted by logits)

Explaining per-position operation in Line 9 via its effect on Output Logits in Line 12

**Output Token: a**

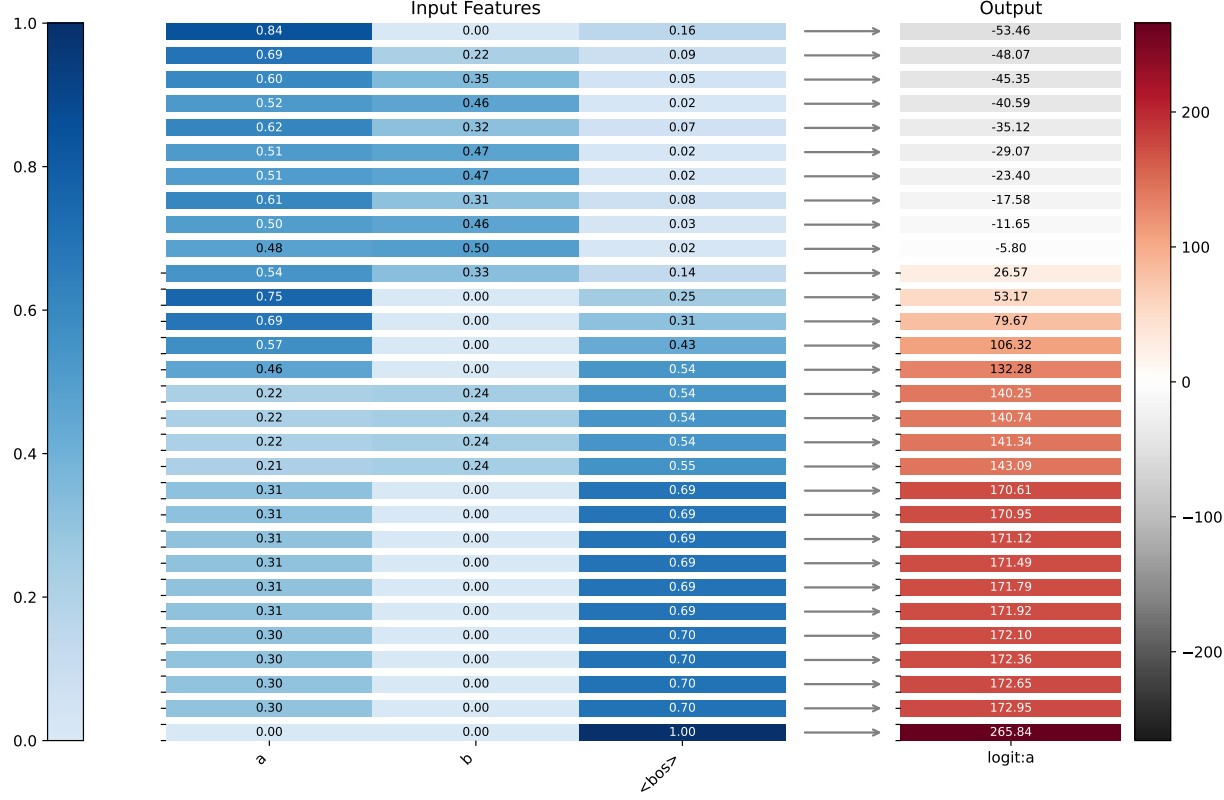

*Figure 71.* MLP Input-Output for token: a (sorted by logits)

**Output Token: b**

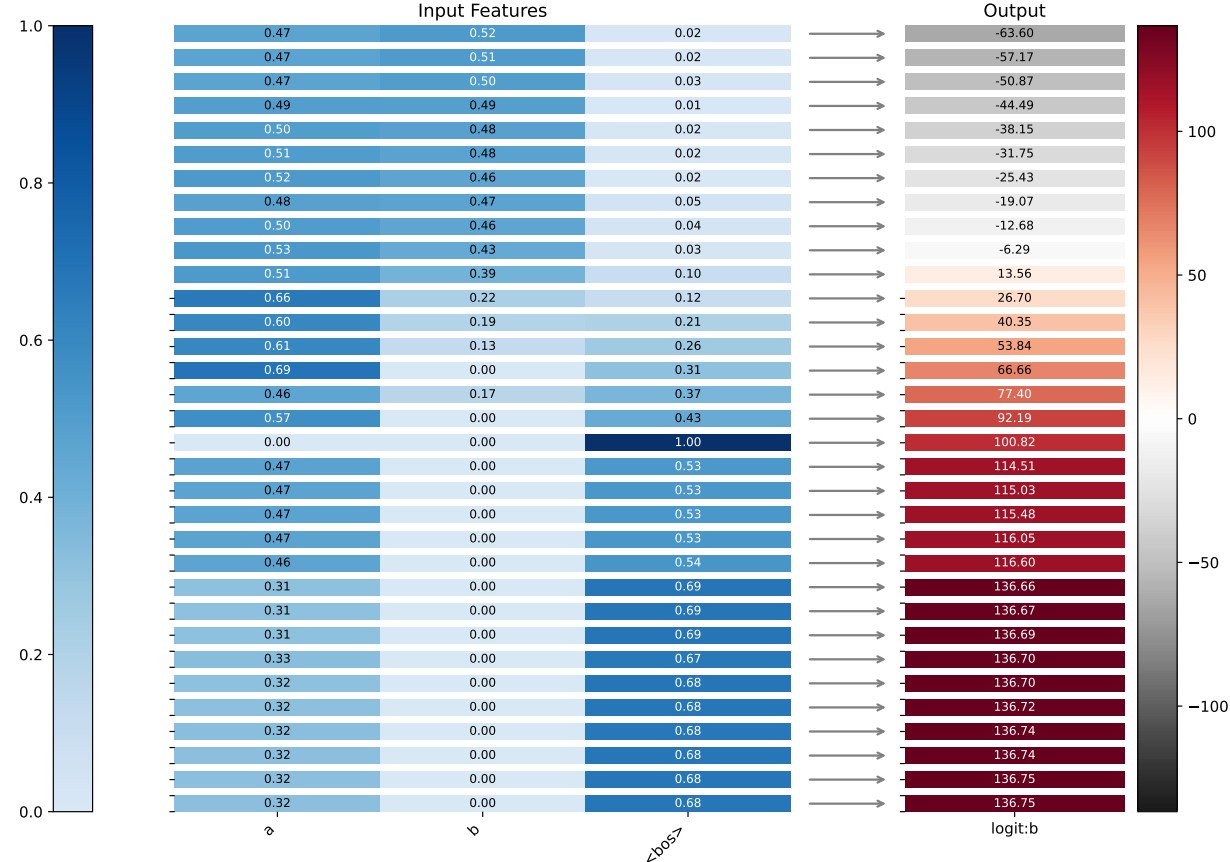

*Figure 72.* MLP Input-Output for token: b (sorted by logits)

**Output Token: EOS**

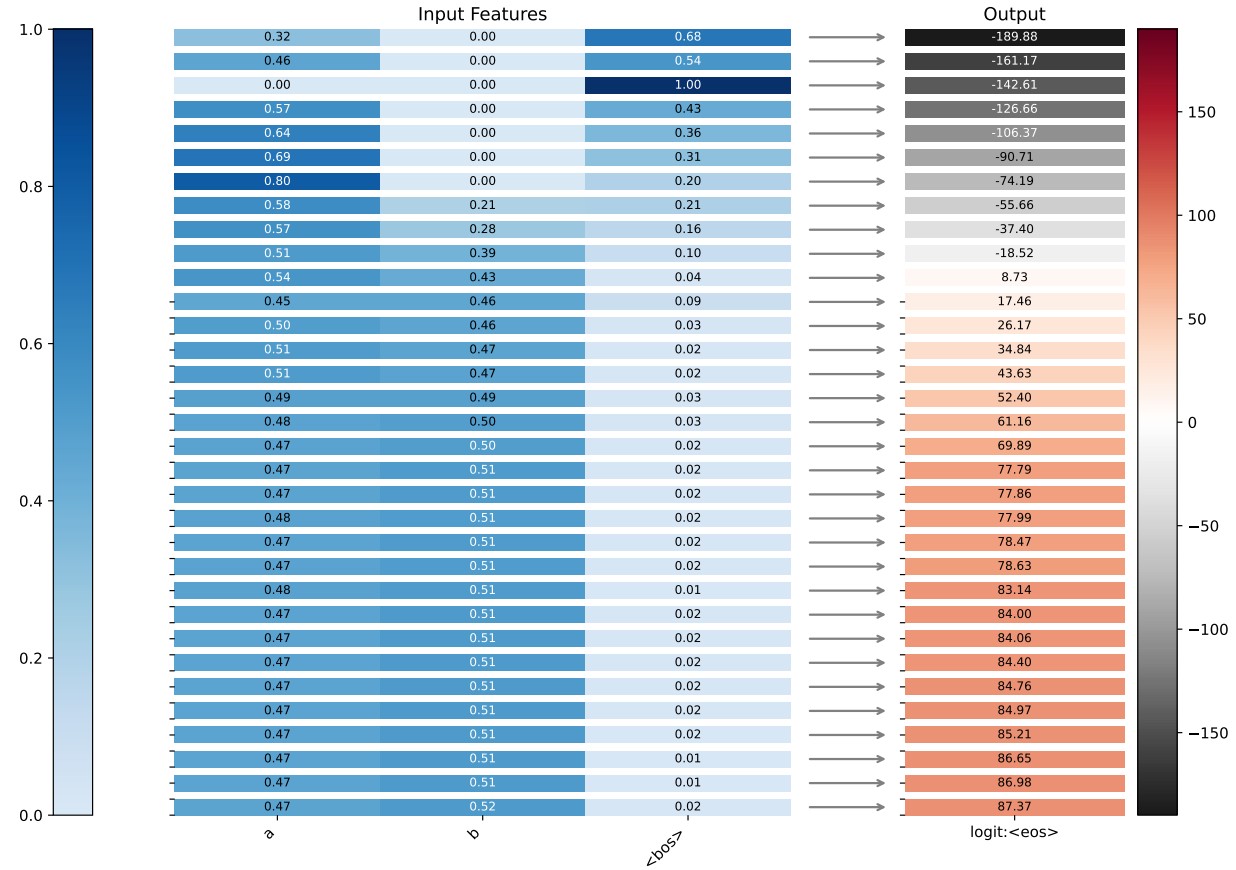

*Figure 73.* MLP Input-Output for token: EOS (sorted by logits)

## K.2. Dyck-2

**Task Description:**

$$\langle\text{bos}\rangle \ (a(ab)^*b)^* \ \langle\text{eos}\rangle$$

**Architecture:** Layers: 1     Heads: 1     Hidden Dim: 16 LR: 0.001     Dropout: 0
**Performance (w/Pruning $\rightarrow$ w/Primitives):** Task Accuracy: $1.00 \rightarrow 1.00$; Match Accuracy: $1.00 \rightarrow 1.00$

**Code**

```
1. s1 = select(q=token, k=token, op=(a))      # layer 0 head 0
2. a1 = aggregate(s=s1, v=token)              # layer 0 head 0
3. new_a1 = element_wise_op(a1)               # layer 0 mlp
4. logits1 = project(inp=token, op=(b))
5. logits2 = project(inp=a1, op=(c))
6. logits3 = project(inp=token, op=(d))
7. logits4 = project(inp=new_a1, op=(inp==out))
8. logits5 = project(op=(e))
9. prediction = sigmoid(logits1+
                        logits2+
                        logits3+
```

```
logits4+
logits5)
```

**Interpretation** This program is a more complex version of the D4 program discussed in Figure 8 the main paper. Of note, the selector in Line 1 assigns unequal weights to "a" and "b" (Figures 74a and 75a). As a result, the output a1 does not simply record the relative counts of "a", "b", and "BOS', but rather records *differently weighted* counts. The element-wise operation in line 3 performs a computation similar to that in D4, as shown by the samples below. The output logits are then composed by complementing the output of the elementwise operation (line 7) with information from the current token (lines 4 and 6), the weighted relative counts (line 5), and a bias favoring "a" over "b"/"EOS" (line 8).

We note that the "unbalanced" weighting of "a" and "b" closely relates to findings from Wen et al. (2023), who found that transformers can recognize bounded-depth Dyck languages with "unbalanced" attention patterns, though such unbalanced-ness may eventually hurt length generalization on *unboundedly* long inputs. That said, our results here show that, at lengths $\leq 150$, the program manages to deal with the unbalanced counts at perfect accuracy.

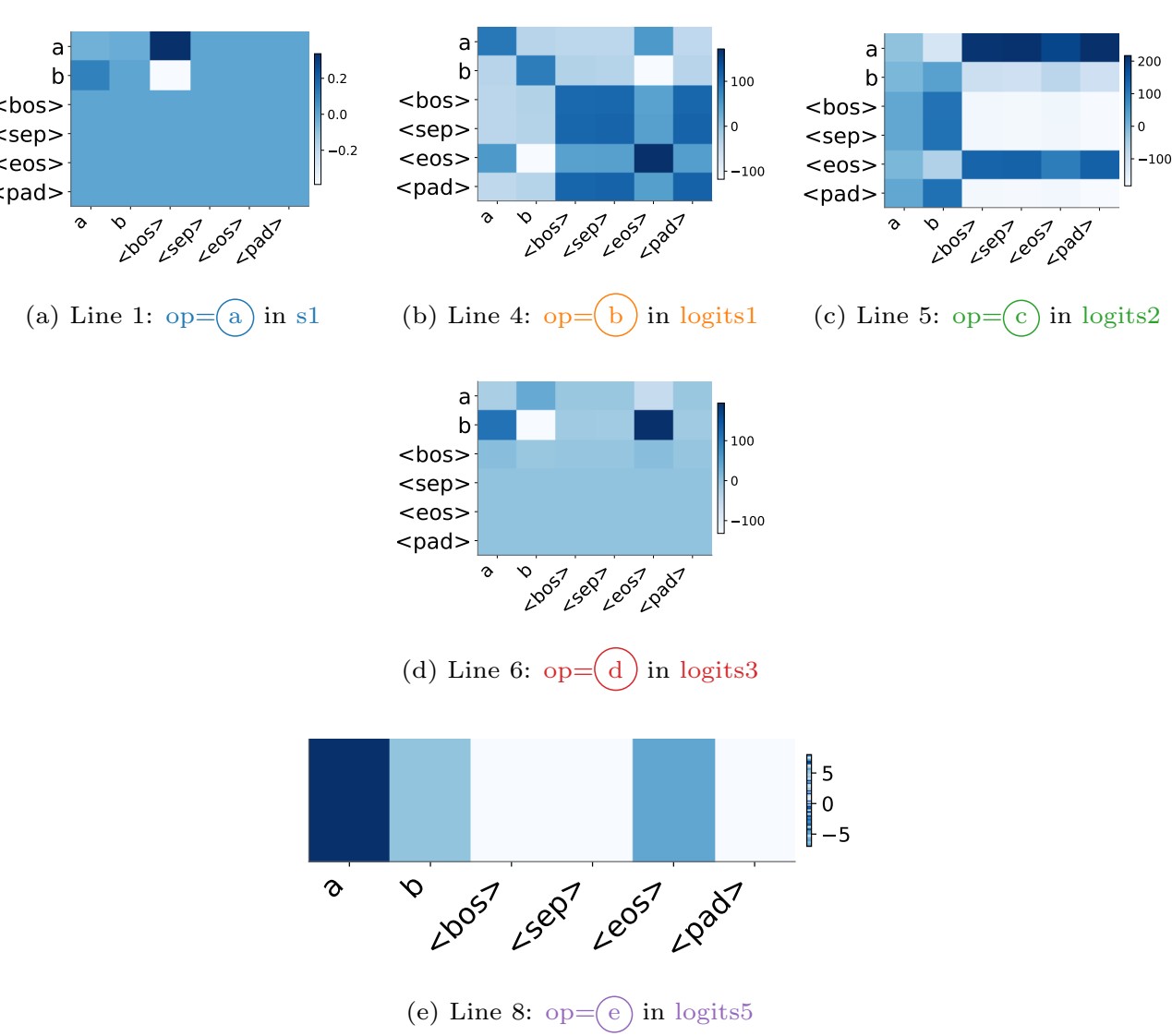

(a) Line 1: op=(a) in s1

(b) Line 4: op=(b) in logits1

(c) Line 5: op=(c) in logits2

(d) Line 6: op=(d) in logits3

(e) Line 8: op=(e) in logits5

*Figure 74.* Heatmaps supporting the program for Dyck-2 model.

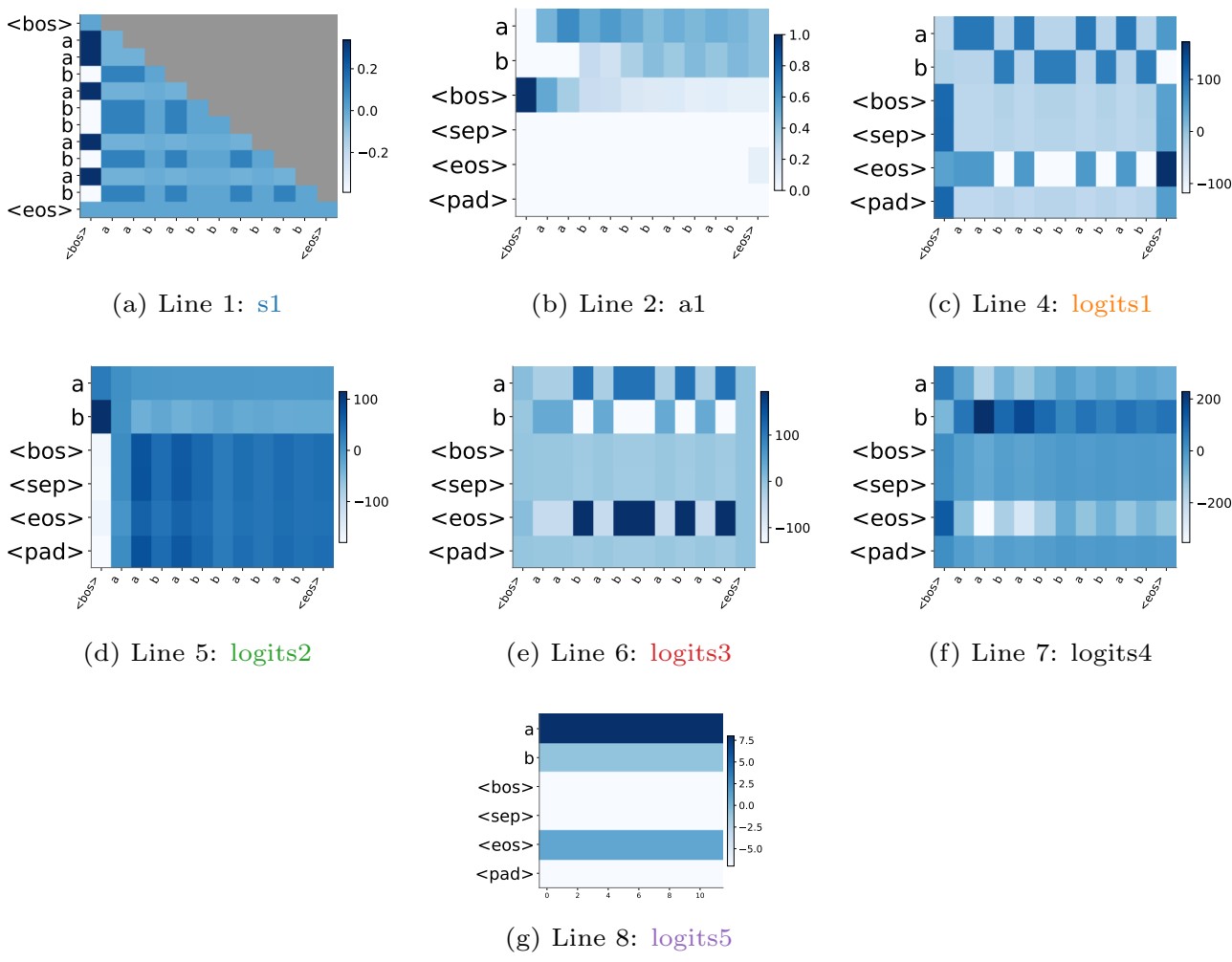

*Figure 75.* Variables Heatmaps for Dyck-2 model on an example input.

**MLP Input-Output Distributions**  Explaining per-position operation in Line 3 via its effect on Output Logits in Line 7

**Output Token: a**

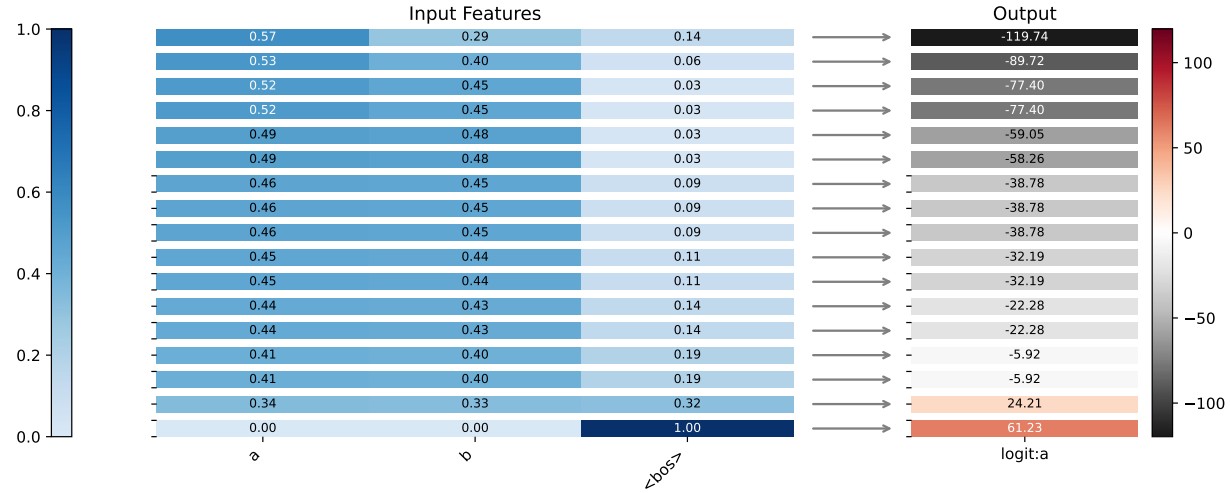

*Figure 76.* MLP Input-Output for token: a (sorted by logits)

## Output Token: b

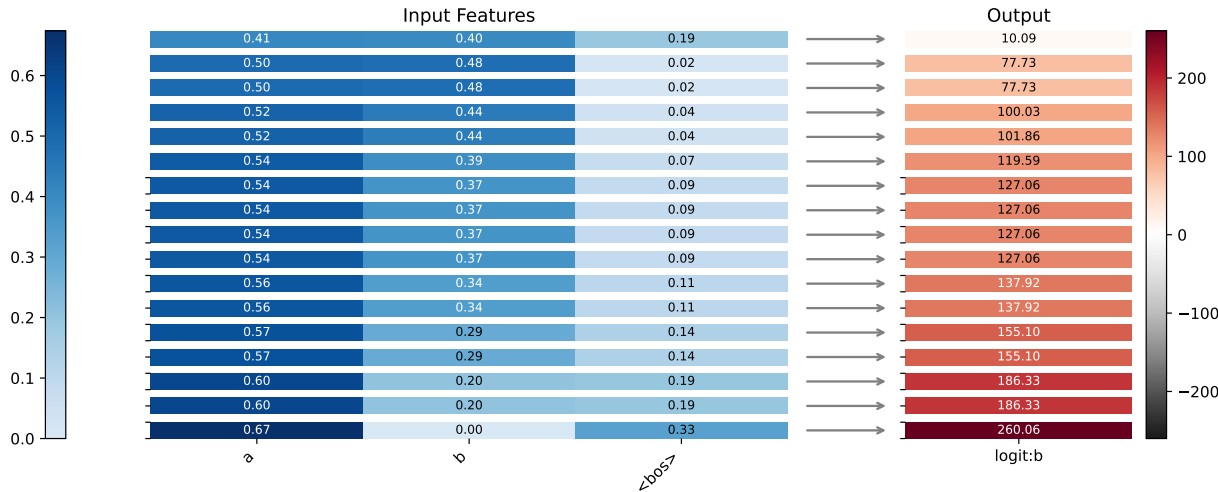

*Figure 77.* MLP Input-Output for token: b (sorted by logits)

## Output Token: EOS

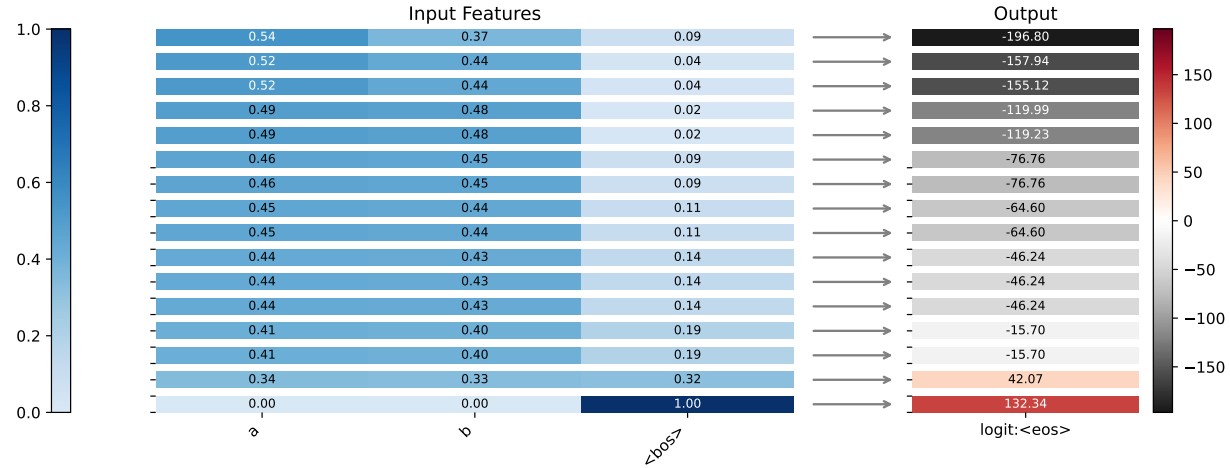

*Figure 78.* MLP Input-Output for token: EOS (sorted by logits)

## K.3. Dyck-4

**Task Description:**

$$\langle \text{bos} \rangle \ (a(a(a(ab)^*b)^*b)^*b)^* \ \langle \text{eos} \rangle$$

**Architecture:** Layers: 1    Heads: 2    Hidden Dim: 256 LR: 0.0001    Dropout: 0
**Performance (w/Pruning → w/Primitives):** Task Accuracy: $1.00 \rightarrow 1.00$; Match Accuracy: $1.00 \rightarrow 0.99$

**Code**

```
1. a1 = aggregate(s=[], v=token)            # layer 0 head 0
2. new_a1 = element_wise_op(a1)             # layer 0 mlp
3. logits1 = project(inp=new_a1, op=(inp==out))
4. prediction = sigmoid(logits1)
```

**Interpretation**    This program is discussed in the main paper, Figure 8. We provide further examples of the elementwise operation below.

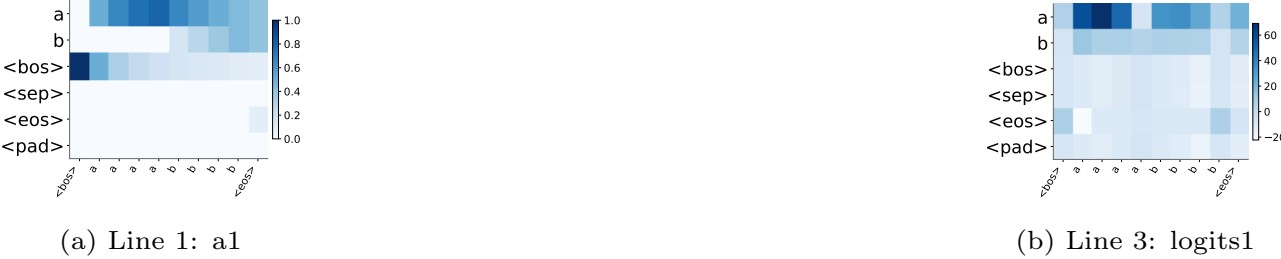

(a) Line 1: a1

(b) Line 3: logits1

*Figure 79.* Variables Heatmaps for Dyck-4 model on an example input. (b) shows the `logits1` generated by the elementwise operation: "a" is allowed as long as the depth doesn't exceed 4; "EOS" is allowed at the beginning and end (as the string is balanced there); "b" is allowed except at the beginning and end (it is only possible when the string is unbalanced).

**MLP Input-Output Distributions**    Explaining per-position operation in Line 2 via its effect on Output Logits in Line 3

**Output Token: a**

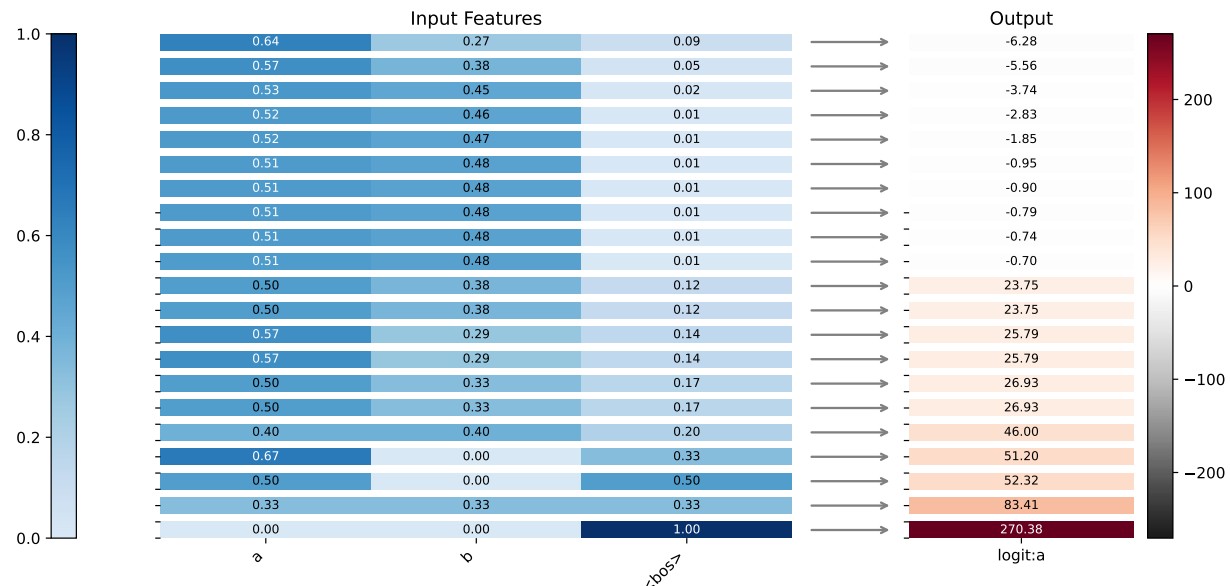

*Figure 80.* MLP Input-Output for token: a (sorted by logits). A condensed version (with just the output signs, and all three output dimensions together) is shown in Figure 8. We see that "a" receives a positive output logit if and only if #a - #b < 4 · #BOS, noting that #BOS = 1 by definition. The operation thus correctly enforces the bounded-depth constraint of D4.

## Output Token: b

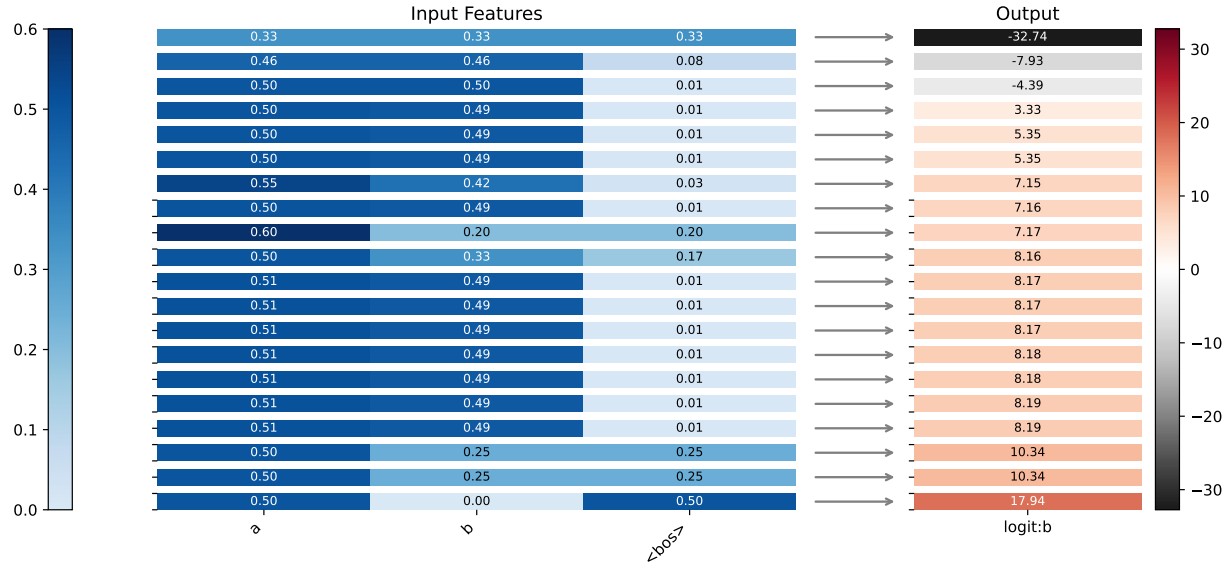

*Figure 81.* MLP Input-Output for token: b (sorted by logits). A condensed version (with just the output signs, and all three output dimensions together) is shown in Figure 8. We see that "b" receives a positive output logit if and only if #a > #b.

## Output Token: EOS

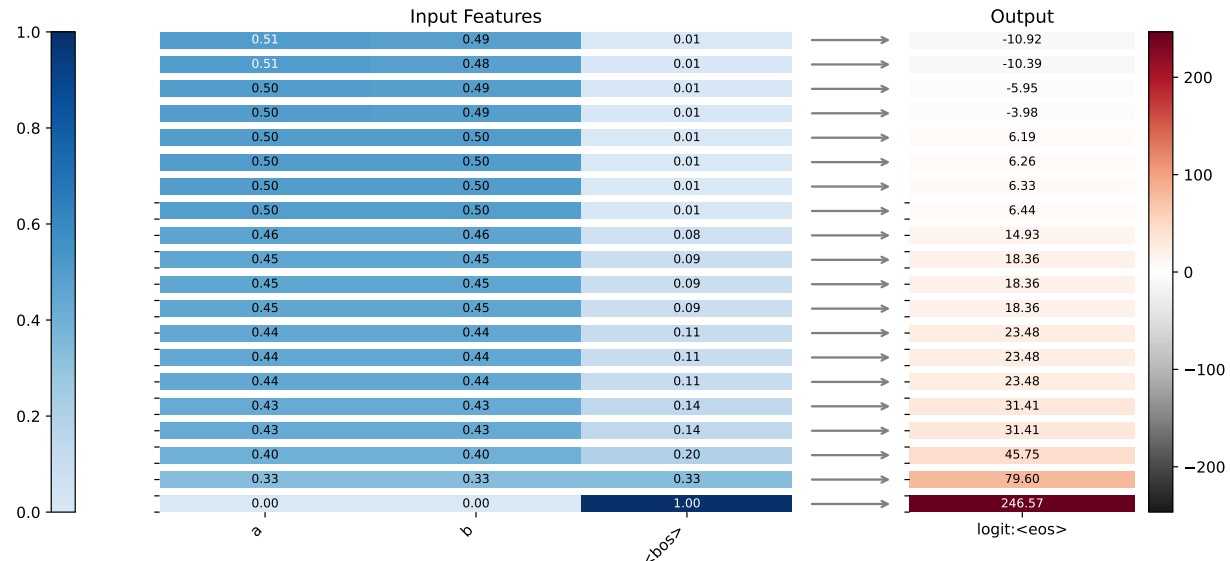

*Figure 82.* MLP Input-Output for token: EOS (sorted by logits). A condensed version (with just the output signs, and all three output dimensions together) is shown in Figure 8. We see that "EOS" receives a positive output logit if and only if #a = #b, i.e., the operation enforces that any string in the language must be balanced.

## K.4. aastar

**Task Description:**

$$\langle\text{bos}\rangle \ (aa)^* \ \langle\text{eos}\rangle$$

**Architecture:** Layers: 1     Heads: 2     Hidden Dim: 16 LR: 0.001     Dropout: 0.1
**Performance (w/Pruning $\rightarrow$ w/Primitives):** Task Accuracy: $1.00 \rightarrow 1.00$; Match Accuracy: $1.00 \rightarrow 1.00$

**Code**

```
1. s1 = select(k=token, op=(k==BOS))
2. a1 = aggregate(s=s1, v=pos)          # layer 0 head 1
3. m1 = element_wise_op(pos+a1)          # layer 0 mlp
4. logits1 = project(inp=m1, op=(inp==out))
5. prediction = sigmoid(logits1)
```

**Interpretation**   Lines 1–2 retrieve the position of BOS (note that, while the position is 0 in this example, this does not generally hold due to the use of random offsets in positional encoding during training). Line 3 now jointly processes the current position with the position of BOS in an elementwise operation. In this case, the elementwise operation was not replaced with a primitive. It directly feeds into the next-token prediction `logits1`; inspecting its values on a sample input (Figure 84b) show that "a" always receives a positive logit, whereas the logit for EOS alternates – which is correct behavior as the number of a's in the input string in the language has to be even. This shows that the elementwise operation in Line 3 effectively checks the parity between the current position of BOS, and outputs a logit for EOS on this basis.

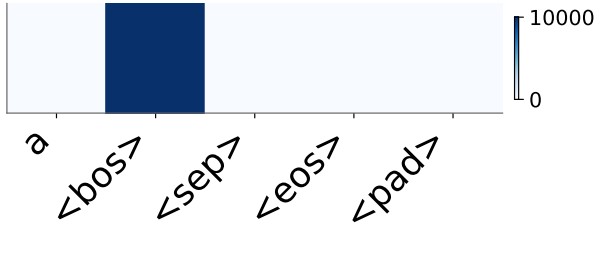

*(a)* Line 1: op=(k==BOS)

*Figure 83.* Heatmaps supporting the program for aastar model.

*(a)* Line 1: s1          *(b)* Line 4: logits1          *(c)* Line 2: a1

*Figure 84.* Variables Heatmaps for aastar model on an example input.

## K.5. abcde

**Task Description:**

$$\langle\text{bos}\rangle\ aa^*bb^*cc^*dd^*ee^*\ \langle\text{eos}\rangle$$

**Architecture:** Layers: 1    Heads: 1    Hidden Dim: 16 LR: 0.001    Dropout: 0
**Performance (w/Pruning → w/Primitives):** Task Accuracy: $1.00 \to 1.00$; Match Accuracy: $1.00 \to 1.00$

**Code**

```
1. logits1 = project(inp=token, op= a )
2. prediction = sigmoid(logits1)
```

**Interpretation**    This program is discussed in the main paper.

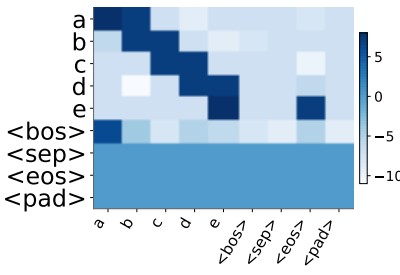

(a) Line 1: op= a  in logits1

*Figure 85.* Heatmaps supporting the program for abcde model.

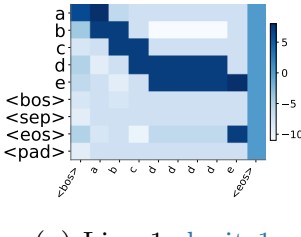

(a) Line 1: logits1

*Figure 86.* Variables Heatmaps for abcde model on an example input.

## K.6. ab_d_bc

**Task Description:**

$$\langle\text{bos}\rangle\ \{a,b\}^*d\{b,c\}^*\ \langle\text{eos}\rangle$$

**Architecture:** Layers: 1    Heads: 1    Hidden Dim: 16 LR: 0.001    Dropout: 0.1
**Performance (w/Pruning → w/Primitives):** Task Accuracy: $1.00 \to 1.00$; Match Accuracy: $1.00 \to 1.00$

**Code**

```
1. s1 = select(k=token, op= b )
2. a1 = aggregate(s=s1, v=token)          # layer 0 head 0
```

```
3. logits1 = project(inp=a1, op=(a))
4. logits2 = project(op=(c))
5. prediction = sigmoid(logits1+
                        logits2)
```

**Interpretation**   Essentially, the model needs to check if "d" has occurred so far: if it hasn't, $\{a, b, d\}$ are valid; if it has, $\{b, c, EOS\}$ are valid. Indeed, lines 1–2 encode this information by assigning weight to "d" if it has occurred. Line 3 translates this information into the next-word expectations described above: e.g., the row for BOS assigns positive weight to a, b, d and negative weight to c, EOS; whereas the row for "d" assigns positive weight to b, c, EOS. Line 4 complements this by assigning negative weight to BOS, SEP, PAD.

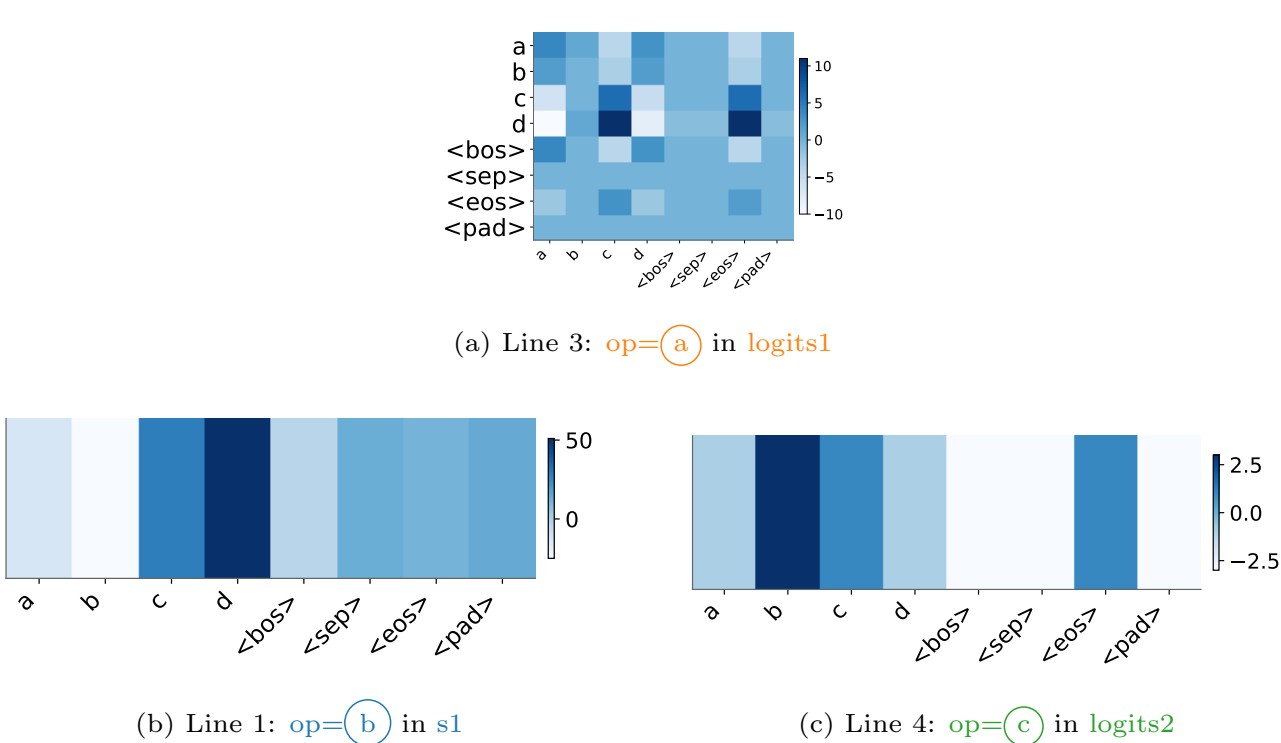

(a) Line 3: op=(a) in logits1

(b) Line 1: op=(b) in s1

(c) Line 4: op=(c) in logits2

*Figure 87.* Heatmaps supporting the program for ab_d_bc model.

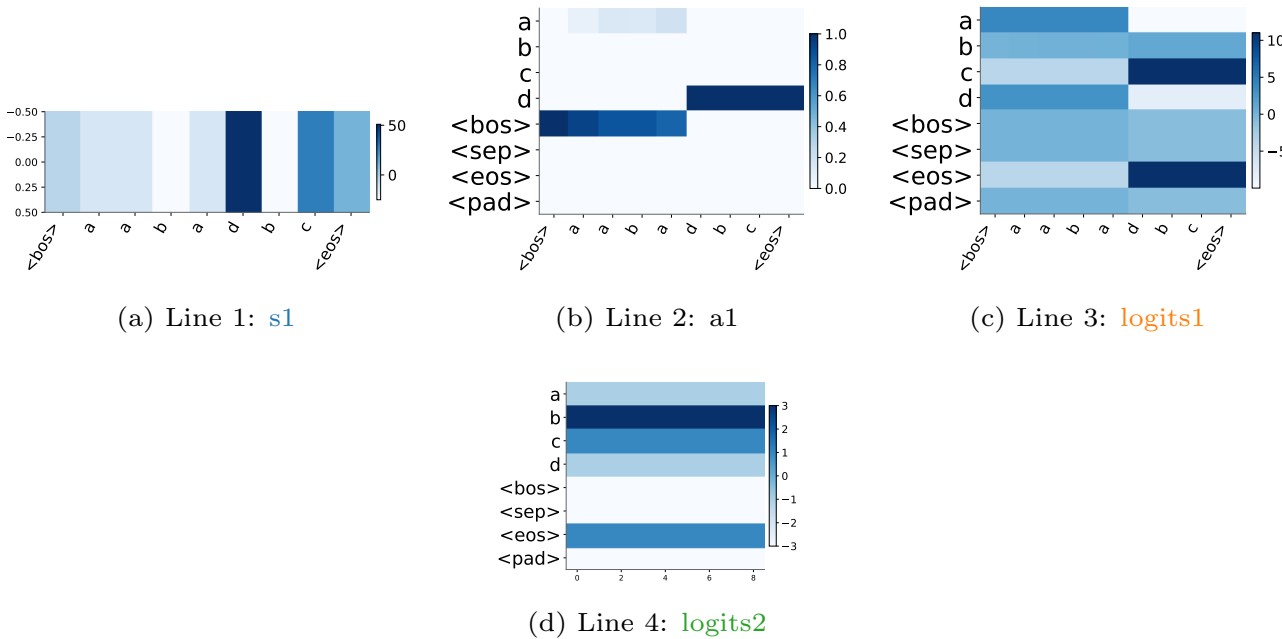

(a) Line 1: s1  (b) Line 2: a1  (c) Line 3: logits1

(d) Line 4: logits2

*Figure 88.* Variables Heatmaps for ab_d_bc model on an example input.

## K.7. tomita1

**Task Description:**

$$\langle bos \rangle \; 1^* \; \langle eos \rangle$$

**Architecture:** Layers: 1    Heads: 1    Hidden Dim: 16 LR: 0.001    Dropout: 0
**Performance (w/Pruning → w/Primitives):** Task Accuracy: $1.0 \to 1.0$; Match Accuracy: $1.0 \to 1.0$

**Code**

```
1. prediction = sigmoid(bias)
```

## K.8. tomita2

**Task Description:**

$$\langle bos \rangle \; (10)^* \; \langle eos \rangle$$

**Architecture:** Layers: 1    Heads: 1    Hidden Dim: 16 LR: 0.001    Dropout: 0
**Performance (w/Pruning → w/Primitives):** Task Accuracy: $1.00 \to 1.00$; Match Accuracy: $1.00 \to 1.00$

**Code**

```
1. logits1 = project(inp=token, op=(a))
2. prediction = sigmoid(logits1)
```

**Interpretation**   This program straightforwardly describes next-token expectations in terms of the current token: BOS is followed by 1 or EOS; 1 is followed by 0; 0 is followed by 1 or EOS.

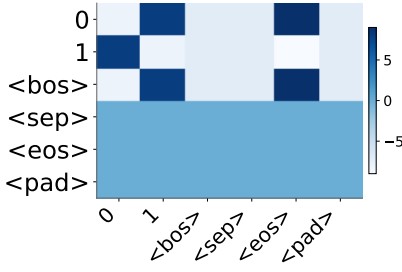

(a) Line 1: op=(a) in logits1

*Figure 89.* Heatmaps supporting the program for tomita2 model.

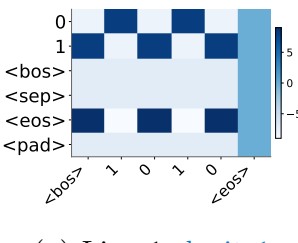

(a) Line 1: logits1

*Figure 90.* Variables Heatmaps for tomita2 model on an example input.

## K.9. tomita7

**Task Description:**

$$\langle bos \rangle \ 0^*1^*0^*1^* \ \langle eos \rangle$$

**Architecture:** Layers: 2    Heads: 1    Hidden Dim: 16 LR: 0.001    Dropout: 0.1
**Performance (w/Pruning → w/Primitives):** Task Accuracy: $1.00 \to 0.98$; Match Accuracy: $1.00 \to 0.98$

**Code**

```
1. s1 = select(q=token, k=token, op=(a))         # layer 0 head 0
2. a1 = aggregate(s=s1, v=token)          # layer 0 head 0
3. s2 = select(q=token, k=a1, op=(k==q),
              special_op=(uniform selection))        # layer 1 head 0
4. a2 = aggregate(s=s2, v=a1)            # layer 1 head 0
5. logits1 = project(inp=a2, op=(c))

6. logits2 = project(op=(d))
7. prediction = sigmoid(logits1+
                        logits2)
```

**Interpretation**    The model essentially prohibits generating a 0 if a $1^*0^*1^*$ subsequence is detected in Lines 1-4. In `s1` (Figure 91a) q=0 pays attention to 1's and q=1 to 0's, and thus `a1` (Figure 92b) holds the count of 0's (for q=1) and 1's (for q=0) before the current token. Then, in Line 4 `a1` is aggregated to form `a2` (Figure 92d), which for q=0 holds the count of 0's preceding 1's preceding current token. Essentially, it detects whether the current symbol is one of the trailing 0's in a substring $0^*1^*0^*$. Similarly, for q=1, `a2` detects the substring $1^*0^*1^*$. In Line 5, the model severely downweights logit 1 if $1^*0^*1^*$ substring was detected (Figure 91c). By default, the model allows output of all the tokens among 0, 1 and $\langle eos \rangle$, as shown in high logits for bias and projection matrix in line 5 (Figure 91c,d).

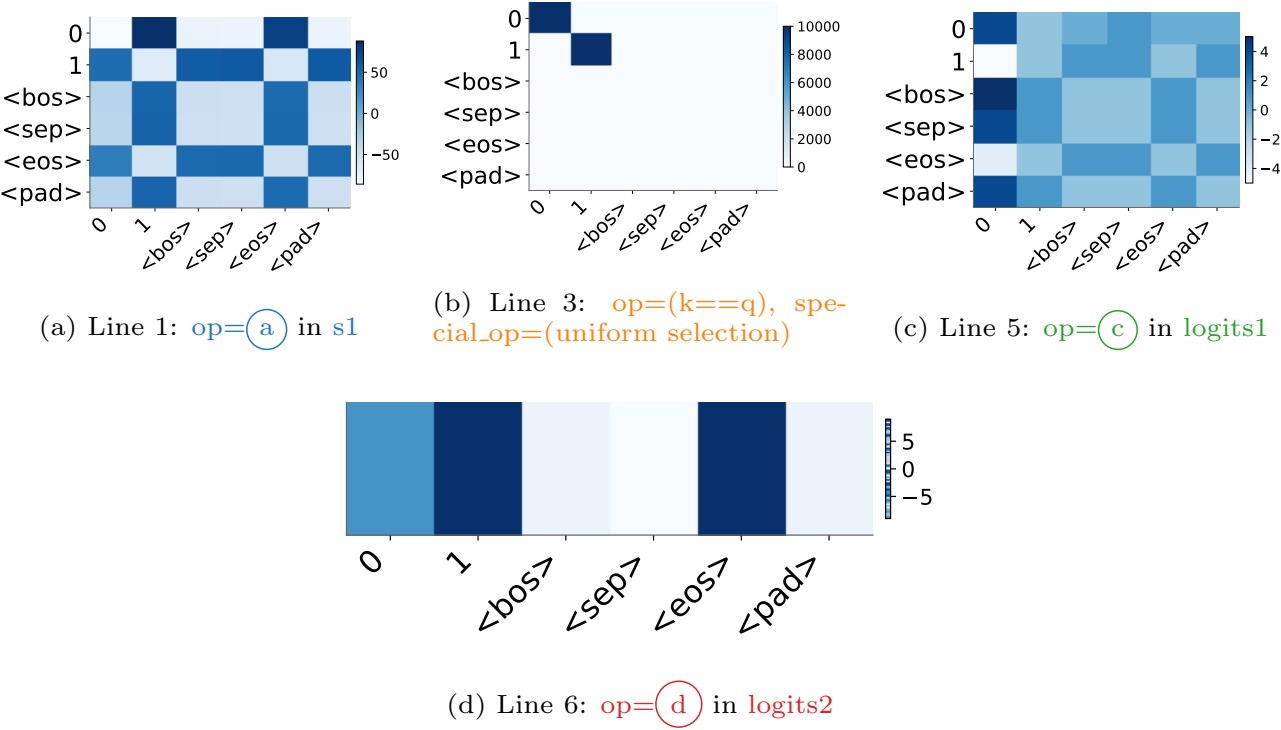

(a) Line 1: op=(a) in s1

(b) Line 3: op=(k==q), special_op=(uniform selection)

(c) Line 5: op=(c) in logits1

(d) Line 6: op=(d) in logits2

*Figure 91.* Heatmaps supporting the program for tomita7 model.

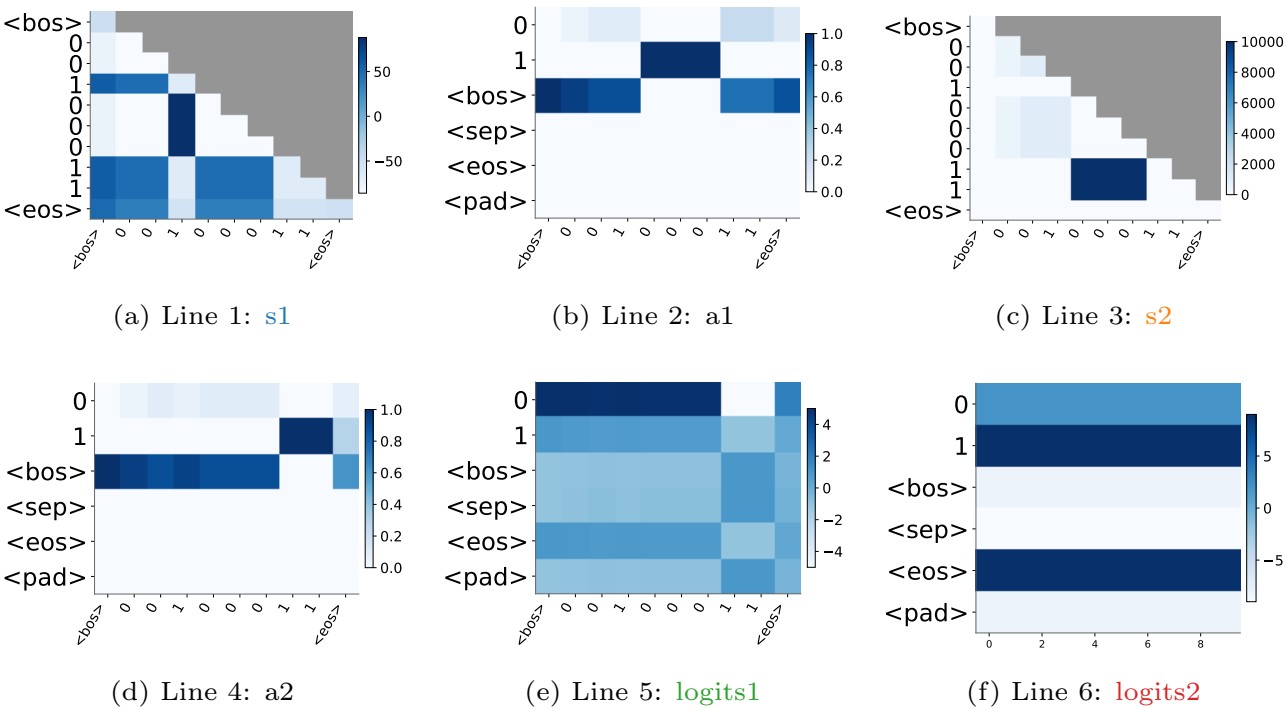

(a) Line 1: s1

(b) Line 2: a1

(c) Line 3: s2

(d) Line 4: a2

(e) Line 5: logits1

(f) Line 6: logits2

*Figure 92.* Variables Heatmaps for tomita7 model on an example input.

