# OpenReview forum: "Discovering Interpretable Algorithms by Decompiling Transformers to RASP"
_ICML.cc/2026/Conference — ICML 2026 regular_

### Official Review · Reviewer_Bgrp · 2026-03-01

**Soundness:** 3
**Presentation:** 2
**Significance:** 2
**Originality:** 3
**Overall Recommendation:** 4
**Confidence:** 4

**Summary:**

This paper introduces a method for decompiling trained Transformer models into compact D-RASP programs. The authors first define D-RASP, a target dialect of RASP, and then present a two-step pipeline: (1) faithfully reparameterizing a model into D-RASP, and (2) simplifying the program by mapping weights to a library of predefined primitives. Extensive experiments demonstrate that length-generalizing models can often be decompiled into short D-RASP programs, providing empirical support for the conjecture that length generalization in Transformers is directly linked to the implementation of simple, RASP-like algorithms.

**Compliance With Llm Reviewing Policy:**

Affirmed.

**Final Justification:**

The rebuttal phase have addressed my main concerns about human-designed primitives, and also non-legnth-generalizing models. Therefore, I will increase my score.

**Key Questions For Authors:**

1. In Step 2.3, the paper states that matrices are optimized to be sparse and integer-valued if they cannot be replaced by a library primitive. If this optimization is effective, what is the fundamental necessity of the predefined matrix library? Is it feasible to optimize all matrices and vectors directly without relying on a human-designed library?

2. Could the authors elaborate on the failure modes for non-length-generalizing models? Specifically, what alternative approaches might be used to understand their internal mechanisms if D-RASP decompilation is inapplicable?

**Limitations:**

Yes.

**Strengths And Weaknesses:**

**Strengths**

1. The paper builds upon prior work by providing concrete empirical methods to bridge the gap between length-generalizing Transformers and RASP programs.

2. The proposed pipeline for reparameterizing and simplifying trained Transformers into readable programs is novel and serves as a valuable methodological tool for the mechanistic interpretability community.

**Weaknesses**

1. In Section 3.1, the authors propose the Linear Layer Norm Assumption (LLNA) to prove Theorem 3.2. However, the formulation of Assumption 3.1 could benefit from greater mathematical rigor. Specifically, the phrase "negligible change to the input-output behavior" should be formally defined and quantified.

2. The simplification process outlined in Section 3.2 maps matrices to a predefined library of primitives. Because these primitives are human-designed, the simplification process inherently introduces a degree of subjectivity into the decompilation pipeline.

3. In Appendix A (FAQ 5), the authors note that decompilation requires properties like LLNA to hold, which often fails for non-length-generalizing models. The inability to successfully decompile these models limits the universality of the method. Providing further analysis or discussion on the mechanisms of these non-generalizing models would strengthen the overall claims about the unique connection between RASP and length generalization.

---

> ### Author Rebuttal · Authors · 2026-03-30
>
> We thank the reviewer for the constructive feedback and acknowledging the novelty and value of our work. We address the reviewer's concerns and questions below.
>
> **Weakness 1: rigor in Linear Layer Norm assumption (LLNA) formulation**
>
> We thank the reviewer for pointing this out. We use "negligible" to denote any threshold on the change to the input-output behavior that the user considers insignificant enough. The choice of the threshold affects the overall faithfulness of the program to the model. In our experiments, we deem LLNA to hold if the model can achieve > 90% match accuracy after linearizing Layer Norm. For the theorem, we will adapt the statement to clarify that we assume perfect match of the linearization (equality between left and right sides of Eq. 5). We will clarify this in the text.
>
> **Weakness 2 and Question 1: human-designed primitive matrices**
>
> Yes, it is feasible to not use the predefined matrix library and optimize all the matrices directly. The benefit of having a predefined library is that replacing a matrix with a predefined operator simplifies and cleans the operation and automatically provides an explanation for it, without any need of a human annotator to look at the matrices. This can be viewed as an additional optimization of a pipeline, rather than a necessity.
>
> Our library of "pre-interpreted matrices" enables us to  **directly obtain an interpretation** avoiding the need for an additional step of manual interpretation after looking at the matrix.
>
> We note that this step is not required for the decompilation to be successful. The automatically constructed matrices without primitive replacement could also be interpretable (please see Appendix I for an example of matrices before primitive replacement). In fact, in most programs shown in Appendices J and K, the human-designed primitives are not used. We introduced these primitives to simplify the programs further, when possible. Intriguingly, the success of primitive replacement also suggests it as a strategy for editing the model's internal algorithm, which will be interesting to explore in future work.
>
> **Weakness 3: inability to decompile non-length-generalizing models**
>
> We emphasize that **we do not claim that the method can be successfully applied to any trained model**. Some models might be internally implementing a heuristic-based non-length-generalizable solution, and a short D-RASP program might not exist for those models, and thus cannot be extracted. We argue that one *should not* hope to decompile non-generalizing models into RASP in the first place, since these would not be expected to implement interpretable and concise RASP algorithms [1]. Empirically, LLNA tends to hold on length-generalizing models. We view the inability to decompile non-generalizing models not as a failure of a method, but as an interesting **scientific insight about transformers**.
>
> We will extend Appendix A, FAQ 5 with this discussion.
>
> **Question 2: failure modes for non-length-generalizing models**
>
> As shown in Figure 12, their internal circuits usually consist of densely connected components, and finding a sparse and faithful circuit for those models is often difficult. However, it might be possible to find a circuit and a D-RASP program for a subtask of the task, or try to use any other interpretability method, like SAEs or probing, to understand parts of the behavior. Finding a comprehensive, causal and faithful explanation for the behavior of those models is an open question.
>
> [1] Huang, Xinting, et al. "A Formal Framework for Understanding Length Generalization in Transformers." The Thirteenth International Conference on Learning Representations.

---

> > ### Author Rebuttal · Reviewer_Bgrp · 2026-04-03
> >
> > Thank you for rebuttal and discussion, which have addressed my concerns. Therefore, I will increase my score.
> >
> > Please also consider include these discussions (e.g., LLNA assumptions, FAQ 5) into the updated paper.

---

### Official Review · Reviewer_e8S2 · 2026-03-12

**Soundness:** 2
**Presentation:** 1
**Significance:** 2
**Originality:** 2
**Overall Recommendation:** 3
**Confidence:** 3

**Summary:**

This paper presents a method for extracting interpretable symbolic programs from trained Transformers via decompilation into a D-RASP (RASP-like) representation. The approach reparameterizes a trained transformer into a symbolic program under certain simplifying assumptions, then applies a simplification phase that prunes and matches primitives to derive a more compact algorithmic representation. Experiments on small Transformer models trained on synthetic algorithmic tasks demonstrate several instances where the extracted programs closely correspond to intuitive algorithms.

**Compliance With Llm Reviewing Policy:**

Affirmed.

**Final Justification:**

I appreciate the authors’ efforts to clarify the paper structure. However, my core concerns about presentation and organization remain unresolved. In particular, I was unable to find explicit references to the FAQ section at the specific locations mentioned (e.g., lines 159, 370, 385), and the main body of the paper does not clearly integrate or reference the FAQ. This reinforces my impression that the overall structure is not well aligned and lacks coherence.

More broadly, the manuscript still does not clearly separate contributions, methodology, and related work, making it difficult to follow. These are structural and presentation issues that would require substantial revision rather than rebuttal clarification.

Therefore, I maintain my score of 3. That said, if the AC considers the above presentation and readability concerns to be less critical, I would not object to the paper being accepted.

**Key Questions For Authors:**

1. How does the interpretability and faithfulness of extracted D-RASP programs compare empirically with circuits discovered using existing circuit discovery approaches?

2. Do you expect the approach to work on models trained for natural language tasks or other real-world datasets? If so, what challenges would arise?

3. Can the authors provide more systematic quantitative metrics evaluating the quality of extracted programs, such as program length distributions, interpretability scores, or comparisons with baseline interpretability methods?

**Limitations:**

yes

**Strengths And Weaknesses:**

**Strengths**

1. The paper studies an interesting problem in mechanistic interpretability: recovering algorithmic structure from trained Transformers.

2. The idea of mapping Transformer computations into a symbolic intermediate representation is conceptually appealing.

3. The examples provide some intuition about how simple algorithms may emerge in trained models.

---

**Weaknesses**

1. The presentation format significantly hurts readability. Important explanations are scattered across different sections, appendices, and an FAQ, making it difficult for readers to locate key information. Several technical details necessary for understanding the method are not clearly explained in the main text.

2. Many figures primarily visualize intermediate tensors or example program steps, but they do not provide a deeper analysis of performance, robustness, or statistical trends. As a result, the experimental section feels more like illustrative demonstrations rather than a rigorous empirical study.

3. The paper mainly presents qualitative examples and descriptive results. Quantitative evaluation metrics, ablation studies, and comparisons across different model architectures or hyperparameters are limited. The results rely heavily on visual examples and manually interpreted cases rather than systematic empirical analysis.

4. The method relies on assumptions such as the Linear Layer Norm Assumption (LLNA) and focuses on specific Transformer architectures with absolute positional embeddings. The paper does not sufficiently analyze how sensitive the method is to violations of these assumptions.

---

> ### Author Rebuttal · Authors · 2026-03-30
>
> We thank the reviewer for the constructive feedback. We address the reviewer's questions and concerns below:
>
> **Weakness 1: presentation**
>
> We thank the reviewer for the suggestion. Due to the great amount of content, we had to move some details into Appendix. We will move some information from Appendix A (FAQ) and Appendix E into the main paper. We would appreciate it if the reviewer can specify which details they think are most critical for understanding, and should be placed in the main paper.
>
> **Weaknesses 2-3, Question 3: quantitative evaluation**
>
> We address these concerns together, since they are closely related.
>
> Firstly, we kindly note that we provide the programs for **all** the decompiled models in Appendices J and K, which goes beyond individual examples. We believe that by transparently allowing the reader to qualitatively verify all information that went into our interpretation, we strongly support our claim about the interpretability of the method.
>
> Secondly, we'd like to kindly point out that we provide some **quantitative measures** already. Figure 4 shows **program length vs. match accuracy** for decompilable models. Figure 10 shows number of edges (can be directly converted to program length) vs match accuracy for models trained with different hyperparameters, architectures and length generalization performance. We will move some of the quantitative insights from Appendix E to the main paper.
>
> To provide a better **analysis of robustness**, we train 10 models on two representative tasks with **different random seeds**, and see that our method extracts highly similar programs. We'd kindly refer the reviewer to our response to reviewer cRWX, (Question 2) for details on this experiment.
>
> For additional quantitative results on stability of the pruning procedure, we kindly refer the reviewer to our response to reviewer cRWX, Question 3.
>
> For **ablation studies**, we look at the gains of each stage of the pipeline to match accuracy. After linearizing layer norm and the first stage of pruning (see Appendix F for details about pruning stages) the average match accuracy over all decompilable models in our evaluation set is 93%, after splitting MLPs and path-based pruning: 93%, after pruning QK products: 95%, after replacing with primitives: 95%. None of the parts of the pipeline significantly decrease faithfulness.
>
> For **performance statistics** and automated interpretability metrics, we report statistics of the usage of human-designed primitives. Among the programs provided in Appendices J and K, 34% of all select operations, 30% of project operations and 19% of MLP operations are replaced with primitives. 88% of all MLP operations have single path as an input, which means that they were successfully splitted.
>
> We hope that this analysis helps resolve the reviewer's concerns.
>
> **Weakness 4: Linear Layer Norm Assumption (LLNA) and Absolute Positional Embeddings (APE)**
>
> We answer in parts.
>
> *LLNA*
>
> We kindly refer the reviewer to our response to reviewer cRWX for a discussion justifying this assumption.
>
> Regarding our method's sensitivity to LLNA, we'd like to kindly point out that we analyzed what happens when LLNA doesn't hold in Figure 11: if LLNA doesn't hold, even circuit discovery usually doesn't find sparse solutions. Furthermore, Tables 1 and 2 show that violation of this assumption correlates with length-generalization, and thus with the model's internal decompilability [1].
>
> We kindly refer the reviewer to our response to reviewer Bgrp (Weakness 3) on the discussion of why we don't view our inability to decompile some models as a negative result.
>
> *APE*
>
> We analyse APE Transformers mainly because the theory of length-generalization [1] is well-developed for APE, allowing us to build directly on past results. Furthermore, a lot of interpretability research has taken GPT-2 as a reference [2]. In fact, many of our programs do not use positional information. We kindly refer the reviewer to Appendix A, FAQ 1 for further discussion.
>
>
> **Question 1**
>
> The first stage of the pipeline gets distinguished from circuit discovery only when the layer norm is linearized. Thus, one can view Figure 12 as a comparison to circuit discovery.
> However, our method is qualitatively different from other common interpretability methods, and can be viewed as circuit discovery with automated explanation of each component's function. We kindly refer the reviewer to Appendix A, FAQ 3 for more discussion on this.
>
> **Question 2: scaling to LLMs**
>
> We kindly refer the reviewer to our response to reviewer rd88 and Appendix A, FAQ 2 (due to space constraints).
>
> [1] Huang, X., et al. "A Formal Framework for Understanding Length Generalization in Transformers." The Thirteenth International Conference on Learning Representations.
>
> [2] Wang, K. R., et al. "Interpretability in the wild: a circuit for indirect object identification in gpt-2 small." In The Eleventh International Conference on Learning Representations, 2023.

---

> > ### Author Rebuttal · Reviewer_e8S2 · 2026-04-04
> >
> > The rebuttal offers helpful clarifications; however, my main concerns remain. The paper lists 11 questions in the FAQ section, but these are not clearly connected to the main paper. Overall, the manuscript lacks a well-defined structure, particularly in distinguishing the contributions, related work, and methodology.
> >
> > Given these structural issues, the paper would require substantial revision to be suitable for researchers to read. Considering the amount of work involved, I assigned a score of 3 in my initial review rather than rejecting it. However, these are structured problems that cannot be resolved through a rebuttal alone. Therefore, I am inclined to maintain my score and lean toward rejection.

---

> > > ### Author Response · Authors · 2026-04-04
> > >
> > > We thank the reviewer for the suggestions to improve the structure of the paper. We are pleased that we could address the reviewer's concerns regarding the core content, and we address the feedback regarding readability below.
> > >
> > > > The paper lists 11 questions in the FAQ section, but these are not clearly connected to the main paper.
> > >
> > > We kindly note that several questions in the FAQ directly extend the discussions within the main text. For instance, Question 1 corresponds to line 159, Question 3 to line 385, and Question 4 to line 370. Other entries, such as Question 2 (application to LLMs), are intended to clarify supplementary questions a reader might have rather than support primary claims.
> > >
> > > To better connect the FAQ with the main text, we will extend the discussion of the primary questions within the main paper and include explicit cross-references for the supplementary questions.
> > >
> > > > the manuscript lacks a well-defined structure, particularly in distinguishing the contributions, related work, and methodology
> > >
> > > We kindly note that we separate our methodology and the insights regarding Transformers derived with it into different sections ("Decompilation Method" and "Results"). Comparison to related work is included in the paragraphs starting on lines 106 and 184, clearly delineating our contributions from the work we build upon. Furthermore, a broader contextualization of related work is provided in the Discussion (line 370 onwards).
> > >
> > > To further highlight the structure, we will:
> > >
> > > 1. Clearly separate the related work from the methodological content in the paragraph on line 184 and label it accordingly.
> > > 2. Explicitly state how the paper is organized and the content of each section at the end of the Introduction.
> > > 3. Move some content from the "Discussion" section to a separate "Related Work" section, allowing the Discussion to focus on the implications of our research. We will also integrate relevant content from the FAQ (e.g., application to LLMs) to the Discussion.
> > >
> > > We acknowledge that the paper is dense, as it has a lot of content and proposes a new methodology bridging two distinct research directions: theory and mechanistic interpretability. We did our best to make the text accessible, and we believe the rebuttal discussions have helped identify several crucial details (e.g., APE, LLNA, circuit discovery comparisons, and quantitative evaluation), an extended discussion of which will further improve clarity.

---

### Official Review · Reviewer_cRWX · 2026-03-16

**Soundness:** 4
**Presentation:** 4
**Significance:** 3
**Originality:** 3
**Overall Recommendation:** 5
**Confidence:** 4

**Summary:**

This paper proposes a two-stage pipeline for extracting symbolic programs from trained transformers. First, it defines a low-level RASP dialect, D-RASP, and shows that a GPT-2 style transformer satisfying a linearized layer-norm assumption can be translated into an equivalent D-RASP program obtained directly from the model parameters. Second, because the resulting program is extremely large, the paper applies causal pruning and primitive replacement to recover a much smaller sufficient subprogram. Empirically, on small GPT-2-like models trained on algorithmic tasks and bounded-depth formal languages, the method often recovers short, human-readable programs, especially for models that length-generalize. The paper argues that this provides direct evidence that some trained transformers implement simple RASP-like algorithms internally.

**Compliance With Llm Reviewing Policy:**

Affirmed.

**Final Justification:**

I maintain my accept score.  This work simultaneously demonstrates impressive decompilation results on standard Transformer architectures and provides a new angle on demonstrating the RASP length-generalization conjecture.

**Key Questions For Authors:**

1. How closely does the linearized model match the original transformer across the evaluated tasks? Quantitative comparisons would clarify how faithfully the extracted programs reflect the true model.

2. If the same task and architecture are trained with different random seeds, how stable are the extracted programs?

3. How sensitive are the recovered programs to the sparsity penalty and pruning procedure?

4. What are the main practical obstacles to applying this method to moderately larger transformers (e.g., models used in standard language modeling benchmarks)? For example, how much time does the pruning step require?

**Limitations:**

The authors adequately address the limitations.

**Strengths And Weaknesses:**

## Strengths
- The work targets standard Transformer architectures trained via standard means, as opposed to prior work which changes the architecture and/or training regime.
- The work provides an automated means of testing the RASP length-generalization conjecture.
- The experiments consistently show that models which length-generalize are the ones from which short programs can be recovered, which is an interesting and potentially useful empirical observation.

## Weaknesses

- The translation theorem applies to a transformer with linearized layer normalization rather than the original model. The method therefore does not strictly decompile the real network, and the approximation already fails in one of the reported experiments.

- All experiments involve very small GPT-2–like models trained on synthetic algorithmic or formal-language tasks. It is unclear whether the approach would scale to larger models or to natural language tasks.

- Models are early-stopped based on test accuracy and representative models are selected based on length-generalization performance. This weakens the empirical claim that decompilability is a robust property rather than a selection artifact.

---

> ### Author Rebuttal · Authors · 2026-03-30
>
> We thank the reviewer for the constructive feedback. We address the concerns and questions below.
>
> **Weakness 1: linearized layer norm assumption (LLNA)**
>
> We believe LLNA is reasonable for the following reasons:
> 1. When LLNA holds, the decompilation is faithful to the original model. We preserve the input-output behavior of the underlying network, making sure that if a program exists, it is *causally faithful* to the *original* model, even though its layer norm (LN) is linearized.
> 2. LLNA tends to hold for length-generalizable models, which are exactly the class of models for which the RASP programs are suggested to exist [1]. Additionally, empirically LNs in even larger fully trained models tend to not implement non-linear mechanisms [2]. Thus, while theoretically LLNA could lead to inability of extracting a program, this would empirically concern cases where a short D-RASP program is anyways not implemented by a model (Tables 1 and 2). Therefore, we believe that our method is *not limited by* this assumption.
>
> We will extend the discussion on this in the main paper.
>
> **Weakness 2: scaling**
>
> We kindly refer the reviewer to our response to reviewer rd88 and Appendix A, FAQ 2.
>
> **Weakness 3: selection bias**
>
> Our goal is not to decompile *any* model, but to test the idea that length-generalizable models tend to implement short and interpretable RASP programs [1]. Hence, we specifically select for length-generalization; notably, we do *not* select for decompilability. We argue that this helps us address our research question, without biasing conclusions. For additional discussion on robustness, we'd kindly refer the reviewer to our response to reviewer e8S2 (Weaknesses 2-3, Question 3).
>
> **Question 1**
>
> In all our experiments, match accuracy between the decompiled model and the original model is set to be >90%. We say that LLNA holds if match accuracy is above this threshold. In Appendices J and K, we report match accuracies between the original and decompiled models for each decompiled program.
>
> **Question 2: robustness to random seeds**
>
> We train 10 models on Most Frequent task with the same architecture, but different random seeds. Our method decompiles all of these models into exactly the same programs as the model in the paper. Notably, the computation graph after pruning is the same only in 40% of the cases, which suggests that our method is *more robust than circuit discovery*. The reason for this advantage of our method is related to two extra steps following pruning in our pipeline: replacing elementwise operations with "zero" or "no-op" primitives, and invariance of the program to specific model components, e.g. if the model has two attention heads that implement the same logic, circuit discovery might find two distinct solutions, but both result in the same lines in D-RASP code. Similarly, when we trained 10 models on Unique Copy task, 3/10 did not generalize and thus were not decompilable. For other 7, pruning results in the same computation graphs as in the example in the paper, but select/project primitives sometimes differ. However, Pearson correlation between operation matrices is quite high: 0.8 on average across operations and models. We will extend this analysis to other tasks from our evaluation set and add these experiments to the paper.
>
> **Question 3: pruning stability**
>
> Sparsity penalty controls the tradeoff between faithfulness to the original model and length of a program. We sweep this hyperparameter for each model and choose the best run on a Pareto frontier (Figure 4).
>
> While varying pruning hyperparameters might result in different programs, empirically they are quite stable. For all the 21 models in our evaluation set, we take all pruning runs that achieve >90% match accuracy. Pruning is structurally consistent: for 17/21 models *all* selected pruning runs find a computation graph which is a subgraph or a supergraph of the computation graph selected for the paper. It often finds exactly the same computation graph regardless of hyperparameters: for 10/21 models *all* runs resulted in the same programs as shown in the paper. We'll add these results to the Appendix.
>
> **Question 4**
>
> There are multiple potential obstacles to applying the pipeline to LLMs: from high computational cost of pruning to the fact that LLMs were trained on complex natural data, which makes it unlikely that they as a whole can be easily translated to RASP. However, they might implement simple algorithms for each subtask. We'd kindly refer the reviewer to Appendix A, FAQ 2 for a detailed discussion on this.
>
> [1] Huang, Xinting, et al. "A Formal Framework for Understanding Length Generalization in Transformers." The Thirteenth International Conference on Learning Representations.
>
> [2] Baroni, Luca, et al. "Transformers Don’t Need LayerNorm at Inference Time: Scaling LayerNorm Removal to GPT-2 XL and Implications for Mechanistic Interpretability." Mechanistic Interpretability Workshop at NeurIPS 2025.

---

> > ### Author Rebuttal · Reviewer_cRWX · 2026-04-03
> >
> > Thank you for addressing my questions.  I maintain my accept score.

---

### Official Review · Reviewer_rd88 · 2026-03-17

**Soundness:** 3
**Presentation:** 3
**Significance:** 2
**Originality:** 3
**Overall Recommendation:** 5
**Confidence:** 3

**Summary:**

This paper presents a general method for extracting RASP programs from trained Transformers. The basic process is to re-parameterize a Transformer as a RASP program and then apply causal interventions to discover a small sufficient subprogram. Through experiments on Transformers trained on algorithmic and formal language tasks, the authors show that the method can often recover simple and interpretable RASP programs. The results provide direct evidence that in the controlled setting of small models length-generalizing Transformers can internally implement simple RASP-like algorithms.

**Compliance With Llm Reviewing Policy:**

Affirmed.

**Final Justification:**

The rebuttal is reasonable, and I maintain my original positive judgment of the submission.

**Key Questions For Authors:**

Suggestion: could you add a summary of some of the important questions and answers in Appendix A towards the end of the main text? For example, 1-3 are good candidates.

**Limitations:**

The authors should have a dedicated limitation section or subsection.

**Strengths And Weaknesses:**

Strengths
- This paper is technically sound and clearly written.
- The decompilation method to my knowledge is original.
- Symbolic program extraction is a promising and interesting topic in machine learning, and it has positive implications for model interpretability.
- Experiments include both algorithmic and formal language tasks.
- Appendix A FAQ is very reader friendly.

Weaknesses
- Experiments are only done on small transformers.
- It is not clear how the method and results would transfer to real language models.

---

> ### Author Rebuttal · Authors · 2026-03-30
>
> We thank the reviewer for the constructive feedback and acknowledging the novelty and value of our work and clarity of our writing. We address the concerns and questions below.
>
> > Experiments are only done on small transformers. It is not clear how the method and results would transfer to real language models.
>
> While our research question is about foundational properties of the architecture, the application of this method to LLMs is an exciting future direction, and in this paper we lay an important foundation for it. Scaling methods that were first introduced for small models is a common path in interpretability research, and established concepts, like induction heads, took that route. We discuss this more in Appendix A, FAQ 2.
>
> Importantly, our primary research question in the paper is whether length-generalizing transformers implement RASP-like algorithms, and this question is answered most cleanly with small models trained from scratch. In particular, we believe that our work provides some vindication for use of the C-RASP for understanding Transformers. The fact that  models that length generalize implement algorithms predicted by C-RASP helps bring credence to the framework. Establishing a connection between a (thus far) theoretical framework and interpretability research can help initiate a strong line of research where work on both domains complement one another to improve our understanding of Transformers.
>
> > Suggestion: could you add a summary of some of the important questions and answers in Appendix A towards the end of the main text? For example, 1-3 are good candidates.
>
> We thank the reviewer for the suggesion and will incorporate it in the final version.

---

> > ### Author Rebuttal · Reviewer_rd88 · 2026-04-03
> >
> > Thanks for addressing my comments. I maintain my already positive score.

---

### Decision · Program_Chairs · 2026-04-30

**Decision:**

Accept (regular)

**Comment:**

This paper presents an automated pipeline for extracting symbolic D-RASP programs from trained transformers without requiring architectural or training modifications. The approach reparameterizes transformer computations via a linear layer norm assumption (LLNA), prunes sparse structures, and simplifies programs using human-designed primitives. The central finding is that length-generalizing models yield short, interpretable programs while non-generalizing models produce dense, uninterpretable circuits.

The rebuttal effectively addressed the main concerns about human-designed primitives and the handling of non-length-generalizing models. While concerns about scalability to larger models and some presentation issues remain, the central finding remains compelling and well-supported, making this a meaningful contribution.